# An ATP-gated molecular switch orchestrates human mRNA export

Ulrich Hohmann[1,2,6 ✉], Max Graf[1,3], László Tirián[2], Belén Pacheco-Fiallos[1,3], Ulla Schellhaas[1,3], Laura Fin[1], Dominik Handler[2], Alexander W. Phillips[1], Daria Riabov-Bassat[1], Rupert Faraway[1,4,5], Thomas Pühringer[1], Michael-Florian Szalay[1], Elisabeth Roitinger[1,2], Julius Brennecke[2 ✉] & Clemens Plaschka[1 ✉]

The nuclear export of mRNA is an important step in eukaryotic gene expression[1]. Despite recent molecular insights into how newly transcribed mRNAs are packaged into ribonucleoprotein complexes (mRNPs)[2,3], the subsequent events that govern mRNA export are poorly understood. Here we uncover the molecular basis underlying key events of human mRNA export, including the remodelling of mRNP-bound transcription–export complexes (TREX), the formation of export-competent mRNPs, the docking of mRNPs at the nuclear pore complex (NPC), and the release of mRNPs at the NPC to initiate their export. Our biochemical and structural data show that the ATPase UAP56 (also known as DDX39) acts as a central molecular switch that directs nucleoplasmic mRNPs from TREX to NPC-anchored TREX-2 complexes through its ATP-gated mRNA-binding cycle. Collectively, these findings establish a mechanistic framework for a general and evolutionarily conserved mRNA export pathway.

Eukaryotic gene expression requires the nuclear export of newly synthesized mRNA through the NPC for translation in the cytoplasm. To prevent the translation of aberrant RNAs, mRNA export is selective for mature mRNA ribonucleoprotein complexes (mRNPs).

Mature mRNPs are marked by specific proteins, which they acquire during the capping, splicing, cleavage and polyadenylation of their precursor mRNAs[1,4]. By recognizing these maturation marks, the transcription–export complex (TREX) assembles on the surface of packaged mRNPs and selects maturing mRNAs for export[1,2,5]. TREX also aids in mRNA packaging and thereby ensures genome integrity by preventing the formation of harmful RNA–DNA hybrids, called R-loops[6]. However, packaged TREX–mRNP complexes cannot be directly exported[7,8]. Instead, they undergo a two-step remodelling process. First, TREX is disassembled to generate export-competent mRNPs[1]. Second, these remodelled mRNPs engage the NPC, where the mRNA export factor, NXF1–NXT1, facilitates mRNP transport across the NPC's selective permeability barrier[9–11]. Although the factors that are required for mRNA export were identified decades ago[9,12–14], the mechanistic basis of the different steps, leading to the remodelling of mRNPs for nuclear export, as well as how mRNPs navigate through these steps, remains unclear.

Here, using a combination of biochemistry, in silico protein–protein interaction screening, cryo-electron microscopy (cryo-EM), and cellular assays, we identify a general mechanism for mRNA export that assigns molecular functions to key mRNA export proteins and their complexes.

## UAP56 bridges the THO complex to mRNPs

Newly made nuclear mRNPs form compact globules[2,3,15], which are decorated with TREX complexes on their surface[2] (Fig. 1a). To investigate how TREX–mRNP complexes are subsequently remodelled for nuclear export, we focused on how TREX interacts with mRNPs after their recognition and packaging. In humans, TREX comprises a tetramer of the six-subunit THO complex, each containing THOC1, THOC2, THOC3, THOC5, THOC6, THOC7, four UAP56 (in yeast, Sub2) molecules and various mRNA export adaptors such as ALYREF (in yeast, Yra1)[9,13]. ALYREF interacts directly with mRNP-bound maturation marks, such as the exon junction complex (EJC) or the cap-binding complex, through its RNA-recognition motif domain[2,16,17]. Moreover, ALYREF binds to UAP56 through its N- and C-terminal UAP56-binding motifs (N- and C-UBM)[9,18], although only the C-UBM had been observed in structures[2,19]. UAP56 is a DExD-box ATPase, whose two RecA lobes, RecA1 and RecA2, can clamp onto RNA together with ATP[19]. In the cryo-EM structures of native TREX–mRNP complexes[2], the four UAP56 molecules are primed for mRNA clamping[2]; the UAP56 RecA1 and RecA2 lobes are coordinated through interactions with their cognate THOC2 subunit[19–22] (Fig. 1b,c), and the UAP56 RecA1 lobe connects to the mRNP by binding to the ALYREF C-UBM[2,18] (Fig. 1b–d).

ALYREF has a UBM at its N terminus[23]. While it is thought that this N-UBM mimics the C-UBM in binding to the RecA1 lobe of UAP56[18,20–22], the amino acid sequences of the two UBMs differ despite each being highly conserved (Extended Data Fig. 1a). When we re-examined published TREX–mRNP maps[2], we identified a low-resolution density consistent with an AlphaFold2 prediction of the ALYREF N-UBM with the UAP56 RecA2 lobe (Fig. 1b,c and Extended Data Fig. 1b–g). Furthermore, cryo-EM of a minimal, reconstituted TREX–mRNP complex revealed the THO–UAP56 protomer at 4.1 Å resolution, enabling us to unambiguously assign the ALYREF N-UBM on the UAP56 RecA2 lobe (Extended Data Fig. 2b–g and Extended Data Table 1). Mutating either

[1]Research Institute of Molecular Pathology (IMP), Vienna BioCenter (VBC), Vienna, Austria. [2]Institute of Molecular Biotechnology of the Austrian Academy of Sciences (IMBA), Vienna BioCenter (VBC), Vienna, Austria. [3]Vienna BioCenter PhD Program, Doctoral School of the University of Vienna and Medical University of Vienna, Vienna, Austria. [4]Max Perutz Labs, Vienna BioCenter (VBC), Vienna, Austria. [5]University of Vienna, Max Perutz Labs, Department of Biochemistry and Cell Biology, Vienna, Austria. [6]Present address: Institute of Molecular Biology (IMB), Mainz, Germany. ✉e-mail: u.hohmann@imb-mainz.de; julius.brennecke@imba.oeaw.ac.at; clemens.plaschka@imp.ac.at

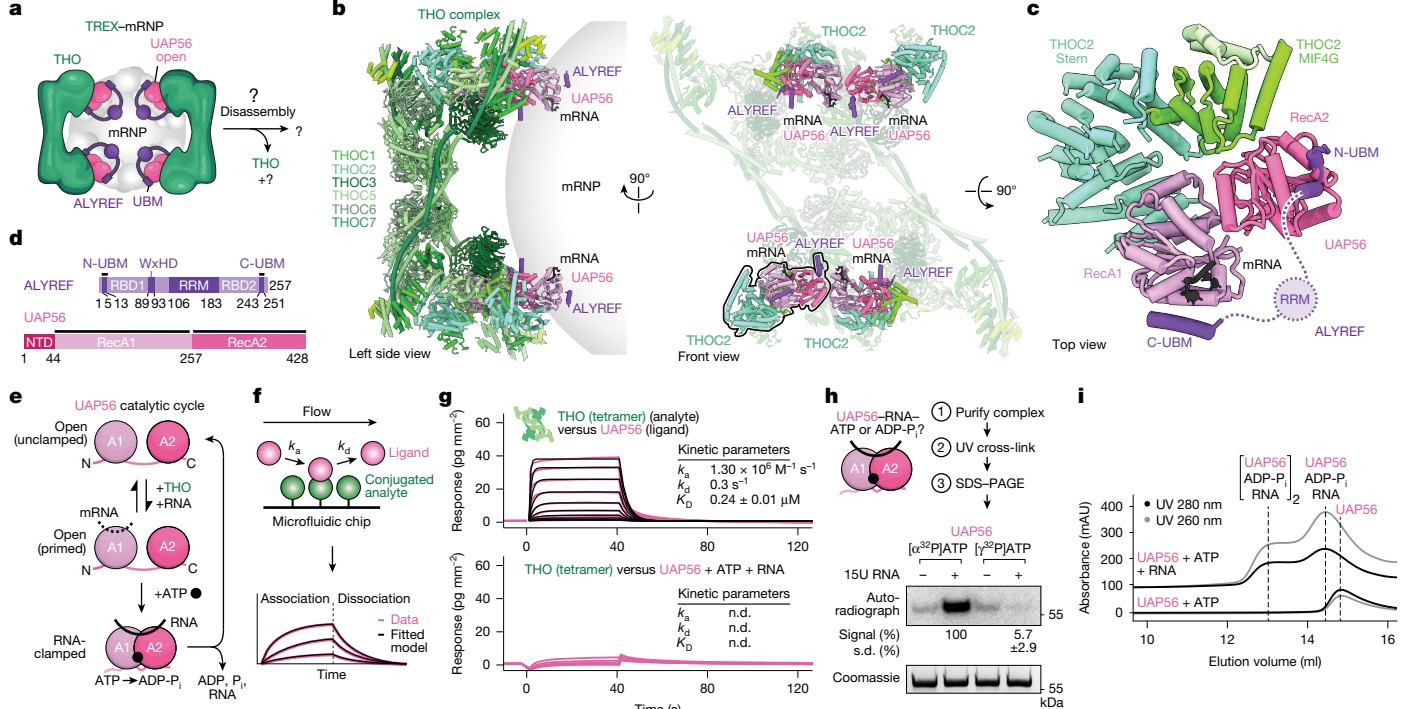

**Fig. 1 | UAP56 controls TREX–mRNP assembly and disassembly. a**, Schematic of the TREX–mRNP complex. For mRNP export, THO must dissociate from the TREX–mRNP complex and the mRNP must be remodelled. **b,c**, The revised cryo-EM structure of a human TREX–mRNA complex contains the ALYREF N- and C-UBMs, shown in two views (**b**), and a magnification of the UAP56 interfaces with THOC2 and ALYREF, viewed from the top (**c**). See also Extended Data Fig. 2g. Shades of green, THO; shades of pink, UAP56; purple, ALYREF; grey sphere, mRNP. **d**, The domain organization of ALYREF and UAP56. Regions included in the atomic model are indicated with a black line. RBD, RNA-binding domain; RRM, RNA recognition motif. **e**, Schematic of the RNA- and ATP-dependent UAP56 catalytic cycle. **f**, Experiment schematic for GCI analysis. **g**, GCI-derived binding kinetics for the recombinant tetrameric THO complex (immobilized) probed with UAP56, or UAP56, ATP and RNA. Sensograms (pink line), fitted model (black) and a binding kinetics summary table are shown. n.d., not detected. **h**, Ultraviolet (UV) cross-linking of UAP56 to radioactive [α$^{32}$P]ATP or [γ$^{32}$P]ATP, with or without 15-nucleotide poly-uridine RNA (experiment scheme on top). After removing excess ATP and RNA from immobilized UAP56, the nucleotide is cross-linked to UAP56 with UV light at 254 nm and visualized by SDS–PAGE using an autoradiograph (middle) and Coomassie-staining (bottom). The radioactive signal in the '+RNA' condition is quantified from three independent experiments. **i**, Size-exclusion chromatography of UAP56 and ATP with or without 15 nucleotide poly-uridine RNA. UV traces at 280 nm (black) and 260 nm (grey) are shown. See Extended Data Fig. 3g for additional controls. mAU, milli-absorbance units.

UBM-binding site in UAP56 selectively abolished the binding to N-UBM or C-UBM peptides in vitro (Extended Data Fig. 3a) and caused severe growth defects in human K562 cells (Extended Data Fig. 3b–d).

Thus, ALYREF binds to two distinct sites on UAP56, forming unique composite surfaces that could be used for mRNA export (see below). Furthermore, we observe UAP56 as the only protein bridging between the THO complex and ALYREF-bound mRNPs (Fig. 1b,c).

## RNA clamping releases UAP56 from the THO complex

To facilitate the nuclear export of mRNPs, TREX disassembles through an unknown mechanism[1,7,8]. Given the central position of UAP56 in TREX, bridging between THO and the mRNP, we investigated whether the ATP-dependent mRNA-clamping of UAP56 (ref. 19) might have a role in TREX disassembly. To test this, we used grating-coupled interferometry (GCI) to measure the binding affinity of the recombinant and surface-immobilized THO complex to either UAP56 alone or to UAP56 preincubated with ATP, or with RNA, or with both ATP and RNA (Fig. 1f,g and Extended Data Fig. 3e,f). Only the latter pre-incubation, with both ATP and RNA, allows UAP56 to adopt its RNA-clamped conformation[24]. While the THO complex bound to isolated UAP56, irrespective of the addition of either ATP or RNA, with $K_D$ values of approximately around 0.24–0.39 μM (Fig. 1g and Extended Data Fig. 3e,f), THO exhibited no measurable binding affinity for RNA-clamped UAP56 formed in the presence of ATP and RNA (Fig. 1g and Extended Data Fig. 3e).

This suggested that TREX dissembles once UAP56 clamps onto RNA. DExD-box family ATPases, including UAP56, act as RNA clampases that clamp rather than translocate on RNA. Some DExD-box ATPases clamp RNA with the highest affinity immediately after ATP hydrolysis, in their ADP- and inorganic phosphate ($P_i$)-bound state[25–27]. Consistently, we observed the near-complete hydrolysis of ATP in RNA-clamped UAP56 complexes (Fig. 1h). The resulting RNA-clamped complexes were stable, including during size-exclusion chromatography (Fig. 1i and Extended Data Fig. 3g), showing that UAP56 can form longer-lived RNA-bound complexes containing ADP-$P_i$. The lack in affinity of UAP56–ADP-$P_i$–RNA complexes for THO indicated that RNA clamping may be important to dissociate UAP56 from the THO complex. Supporting this model, mutation of the DExD-box ATPase motif in UAP56, which prevents ATP hydrolysis and RNA binding in vitro[24], leads to mRNA export defects in yeast[28] and impairs human cell viability (Extended Data Fig. 3b–d).

Notably, when we pre-formed THO–UAP56 complexes using recombinant proteins, we observed inefficient complex disassembly after ATP and RNA addition (Extended Data Fig. 3h). Given that TREX disassembly would require the coordinated release of all four THO-bound UAP56 molecules and because we observed no cooperative binding between UAP56 and tetrameric, dimeric or monomeric THO complexes (Fig. 1g and Extended Data Fig. 3e), we hypothesized that other factors, in addition to RNA-clamping by UAP56, may assist the efficient disassembly of multivalent TREX–mRNPs (Fig. 2a).

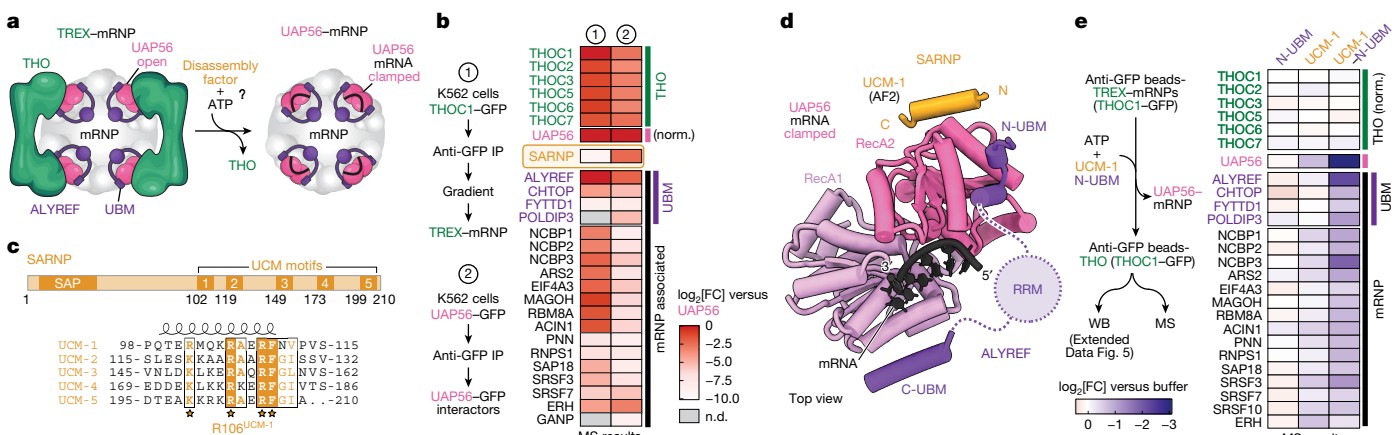

**Fig. 2 | SARNP assists TREX–mRNP disassembly. a**, Schematic of TREX–mRNP disassembly, which may require additional disassembly factors for efficient remodelling. **b**, Comparison of the abundance of mRNP-associated proteins in purified TREX–mRNPs[2] versus the immunoprecipitation (IP) of nuclear GFP–UAP56 obtained from MS. The heat map is coloured according to the log2-transformed fold change (log2[FC]) after normalizing (norm.) to UAP56 levels. **c**, Domain organization (top) and multiple-sequence alignment (bottom) of human SARNP (UniProt: P82979) and its five UCMs. Residues invariant or conserved among the five UCMs (UCM-1–5) are highlighted in orange. **d**, Model of a clamped UAP56–ADP-P$_i$–RNA complex bound to SARNP and ALYREF.

The model was obtained by superposing structures of RNA-clamped UAP56 (Protein Data Bank (PDB): 8ENK)[33], open UAP56–ALYREF N- and C-UBMs (Fig. 1) and a UAP56–SARNP UCM-1 AlphaFold2 Multimer prediction (Extended Data Fig. 3i–l). Pink, UAP56; black, RNA; orange, SARNP; purple, ALYREF. **e**, Native TREX–mRNP disassembly assay. Experiment schematic (left) and MS results (right, heat map) of bead-retained mRNP-associated proteins after adding the ALYREF N-UBM, SARNP UCM-1 or an UCM-1–N-UBM fusion are shown. The heat map is coloured according to the log2[FC] compared with the buffer control, after normalizing to the mean THO complex subunit levels. Additional details are provided in Extended Data Fig. 5e,f. WB, western blot.

## SARNP is a multivalent TREX disassembly factor

To identify factors that assist TREX disassembly, we compared the protein composition of endogenously purified TREX-bound mRNPs[2] with the nuclear UAP56 protein interactome in human cells (Fig. 2b and Supplementary Table 1). This revealed the protein SARNP (CIP29; in yeast, Tho1), which was absent from purified TREX–mRNPs but highly enriched in the UAP56 interactome. SARNP is broadly conserved across eukaryotes and has been implicated in mRNA export in yeast[29], plants[30,31] and humans[32,33]. Moreover, SARNP binds to human mRNAs in vivo[34,35] and RNA-clamped UAP56 in vitro[32,33] in the absence of the THO complex[31], consistent with the anticipated activities of a factor that aids TREX disassembly.

Using AlphaFold2, we predicted a direct interaction between the human UAP56 RecA2 lobe and SARNP, which was indistinguishable from a recent crystal structure of a chimeric human UAP56–yeast SARNP complex[33] (Fig. 2c,d and Extended Data Figs. 3i–l and 4a,b). Mutation of the highly conserved residue R106D in the SARNP motif (residues 81–115) or of D283R in the UAP56 RecA2 lobe disrupted their interaction in vitro (Extended Data Fig. 4a–c). Consistently, the UAP56(D283R) mutant caused a growth defect in human K562 cells (Extended Data Figs. 3b–d and 4d,e) and impaired the cellular UAP56–SARNP interaction (Extended Data Fig. 4f). Owing to its biochemical activities described below, we refer to SARNP's UAP56-interacting peptide as the UAP56-clamping motif (UCM). The UCM is found five times in human SARNP[33] with the consensus sequence R/KxxxRAxRFG, and all five UCMs bind to UAP56 (Fig. 2c and Extended Data Fig. 4g,h). Mutation of the central R in these five UCMs abolished the interaction of full-length SARNP with UAP56, while truncation of SARNP's annotated N-terminal SAP domain had no effect (Extended Data Fig. 4i).

The binding site for the SARNP UCMs on the UAP56 RecA2 lobe is located directly adjacent to the newly identified binding site for the ALYREF N-UBM (Figs. 1b,c and 2d), suggesting that UCM and N-UBM might bind synergistically. To investigate this, we first showed that purified SARNP UCM-1 (hereafter UCM), ALYREF N-UBM and, additionally, ALYREF C-UBM peptides could bind to UAP56 simultaneously in vitro (Extended Data Fig. 4j–l). We next determined the affinities between UAP56 and isolated UCM-1 peptide (10 ± 2 μM) or N-UBM

peptide (28 ± 8 μM) (Extended Data Figs. 3a and 5a). We then generated different structure-guided 'single-chain fusions' with UAP56 to stabilize low-affinity peptide–UAP56 interactions (Extended Data Fig. 5b,c). We observed enhanced binding of an ALYREF N-UBM peptide to a UAP56–UCM fusion protein (Extended Data Fig. 5b,c), indicating that the synergistic binding of N-UBM and UCM peptides could be important in mRNPs, examined further below.

Notably, binding of a SARNP UCM to UAP56 would sterically clash with the interaction between UAP56 and the THOC2 MIF4G domain in TREX–mRNPs (Extended Data Fig. 4a,b). UCM binding to UAP56 could thereby prevent RNA unclamping and rebinding of UAP56 to THOC2, keeping UAP56 mRNA-clamped, and promoting TREX disassembly and the directionality of these steps. To test this model, we reconstituted the THO–UAP56 complex on beads and examined its integrity after addition of purified ALYREF N-UBM, SARNP UCM peptide, a SARNP UCM-1–ALYREF N-UBM fusion peptide or full-length SARNP, in the presence of ATP and RNA. Among these conditions, the recombinant THO–UAP56 complex disassembled most efficiently when adding the UCM–N-UBM fusion peptide (Extended Data Fig. 5d). These combined data indicate that the synergistic and multivalent binding of SARNP and the ALYREF N-UBM to RNA-clamped UAP56 promotes the efficient dissociation of UAP56 from the THO–UAP56 complex, thus disassembling TREX.

To challenge this model in a more native setting, we assessed whether SARNP and mRNA-clamping by UAP56 could disassemble TREX on endogenous mRNPs. We immobilized TREX–mRNPs purified from the nuclear extract of human K562 cells through the GFP-tagged THO subunit THOC1 (refs. 2,20) and added ATP, or ATP together with either the ALYREF N-UBM, the SARNP UCM or a UCM–N-UBM fusion peptide (Fig. 2e). We measured the release of UAP56–mRNPs from immobilized THO complexes using mass spectrometry (MS) and western blotting (Fig. 2e, Extended Data Fig. 5e,f and Supplementary Table 2). While the addition of the N-UBM alone did not result in mRNP release (Fig. 2e and Extended Data Fig. 5e), addition of the SARNP UCM or of full-length SARNP resulted in detectable mRNP release (Fig. 2e and Extended Data Fig. 5e,f). We note that this release may have been aided by endogenous ALYREF, which co-purifies with mRNPs[2]. This release effect was further enhanced by the addition of the UCM–N-UBM fusion (Fig. 2e and Extended Data Fig. 5e). Consistent with a role downstream of THO,

when we acutely depleted SARNP in human K562 cells using the dTAG system, we observed no decrease in UAP56's interaction levels with either ALYREF or THO (Extended Data Fig. 5g,h). Collectively, these data support a role for UAP56 as the central bridge between the THO complex and the mRNP and demonstrate that the ATP-dependent RNA-clamping of UAP56, assisted by ALYREF and SARNP, is sufficient to disassemble TREX–mRNPs.

## SARNP stabilizes UAP56–RNA complexes in vitro

To prevent reassociation of the mRNP with THO, UAP56 must stably clamp onto mRNA. We hypothesized that SARNP and ALYREF may stabilize RNA-clamped UAP56. Indeed, we observed that SARNP, as well as individual or joint fusions of ALYREF N-UBM and a SARNP UCM to UAP56 enhanced the UAP56–RNA interaction in vitro with up to around sixfold higher RNA affinity ($K_D = 0.12 \pm 0.03\,\mu M$) compared with wild-type (WT) UAP56 ($K_D = 0.78 \pm 0.09\,\mu M$) (Extended Data Fig. 5i,j). SARNP and UAP56 RNA clamping may thereby not only promote THO release but also stabilize UAP56–mRNP complexes. RNA-clamped UAP56 could therefore determine the downstream fate of the mRNP, such as mRNP docking at the NPC.

## UAP56–RNA binds to the NPC-anchored TREX-2 complex

Single-molecule tracking experiments of mRNAs in yeast and human cells revealed that mRNPs transiently dock at NPCs before their nuclear export[36–39], but the mechanism underlying these events is unclear (Fig. 3a). To identify proteins that might engage with UAP56–mRNPs after TREX disassembly, we generated a list of putative UAP56 interactors based on their greater than twofold enrichment in UAP56 immunoprecipitates from K562 cell nuclear extract (Fig. 2b). We then performed a pairwise AlphaFold2 Multimer interaction screen between each of these candidates and UAP56 and ranked the results by their interface prediction TM (ipTM) scores (Fig. 3b and Supplementary Table 3). Top-scoring candidates included known UAP56 interactors, such as THOC2, SARNP, and the N- or C-UBM containing export adaptors ALYREF, CHTOP, UIF and PHAX[18], as well as new putative interactors with roles in nuclear mRNA metabolism, including RBM26, RBM27 and NCBP3[40,41] (Extended Data Fig. 6).

Among the top-ranking predicted UAP56 interactors were GANP and PCID2, which are two of the five subunits of the NPC-anchored TREX-2 complex[14,42–46] (Fig. 3b,c). TREX-2 is required for mRNA export, but its molecular functions are unclear[14,47]. The human TREX-2 complex consists of GANP, PCID2, SEM1, ENY2, and CETN2 or CETN3 (yeast, Sac3, Thp1, Dss1, Cdc31 and Sus1, respectively)[14,42–46]. The GANP subunit scaffolds the four other subunits and anchors TREX-2 to the nuclear basket of the NPC[14,48,49]. The predicted interaction between UAP56 and TREX-2 therefore suggested a model in which TREX and TREX-2 act in a linear pathway: UAP56–mRNPs, after their release from THO, might dock at the NPC through TREX-2 to facilitate mRNA nuclear export.

To investigate whether UAP56 binds to GANP and PCID2, we purified a minimal recombinant TREX-2 complex (previously termed TREX-2[M]; ref. 14) comprising the GANP Sac3 domain (residues 582–1004), PCID2 and SEM1 (Fig. 3c). In in vitro pull-down experiments, TREX-2[M] bound to UAP56 in a stochiometric complex, confirming their direct interaction (Fig. 3d and Extended Data Fig. 7a). This UAP56–TREX-2[M] complex could still bind to the ALYREF N-UBM, C-UBM or SARNP UCM peptides (Extended Data Fig. 7a,b).

To reveal the molecular interfaces of the UAP56–TREX-2 complex, we determined cryo-EM structures of the TREX-2[M] complex in isolation (3.5 Å resolution) and bound to UAP56 (3.5 Å resolution) (Fig. 3e, Extended Data Figs. 7c–g and 8 and Extended Data Table 1). The cryo-EM structure of the human TREX-2[M] complex in isolation was similar to reported structures of the yeast TREX-2[M] complex[50–53] (Extended Data

Fig. 9a,b), exhibiting a V-shaped architecture made of the GANP Sac3 domain and PCID2 (Fig. 3e). SEM1 is largely unstructured and binds to PCID2 (Fig. 3e). In our UAP56–TREX-2[M] structure, the N-terminal half of PCID2 rotates slightly outwards to accommodate UAP56, and regions in the GANP N terminus become ordered compared with apo TREX-2[M] (Fig. 3e). Although we performed these experiments with RNA-bound UAP56, the cryo-EM structure shows UAP56 in an RNA-unbound conformation. While the UAP56 RecA1 lobe is well resolved and binds to the 'V' formed by GANP and PCID2, the RecA2 lobe is mobile (Fig. 3e and Extended Data Fig. 8b,d,e). These findings suggested that UAP56 might facilitate the docking of its bound mRNPs at the NPC through interactions with TREX-2, and that TREX-2 may subsequently remodel UAP56–mRNP complexes (Fig. 4a).

## The conserved NTD of UAP56 binds to TREX-2

UAP56 has an unstructured N-terminal domain (NTD) that is conserved from yeast to humans (Extended Data Fig. 9c). Although required for mRNA export[28], the molecular function of the NTD is unclear. In the UAP56–TREX-2[M] structure, the UAP56 NTD binds between the GANP and PCID2 winged-helix (WH) domains (through UAP56 residues 10–15, NTD interface I) and along the GANP Sac3 domain (through UAP56 residues 39–44, NTD interface II; Fig. 4b). Consistent with the structure, deletion of UAP56 residues 1–28, which includes NTD interface I, resulted in an approximately 2.5-fold reduction in UAP56–TREX-2[M] affinity ($K_D = 0.15 \pm 0.02\,\mu M$) compared with full-length UAP56 ($K_D = 0.07 \pm 0.01\,\mu M$) (Fig. 4c and Extended Data Fig. 9d). UAP56 lacking the entire NTD (residues 1–43, UAP56ΔNTD) displayed a more than tenfold reduced affinity ($K_D = 0.95 \pm 0.05\,\mu M$) (Fig. 4c and Extended Data Fig. 9e). Moreover, the isolated UAP56 NTD peptide (residues 1–21) was sufficient to bind to TREX-2[M] in vitro and in nuclear cell extracts, whereas mutated NTD peptides were not (Extended Data Fig. 9f–h). Furthermore, mutations affecting conserved residues in the WH domains of GANP and PCID2 have been shown to lead to mRNA export defects in yeast in vivo[50]. These TREX-2[M] mutations would critically contribute to the newly identified interface between TREX-2[M] and the negatively charged UAP56 NTD (Fig. 4b).

Four experimental assays demonstrate that the UAP56–TREX-2 interfaces are functionally relevant in cells: first, we examined the impact of different UAP56 mutations in a cell-based RNA export tethering assay. Aptamer-mediated tethering of UAP56 to a reporter pre-mRNA promoted its export to a degree comparable to the tethering of the mRNA export factor NXF1 (refs. 54,55) (Fig. 4d and Extended Data Fig. 9i,j), consistent with UAP56 promoting mRNP export after TREX disassembly. The combined mutation of critical residues in the UAP56 NTD interface I (L10S, L11S, D12K, Y13S) reduced the export-promoting effect compared to WT UAP56, while removal of the entire UAP56 NTD strongly reduced its export-promoting ability (Fig. 4d and Extended Data Fig. 9i,j), in agreement with our in vitro results (Fig. 4c). This reduction in the export-promoting effect was comparable to mutations of UAP56 RecA1 residues that face PCID2 in our UAP56–TREX-2[M] structure (D49A, L51W, Q78A, L81K) (Fig. 4d and Extended Data Fig. 9i). As expression levels and nuclear import of the different λN-tagged UAP56 constructs were unaffected (Extended Data Fig. 9j), the observed export defects are most likely due to an impaired UAP56–TREX-2 interaction. Tethering of the TREX-2 subunits PCID2, CETN3 or ENY2 to the reporter pre-mRNA did not promote export (Fig. 4d), presumably because the human TREX-2 complex is constitutively anchored to the NPC basket[56]. In a second experiment, we truncated the UAP56 NTD in a CRISPR–Cas9 knockout–rescue assay, leading to a severe growth defect in human K562 cells (Extended Data Fig. 4d,e). Third, we probed the UAP56 NTD–PCID2 interface by mutating PCID2. We generated a human K562 cell line to acutely deplete PCID2 using the dTAG system (Extended Data Fig. 10a,b). While the ectopic expression of WT PCID2 fully rescued PCID2–dTAG depletion, expression of the PCID2 mutant

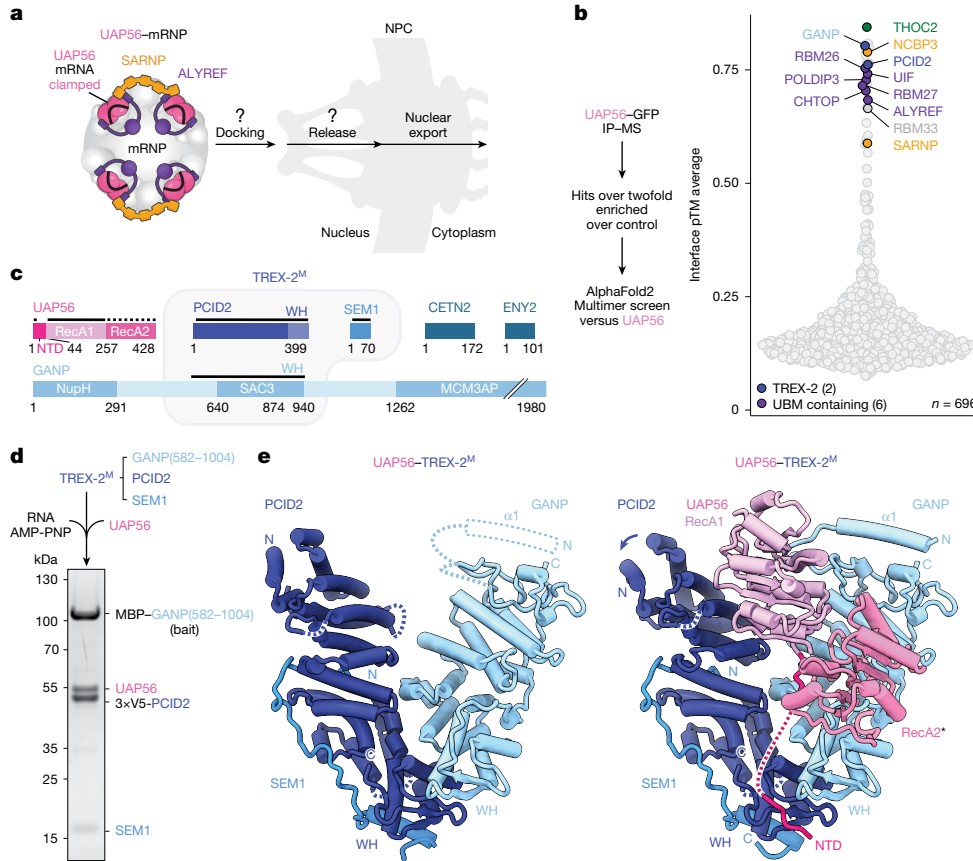

**Fig. 3 | UAP56 binds to the NPC-anchored TREX-2 complex. a**, Schematic of the UAP56–mRNP complex, which docks at the NPC through an unknown mechanism before export. **b**, AlphaFold2 Multimer in silico screen predicts novel UAP56 interactors. An experiment schematic (left) and the screen results (right) are shown. UAP56–candidate predictions are ranked by the average interface pTM score (ipTM average). Predicted interactors of interest are highlighted with colours. **c**, The domain organization of UAP56 (pink) and TREX-2 complex subunits (blue). Regions included in the atomic model are indicated with a black line, and rigid-body fits are shown with a dotted line. NupH, nucleoporin homology. **d**, Reconstitution of a recombinant UAP56–TREX-2^M complex (Extended Data Fig. 7c). SDS–PAGE analysis (Coomassie stain) of a representative in vitro protein pull-down is shown. **e**, Cryo-EM structures of human TREX-2^M (left) and UAP56–TREX-2^M (right) viewed from the front. The UAP56 RecA2 lobe, marked with an asterisk, is putatively fitted based on a low-resolution density (Extended Data Fig. 8). Blue, SEM1; dark blue, PCID2; light blue, GANP; shades of pink, UAP56.

(K374D and K388D) in the UAP56 NTD interface was lethal (Extended Data Fig. 10c,d). This mutant PCID2 protein was also impaired in binding cellular UAP56 (Extended Data Fig. 10e). Fourth, we carried out poly(A) RNA FISH in human cells using the PCID2–dTAG cell line (Fig. 4e and Extended Data Fig. 11). Nuclear poly(A) RNA FISH signal accumulated after PCID2–dTAG depletion, consistent with a block in mRNA nuclear export. This effect was of a comparable magnitude to the independent GANP–dTAG depletion (Fig. 4e and Extended Data Fig. 11). The ectopic expression of WT PCID2 could fully rescue the poly(A) RNA signal after PCID2–dTAG depletion, but a PCID2 mutant in the UAP56 NTD interface could not.

Collectively, these data suggest that the interaction of UAP56 with the NPC-anchored TREX-2 complex is important for mRNA nuclear export. In cells, the efficient docking of mRNPs at the NPC may be further enhanced by multivalent interactions between multiple UAP56 molecules of the mRNP and multiple TREX-2 complexes at the NPC, owing to the NPC's eightfold symmetry.

## TREX-2 triggers RNA release from UAP56

For export across the NPC, mRNPs must eventually dissociate from TREX-2, which is anchored to the NPC's basket (Fig. 4a). A clue as to how this might happen came from our UAP56–TREX-2^M structure. Although we prepared the UAP56–TREX-2^M cryo-EM sample in the presence of RNA and non-hydrolysable AMP-PNP, UAP56 is not RNA-clamped in the structure (Figs. 3e and 4b). Instead, the UAP56 RecA1 lobe is sandwiched between PCID2 and a highly conserved loop within GANP (residues 674–686) (Figs. 3e and 4b and Extended Data Figs. 9b and 10f). This GANP loop, which we named the wedge, is visible only in the UAP56–TREX-2^M structure, and not in the isolated human TREX-2^M (Fig. 3e) or in a published yeast TREX-2^M cryo-EM structure[53] (Extended Data Fig. 9a). In our UAP56–TREX-2^M structure, the GANP wedge adopts a position near the UAP56 RecA1 lobe, which would be occupied by the RecA2 lobe in RNA-clamped UAP56[19]. Notably, the UAP56–TREX-2^M complex contains the AMP-PNP nucleotide, which is bound between UAP56 RecA1 residue F65 and the evolutionarily invariant GANP wedge residue R678 (Fig. 4b and Extended Data Fig. 10f,g). At this location, GANP R678 substitutes for F381 of UAP56 RecA2, which would coordinate the nucleotide in RNA-clamped UAP56 (refs. 19,33) (Fig. 4b and Extended Data Fig. 10g). These data suggest that the GANP wedge could promote the release of RNA from UAP56, consistent with a previous observation implicating TREX-2 in the removal of UAP56 from yeast mRNPs[44].

As the release of RNA from DExD-box ATPases is coupled to ADP and P_i release, we investigated whether TREX-2^M affects the apparent ATPase activity of UAP56. Using an in vitro ATPase assay (Fig. 4f and Extended Data Fig. 10h), we observed that recombinant TREX-2^M stimulates the ATPase activity of UAP56 by more than fiftyfold in the presence of RNA (Fig. 4f and Extended Data Fig. 10h). A single point mutation of the

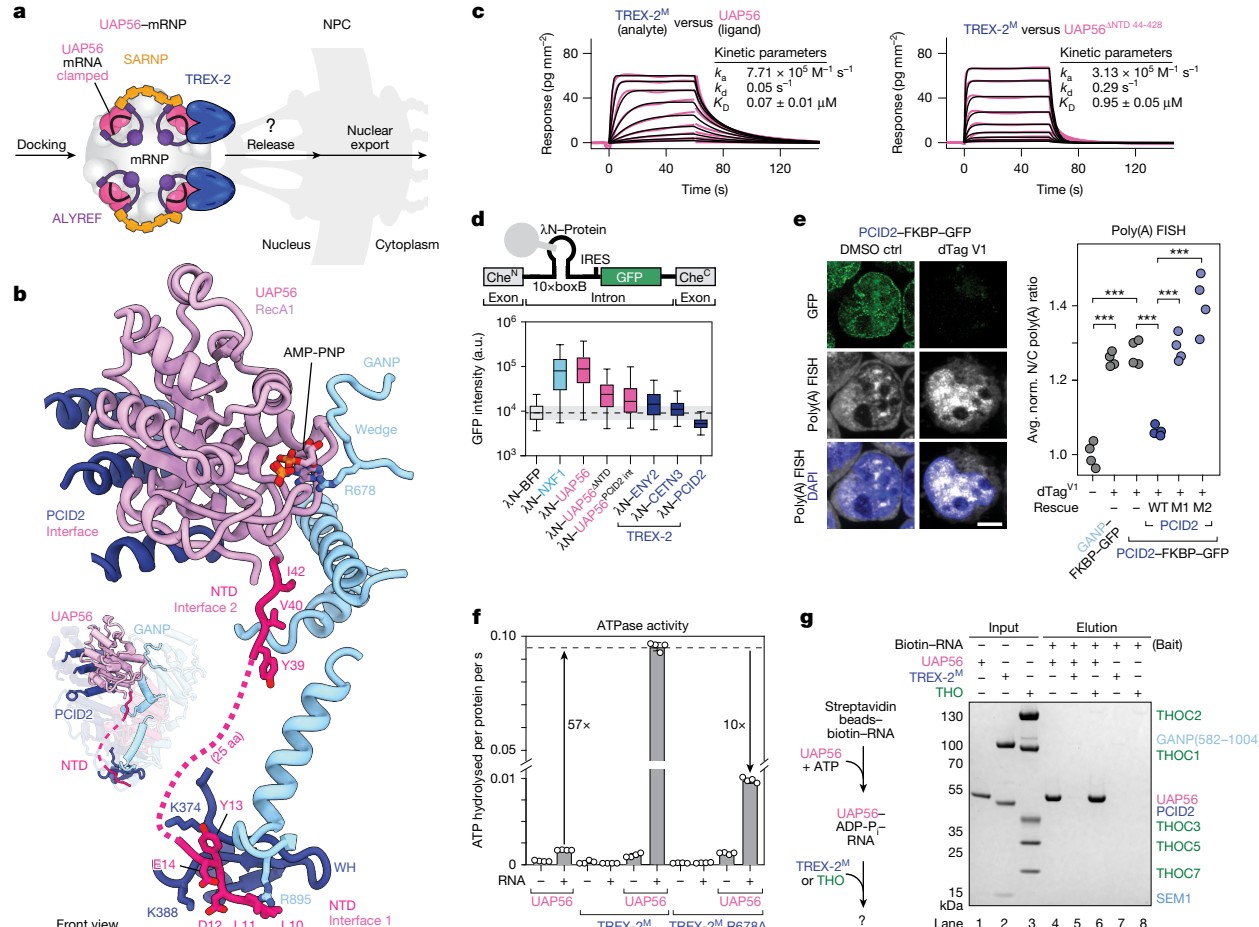

**Fig. 4 | UAP56–TREX-2 interfaces and mRNA release. a**, Schematic of an UAP56–mRNP docked at the NPC, which needs to be released from TREX-2 for mRNP export. **b**, Details of UAP56–TREX-2$^M$ interfaces. TREX-2 regions that are not involved in UAP56 binding (inset) are omitted for clarity. Side chains of key interface residues and AMP-PNP are shown as sticks. Colours are as defined in Fig. 3. **c**, GCI-derived binding kinetics for TREX-2$^M$ (immobilized) probed with UAP56 or UAP56ΔNTD (residues 44–428). Sensograms (pink line), the fitted model (black) and a binding kinetics summary table are shown. **d**, RNA-export tethering assay. λN-tagged proteins bind to a reporter RNA construct containing boxB RNA aptamers (top). When exported, the reporter RNA is translated into GFP, which is quantified by fluorescence-activated cell sorting (FACS) (top) (Extended Data Fig. 9i,j). The box plots show the median (centre line), interquartile range (25th–75th percentiles; box limits), and the whiskers extend to the 5th and 95th percentiles. Minimum $n = 40,000$ cells in three independent experiments. a.u., arbitrary unit; PCID2 int, PCID2 interface. **e**, Mutation of the UAP56 NTD–PCID2 interface in PCID2 accumulates nuclear poly(A) RNA. Shown are representative cells (left; z-projection; Extended Data Fig. 11b) and the ratios

of the nucleocytoplasmic (N/C) poly(A) RNA FISH signal (right) after the depletion of endogenous PCID2 or GANP for 16 h, or after the depletion and rescue of endogenous PCID2 with WT or mutant PCID2 constructs M1 (PCID2 NTD-binding mutant (K374D/K388D)) and M2 (PCID2 GANP-binding mutant (D356R/A365F)). Scale bar, 10 µm. Four replicates per condition, with >70 cells per replicate. Pairwise significance testing was performed using two-sided Welch t-tests, with false-discovery rate (FDR) correction for multiple testing; ***$P < 0.001$. Details and exact P values are provided in Extended Data Fig. 11c,d. Avg., average. **f**, TREX-2$^M$ stimulates UAP56's ATPase activity in vitro, measured as ATPase rates (molecules of ATP hydrolysed per protein per second) with and without 15 nucleotide poly-uridine RNA. Data are mean ± s.d. from four independent samples. **g**, UAP56 RNA-unclamping assay. Bead-immobilized 15-nucleotide poly-uridine RNA was incubated with UAP56 and ATP to form UAP56–ADP-P$_i$–RNA complexes, which were then challenged with recombinant THO or TREX-2$^M$ complexes. Bead-remaining UAP56–ADP-P$_i$–RNA complexes were analysed using SDS–PAGE (Coomassie-staining).

GANP wedge residue R678 to alanine reduced the stimulatory effect of TREX-2$^M$ approximately tenfold (Fig. 4f and Extended Data Fig. 10h), without affecting UAP56–mutant TREX-2$^M$ binding (Extended Data Fig. 9e). As RNA-clamped UAP56 complexes contain ADP and Pi (Fig. 1h), TREX-2 would probably stimulate UAP56 by dissociating RNA, ADP and P$_i$ from UAP56, rather than promoting ATP hydrolysis itself. To test this, we immobilized UAP56–ADP-P$_i$–RNA complexes through RNA on beads and incubated these with either TREX-2$^M$, the TREX-2$^M$ GANP wedge mutant (R678A) or with the THO complex (Fig. 4g and Extended Data Fig. 10i). While TREX-2$^M$ unclamped all UAP56 from the RNA, the TREX-2$^M$ GANP mutant was less efficient, consistent with the ATPase assay (Fig. 4f). By contrast, the THO complex had no measurable effect on the unclamping of UAP56–ADP-P$_i$–RNA complexes, consistent with

our GCI data (Fig. 1g and Extended Data Fig. 2e) and the proposed role of THO in the loading, but not unloading of UAP56 from RNA.

Taken together, these data suggest that TREX-2 may function not only as the nuclear docking site for UAP56–mRNPs at the NPC, but also as the site at which UAP56 dissociates from mRNPs.

## A general model for mRNA export

The data presented in this study offer a framework for understanding mRNA export (Fig. 5a). Central to this model is the ATPase UAP56, which orchestrates a linear process that guides mRNAs through distinct molecular complexes, from the completion of mRNP biogenesis to mRNP docking and remodelling at the NPC before export. Synthesizing

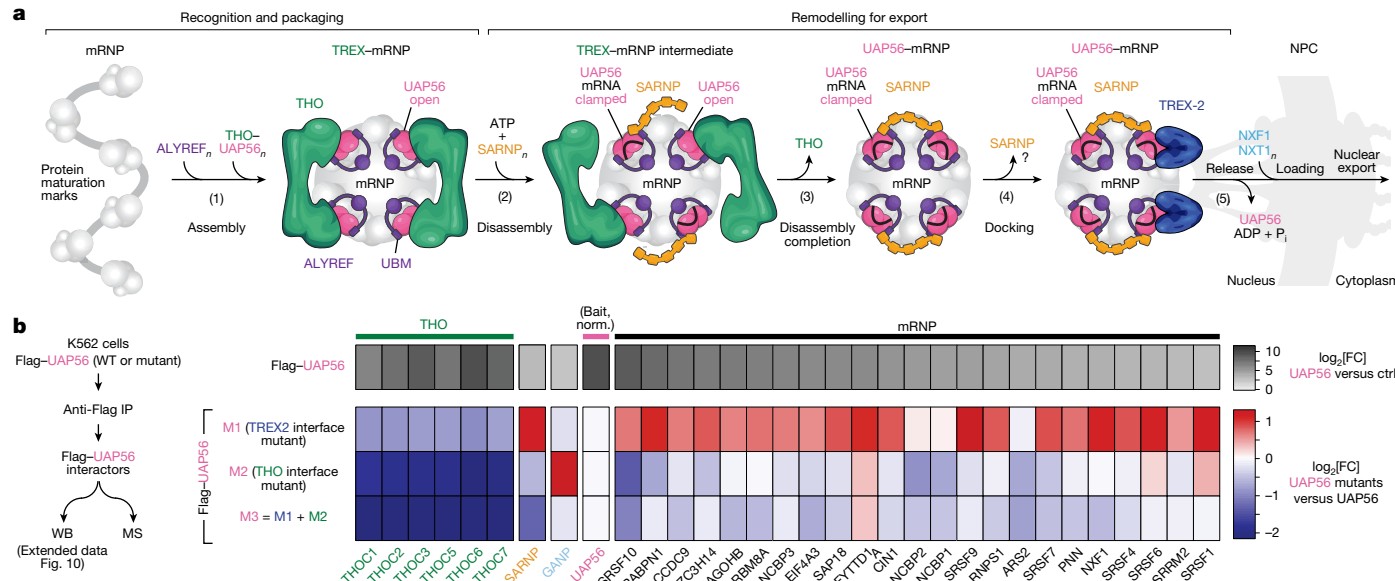

**Fig. 5 | A general model for mRNA nuclear export. a**, The RNA clampase UAP56 acts as a molecular switch to direct human mRNAs to (1) assemble and (2) disassemble TREX–mRNPs; (3) form UAP56–mRNPs aided by SARNP; (4) dock; and (5) release at the NPC through TREX-2. Loading of the mRNA export factor NXF1–NXT1 onto mRNPs may occur in the nucleoplasm[73] or at the NPC[74,75], initiating mRNP nuclear export. These illustrated steps may occur co-transcriptionally. **b**, Immunoprecipitations followed by quantitative MS analysis of WT or three UAP56 protein mutants probe the mRNA export model. Left, experiment schematic. The heat map at the top (greyscale) shows $\log_2$-transformed fold changes in protein enrichment of WT UAP56 versus a control (ctrl). The bottom heat maps (blue–white–red scale) show the fold changes in three UAP56 mutants versus WT UAP56. The enrichment of GANP in UAP56 mutant M2 is likely due to the binding of free nuclear UAP56 mutant M2 protein to TREX-2. Experiment outcomes are discussed in detail in Extended Data Fig. 12a.

previous insights, we propose a five-step pathway for the sequential events governing mRNA export (Fig. 5a).

First, during mRNA transcription and maturation, ALYREF and other mRNA export adaptors[18,55] bind to the newly made mRNP through specific protein marks, which initiates the selective packaging of mRNA into mRNP globules through low affinity and multivalent protein–protein and protein–mRNA interactions[1–4,57].

Second, these mRNPs acquire a high density of N-UBMs and C-UBMs on their surface, which recruit the tetrameric THO complex through four UAP56 molecules, assembling TREX on the mRNP surface[2]. TREX thereby aids further mRNP compaction and chaperones the mRNA, preventing the formation of harmful R-loops.

Third, THO dissociates from these multivalent TREX–mRNPs when UAP56 clamps onto mRNA together with ATP. SARNP may bind together with ALYREF in the resulting UAP56–ADP-$P_i$–mRNP complexes to stabilize RNA-clamped UAP56 and prevent UAP56 from reassociating with THO, thereby increasing the efficiency of TREX disassembly.

Fourth, these remodelled UAP56–mRNPs would diffuse in the nucleus[36,37] before docking at the NPC-anchored TREX-2 complex through UAP56. Once docked, UAP56–mRNPs could bind to the mRNA export factor NXF1–NXT1 that is enriched at the NPC by several FG repeat-containing proteins[58,59], including the TREX-2 subunit GANP[14,60].

Fifth, TREX-2 unclamps UAP56 from mRNA, releasing these mRNPs for their export through the NPC through the mRNA export factor NXF1–NXT1. Consistent with this model, overexpression of the isolated GANP Sac3 domain in yeast leads to an mRNA export defect[14], presumably because nucleoplasmic TREX-2 prematurely releases UAP56 from mRNPs. In cells, these five steps might occur during or after transcription.

This general mRNA export model relies on UAP56 as the central molecule, which would functionally and sequentially connect TREX and TREX-2 complexes. This predicts that the interactions of UAP56 with THO or TREX-2 differentially regulate UAP56 binding to mRNPs. To test this, we designed three UAP56 mutants (M1–M3): UAP56 mutant M1 (D49R/L51D) impairs binding to TREX-2; mutant M2 (F336E/R339D) impairs binding to THO; and mutant M3 (M1 + M2, D49R/L51D/F336E/ R339D) impairs binding to both THO and TREX-2. As the THO- and TREX-2-binding surfaces of UAP56 partially overlap, we confirmed the expected binding specificities of each mutant in vitro (Extended Data Fig. 10j,k). We then expressed WT or mutant UAP56 proteins in human K562 cells and analysed their protein interactomes by quantitative MS (Fig. 5b, Extended Data Figs. 10l and 12a and Supplementary Table 4). For UAP56 mutant M1, which is defective in TREX-2 binding, SARNP and mRNP proteins were enriched. By contrast, for UAP56 mutants M2 and M3, which are defective in THO- or THO- and TREX-2-binding, SARNP and mRNP proteins were depleted. These results support that (1) the THO complex promotes the binding of UAP56 to mRNPs; (2) SARNP associates with UAP56–mRNPs downstream of THO but upstream of TREX-2; and (3) TREX-2 removes UAP56 from mRNPs. While we do not exclude that UAP56 molecules or mRNAs could bypass individual steps in the proposed model (Fig. 5a), our data support the sequential actions of the THO and TREX-2 complexes on UAP56.

## Discussion

Here we describe a general model for mRNA nuclear export involving a conserved set of factors, which depends on a series of regulated protein–protein and protein–mRNA interactions. Notably, the in silico UAP56 protein interaction screen identified additional UBM-containing and UCM-containing proteins, including a protein of viral origin (Extended Data Fig. 6i,j and Supplementary Table 3). Thus, while SARNP appears less important for mRNA nuclear export than other pathway factors (UAP56, THO, ALYREF, TREX-2)[33], its function may be partially redundant with other UCM- or non-UCM-containing factors or might in some cases be bypassed entirely. Taken together, we speculate that the mRNA export pathway provides additional levels of regulation for mRNA biogenesis and quality control that remain to be identified.

Our mechanistic insights into UAP56 as an RNA clampase show parallels to its close DExD-box ATPase homologue, EIF4A3—a member of the splicing-dependent EJC. Both ATPases bind to a cognate MIF4G-containing protein for their loading onto mRNA (here and previously[19,61]), both can form stable ATPase–ADP-P$_i$–RNA complexes (here and previously[26]) and both are mRNA-bound for prolonged periods (here and previously[62,63]). UAP56 would clamp onto mRNA for minutes, owing to the high rates of mRNA nuclear export[37,64–66], aided by ALYREF, SARNP or other proteins[18,67]. EIF4A3 would clamp onto mRNA, in some cases for days[62], until the first round of translation, helped by other proteins and the two EJC subunits, MAGOH and Y14[16,68]. Other DExD-box ATPase–MIF4G systems may be regulated by related mechanisms to control other RNA processes.

The UAP56–TREX-2 interaction also provides insights into 'gene gating'. By generating chromatin-tethered UAP56–mRNPs, transcribed genes could enrich at the NPC-tethered TREX-2 complex[10,13], enhancing gene expression efficiency[42,43,48,69,70]. Supporting this model, mutations in yeast GANP that affect gene gating[51] map onto the UAP56–TREX-2 interface in our (Extended Data Figs. 9b and 10f) and other UAP56–TREX-2 complex structures, reported while this Article was under review[71,72].

In conclusion, we reveal a mechanistic framework for the selective and efficient nuclear export of mRNA and the molecular functions of conserved proteins and complexes that control individual steps. At the core of this pathway lies the protein UAP56, which orchestrates the nuclear export of mRNA as an ATP-gated molecular switch.

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

# Methods

## Vectors and sequences
All vectors and sequences are described in Supplementary Table 5.

## Protein purification

**THO complex, EJC and ALYREF.** Recombinant THO complex tetramer (THOC1, THOC2 residues 1–1203, THOC3, THOC5, THOC6 and THOC7), dimer (same as tetramer but lacking THOC6) and monomer (THOC1, THOC2 residues 1–1203, THOC3, THOC5 residues 1–224, THOC7 residues 1–159) as well as the EJC subunits eIF4A3, MAGOH–Y14 and ALYREF[N] (residues 1–182), ALYREF[C] (residues 106–257) and full-length ALYREF were purified as described previously[2,20].

**UAP56 and UAP56 fusion proteins.** 6×His-TwinSTREPII-3C-UAP56, 6×His-MBP-3C-UAP56, 6×His-3C-UAP56ΔNTD (residues 44–428) and 10×His-3C-UAP56 WT and mutant constructs were purified as described previously for UAP56[20]. The fusion proteins 10×His-UAP56–UCM-1, 10×His-UAP56–N-UBM and 10×His-UAP56–UCM-1–N-UBM were expressed in *Escherichia coli* BL21 DE3 RIL cells grown in LB medium, induced at an optical density at 600 nm ($OD_{600}$) of 1.0 with 0.5 mM IPTG and incubated at 37 °C for 3 h. Cells were resuspended in lysis buffer (25 mM HEPES pH 7.9, 5% (v/v) glycerol, 300 mM NaCl, 20 mM imidazole, 0.05% Tween-20 and cOmplete EDTA-free protease inhibitor cocktail) and lysed by sonication. The lysate was clarified by centrifugation and the supernatant was filtered through 1 µm and 0.45 µm filters and applied to a HisTrap HP 5 ml column (Cytiva) pre-equilibrated with buffer A (25 mM HEPES pH 7.9, 5% (v/v) glycerol, 300 mM NaCl, 20 mM imidazole). The column was washed with buffer A containing 44 mM imidazole and proteins were eluted with a linear gradient from 50 mM to 300 mM imidazole. The peak fractions were diluted with buffer C (25 mM HEPES pH 7.9, 5% (v/v) glycerol, 1 mM DTT) to 100 mM NaCl and further purified by anion-exchange chromatography using a HiTrapQ 5 ml column (Cytiva), pre-equilibrated in buffer C. The column was washed with buffer C containing 100 mM NaCl and eluted using a linear gradient from 200 mM to 400 mM of NaCl. Peak fractions were concentrated and loaded on a HiLoad 16/600 Superdex 200 pg column (Cytiva) equilibrated using buffer D (25 mM HEPES pH 7.9, 5% (v/v) glycerol, 250 mM NaCl, 1 mM DTT). The purified proteins were concentrated, flash-frozen and stored at −80 °C.

**SARNP UCM-1 and ALYREF N-UBM.** 10×His-SUMO-3V5-tagged ALYREF N-UBM, SARNP UCM-1, UCM-1–N-UBM and UCM-1(R106D)–N-UBM were expressed in *E. coli* BL21 DE3 RIL cells. UCM-1, N-UBM and UCM-1 (R106D) were expressed in LB medium at 37 °C for 3 h after induction with 0.5 mM IPTG at an $OD_{600}$ of 1.0. UCM-1–N-UBM and UCM-1(R106D)–N-UBM were incubated at 18 °C overnight after induction. Cell pellets were resuspended in lysis buffer and lysed by sonication. Lysates were clarified by centrifugation, filtered through 1 µm and 0.45 µm filters and loaded onto a HisTrap HP 5 ml column equilibrated in buffer A. The column was washed with buffer A and proteins were eluted at 350 mM imidazole. The peak fractions were diluted to 50 mM NaCl with buffer C and loaded onto the HiTrapQ HP 5 ml column equilibrated in buffer C. The column was washed with buffer C supplemented with 50 mM NaCl and eluted using a linear gradient from 50 mM to 500 mM NaCl. The peak fractions were concentrated and applied to a HiLoad 16-600 Superdex 75 pg column (Cytiva) equilibrated using buffer E (10 mM HEPES pH 7.9, 500 mM NaCl, 10% (v/v) glycerol, 1 mM DTT, 20 mM imidazole). The peak fractions were concentrated again, flash-frozen in liquid nitrogen and stored at −80 °C. Buffer A and B for the purification of UCM-1–N-UBM contained 500 mM NaCl.

The UCM-1 R106D mutant was purified using a similar strategy with the following exceptions: two wash steps were performed during His-Trap using buffer A including a high-salt wash (25 mM HEPES pH 7.9, 5% (v/v) glycerol, 1 M NaCl) and buffer A supplemented with 50 mM

imidazole. During the anion-exchange step, the column was washed with 100 mM NaCl and eluted by a linear gradient from 100 mM to 400 mM NaCl.

**SARNP.** SARNP-6×His or SARNP[5xRtoD]-6×His were expressed in *E. coli* BL21 DE3 RIL cells grown in LB medium overnight at 18 °C after induction with 0.5 mM IPTG at $OD_{600}$ = 1.0. MBP-SARNP[47–210]-3C-3V5-SUMO-10×His was expressed in *E. coli* BL21 DE3 RIL cells grown in LB medium for 3 h at 37 °C after induction with 0.5 mM IPTG at $OD_{600}$ = 1.0. Cell pellets were resuspended in lysis buffer (50 mM HEPES pH 7.9, 500 mM NaCl, 10% (v/v) glycerol, 20 mM imidazole, 1 mM DTT, 0.5 mM PMSF, cOmplete EDTA-free protease inhibitor cocktail and 0.1% Tween-20), lysed by sonication and centrifuged. The supernatant was filtered through a 0.4-µm filter and loaded onto a HisTrap HP 5 ml column equilibrated using buffer E. The column was washed with 15 mM imidazole and SARNP eluted using a linear gradient from 15 to 350 mM imidazole. The peak fractions were diluted to 100 mM NaCl using buffer F (25 mM HEPES pH 7.9, 10% (v/v) glycerol and 2.5 mM DTT) and applied to a HiTrapQ HP 5 ml column equilibrated using buffer F (200 mM NaCl). The column was washed and bound protein was eluted over a linear gradient from 100 mM to 800 mM NaCl. The peak fractions were concentrated and applied to the HiLoad 16-600 Superdex 200 pg column, pre-equilibrated using buffer D containing 2.5 mM DTT and 250 mM salt. The purified protein was concentrated, flash-frozen and stored at −80 °C.

**TREX-2[M] and TREX-2[M] (R678A).** TREX-2[M] and TREX-2[M] (R678A) were expressed in *E. coli* BL21 DE3 RIL cells grown in LB medium at 37 °C until $OD_{600}$ at 1.0. Expression was induced by addition of 0.5 mM IPTG and cells were incubated at 18 °C overnight. Cells were collected by centrifugation and resuspended in lysis buffer (containing 500 mM NaCl and no Tween-20 for the TREX-2[M] purification). Cells were lysed by sonication and lysates were centrifuged. The supernatant was filtered through 1-µm and 0.45-µm filters and applied to a HisTrap HP 5 ml column equilibrated with buffer A, washed with buffer A (50 mM NaCl) and eluted over a linear gradient to 350 mM imidazole. The complex was diluted in buffer C to 100 mM NaCl and loaded on a HiTrapQ HP 5 ml column equilibrated with buffer C containing 100 mM NaCl. After a wash step with the buffer C containing 100 mM NaCl, the complex was eluted from the HiTrapQ column using a linear gradient to 800 mM NaCl (500 mM NaCl for TREX-2[M] (R678A)). The peak fractions were concentrated and applied to a HiLoad 16-600 Superdex 200 pg column equilibrated with buffer D. The purified complex was concentrated, flash-frozen and stored at −80 °C.

**MBP–GANP and PCID2–UAP56–UCM-1–N-UBM.** MBP–GANP (residues 582–1004) and 10×His–PCID2–UAP56–UCM-1–N-UBM – SEM1 were expressed in *E. coli* BL21 DE RIL cells. MBP–GANP was expressed in LB medium at 18 °C overnight after induction with 0.5 mM IPTG at $OD_{600}$ at 1.0 and 10×His-PCID2–UAP56–UCM-1–N-UBM – SEM1 was expressed in autoinduction medium at 30 °C. Bacterial cell pellets for MBP–GANP were lysed in buffer A containing 500 mM NaCl by sonication and centrifuged. The supernatant was loaded on a HisTrap HP 5 ml column equilibrated using buffer A containing 500 mM NaCl and 50 mM imidazole. The column was washed with this buffer A and eluted over a linear gradient to 350 mM imidazole using buffer B contained 500 mM NaCl. Peak fractions were diluted to 100 mM NaCl using buffer C, applied to a HiTrapQ HP 5 ml column and washed with buffer C containing 100 mM NaCl. The proteins were eluted using a linear gradient to 800 mM NaCl. The flow-through of the anion-exchange step was concentrated and loaded on a HiLoad 16-600 Superdex 200 pg column equilibrated with buffer E. Peak fractions were concentrated, flash-frozen and stored at −80 °C.

10×His-PCID2–UAP56–UCM-1–N-UBM – SEM1 was purified using a similar strategy with the following changes: buffers A and B contained 300 mM NaCl and the co-expressed complex was eluted from HisTrap

using a linear gradient from 50 to 300 mM imidazole. Moreover, the column was washed with buffer C containing 160 mM NaCl during the anion-exchange and eluted over a linear gradient from 160 to 400 mM NaCl.

**MBP–MCP.** MBP–MCP was expressed in *E. coli* Rosetta2 pLysS cells, grown in LB medium at 37 °C until $OD_{600}$ at 0.7, induced by addition of 0.5 mM IPTG and incubated at 37 °C for 3 h. Cells were resuspended in lysis buffer (20 mM HEPES pH 7.9, 200 mM KCl, 1 mM EDTA, 0.5 mM PMSF) and lysed by sonication. The lysate was centrifuged, filtered through a 0.45-µm filter and loaded on a MBP Trap HP column (Cytiva) equilibrated with buffer G (20 mM HEPES pH 7.9, 200 mM KCl, 1 mM EDTA). The column was washed first with buffer G and then with buffer H (20 mM HEPES pH 7.9, 20 mM KCl, 1 mM EDTA) and the protein was eluted using buffer H containing 10 mM maltose. The protein was further purified using a HiTrap Heparin HP 5 ml column and washed with buffer H (no EDTA). The protein was eluted over a linear gradient to 400 mM KCl. Peak fractions were flash-frozen in storage buffer (10 mM HEPES pH 7.9, 57 mM KCl, 1 mM EDTA, 10% (v/v) glycerol) and stored at −80 °C.

## Pull-down experiments using recombinant proteins

**In vitro THO–UAP56 disassembly assay.** Recombinant MBP–UAP56 (6.75 µg per reaction) was combined with a twofold molar excess of monomeric THO complex (10 µg per reaction) in buffer I (20 mM HEPES pH 7.9, 50 mM KCl, 1 mM $MgCl_2$, 5% glycerol, 0.1% Igepal CA-630) and incubated with 30 µl of amylose resin (E8021S, NEB), pre-equilibrated in buffer I, for 30 min at room temperature. The resin was then separated from the supernatant by centrifugation, washed three times with buffer I, resuspended in 40 µl of buffer I per reaction and split into individual tubes for each THO–UAP56 disassembly reaction. Components for the release reaction were prepared in a final volume of 40 µl buffer I (200 µM 15 U RNA, 0.1 mM ATP, and 55, 55 and 60 µg of Sumo-V5-N-UBM, Sumo-V5-UCM-1 and Sumo-V5-UCM-1-N-UBM, respectively), combined with the amylose resin with immobilized UAP56–THO complex and incubated for 60 min at room temperature. After four washes with buffer I, the bead-retained complexes were then eluted in buffer I supplemented with 100 mM maltose for 20 min at room temperature. Elutions and input samples of the individual recombinant proteins were separated on 4–12% gradient SDS–PAGE gels and visualized by Coomassie staining. The amount of bead-retained THO complex in each reaction was analysed in Fiji[76]. The intensity of the THOC2 band was measured, normalized to the intensity of the MBP–UAP56 band, and the background subtracted; THOC2 in the reaction incubated with buffer I without supplements was set to 100%.

**RNA-clamping assay.** In step 1, for each reaction 1 µg of in vitro transcribed 450 nucleotides AdML RNA and 5.1 µg of MBP–MS2 (equimolar with the RNA) in buffer J (20 mM HEPES pH 7.9, 100 mM KCl, 2 mM $MgCl_2$, 5% glycerol, 0.1% Igepal CA-630) were incubated with 20 µl of amylose resin (E8021S, NEB), pre-equilibrated in buffer J for 30 min at room temperature. The resin was then collected by centrifugation, the supernatant containing unbound components was removed, and three washes with buffer J were conducted before the resin with immobilized MBP–MS2–RNA was resuspended in 40 µl buffer J supplemented with 1 mM AMP-PNP and split into the desired number of reactions. Components to be tested for RNA binding in step 2 (23 µg UAP56 or UAP56-N-UBM, UAP56-UCM, UAP56-UCM-1-N-UBM (twofold molar excess over the RNA), 24 µg SARNP (twofold molar excess over UAP56)) were prepared in buffer J containing 1 mM AMP-PNP, combined with the resin prepared in step 1 and incubated for 90 min at room temperature. The resin was then again collected by centrifugation, washed three times with buffer J and incubated with 40 µl buffer J containing 0.4 µg benzonase to elute RNA-bound proteins. Elutions and input samples of the individual recombinant proteins were separated on 4–12% gradient

SDS–PAGE gels and visualized by Coomassie staining. To assess the amount of RNA-bound UAP56 in Fiji[76] we measured the intensity of the UAP56 band, subtracted the background and normalized to UAP56 in the presence of SARNP set to 100%.

**SARNP UCM-1 and ALYREF N-UBM–UAP56 pull-down.** To assess the interaction of UAP56 and the SARNP UCM-1 or the ALYREF N-UBM 7.5 µg of Sumo-V5-3C-tagged UCM-1, N-UBM or UCM-1–N-UBM were combined with a fourfold molar excess of UAP56 in buffer K (25 mM HEPES pH 7.9, 40 mM KCl, 5% glycerol, 0.01% Igepal CA630, 1 mM $MgCl_2$, 1 mM TCEP) in the presence of 50 µM 15 U RNA and 1 mM AMP-PNP, and incubated for 1 h at 4 °C before being added to 10 µl magnetic V5 beads (v5tma, Chromotek), pre-equilibrated in buffer K. After incubation for another hour rotating at 4 °C, the beads were centrifuged briefly to recover beads from the lid (1,300*g*, 2 min, 4 °C) and washed three times with buffer K on a magnetic rack. The samples were eluted using 30 µl 200 mM glycine (pH 2.52) for 5 min at room temperature. Eluates were neutralized using 2.5 µl 1 M Tris pH 10.4, and, together with input samples of the individual recombinant proteins, separated on 4–12% gradient SDS–PAGE gels and visualized by Coomassie staining.

**UAP56–TREX-2$^M$ pull-down.** To analyse the interaction of TREX-2$^M$ and UAP56, TREX-2$^M$ with an MBP-tag on the GANP subunit, was combined with a fourfold molar excess of UAP56 and a tenfold molar excess of UCM–UBM fusion peptide in buffer K, with or without 50 µM 15 U RNA and 1 mM AMP-PNP, and incubated rotating for 1 h at 4 °C. The samples were added to 30 µl pre-equilibrated amylose resin (E8021S, NEB), and incubated for another hour with rotating at 4 °C. Beads were centrifuged (1,300*g*, 2 min, 4 °C) to remove the unbound fraction, washed three times with buffer K, and bead-bound complexes were eluted for 1 h at room temperature in 30 µl buffer K supplemented with 100 mM maltose. Elutions and input samples of the individual recombinant proteins were separated on 4–12% gradient SDS–PAGE gels and visualized by Coomassie staining.

**UAP56 NTD –TREX-2$^M$ pull-down.** To test the interaction of the isolated UAP56 NTD and TREX-2$^M$, 30 µl of Pierce High Capacity NeutrAvidin Agarose beads (29202, Thermo Fisher Scientific) were pre-equilibrated with buffer K and incubated with or without 30 µg of biotinylated UAP56 NTD peptide (residues 1–24, WT or mutant, with biotin on the C terminus) in buffer K for 1 h at room temperature. The beads were then washed three times to remove unbound peptide and incubated with protein samples (set up in a 50 µl reaction containing 50 µM 15U RNA and 1 mM AMP-PNP and, as applicable: 7.5 µg TREX-2$^M$ with or without a fourfold molar excess of UAP56; 7.5 µg GANP(582–1004) with a 2.5-fold molar excess of the PCID2–UAP56–UCM–N-UBM – SEM1). After an incubation of 1 h rotating at 4 °C, the beads were again collected by centrifugation, washed three times with buffer K and bead-bound material eluted for 5 min at room temperature in 30 µl of 200 mM glycine pH 2.52. The elutions were neutralized with 100 mM Tris pH 10.4, separated alongside input samples of isolated recombinant proteins on 4–12% gradient SDS–PAGE gels and visualized by Coomassie staining.

**RNA-unclamping assay.** Biotinylated 15U RNA (33 µM), recombinant UAP56 (10 µM) and 1 mM ATP were incubated in buffer A2 (20 mM HEPES pH 7.9, 40 mM KCl, 2 mM $MgCl_2$, 5% glycerol, 0.1% Igepal CA630) with 20 µl Pierce High Capacity NeutrAvidin Agarose beads (29202, Thermo Fisher Scientific), pre-equilibrated in buffer A2, for 30 min at room temperature. After three washes with buffer A2 to remove unclamped UAP56 and excess ATP, the beads were resuspended in buffer A2 and split into the desired number of reactions. Next, 2.2 µM/0.44 µM WT or GANP R678A TREX-2$^M$ or 5 µM THO complex monomer were added in buffer A2 and the reactions incubated for 10 min at room temperature. Unbound proteins were then removed through washes twice in buffer (20 mM HEPES pH 7.9, 500 mM KCl, 2 mM $MgCl_2$, 5% glycerol, 0.1% Igepal

CA630) and twice in buffer A2 before elution of RNA bound proteins (0.4 μg benzonase in buffer A2) for 10 min at room temperature. Elutions were then analysed by Coomassie-stained SDS–PAGE and the amount of remaining RNA clamped UAP56 quantified in Fiji.

**UCM/UBM–UAP56 and –UAP56–TREX-2 pull-down.** Biotinylated peptides were immobilized in buffer A2 on 20 μl Pierce High Capacity NeutrAvidin Agarose beads (29202, Thermo Fisher Scientific), pre-equilibrated in buffer A2. The beads were washed three times to remove excess peptide and resuspended in buffer A2 before adding 20 μM UAP56 or 3.2 μM UAP56 with 6.4 μM TREX-2[M] and incubating at room temperature for 30 min (for UAP56 alone) or at 4 °C for 1 h (for UAP56–TREX-2[M]). Unbound UAP56 was then removed, the beads washed three times and bead-bound UAP56 was eluted in low-pH buffer for 10 min at room temperature. The elutions were neutralized and analysed by Coomassie-stained SDS–PAGE.

**UAP56–SARNP pull-down.** Magnetic anti-Flag M2 Beads (Merck, M8823; 20 μl per reaction) were equilibrated in buffer A2. Flag-tagged UAP56 (5 μM) and WT or mutant SARNP (20 μM) were added to the beads and incubated for 1 h at room temperature. Subsequently, unbound protein was removed, beads were washed three times and bead-bound complexes were eluted in low-pH buffer for 10 min at room temperature. The elutions were neutralized and analysed by SDS–PAGE and Coomassie staining.

**ALYREF–UAP56–SARNP pull-downs.** MBP tagged full-length or truncated ALYREF (2.5 μM) was immobilized in buffer A2 on 20 μl buffer-equilibrated amylose resin (NEB, E8021) with, as applicable, 12.5 μM UAP56, UCM-1, N-UBM or C-UBM or 6 μM full-length SARNP, and with or without 1 mM ATP and 200 μM 15 U RNA for 1 h at room temperature. Unbound protein was removed, beads were washed twice and bound complexes were eluted by incubating the beads for 5 min in SDS sample buffer at 95 °C before analysing the elutions using Coomassie-stained SDS–PAGE.

## GCI analysis

For GCI[77] experiments, the analyte is immobilized on a microfluidic chip and a putative ligand is flown in at increasing concentrations (association) and subsequently washed out with buffer (dissociation) (Fig. 1f). Binding is recorded as a change in the refractive index, yielding sensograms, which are fitted with a 1-to-1 binding kinetic model. GCI experiments were performed on a Creoptix WAVE system (Creoptix) using 4PCP WAVEchips (quasi-planar polycarboxylate surface; Creoptix). Chips were conditioned with borate buffer (100 mM sodium borate pH 9.0, 1 M NaCl), and either streptavidin (10 μg ml⁻¹ in 10 mM sodium acetate pH 5.0) or a monoclonal anti-V5 antibody (R960252, Invitrogen; 2 μg ml⁻¹ in 10 mM sodium acetate pH 5.0) immobilized using a standard amine coupling protocol, followed by passivation of the surface with BSA (0.5% in 10 mM sodium acetate pH 5.0) and final quenching with 1 M ethanolamine pH 8.0. Biotinylated 15 U RNA, UCM or UBM peptides or V5-tagged THO or TREX-2[M] complexes were captured on the prepared chip until the desired density was reached. UAP56 was injected in a 1:2 dilution series, starting from a highest concentration of 5 μM with or without 200 μM 15 U RNA, in 25 mM HEPES pH 7.9, 50 mM KCl, 1 mM MgCl₂, 1 mM TCEP, with and without 1 mM ATP at 25 °C. Blank injections were used for double referencing and a DMSO calibration curve was used for bulk correction. Analysis and correction were performed using the Creoptix WAVEcontrol software (applied corrections: x and y offset, DMSO calibration, double referencing) using a one-to-one binding model. The data and fitted models were plotted in R.

## UAP56–ATP cross-linking

Recombinant UAP56 with an N-terminal 10×His-2×Strep-3C tag (8 μM) was incubated in a total reaction volume of 15 μl in buffer A3 (25 mM

HEPES pH 7.9, 50 mM KCl, 2 mM MgCl₂) including 0.025 μM radioactive [α³²P]ATP or [γ³²P]ATP (around 3,000 Ci mmol⁻¹), 5 μl of magnetic nickel particles (Promega, V8560) and with or without 120 μM 15 U RNA for 30 min at room temperature. Unbound UAP56 and excess ATP and RNA were removed and the beads washed three times before being resuspended in 15 μl of buffer A3 and crosslinked for 2 min at a distance of 7 cm in a Stratagene Stratalinker UV1800 at λ = 254 nm. After the cross-linking reaction 5 μl of 5× SDS loading dye was added and the beads boiled for 2 min at 92 °C. The samples were then analysed on homemade 10% SDS–PAGE gels, stained with Coomassie-stain and the radioactive signal visualized using storage phosphor-screens and an Amersham Typhoon laser scanner.

In this experiment the ATP base is cross-linked to UAP56. Using [α³²P]ATP radioactive signal can be observed for UAP56-cross-linked nucleotide independently of whether or not ATP is hydrolysed in the UAP56–RNA complex, because the radioactive α³²P is present in both ADP and ATP. By contrast, when using [γ³²P]ATP, radioactive signal would only be observed for the UAP56–RNA complex if intact ATP had been crosslinked. If, as observed here, ATP is hydrolysed in the UAP56–RNA complex, the radioactive γ³²P is lost after the denaturation of the complex and no radioactive signal is observed.

## Size-exclusion chromatography

UAP56 (10His-Twinstrep-3C tagged, 20 μM) was incubated in a 500 μl reaction in buffer X3 (25 mM HEPES pH 7.9, 100 mM KCl, 1 mM MgCl₂, 1 mM TCEP) with 1 mM ATP or ADP and with or without 120 μM 15 U RNA for 1 h at room temperature. The samples were then analysed on a Superdex 200 increase 10/300 GL size-exclusion chromatography column, equilibrated in buffer X3, with monitoring of the UV absorption at 260 nm and 280 nm.

## IP experiments

**GFP–UAP56 IP for MS analysis.** GFP IPs were performed in triplicates from nuclei of GFP–UAP56 or WT K562 cells. For each replicate, 200 million cells were fractionated into nuclei and cytoplasm as previously described[2,78], nuclei were lysed in buffer L (50 mM Tris pH 7.5, 100 mM KCl, 3 mM MgCl₂, 0.25% Triton X-100, 0.25% Igepal CA630, 10% glycerol, 1× protease inhibitor cocktail, 1 mM DTT) supplemented with 1 μg ml⁻¹ benzonase and 0.1% deoxycholate and the lysates were incubated for 15 min at 4 °C on a rotating wheel followed by a centrifugation step to pellet chromatin (21,000g, 10 min, 4 °C). The supernatant was then incubated with 20 μl magnetic GFP-Trap MA-Agarose beads (Chromotek), pre-equilibrated in buffer L and incubated on a rotating wheel for 4 h at 4 °C. The beads were then collected on a magnetic rack, washed four times in 1 ml buffer L, and four times with 20 mM Tris pH 7.5, 100 mM KCl. After the final wash step, all buffer was removed and the beads were snap-frozen in liquid nitrogen. The samples were analysed by MS starting from an on-bead digest of bound protein complexes.

**Flag–UAP56 IP for western blot and quantitative MS.** V5-flag-TurboID-tagged UAP56 constructs (WT and mutants M1, M2 and M3) as well as V5-flag-TurboID-eGFP-NLS as a control were cloned under the TRE-tight promoter into the PiggyBac system ePB vector backbone, featuring in addition the expression of rtTA-Advanced-P2A-mScarlet under the human UbC promoter. Plasmids were electroporated into WT human K562 cells together with a plasmid encoding a PiggyBac transposase. mScarlet expression was used to identify a transgene harbouring cell population by FACS, and transgene expression was induced by the addition of 0.2 μg ml⁻¹ doxycycline 2 days before collecting the cells.

Flag IPs were performed in triplicates from 60 million nuclei per replicate as described above for GFP–UAP56 IP, but with 20 μl magnetic anti-flag M2 magnetic beads (Millipore, M8823) and 10% of the beads were analysed by western blot using anti-THOC2 (ab129485, Abcam, 1:1,000), anti-UAP56 (ab181059, Abcam, 1:1,000), anti-histone

H3 (ab1791, Abcam, 1:1,000) and goat-anti-rabbit antibody coupled to HRP (Thermo Fisher Scientific, 31466, 1:5,000) antibodies.

**UAP56 IP western blotting.** UAP56 IP experiments were performed as outlined above with the following changes: UAP56 was precipitated with anti-UAP56 antibody (Cell Signaling Technology, 47258) coupled to magnetic protein G beads (Thermo Fisher Scientific, 88802, the control reaction was performed with protein G beads without antibody) from 1.5 million K562 cell nuclei. To analyse the UAP56 interactome after SARNP depletion (Extended Data Fig. 5g,h) we used a SARNP-FKBP12[F36V] cell line. dTAG-V1 was added 6 h before collecting the cells to deplete SARNP. Elutions were analysed by standard western blot procedures and probed with anti-GANP (ab113295, Abcam, 1:1,000), anti-THOC2 (ab129485, Abcam, 1:1,000), anti-UAP56 (ab181059, Abcam, 1:1,000), anti-SARNP (PA5-56585, Invitrogen, 1:1,000), anti-ALYREF (ab202894, Abcam, 1:1,000) and goat-anti-rabbit antibody coupled to HRP (Thermo Fisher Scientific, 31466, 1:5,000).

**UAP56 NTD IP and analysis.** UAP56 NTD peptide (residues 1–24, biotin on the C terminus, WT or scrambled control, 75 µg per experiment) was immobilized on 30 µl of Pierce Strepdavidin magnetic beads (88817, Thermo Fisher Scientific, pre-equilibrated in PBS + 0.1% Igepal CA630) for 10 min at room temperature. Beads were then washed three times in buffer K and added to K562 nuclear lysate (see below). For the K562 nuclear lysate, 70 million K562 cells were fractionated in nuclei and cytoplasm (see above). Nuclei were resuspended in 700 µl buffer L supplemented with 0.1% deoxycholate and incubated on a rotating wheel for 1 h at 4 °C. The lysates were then briefly sonicated and centrifuged for 5 min at 3,000g and 4 °C. The supernatant was united with the peptide-bound beads and incubated with rotation for 2 h at 4 °C, after which the beads were essentially washed and analysed as described above (GFP–UAP56 IP), with the difference that 10% of the beads were used for western blotting and probed with an anti-GANP (ab113295, Abcam, 1:1,000) antibody and goat-anti-rabbit antibody coupled to HRP (Thermo Fisher Scientific, 31466, 1:5,000). Any specific interactor of the UAP56 NTD is expected to also interact with full-length UAP56, based on which we intersected the MS results of the UAP56 NTD peptide IP with all proteins enriched above a $\log_2[\text{FC}]$ cut-off of 0.5 in the GFP–UAP56 IP before further analysis.

**V5-PCID2 IP western blotting.** V5-PCID2 IP experiments were performed as outlined above with the following changes: V5-PCID2 was precipitated from 1.5 million K562 cells using magnetic V5-Trap beads (v5tma, Chromotek). We used cells containing a FKBP12[F36V] tag[79,80] on endogenous PCID2, allowing for the rapid depletion of the endogenous protein after addition of the dTAG-V1 compound 6 h before collecting the cells, and which expressed dox-inducible mScarlet-V5-PCID2 WT or mutant proteins. Elutions were analysed using standard western blotting procedures and probed with anti-GANP (ab113295, Abcam, 1:1,000), anti-V5 (2F11F7, Invitrogen, 1:1,000), anti-UAP56 (ab181059, Abcam, 1:1,000), anti-histone H3 (17168-1-AP, Proteintech, 1:1,000) antibodies, and goat-anti-rabbit antibody coupled to HRP (Thermo Fisher Scientific, 31466, 1:5,000) and goat-anti-mouse antibody coupled to HRP (Thermo Fisher Scientific, G-21040, 1:5,000).

## Endogenous TREX-disassembly assay

Nuclear extracts from a THOC1-3C-GFP overexpressing K562 cell line were prepared as previously described[20]. In total, 3.6 ml of nuclear extracts was supplemented with protease inhibitor cocktail and incubated with GFP-Trap Agarose resin (Chromotek), pre-equilibrated with buffer M (20 mM HEPES pH 7.9, 100 mM KCl, 2 mM MgCl$_2$, 8% glycerol, 0.05% (v/v) Igepal CA-630, 0.5 mM TCEP) for 3 h at 4 °C. The beads with immobilized endogenous TREX-mRNPs were then washed five times with 1.5 ml buffer M, aliquoted in 12 individual reactions and collected by centrifugation for 1 min at 1,000g to remove the supernatant.

Meanwhile, 10×His-Sumo-3V5-3C-UBM, 10×His-Sumo-3V5-3C-UCM or 10×His-Sumo-3V5-3C-UCM-N-UBM were prepared at a final concentration of 19 µM in buffer G (25 mM HEPES pH 7.9, 200 mM NaCl, 10 mM MgCl$_2$, 10% glycerol, 5 mM ATP, 1 mM TCEP) and incubated at room temperature for 30 min. Next, the beads with immobilized endogenous TREX-mRNPs were incubated either with buffer G or supplemented with UCM and/or UBM peptide in buffer G, as described above, for 1 h at room temperature with rotation. After addition of a final concentration of 50 µg ml⁻¹ benzonase and a further 30 min incubation, the beads were centrifuged for 1 min at 1,000g and washed twice with buffer M. Complexes remaining on the beads were eluted by boiling in 2× SDS sample buffer, loaded onto an SDS–PAGE gel and run for 3 min at 180 V in 1× MOPS buffer. The gels were stained with Coomassie blue, and the bands containing bead-retained protein were excised for MS analysis.

An aliquot of the elutions was analysed by western blotting according to standard protocols. We used anti-GFP (CAS A11122, Thermo Fisher Scientific, 1:1,000), anti-UAP56 (AB181059, Abcam, 1:1,000) and anti-EIF4A3 (AB180519, Abcam, 1:1,000). Primary antibodies were incubated overnight at 4 °C. For detection, we used a secondary goat-anti-rabbit antibody coupled to HRP (CAS 31466, 1:5,000).

## Cryo-EM sample preparation, imaging and analysis

**Model building for the endogenous human TREX complex including the ALYREF N-UBM.** The structure of the human endogenous TREX complex (PDB: 7ZNK)[2] was analysed together with the THO monomer 2B map (Electron Microscopy Data Bank (EMDB): EMDB-14806)[2]. Manual inspection revealed additional density on the UAP56 RecA2 lobe, which we hypothesized to be the ALYREF N-UBM. The ALYREF N-UBM was modelled in Coot based on the superposition of an AlphaFold2 Multimer prediction model of a UAP56–ALYREF complex on UAP56 chain p. All structural figures were prepared using UCSF Chimera X[81,82].

**TREX–EJC–RNA complex reconstitution and sample preparation.** TREX–EJC–RNA complexes were reconstituted as described previously[2] with small modifications. We used a 15 U ssRNA to assemble the ALYREF[N]–EJC–RNA and ATPγS was omitted from buffer U. The eluted sample was loaded onto a 15–40% sucrose density gradient and centrifuged at 23,000 rpm for 16 h in a SW60Ti rotor. We collected fractions and analysed every other fraction using SDS–PAGE stained with Coomassie blue.

For Cryo-EM sample preparation, we followed the described methodology described previously[2] with the following variations: the 15–40% sucrose density gradient was supplemented with a glutaraldehyde gradient from 0 to 0.05% to stabilize the complexes and it was centrifuged at 23,000 rpm for 16 h in a SW60Ti rotor, and we applied the sample to glow discharged Quantifoil Cu 200 2/1 grids.

**Cryo-EM data acquisition of TREX–EJC–RNA complex reconstituted on 15 U RNA.** Data were collected at IST Austria on the Thermo Fisher Titan Krios G3i system operated at 300 keV, equipped with a Gatan K3 direct electron detector operated in counting mode and a BioQuantum post-column energy filter set to a slit width of 10 eV. The objective aperture was retracted and a 50 µm C2 aperture was inserted. Data were collected at pixel size of 0.84 Å px⁻¹, a total dose of 60 e⁻ fractionated over 40 frames and a defocus range of −0.75 to −1.25 µm using EPU. The dataset was collected at a dose rate of 33.914 e⁻ px⁻¹ s⁻¹. We acquired 5 images per hole and collected a total of 10,510 micrographs.

Data were preprocessed using Warp (v.1.09)[83]. CTF parameters were estimated with a spatial resolution of 6 × 4 and motion correction was performed with a spatial resolution of 6 × 4. We picked 470,103 particles in Warp using a custom BoxNet model and extracted them in RELION (v.3.1)[84] in a box size of 672 Å. For initial classification, particles were binned to 3.42 Å pixel⁻¹.

3D classification with six classes was performed on the extracted particles using a reference volume of a TREX–EJC–RNA on 15U

reconstruction from a dataset collected on a Glacios TEM microscope low-pass filtered to 60 Å and a spherical mask of 550 Å diameter. Class 5 was selected with 84,300 octamer particles. To increase dataset size, we separately extracted four THO–UAP56 dimers from each octamer, yielding a total of 329,826 dimers after removal of duplicates and re-extraction in CryoSPARC (box size, 436 ; 1.24 Å, pixel$^{-1}$). After another round of heterogeneous refinement with three classes the 204,147 particles of the best class were further refined through (1) a local refinement and non-uniform refinement using a TREX complex mask, yielding the 5.89 Å TREX complex Map A and (2) a local refinement using a mask including THO monomer 1A and UAP56, yielding the 4.12 Å THO–UAP56–ALYREF–N-UBM complex map B.

**Model building for the THO–UAP56–ALYREF–N-UBM complex.** The structure of the human THO–UAP56 complex (PDB: 7ZNL)[2] was docked into the THO–UAP56–ALYREF–N-UBM complex MapB. For UAP56, both RecA lobes were individually rigid-body fitted into the new map in Coot[85,86]. The ALYREF N-UBM was fitted into the density based on an AlphaFold2 Multimer prediction of a UAP56–ALYREF complex and manually adjusted in COOT and the resulting structure was refined in phenix[87,88] using the phenix.real_space_refine routine with secondary structure and rotamer restraints.

**UAP56–UCM-1–N-UBM–TREX-2$^M$ complex reconstitution and sample preparation.** A PCID2–UAP56–UCM-1–N-UBM fusion protein in complex with SEM1 was combined with a 1.2× molar excess of MBP–GANP(582–1004) in buffer N (25 mM HEPES pH 7.9, 5% glycerol, 1 mM MgCl$_2$, 1 mM TCEP, 200 μM 15 U RNA, 1 mM AMP-PNP) and incubated on ice for 1 h. The sample was then centrifuged (21,130$g$, 15 min, 4 °C) and applied to a Superdex 200 increase 10/300 size-exclusion column, pre-equilibrated in buffer N, to separate the PCID2-UAP56–UCM-1–N-UBM – MBP–GANP(582–1004) complex from isolated components. The peak fractions were analysed by SDS–PAGE and Coomassie staining to confirm stochiometric complex formation of the three proteins. The peak fraction was then diluted with buffer N to 0.8 mg ml$^{-1}$ and cryo-EM grids were prepared by applying 4 μl of the sample to glow-discharged Cu R1.2/1.3 300-mesh holey carbon grids (Quantifoil). The grids were prepared, blotted at 8 °C under 90% humidity and plunged into liquid ethane using a Leica EM GP2.

**Cryo-EM data acquisition of a UAP56–UCM-1–N-UBM–TREX-2$^M$ complex.** Two datasets were collected on a 300 kV Titan Krios G4 equipped with a cold field-emission gun, a post-column Selectris energy filter (Thermo Fisher Scientific) with a 5 eV slit width and a Falcon 4i direct electron detector (Thermo Fisher Scientific). The objective aperture was retracted and a 50 μm C2 aperture was inserted. For dataset 1, we collected 6,839 micrographs using EPU in the .eer format, with five images per hole, a pixel size of 0.749 Å px$^{-1}$, a total dose of 50 e⁻ Å$^{-2}$ and a defocus range of −1 to −2.5 μm. Dataset 2 consists of 9,374 micrographs collected at a tilt angle of 20° and otherwise identical settings.

We performed on-the-fly preprocessing (patch motion correction and CTF estimation) using the CryoSPARC[89] live routine. For dataset 1, we initially picked 3.8 million particles in CryoSPARC live, extracted them with a 225 Å box and binned to 1.755 Å px$^{-1}$ and performed 2D classification. We then generated ab initio models for TREX-2$^M$ and UAP56–TREX-2$^M$, which were further refined through heterogeneous refinements and non-uniform refinements. These models were used as the initial reference maps for three rounds of heterogeneous refinement of 1.67 million particles picked in WARP and extracted with a 225 Å box and binned to 1.755 Å px$^{-1}$, yielding 199,358 UAP56–TREX-2$^M$ particles in the best class. These were re-extracted with a 225 Å box and binned to 0.877 Å px$^{-1}$. Further heterogeneous refinement and 3D variability analysis yielded 19,188 UAP56–TREX-2$^M$ particles in the best two classes.

For dataset 2 we picked 2.4 million particles in WARP and extracted them with a 225 Å box and binned to 1.755 Å px$^{-1}$. After 2D classification, we obtained 660,903 TREX-2$^M$ and UAP56–TREX-2$^M$ particles. Two rounds of heterogeneous refinement yielded 316,490 TREX-2$^M$ and 120,526 UAP56–TREX-2$^M$ particles.

The 316,490 TREX-2$^M$ particles were re-extracted with a 225 Å box and binned to 0.877 Å pixel$^{-1}$, and subjected to a non-uniform refinement followed by 3D variability analysis. Then, 57,499 particles from the best two clusters were refined through a local CTF refinement and a final local refinement with TREX-2$^M$ mask yielded the 3.5 Å TREX-2$^M$ complex Map C.

The UAP56–TREX-2$^M$ particles were re-extracted with a 225 Å box and binned to 0.877 Å pixel$^{-1}$ and subjected to another round of heterogeneous refinement, a non-uniform refinement, and a 3D variability analysis, resulting in 18,304 UAP56–TREX-2$^M$ particles, which were combined with the UAP56–TREX-2$^M$ particles from dataset 1. The combined 37,692 particles were subjected to a local CTF refinement and a final local refinement, leading to the 3.5 Å UAP56–TREX-2$^M$ complex map D. A further 3D classification with 20 classes and a GANP–UAP56-RecA2 mask, followed by local refinement of a class with 7,741 particles with bound UAP56-RecA2 lobe yielded the 4.22 Å UAP56–TREX-2$^M$ complex map E.

**Model building for the TREX-2$^M$ complex and the UAP56-UCM-1-N-UBM–TREX-2$^M$ complex.** An Alphafold2 Multimer prediction of TREX-2$^M$ or UAP56–TREX-2$^M$ was used as an initial model and docked into map C and map D densities, respectively. Model building was then manually adjusted in COOT and refined in phenix[87,88] using the phenix.real_space_refine routine with secondary structure and rotamer restraints. The model of the UAP56 RecA2 lobe was obtained from an AlphaFold2 Multimer prediction and manually fitted into the UAP56 RecA2 density in map E.

### ATPase assay

Steady-state UAP56 ATPase activity was measured using a NADH-coupled ATPase assay[20,90], with final concentrations of 5 U ml$^{-1}$ rabbit muscle pyruvate kinase, type III (Sigma-Aldrich), 5 U ml$^{-1}$ rabbit muscle L-lactic dehydrogenase, type XI (Sigma-Aldrich), 500 μM phosphoenolpyruvate and 50 μM NADH. The reactions were prepared in a final volume of 10 μl in a 1,536-well plate and in buffer O (25 mM HEPES pH 7.9, 40 mM KCl, 0.5 mM MgCl$_2$, 5% (w/v) glycerol, 0.5 mM ATP) with 0.5/2 μM UAP56 (when measured with TREX-2$^M$ or in isolation), 2 μM TREX-2$^M$ (WT or R678A mutant), 200 μM 15 U RNA. The NADH emission signal decay was monitored over time at 37 °C in a PHERAstar FS (BMG LABTECH), with a 0.03–100 μM NADH dilution series as a calibration standard. UAP56 ATPase rates were determined by linear regression of the NADH decay, corrected for ATP decay, as hydrolysed molecules of ATP per s per enzyme. Input samples of the individual reactions were separated on 4–12% gradient SDS–PAGE gels and visualized by Coomassie staining.

### Human K562 cell line experiments
**Generation of an endogenously tagged GFP–3C–UAP56 cell line.** Human K562 cells (DSMZ) were edited to express an eGFP–3C–DDX39B fusion protein using a modification of a previously described CRISPR–Cas9 knockin protocol[91]. In brief, the gRNA was designed using the Benchling.com CRISPR gRNA design tool (Benchling; AAAC TAACTGGGCCGGCAGGGGAAC) and cloned into the plasmid pLCG (hU6-sgRNA-EFSSpCas9-P2A-mCherry)[92], a gift from J. Zuber. The 500 bp sequences flanking the DDX39B start codon were obtained by PCR on genomic K562 cDNA (using 5′ homology genomic primers: ATCCTCAA GTAAGGGGGTACCAGGACTCTACTTGTCATCTCCATTTTCC, GAGA TGTTGAAGGTCTTCATAACTGGGCCAGCAGGGGA; and 3′ homology genomic primers: AGGGCCCGGGTGGAGGTTCCGCTGGAGCAGAGAAC GATGTGGACAATG, ATCCCCCCTTTTCTTTTAAAGAATTCTGATCTAGC

CTTAAGTATAAACCC) and subcloned into the pLPG vector[92], a gift from J. Zuber, digested with MluI using Gibson Assembly (NEB), yielding the final vector pLPG-GFP-AID (5′-Blast$_R$-P2A-eGFP-AID-3C).

K562 cells were grown in RPMI medium supplemented with 10% FBS (Sigma-Aldrich), 2% L-glutamine (Gibco), 1% sodium pyruvate (Sigma-Aldrich) and 1% penicillin–streptomycin (Sigma-Aldrich) and transfected with the HDR donor and the Cas9 plasmids using Neon electroporation device (Invitrogen) according to user guide manual (for suspension cells). Then, 6 days after transfection, after several passages, cells were subjected to FACS using the BD FACSAria III (BD Biosciences) system. Cells expressing the eGFP-tag were sorted into 96-well plates. After approximately 2 weeks, wells with homogeneous fluorescence were genotyped (primers: TGCTAATTACACAAGGCTT, ACCTGCCACAGACCACTTCT), homozygous clones were further analysed by western blotting for homozygous knockin of the tag using anti-UAP56 (ab181059, Abcam, 1:1,000), anti-GFP (A11122, Invitrogen, 1:1,000), goat-anti-rabbit antibody coupled to HRP (Thermo Fisher Scientific, 31466, 1:5,000) and goat-anti-mouse antibody coupled to HRP (Thermo Scientific G-21040, 1:5,000).

**Generation of an endogenously tagged PCID2 cell line.** K562 cells with PCID2 endogenously tagged with an N-terminal eGFP–FKBP12$^{F36V}$–3C tag[79,80] were generated as outlined above for UAP56 with the following changes: the gRNA (TCCGTTCGGCGGCGCTCCCA) was designed using the CHOPCHOP web tool and gRNA ordered as a crRNA from IDT. Cas9–gRNA ribonuclein particles were generated according to the manufacturer's instructions (https://eu.idtdna.com). Repair template DNA molecules with 50-bp-long homology arms (HA) were generated by PCR using 5′-end biotin modified oligonucleotides (Sigma-Aldrich) (5′-HA primer (mutated to eliminate the PAM in the modified locus): TGACGCCAGCTGGCCCGCTTGAGGCGTAGGGGGTGGCGCTCTCCGTT GCGCGGCGCTCCCATGAAGACCTTCAACATCTCTCAGCAGGAC, and biotin-TGACGCCAGCTGGCCCGCTT; 3′-HA primer: GCGCGCTCCCC GGCTAGGACCCACCTGCTGCAGGTACTGGTTAATGGTAATGTGCGCC ATGGAACCTCCACCCGGGCCCTGAAA; and biotin-GCGCGCTCCCCG GCTAGGA). K562 cells were transfected with the repair templates and ribonuclein complexes using a MaxCyte ATx electroporator. Cells expressing the eGFP-tag were sorted into 96-well plates by FACS and, after approximately 2 weeks, cell clones with homogeneous fluorescence were genotyped (primers, GAGGGGACACACGGAACA and CCGAACACACAATCAGAGCC) and further analysed by western blotting for homozygous tagging and for degradation efficiency upon the addition of dTAG-V1.

**Generation of an endogenously tagged SARNP cell line.** SARNP was endogenously tagged with a C-terminal 3C–FKBP12$^{F36V}$–eGFP tag as outlined for PCID2 with the following reagents: gRNA: AG-TATCAGGAACTTTTCATC; homology arm PCR: 5′-HA primer CTTCT TTACAGGCAAAGAAGAGGAAAAGAGCAGAGCGCTTTGGGATTGCCCT GGAAGTTCTGTTTCAGGGCC and biotin-CTTCTTTACAGGCAAAG AAGAGGAAAAG, 3′-HA primer AGAAGGAGAGAAATGGAAAACAC TGGAGAACAGAAAGTATCAGGAACTTTTCAGCACGGGCTTGCG and biotin-AGAAGGAGAGAAATGGAAAACACTGG; genotyping primers: AACCCAGGCAACTATTGTCTTC and CAGCAATAAGTCAAACTGCTGC.

**Generation of an endogenously tagged GANP cell line.** GANP was endogenously tagged with an internal FKBP12$^{F36V}$–eGFP tag as outlined for PCID2 with the following reagents: gRNA: CGTGCCCATGTACTCT GACG; homology arm PCR: 5′-HA primer CTTCCAGCTGTCTGTGCAGC CTGAACCACCGCCTCCAGAGCCCGTGCCCGGAGGTGGATCGATGGGA GT and biotin- CTTCCAGCTGTCTGTGCAG, 3′-HA primer CTTCCCA GAGTCCAGACCTAGAAAAAAAGAGTCCCTACCTCGTCAGAGTAGGAA CCTCCACCCTTGTACAG and biotin-CTTCCCAGAGTCCAGACCTAGA; genotyping primers: TGCAGCTATGTTTT GTCCTGT and TGGGGTGAT GACTAAGGACG.

**Inducible UAP56/DDX39A CRISPR knockout cell line.** Dual sgR-NAs were designed against both UAP56 (GGACATCCATTCCCAGAA and GAACAGCTGGAGCCAGTTACT) and DDX39A (GCTGGCCTTC CAGATCAGCA and GCATGTCGTGGTGGGGACCCC) and cloned into modified Dual-sgRNA_hU6-mU6 vectors[93] (gift from J. Zuber) also expressing eBFP2 (for UAP56 sgRNAs) or iRFP670 (for DDX39A sgRNAs). Both Dual-sgRNA expression vectors were packaged in lentiviruses as previously described[94]. Lentiviruses were then used to infect K562 cells, which allow for the doxycycline-inducible expression of Cas9[93] (gift from J. Zuber), and a cell population containing both Dual-sgRNA constructs was selected by FACS sorting for eBFP- and iRFP670-positive cells.

**Expression of rescue constructs using the PiggyBac system.** Rescue constructs for UAP56 and PCID2 were generated by cloning of the CDSs under the TRE-tight promoter into the PiggyBac system ePB vector backbone, featuring in addition the expression of rtTA-Advanced-T2A-puromycin resistance under the human UbC promoter. UAP56 was fused at the C terminus to a P2A site and mScarlet to monitor transgene expression. PCID2 was expressed with an N-terminal mScarlet-3×V5-3C tag. Rescue constructs were electroporated into UAP56/DDX39A inducible CRISPR KO cells for UAP56 and into FKBP12$^{F36V}$-PCID2 cells for PCID2 together with a plasmid encoding a PiggyBac transposase. Transgene expression was induced by the addition of 0.2 µg ml$^{-1}$ doxycycline.

**Cell growth competition experiments.** The depletion of essential genes, such as *UAP56* or *PCID2*[95], leads to a severe growth phenotype, enabling cell growth competition experiments. Knockout of the *DDX39A* and *DDX39B* genes and the expression of *DDX39B* rescue constructs together with mScarlet were induced 6 days after electroporation of the rescue constructs with 0.2 µg ml$^{-1}$ doxycycline, followed by a cell sorting for the presence of all four fluorophores (GFP for inducible Cas9, BFP and iRFP for the expression of the dual gRNAs, mScarlet for the rescue construct) one day after doxycycline induction. The quadruple-positive cells (and the respective controls) were mixed with WT K562 competitor cells on the second day after doxycycline induction and the ratio of BFP-positive to BFP-negative cells was determined using the BD Fortessa cytometer on several days until the BFP positive cells perished in the control sample without rescue protein. Cell loss was normalized to the mean of the samples containing the untreated maternal line.

For PCID2, the expression of rescue proteins fused to mScarlet was induced 6 days after the electroporation of the rescue construct with 0.2 µg ml$^{-1}$, followed by a cell sorting for mScarlet (rescue construct) and eGFP (for endogenous eGFP–FKBP12$^{F36V}$–3C–PCID2) 1 day after doxycycline induction. The double-positive cells (and the respective controls) were mixed with a BFP-expressing competitor cell line followed by degradation of the endogenous eGFP–FKBP12$^{F36V}$–3C–PCID2 protein after dTAG-V1 treatment (0.25 µM dTAG-V1). The ratio of BFP-positive to BFP-negative cells was determined using the BD Fortessa cytometer on several days until the BFP-negative cells perished in the control sample without rescue protein. Cell loss was normalized to the mean of the samples containing the untreated maternal line.

**Generation of the K562 export reporter cell line.** The full reporter sequence, consisting of the mCherry coding sequence (CDS) with a single intron containing ten boxB sites, an IRES and the GFP-puromycin resistance ORF (mCherry$^{1/2}$-5′SS-10×boxB-IRES-GFP-Puro$^R$-3′SS-mCherry$^{2/2}$), was synthesized (Genewiz) and cloned into a lentiviral vector backbone[96] (pRRL SFFV d20GFP.T2A.mTagBFP Donor was a gift from A, Scharenberg; Addgene plasmid, 31485), yielding the plasmid containing pRRL-SFFV-reporter plasmid. Viral particles were generated by polyethylenimine transfection (Polysciences) of the pRRL-SFFV-reporter plasmid, together with the helper plasmids pCMVR8.74 (a gift from

D. Trono (Addgene plasmid, 22036) and pCMV-VSV-G[97] (a gift from B. Weinberg; Addgene plasmid, 8454) into LentiX-cells (Takara) according to standard procedures. K562 (DMSZ) cells were infected at limiting dilutions and mCherry-positive single cells were isolated using the FACSAria III cell sorter (BD Biosciences). Viral integration of the entire reporter sequence was assessed by genotyping PCR. LentiX and K562 cells were maintained at 37 °C under 5% $CO_2$ and tested negative for mycoplasma.

**Plasmid transfection into K562 export reporter cell line for λN-mediated tethering.** The CDS of a protein of interest was cloned into an acceptor plasmid containing the λN-BC2-Flag tag, a P2A site and the BFP CDS (plasmid nLV-Ef1a, a gift from S. Ameres) using Gibson assembly[98]. For each protein of interest that promoted export, a control plasmid lacking the λN-tag was created (Supplementary Table 5). Plasmids were transfected into the K562 reporter cell line using the Neon Transfection System 10 µl Kit (Invitrogen, MPK1025) according to the manual with 3 µg plasmid per $2 \times 10^6$ cells (pulse voltage (V) = 1,450, pulse width (ms) = 10 and pulse number = 3) in three replicates on different days. Then, 48 h after transfection, cells were analysed using an iQue Screener Plus (Sartoriuos). Flow cytometry data were filtered for good events using FlowAI[99], transfected K562 cells were selected by gating for BFP-positive cells, and their GFP intensity extracted and plotted using GraphPad Prism (v.8).

To control for the expression and nuclear localization of λN-UAP56 and λN-UAP56 ΔNTD aliquots of one million cells were fractionated as previously described[2,78] and analysed by western blotting using anti-UAP56 (ab181059, Abcam, 1:1,000), anti-histone H3-HRP (5192S, Cell Signalling Technologies, 1:1,000) and goat-anti-rabbit antibody coupled to HRP (Thermo Fisher Scientific, 31466, 1:5,000).

**Poly(A) RNA FISH.** For poly(A) RNA FISH experiments we used the cell lines generated for the PCID2-dTAG depletion-rescue experiment as well as a GANP-dTAG cell line (as described above). Specifically, we used the PCID2-FKBP-GFP clonal cell line and populations expressing the respective WT and mutant rescue constructs in the PCID2-FKBP-GFP cell line background. Expression of the rescue constructs was induced 7 days before the experiment, whereby depletion of endogenous PCID2-FKBP-GFP was induced by the addition of dTAG-V1 for 16 h.

Cells were added to 8-well slides (µ-Slide 8 Well high, 80806, Ibidi) precoated with 0.5 µg ml$^{-1}$ concanavalin A and allowed to adhere for 1 h before fixation in 4% PFA for 10 min at room temperature. The slides were then washed in PBS and incubated in 70% ethanol for 1 h at 4 °C. Subsequently, cells were washed with 5× SSC-T (5× SSC, 0.1% Tween-20) and then incubated first in hybridization buffer (30% formamide, 5× SSC, 1× Denhardt's solution, 50 µg ml$^{-1}$ heparin, 0.1% Tween-20, 10% dextran sulfate) for 30 min at 37 °C and then for with hybridization buffer supplemented with 100 nM oligo-dT FISH probe for 2 h at 37 °C. The slides were then washed three times in wash buffer (30% formamide, 5× SSC, 0.1% Tween-20), twice with 5× SSC-T and then incubated for 15 min at room temperature with 5× SSC-T containing 200 ng ml$^{-1}$ DAPI. After three additional washes slides were imaged on an Olympus IX83 based spinning disc confocal microscope with a ×40 air objective (for image quantification, examples are shown in Extended Data Fig. 11b) or a ×100 oil-immersion objective (to record representative images, examples are shown in Fig. 4e).

For each sample, we prepared four replicates and collected five images each, which were analysed using a Python pipeline using Stardist[100] and Cellpose[101] for image segmentation of the nucleus and cytoplasm. The PCID2 rescue constructs were expressed in populations consisting of PCID2-FKBP-GFP cells that either did or did not express the rescue construct. The expression of the rescue constructs can be distinguished due to the mScarlet fused to PCID2 in the rescue constructs (Extended Data Fig. 11b). This enables us to analyse the effect of the rescue construct compared with no rescue construct directly within the same image by grouping cells according to mScarlet levels for the analysis. Data were analysed using R v.4.0.

### AlphaFold2 Multimer screening

Protein interaction prediction screening was performed using a custom pipeline (HT-Colabfold) based on Colabfold, which uses AlphaFold2 Multimer[102–104]. This pipeline was used to predict interactions between UAP56 and 696 proteins that were designated putative UAP56 interactors based on their at least twofold enrichment over a WT control in UAP56–GFP immunoprecipitates. HT-Colabfold manages the pairing, scheduling and data collection for large-scale structure prediction and interaction screens. The pipeline executes pairwise predictions utilizing MMseqs (git@92deb92) for local multiple sequence alignment generation (CPU-node) and Colabfold (git@7227d4c) for structure prediction (GPU-nodes). Each prediction involved the generation of five models, omitting structure relaxation. Predictions with an average iPTM score of >0.5 were considered to be putative hits and diagnostic plots (PAE plot, pLDDT plot and sequence coverage) as well as the generated structures were manually inspected.

### Reproducibility

All experiments, except for cryo-EM data collection and processing, have been repeated at least three times with similar results.

### Reporting summary

Further information on research design is available in the Nature Portfolio Reporting Summary linked to this article.

## Data availability

3D cryo-EM density maps of the TREX–EJC–ALYREF complex, TREX-2M and UAP56–TREX-2M have been deposited into the Electron Microscopy Data Bank under accession numbers EMD-18980 (map A) and EMD-18979 (map B), EMD-18977 (map C), EMD-18978 (map D) and EMD-18981 (map E). The coordinate files of the TREX–EJC–ALYREF, TREX-2M and UAP56–TREX-2M have been deposited at the Protein Data Bank under the accession numbers 8R7L, 8R7J and 8R7K. The coordinate file of the TREX–mRNA complex was updated in the Protein Data Bank under the accession number 7ZNK. Proteomics data have been deposited to the ProteomeXchange Consortium via the PRIDE[105] partner repository under the accession number PXD069399.

## Code availability

HT-Colabfold is free open-source software (MIT) and is available at GitHub (https://gitlab.com/BrenneckeLab/ht-colabfold).

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

**Acknowledgements** We thank our colleagues in the Brennecke and Plaschka groups for their support and discussions; the staff at the Protein Technologies Facility at the Vienna BioCenter Core Facilities (VBCF), a member of the Vienna BioCenter (VBC), for assistance with protein production; the staff at the VBCF Electron Microscopy Facility, in particular T. Heuser and H. Kotisch, for support, data collection and maintaining facilities; V.-V. Hodirnau at the Institute of Science and Technology Austria EM facility for cryo-EM data collection; staff at the VBCF Proteomics facility, in particular O. Hudecz, for MS analysis; R. Zimmermann and his team as well as staff at the CLIP cluster (https://clip.science) for computational support; J. Ahel for local installation of AlphaFold2; M. Madalinski for peptide synthesis; staff at the in-house Molecular Biology Service for reagents; S. Ameres, J. Zuber and J. Santiago Cuellar for sharing reagents; S. Ameres and U. Elling for help with export reporter design; R. Kalis and J. Zuber for advice on CRISPR knockout generation; M. Vorländer for help with data analysis; C. Bernecky, L. Lorenzo-Orts and I. Patten for discussions; and A. Anderson, C. Bernecky, T. Heick-Jensen, L. Lorenzo-Orts, A. Stark and the members of the Brennecke and Plaschka groups for feedback on the manuscript. U.H. was supported by a Marie Skłodowska-Curie fellowship (896416) and an EMBO long-term fellowship (ALTF_1175-2019); D.R.-B. and R.F. by Marie Sklodowska-Curie fellowships (101028744 and 101150910); J.B. by the Austrian Academy of Sciences and Austrian Science Fund (W1207). Research in the laboratory of C.P. is supported by Boehringer Ingelheim, the European Research Council under the Horizon 2020 Research and Innovation Programme (ERC-2020-STG 949081 RNApaxport) and by the Austrian Science Fund (FWF) doc.funds program DOC177-B (RNA@core: Molecular mechanisms in RNA biology). The funders had no role in study design, data collection and analysis, decision to publish or preparation of the manuscript. For the purpose of open access, the author has applied for a CC BY public copyright license to any author accepted manuscript version arising from this submission.

**Author contributions** U.H. designed research and, with M.G., L.F. and T.P., purified recombinant proteins. U.H. and M.G. performed biochemical experiments and collected TREX-2 cryo-EM data. B.P.-F. performed the native TREX disassembly assay and collected TREX–EJC–ALYREF cryo-EM data. U.H. and C.P. carried out cryo-EM data analysis and structure determination. U.S. performed the export tethering assay, for which M.-F.S. engineered and M.-F.S. and C.P. designed the cell line. L.T., A.W.P. and D.R.-B. engineered and grew mammalian cell lines. L.T. carried out growth competition experiments. L.T., R.F. and U.H. performed poly(A) RNA FISH experiments. U.H. performed structural predictions with an AlphaFold2 pipeline implemented by D.H.; U.H., J.B. and C.P. prepared the manuscript with input from all of the authors. E.R. supervised MS analysis. J.B. supervised L.T. and D.H. and designed research. J.B. and C.P. supervised U.H.; C.P. supervised M.G., U.S., B.P.-F., L.F., D.R.-B., R.F., T.P. and M.-F.S., designed research and initiated the mRNA export project.

**Funding** Open access funding provided by Research Institute of Molecular Pathology (IMP)/Institute of Molecular Biotechnology (IMBA)/Gregor Mendel Institute of Molecular Plant Biology (GMI).

**Competing interests** The authors declare no competing interests.

**Additional information**
**Correspondence and requests for materials** should be addressed to Ulrich Hohmann, Julius Brennecke or Clemens Plaschka.

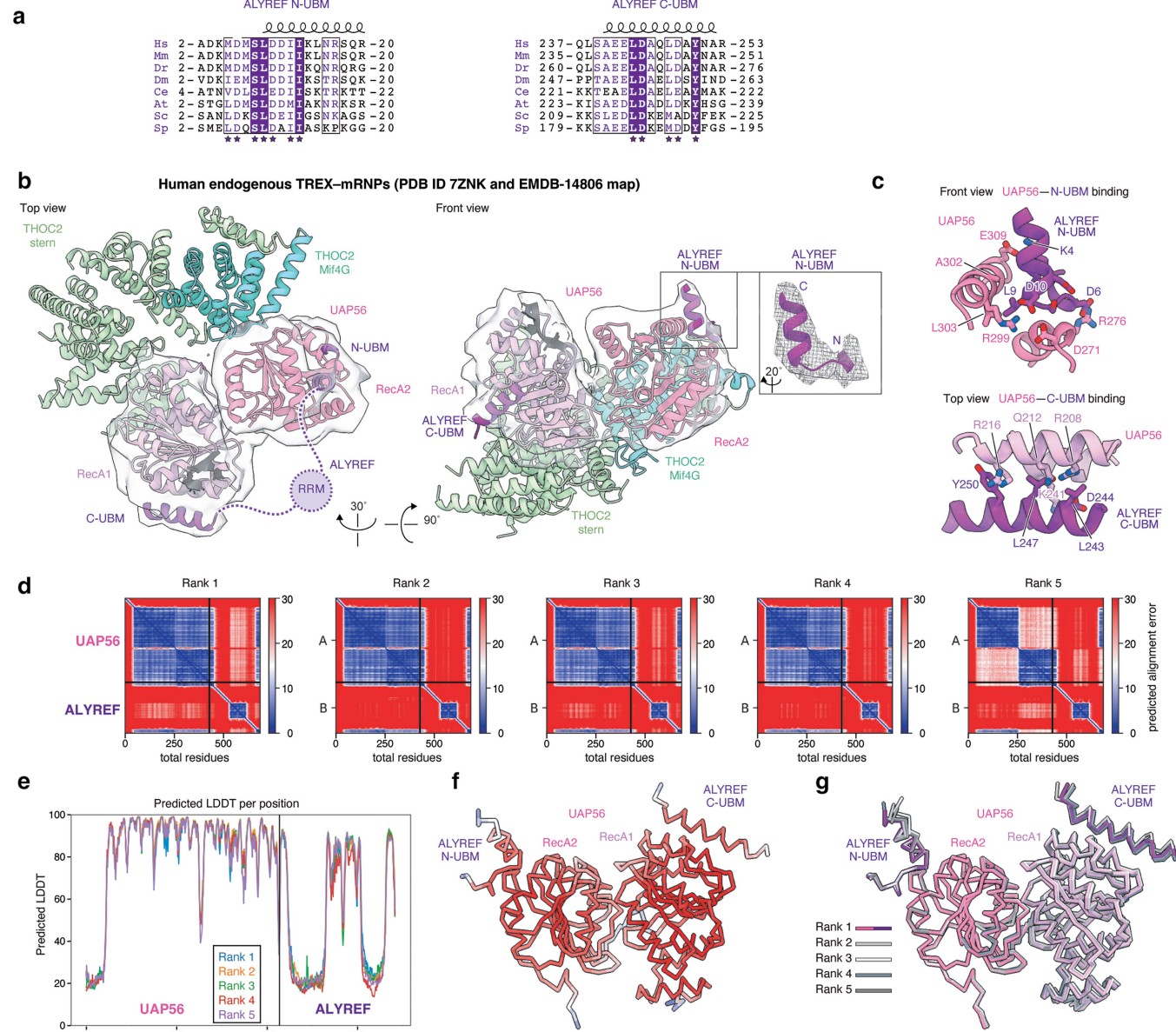

**Extended Data Fig. 1 | Identification of the ALYREF N-UBM binding site in UAP56. a**, Multiple sequence alignment of the ALYREF N-UBM (left) and C-UBM (right) from *H. sapiens* (Hs, Uniprot ID Q86V81), *M. musculus* (Mm, O08583), *D. rerio* (Dr, F1Q9D1), *D. melanogaster* (Dm, Q9V3E7), *C. elegans* (Ce, Q21559), *A. thaliana* (At, Q8L773), and *S. cerevisiae* (Sc, Q12159) and *S. pombe* (Sp, Q09330), coloured by conservation (purple background, invariant residue). Purple stars mark key interface residues with UAP56. **b**, Re-analysis of a human endogenous TREX–mRNP structure reveals an unidentified binding site for a N-UBM on UAP56. Front (left) and top views (middle) of UAP56 bound to THO and mRNPs with the THOC2 MIF4G and Stern domains (green), UAP56 (pink) and the ALYREF C-UBM (purple) (PDB ID 7ZNK), with cryo-EM density around UAP56 and ALYREF shown (EMDB-14806). An additional density at low resolution on UAP56's RecA2 lobe fits with the AlphaFold2 predicted binding site of ALYREF's

N-UBM (purple). **c**, Details of UAP56–N-UBM (top) and UAP56–C-UBM (bottom) interfaces with key residues labelled and shown as sticks. Colours as in **a**. **d-g**, Diagnostic plots for the AlphaFold2 Multimer prediction of ALYREF and UAP56. The N- and C-UBM are predicted with high confidence to bind distinct binding sites: the C-UBM is predicted in the previously experimentally determined binding site on the RecA1 lobe, the N-UBM on a novel binding site on RecA2. Shown are **d**, the PAE plot, **e**, the pLDDT plot, **f**, the structure of the top ranked model in Cα trace coloured by pLDDT (shown are only the ordered and interacting elements: UAP56 RecA1 and RecA2, residues 40-428; ALYREF N-UBM and C-UBM, residues 1-24 and 236-257), and **g**, a superposition of the structures of all five models, rank 1-5, as in **f**, but coloured for rank 1 with UAP56 in pink and ALYREF in purple, and in shades of grey for rank 2-5.

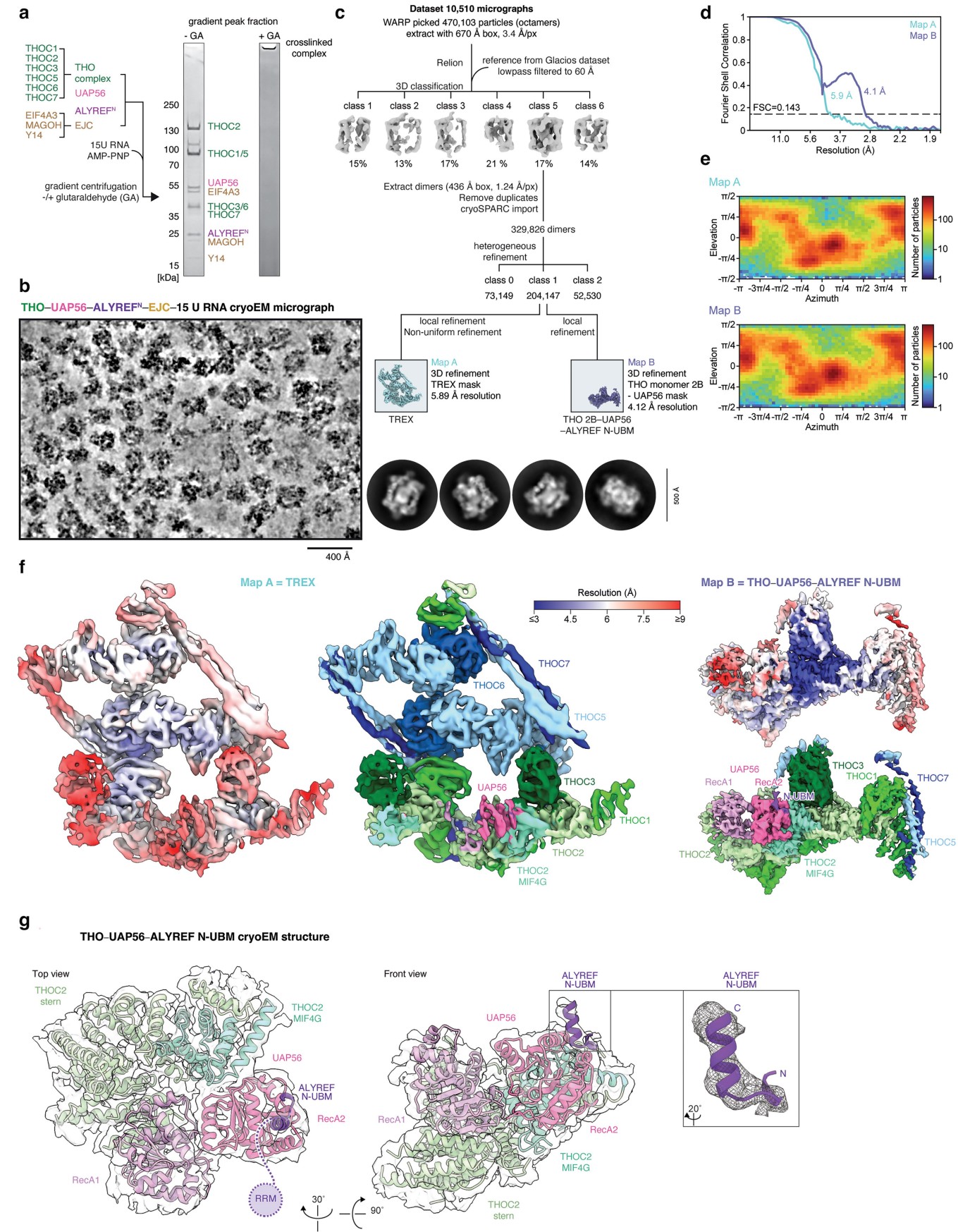

**Extended Data Fig. 2 |** See next page for caption.

**Extended Data Fig. 2 | TREX–EJC–RNA complex reconstitution and cryo-EM analysis. a**, Complex reconstitution scheme for TREX (= THO–UAP56–ALYREF$^N$)–EJC–RNA from recombinant proteins. Complex components were mixed and separated on a 15-40% sucrose gradient with or without 0.05% glutaraldehyde, fractionated and analysed by Coomassie-stained SDS-PAGE. Shown are peak lanes of the crosslinked complex used for cryo-EM sample preparation (right) and the corresponding fraction from the non-crosslinked gradient (left). **b**, Denoised cryo-EM micrograph of TREX–EJC–RNA. Scale bar, 400 Å. **c**, Three-dimensional image classification tree, with representative 2D classes shown below. The dataset contains 10,510 micrographs, of which 470,103 particles (THO octamer) were picked with WARP and processed in RELION and cryoSPARC. The final particle stack contained 204,147 particles and was refined to 5.89 Å for the entire TREX complex (Map A) and to 4.12 Å for a THO monomer (Map B). **d**, Gold-standard Fourier shell correlation (FSC = 0.143) of the TREX complex (Map A) and THO-monomer–UAP56–N-UBM (Map B) cryo-EM maps. **e**, Orientation distribution plots, as visualized in cryoSPARC, for all particles contributing to the TREX complex (Map A) and THO-monomer–UAP56–N-UBM (Map B) cryo-EM maps. **f**, TREX complex (Map A) and THO-monomer–UAP56–N-UBM (Map B) cryo-EM maps coloured by local resolution, alongside the same maps coloured by subunit as in Fig. 1b. **g**, Structure of the THO-monomer–UAP56–N-UBM complex (left, top view; middle and right, front view), with the THOC2 MIF4G and Stern domains (green), UAP56 (pink) and the N-UBM (purple) shown together with the Map B density (inset on the right). The structure reveals density for the ALYREF N-UBM at the newly identified binding site on the UAP56 RecA2 lobe.

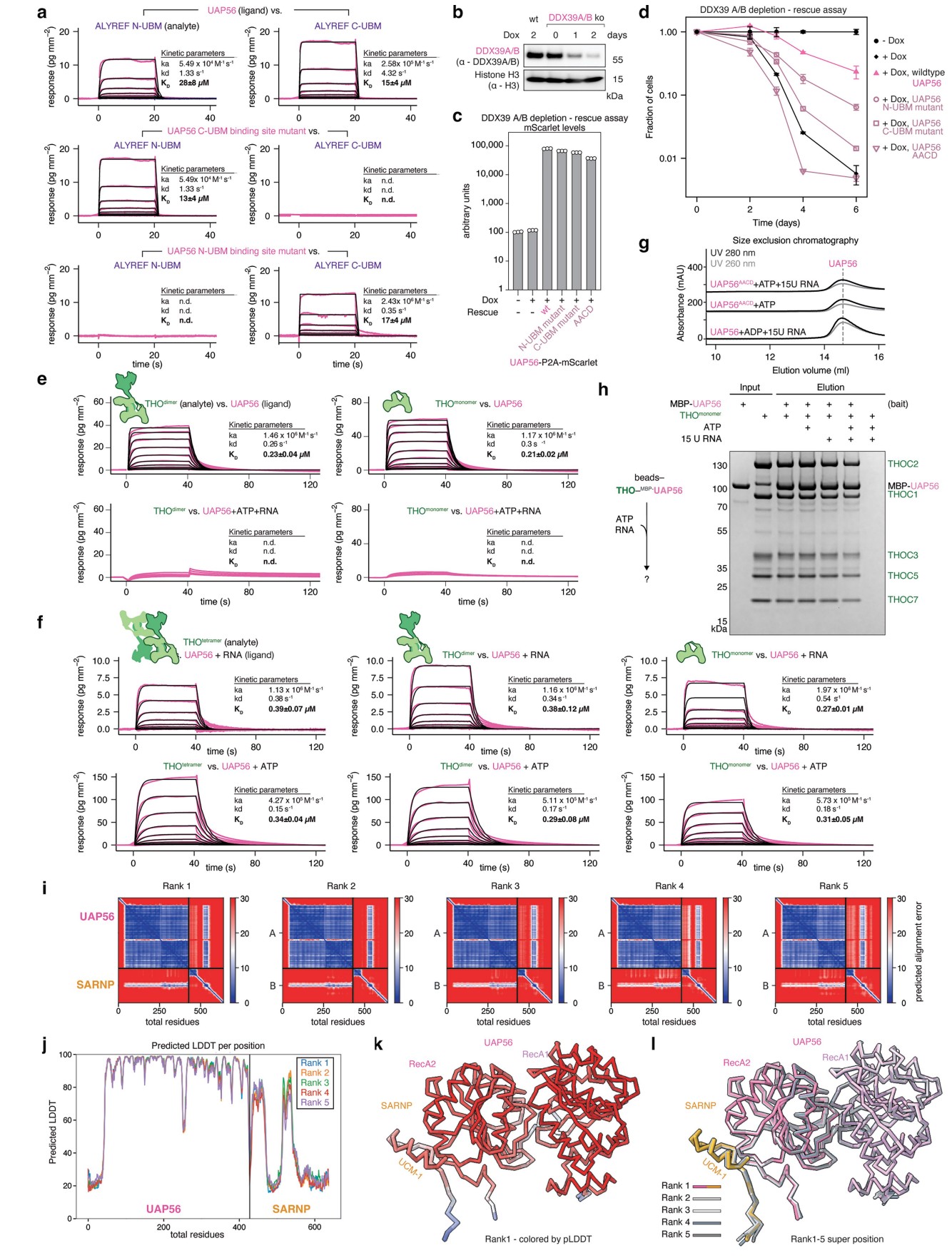

**Extended Data Fig. 3** | See next page for caption.

**Extended Data Fig. 3 | UAP56–ALYREF interfaces are important in vivo, ATP and RNA alone do not dissociate the THO–UAP56 complex, and prediction of a UAP56–SARNP complex. a**, Biochemical evidence that the N-UBM binds a binding site on UAP56 distinct from the C-UBM. GCI-derived direct binding kinetics for UBM peptides as the analyte and UAP56 as the ligand (see Fig. 1f). Sensograms (pink), fitted traces (black), and summary tables are shown. UAP56 binds N- and C-UBM peptides with mid-micromolar affinity (top row). Mutating key residues in the UAP56 C-UBM binding site (R208S, Q212A, R216S, K241S; RQRK) prevents C-UBM binding but leaves N-UBM binding unaffected (middle row). Mutating key residues in the newly identified N-UBM binding site (R276E, A302R, E309S) abolishes N-UBM binding without affecting C-UBM binding (bottom row). **b**, Depletion efficiency of DDX39A and DDX39B after their dox-induced CRISPR-knockout in a human K562 cell line. Shown is a western blot probing for DDX39A/B in wildtype and CRISPR knockout cells at zero, one, or two days after induction of the knockout. A histone H3 western blot loading control is shown alongside. **c**, mScarlet levels, as determined by FACS, assess UAP56 rescue construct expression for the experiment in panel **d**, at two days after the induction of the DDX39A/B knockout and the expression of UAP56-P2A-mScarlet rescue constructs; $n = 3$ samples. **d**, Human K562 cell growth competition assays to probe UAP56–ALYREF interfaces and the UAP56 ATPase activity. Wild type or mutant UAP56 (DDX39B) is ectopically expressed from a doxycycline (dox)-inducible promoter in human K562 cells, which also express dual CRISPR guide RNAs against UAP56 and DDX39A and carry a dox-inducible Cas9. Addition of dox leads to the simultaneous CRISPR-Cas9 knockout of UAP56 and DDX39A and initiates the ectopic expression of the respective rescue construct. See Methods for UAP56–N-UBM, and –C-UBM mutation details. To probe UAP56's ATPase activity we mutated the DECD-box motif to AACD.

Dox-induced cells are mixed at a 1:1 ratio with wild type K562 cells and cell growth is monitored by FACS (see Methods for details); error bars, mean ± s.d. from $n = 3$ independent samples. **e**, GCI-derived direct binding kinetics for a THO dimer or monomer as the analyte and UAP56 as the ligand (compare to Fig. 1f, g). Sensograms (pink), fitted traces (black), and summary tables are shown. **f**, As in **e**, but probing THO tetramer (left), dimer (middle) or monomer (right) as the analyte and UAP56 with either only RNA (top) or only ATP (bottom) as the ligand. Because experiments using UAP56 + ATP or UAP56 + RNA were done on separate GCI chips, the numerical and absolute response levels differ between these conditions. Sensograms (pink), fitted traces (black), and summary tables are shown. **g**, Size exclusion chromatography traces of wild type UAP56 with ADP and a 15 nucleotide poly-Uridine RNA (15U RNA), or of the ATPase mutant UAP56 AACD (D196A, E197A) with ATP and with or without the 15U RNA. Shown are the UV-traces measured at 280 nm (black) and 260 nm (grey). **h**, ATP and RNA lead to the partial dissociation of THO from UAP56 in vitro. Experimental setup (left) and Coomassie stained SDS-PAGE gel of a representative experiment (right). **i-l**, Diagnostic plots for the AlphaFold2 Multimer prediction of SARNP–UAP56. The UCM-1 is predicted with high confidence to bind the UAP56's RecA2 lobe. Shown are **i**, the PAE plot, **j**, the pLDDT plot, **k**, the structure of the top ranked model in Cα trace coloured by pLDDT (shown are only the ordered and interacting elements: UAP56 RecA1 and RecA2, residues 40-428; SARNP UCM-1 region, residues 84-110), and **l**, a superposition of the structures of all five models, rank 1-5, as in **k**, but coloured for rank 1 with UAP56 in pink and SARNP in yellow, and in shades of grey for rank 2-5. The predicted complex is highly similar to a recent crystal structure of a human UAP56–yeast SARNP complex[22], exhibiting an RMSD of 0.957 Å across 391 atom pairs.

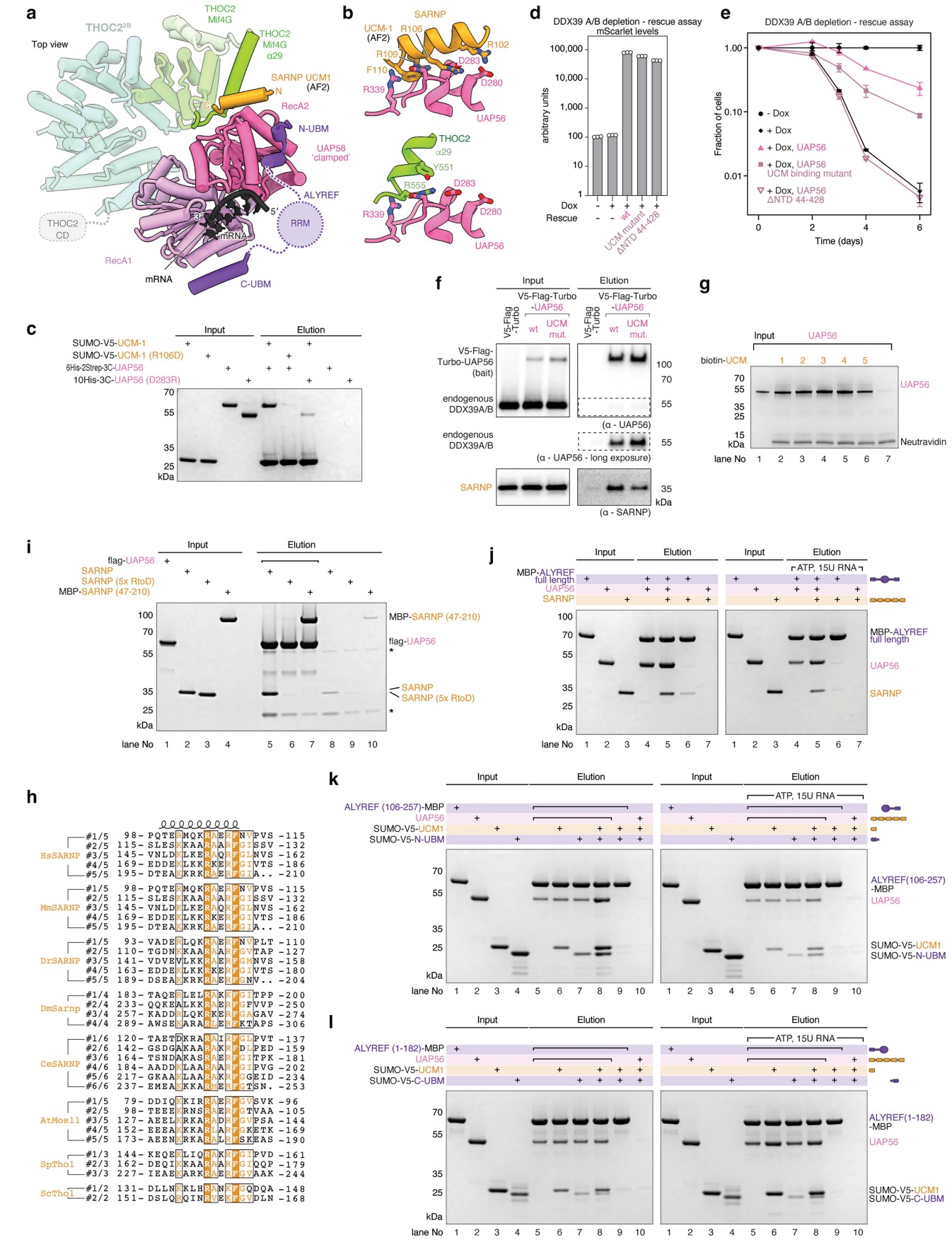

**Extended Data Fig. 4** | See next page for caption.

**Extended Data Fig. 4 | SARNP UCMs bind UAP56 and UAP56 can simultaneously bind the ALYREF N-UBM, C-UBM and SARNP UCM-1. a**, Structural modelling of a UAP56–RNA–SARNP-UCM-1–ALYREF-N-UBM complex (see Fig. 2d) superimposed on THO–UAP56 reveals a clash between the SARNP UCM-1 and the MIF4G domain in THOC2. UAP56, pink; RNA, black; SARNP, yellow; ALYREF UBMs, purple; THOC2 MIF4G and Stern in green and, except for the clashing THOC2 α-helix 29, transparent. **b**, Details of the UAP56–SARNP UCM-1 model with key interface residues shown as sticks (top). Shown below is the same view of UAP56 in the THO–UAP56 complex (THOC2, green)[20], revealing a steric clash between SARNP and THOC2. Colours as in panel **a**. **c**, UAP56 interacts with a SARNP UCM. Recombinant SUMO-V5 tagged SARNP UCM-1 (residues 82-115) is immobilized on magnetic V5 beads and incubated with recombinant full length UAP56. After washing out unbound UAP56, bead-bound complexes are eluted at low pH and analysed by SDS-PAGE followed by Coomassie staining, revealing that UAP56 interacts with the UCM-1 (lane 5). Mutating key residues in the predicted interface (SARNP UCM-1 R106D or UAP56 D283R) prevents the interaction almost completely (lanes 6 and 7). UAP56 shows little background binding (lanes 8 and 9). **d**, mScarlet levels, as determined by FACS, assess UAP56 rescue construct expression for the experiment in panel **e**, at two days after the induction of the DDX39A/B knockout and the expression of UAP56-P2A-mScarlet rescue constructs. Samples for '−dox', '+dox' and '+dox, wildtype rescue' are reproduced from Extended Data Fig. 3c, d for a direct comparison; $n = 3$ independent experiments. **e**, Human K562 cell growth competition assays to probe UAP56–SARNP and –TREX-2 interfaces. Assay as carried out in Extended Data Fig. 3d, but for the UAP56–UCM mutant or UAP56 truncated for its N-terminal domain (ΔNTD). The growth curves for the '−dox', '+dox' and '+dox, wildtype rescue' are reproduced from Extended Data Fig. 3d; error bars, mean ± s.d. from $n = 3$ independent samples. **f**, Flag-immunoprecipitation coupled to a western blot experiment of wildtype or UCM-binding mutant (D283R) V5-flag-TurboID-tagged UAP56. V5-flag-TurboID-UAP56 was immunoprecipitated using magnetic FLAG-beads from the nuclear extract of human K562 cells, two days after the expression of the exogenous UAP56 construct was induced by the addition of doxycycline. After on-bead washes, UAP56 and co-precipitating proteins were eluted from the beads by boiling in SDS-PAGE sample buffer and were probed by western blotting for UAP56 and SARNP (right). Input samples are shown alongside (left). **g**, UAP56 binds the five UCMs of human SARNP. Biotinylated UCM peptides are immobilized on neutravidin agarose beads and incubated with recombinant UAP56. Unbound UAP56 is removed through washes and bound UAP56 eluted at low pH before

analysis on a Coomassie-stained SDS-PAGE gel (lanes 2-6). Input UAP56 is shown in lane 1. UAP56 shows low background binding (lane 7). **h**, Multiple sequence alignment of the SARNP UCMs from *H. sapiens* (Uniprot ID P82979), *M. musculus* (Mm, Q9D1J3), *D. rerio* (Dr, Q504C3), *D. melanogaster* (Dm, Q9VHC8), *C. elegans* (Ce, Q9N3G0), *A. thaliana* (At, Q9LZ08), *S. pombe* (Sp, O74871) and *S. cerevisiae* (Sc, P40040) (bottom), coloured by conservation (orange background, invariant residue). SARNP is multivalent in all eukaryotes, with multiple UCMs connected by low-complexity linkers (Fig. 2c). We speculate that UCM-multivalency may have evolved to disassemble multivalent tetrameric THO–UAP56 complexes in native mRNPs and/or to increase the efficiency of SARNP binding to UAP56 in native mRNPs. **i**, The SARNP UCMs are required for UAP56 interaction, but not the SARNP SAP domain. In vitro pulldown where flag-tagged UAP56 is immobilized on M2 anti-flag resin and incubated with recombinant SARNP constructs. Unbound protein is removed through washes and bead-bound complexes are eluted at low pH prior to analysis on a Coomassie-stained SDS-PAGE gel. Recombinant full length SARNP with UAP56 (lane 5), as does SARNP lacking the N-terminal SAP domain (MBP-SARNP residues 47-210, lane 7). Mutating a key arginine (R) in all five SARNP UCMs abolishes binding to UAP56 (SARNP 5x RtoD, lane 6). Protein inputs (lane 1-4) and background binding controls are shown (lanes 8-10). The asterisks mark the M2 flag antibody heavy and light chains. **j-l**, UAP56 can simultaneously bind the ALYREF N-UBM, C-UBM and the SARNP UCM. In vitro pulldown experiments, where recombinant MBP-tagged ALYREF is immobilized on amylose beads and incubated with recombinant UAP56 and SARNP, UCM or N- or C-UBM containing constructs. Unbound proteins are removed through washes before bead-bound complexes are eluted by boiling in SDS sample buffer and analysed on Coomassie-stained SDS-PAGE gels. Experiments are performed in the absence (left) or presence (right) of ATP and 15 nucleotide poly-Uridine RNA (15U RNA). **j**, Full length ALYREF interacts with UAP56 (lane 4) or UAP56–SARNP (lane 5). SARNP does not bind ALYREF (lane 6), and neither UAP56 nor SARNP show background binding to the beads (lane 7). Protein inputs are shown in lanes 1-3. **k**, ALYREF harbouring only a C-UBM (residues 106-257) interacts with UAP56 (lane 5), and with an UAP56–UCM-1 complex (lane 6), an UAP56–N-UBM complex (lane 7), or an UAP56–UCM–N-UBM complex (lane 8). UCM-1 and N-UBM do not interact with ALYREF (lane 9) and show no background binding (lane 10). Protein inputs are shown in lanes 1-4. **l**, as in **k**, but using an ALYREF construct harbouring only an N-UBM (residues 1-182) and using isolated ALYREF N-UBM instead of C-UBM constructs.

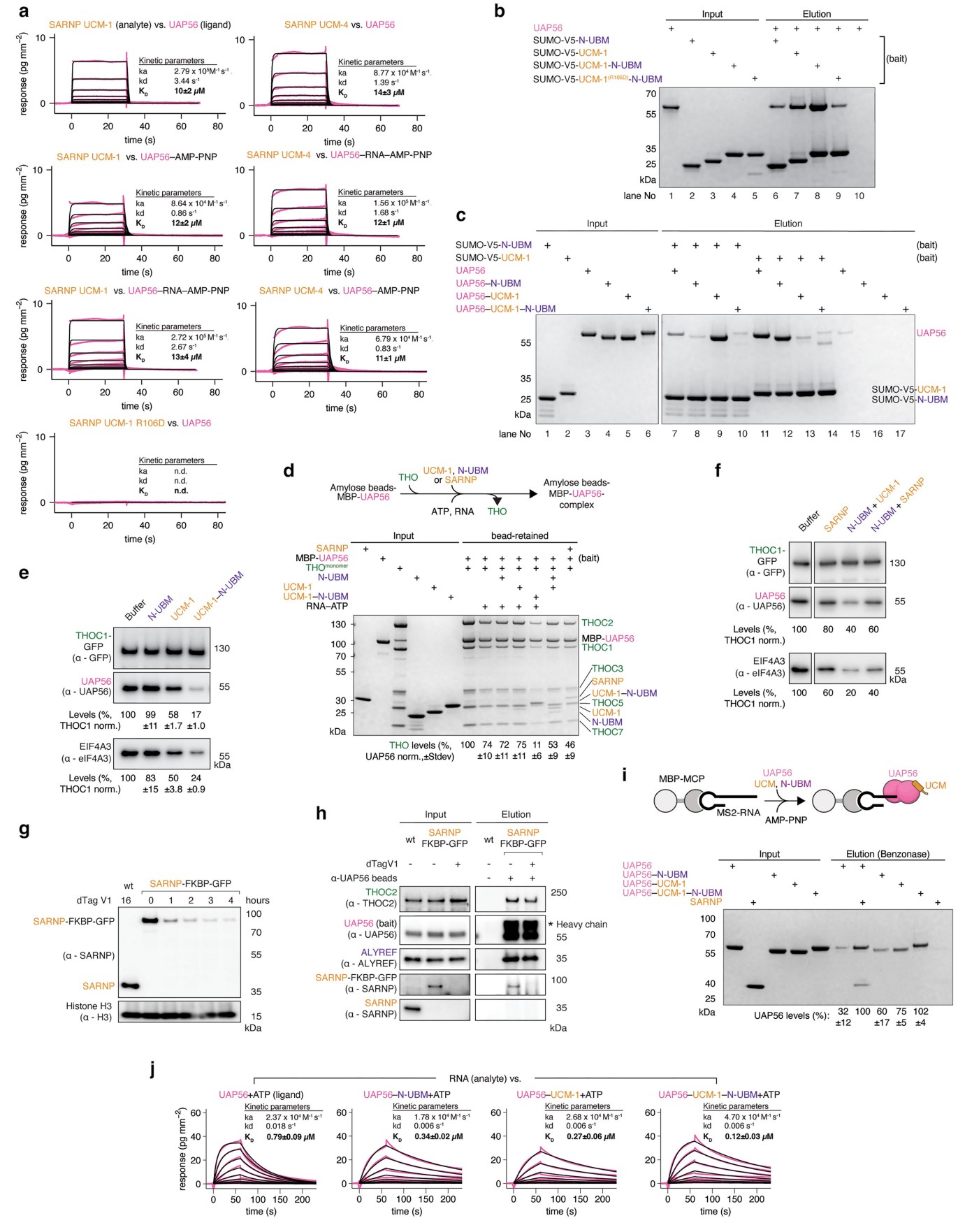

**Extended Data Fig. 5 | See next page for caption.**

**Extended Data Fig. 5 | ALYREF N-UBM and SARNP UCM cooperatively bind UAP56 and SARNP assists TREX disassembly and stabilizes mRNA-clamped UAP56. a**, GCI-derived binding kinetics for SARNP UCM peptides as the analyte and UAP56 as the ligand. UAP56 binds the SARNP UCM-1 (top row) and UCM-4 (middle row) with similar affinities ($K_D$s of 10 to 20 μM), and binding is not affected by the addition of the non-hydrolysable ATP analogue AMP-PNP or by AMP-PNP and a 15 nucleotide poly-Uridine RNA. The mutation R106D in UCM-1 prevents UAP56 binding (bottom). Sensograms (pink), fitted traces (black), and summary tables are shown. **b**, The SARNP UCM-1 and ALYREF N-UBM bind UAP56 simultaneously in a UCM-1–N-UBM fusion protein. SUMO-V5-3C-N-UBM, SUMO-V5-3C-UCM, or a SUMO-V5-3C-UCM–N-UBM fusion (with or without the SARNP R106D) mutation are immobilized on magnetic anti-V5 beads, incubated with UAP56 and washed, eluted and analysed as by Coomassie-stained SDS-PAGE. UAP56 binds the isolated UCM-1 with apparent higher affinity than the N-UBM (lanes 6 and 7). The binding to the UCM-1–N-UBM fusion is greater than for individual peptides (lane 8 vs. lanes 6 and 7), suggesting that both peptides can bind UAP56 simultaneously. Mutating R106 in the UCM–N-UBM fusion reduces UAP56 binding to the level of the N-UBM alone (lane 9), and UAP56 shows little background binding (lane 10). **c**, The SARNP UCM-1 and ALYREF N-UBM bind their cognate binding sites in UAP56 fusion proteins. SUMO-V5-3C tagged N-UBM or UCM-1 are immobilized on magnetic V5 beads and incubated with UAP56 or UAP56–N-UBM, UAP56–UCM or UAP56–UCM–N-UBM fusion proteins. Beads were washed, eluted, and analysed as in panel **b**. UAP56 binds both N-UBM and UCM (lanes 7 and 11). The UAP56–N-UBM fusion binds the UCM like wild type UAP56, but not the N-UBM (lanes 12 and 8). UAP56–UCM does not bind to immobilized UCM and binds the N-UBM with apparent higher affinity than UAP56 alone (lanes 13 and 9), suggesting that N-UBM and UCM bind synergistically. We note that the synergistic binding of the UAP56–N-UBM to the UCM was not apparent, which may be specific to this condition because of the lower affinity of the isolated N-UBM versus UCM-1 to UAP56. The UAP56–UCM–N-UBM fusion binds neither immobilized N-UBM nor UCM (lanes 10 and 14), suggesting that both peptides are bound to their cognate binding site in the fusion protein. None of the UAP56 proteins exhibits relevant background binding (lanes 15-18). **d**, THO–UAP56 disassembly assay with recombinant proteins. Experiment schematic (top) and SDS-PAGE analysis of the results are shown (bottom, Coomassie stain). The amount of bead-retained THO complex is quantified underneath from three independent experiments. The addition of ATP and RNA alone or together with either an ALYREF N-UBM peptide or a SARNP UCM-1 peptide resulted in the comparable, but only partial, dissociation of THO from UAP56. The THO–UAP56 complex disassembled more efficiently when we added the ALYREF N-UBM together with the SARNP UCM-1 peptide or when we used full-length SARNP with its five UCMs. When we instead added a peptide comprising the SARNP UCM fused to the ALYREF N-UBM, the THO–UAP56 complex

disassembled almost completely. To account for variations in MBP-UAP56 immobilization, the band intensities of each lane are normalized by the MBP-UAP56 levels prior to THO quantification. **e-f**, Western blot analysis for endogenous TREX disassembly experiments. Shown are western blots for the experiment in Fig. 2e in **e.**, and for a separate experiment using full-length recombinant SARNP, with or without the ALYREF N-UBM, or non-fused N-UBM and SARNP UCM-1 peptides in **f**. We blotted for the EJC subunit EIF4A3 as a proxy for mRNPs, owing to its high abundance in purified human TREX–mRNPs according to mass spectrometry[2]. **g**, Western blot analysis of SARNP levels in the SARNP-FKBP-GFP cell line after 0, 1, 2, 3, or 4 h of adding dTAG-V1. SARNP is homozygously tagged and depletes rapidly upon the addition of dTAG-V1. A histone H3 western blot is shown alongside as a loading control. **h**, UAP56 is immunoprecipitated with or without the rapid, dTAG-V1-dependent depletion of cellular SARNP. Human K562 wild type or dTAG-degron containing SARNP cells are harvested after 8 h of DMSO or dTAG-V1 treatment. The obtained cells are fractionated into nuclei and cytoplasm and analysed as in panel **h**, with western blotting for the THO subunit THOC2, UAP56, ALYREF, and SARNP. Notably, no increase in THO levels is observed in the UAP56 interactome upon SARNP depletion. This may be explained by the previous observation that nearly all THO complexes are engaged with mRNPs at steady-state and are therefore UAP56-bound[2]. Consistently, the cellular concentration of UAP56 exceeds THO complex levels by around 40-fold[106]. Thus, perturbing TREX disassembly would not increase the levels of UAP56-bound THO complexes. **i**, UAP56 is stabilized on RNA by SARNP. A 450 nucleotide long MS2-loop containing AdML RNA is immobilized on amylose beads through an MBP-MCP fusion protein. The RNA is incubated in the presence of the non-hydrolysable ATP analogue AMP-PNP with UAP56, UAP56 and full length SARNP or with UAP56–N-UBM, UAP56–UCM or UAP56–UCM–N-UBM fusion proteins. Unbound proteins are washed out, and RNA-bound proteins are eluted through RNase (benzonase) digestion of the RNA and visualized on Coomassie stained SDS-PAGE gels, with the quantification of bound UAP56 shown alongside (UAP56 with SARNP is set to 100% bound). UAP56 alone shows little RNA binding. The presence of a N-UBM or UCM increases the RNA bound fraction, and the presence of the UCM–N-UBM fusion or full length SARNP leads to maximum RNA binding. **j**, GCI-derived binding kinetics for a 15 nucleotide poly-Uridine RNA (immobilized) probed with UAP56, UAP56–UCM-1, UAP56–N-UBM or an UAP56–UCM-1–N-UBM fusion, with ATP in all buffers. Sensograms (pink line), the fitted model (black), and a binding kinetics summary table are shown, revealing that N-UBM and UCM-1 increase the affinity of UAP56–ATP to RNA approximately 6-fold. Thus, SARNP and UAP56 mRNA-clamping do not only promote THO release but also stabilize UAP56–mRNP complexes. This is consistent with cellular data for a function of SARNP downstream of TREX–mRNPs (see also panels **g-i** and Fig. 5b), and the THO-dependent association of yeast SARNP (Tho1) with nascent RNA[29].

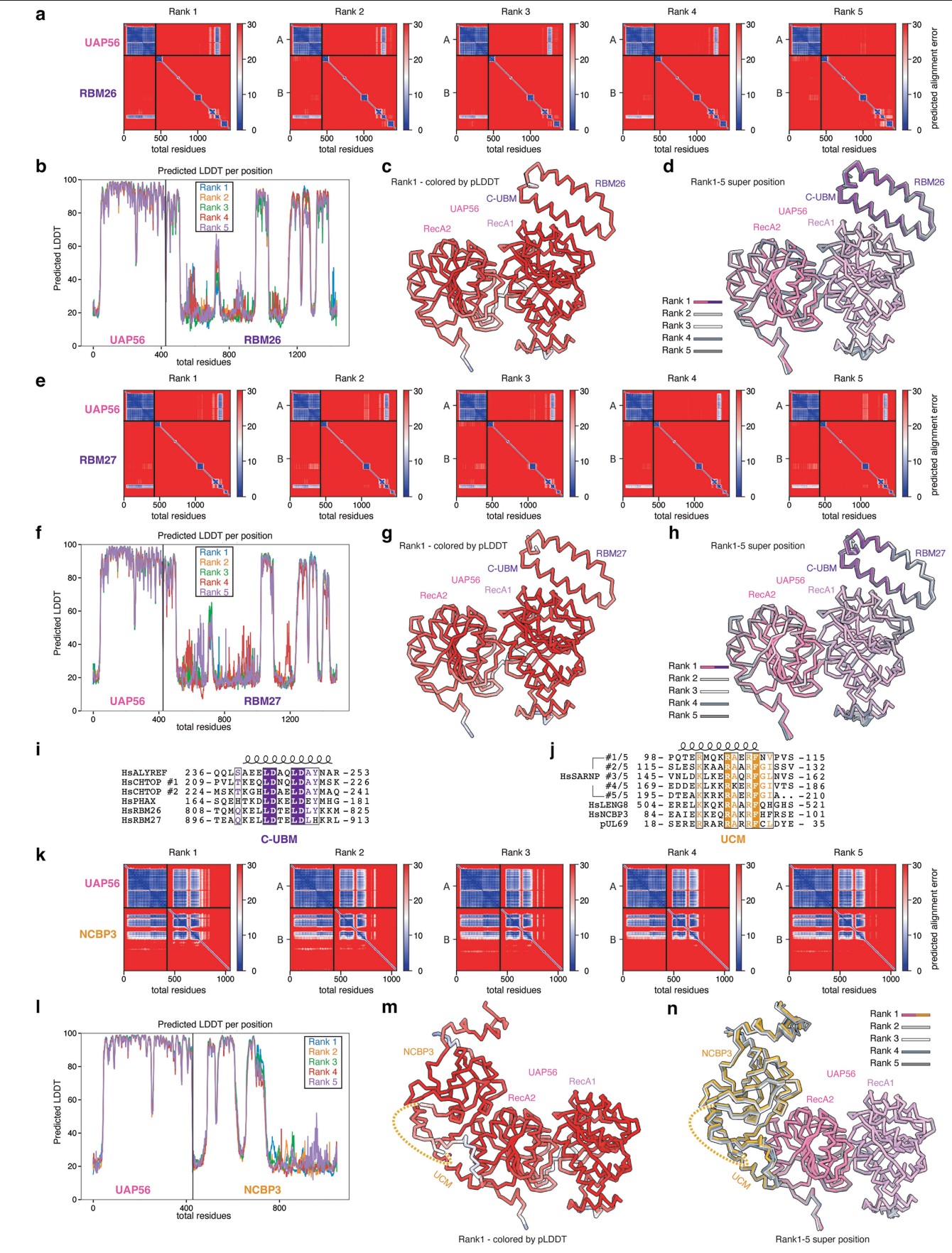

**Extended Data Fig. 6** | See next page for caption.

**Extended Data Fig. 6 | AlphaFold2 Multimer identifies putative UAP56 interactors. a-h**, Diagnostic plots for the AlphaFold2 Multimer prediction of the paralogs RBM26 **a-d**, or RBM27 **e-h**, and UAP56. A putative C-UBM is predicted for RBM26 and RBM27 with high confidence to bind the characterized C-UBM binding site on the RecA1 lobe of UAP56. RBM26 and RBM27 have been implicated in RNA decay[41]. Shown are **a,e**, the PAE plots, **b,f**, the pLDDT plots, **c,g**, the structures of the top ranked model in Cα trace coloured by pLDDT (shown are only the ordered and interacting elements: UAP56 RecA1 and RecA2, residues 40-428; RBM26/27 C-UBM region residues 803-852/891-940), and **d,h**, a superposition of the structures of all five models, rank 1-5, as in **c,g**, but coloured for rank 1 with UAP56 in pink and RBM26/27 in purple, and in shades of grey for rank 2-5. **i**, Multiple sequence alignment of known and novel human C-UBM containing proteins (HsALYREF, Uniprot ID Q86V81, HsCHTOP Q9Y3Y2, HsPHAX Q9H814, HsRBM26 Q5T8P6, HsRBM27 Q9P2N5), coloured by conservation (purple underground = invariant residue). **j**, Multiple sequence alignment of putative UCM motifs in human SARNP, LENG8, NCBP3 and the Human cytomegalovirus protein pUL69 (HsSARNP, Uniprot ID P82979, HsLENG8 Q96PV6, HsNCBP3 Q53F19, pUL69 P16749), coloured by conservation (orange underground = invariant residue). **k-n**, Diagnostic plots for the AlphaFold2 Multimer prediction of NCBP3 and UAP56. A UCM containing region is predicted with high confidence to bind to the RecA2 lobe of UAP56. NCBP3 has been implicated in mRNA biogenesis[40]. Shown are **k**, the PAE plot, **l**, the pLDDT plot, **m**, the structure of the top ranked model in Cα trace coloured by pLDDT (shown are only the ordered and interacting elements: UAP56 RecA1 and RecA2, residues 40-428; NCBP3, residues 59-184 and 231-295), and **n**, a superposition of the structures of all five models, rank 1-5, as in **m**, but coloured for rank 1 with UAP56 in pink and NCBP3 in yellow, and in shades of grey for rank 2-5.

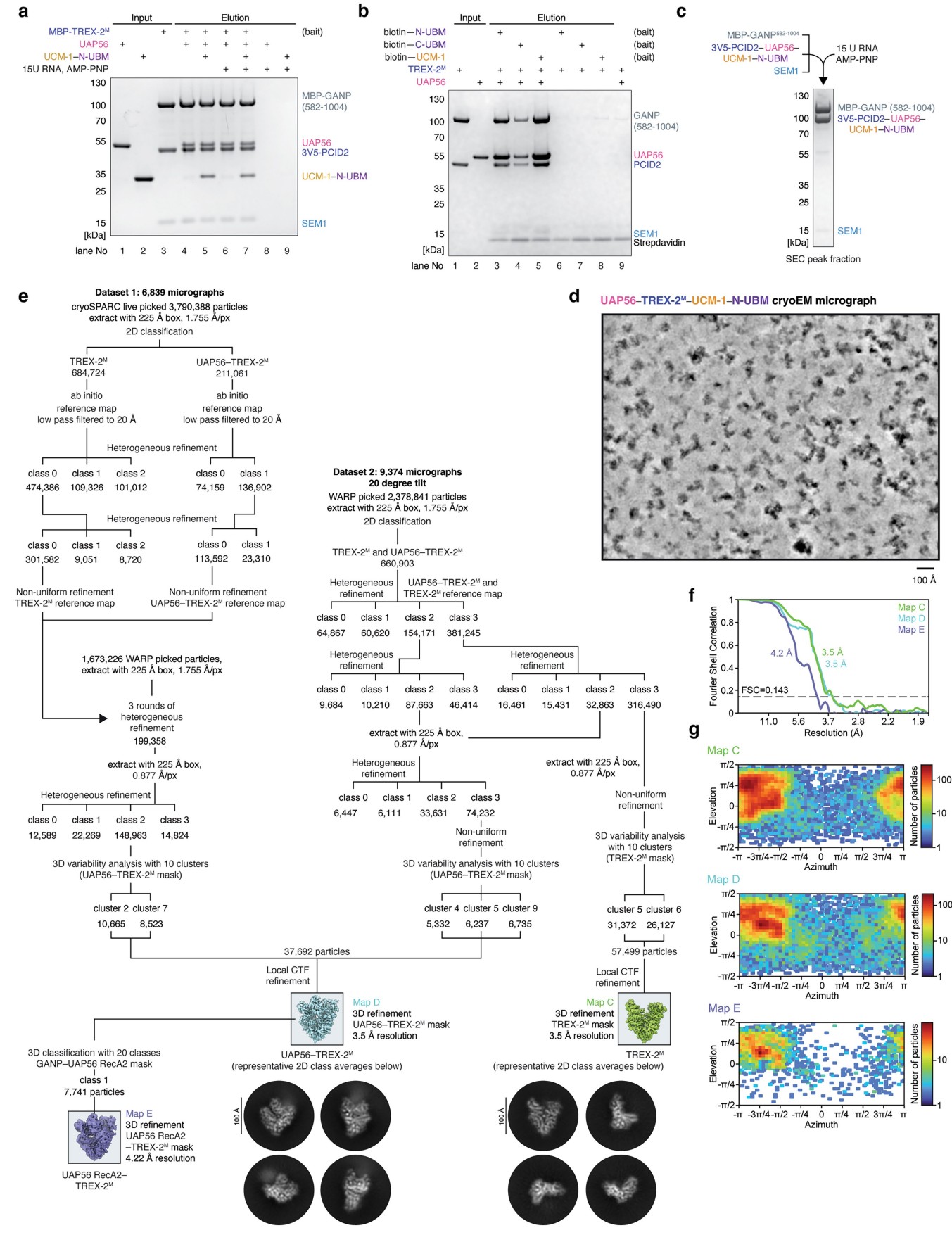

**Extended Data Fig. 7 | See next page for caption.**

**Extended Data Fig. 7 | TREX-2^M and UAP56–TREX-2^M complex cryo-EM analysis. a**, In vitro pulldown assay probing the UAP56–TREX-2^M interaction. Recombinant TREX-2^M complex (MBP-GANP residues 582-1004, PCID2, SEM1) is immobilized on amylose beads through the MBP on the GANP subunit and incubated with UAP56 in the presence or absence of UCM-1–N-UBM peptide and/or 15 U RNA and AMP-PNP. After washes the bead bound proteins are eluted with a maltose containing buffer and visualized in Coomassie stained SDS-PAGE gels. UAP56 forms a near stochiometric complex with TREX-2^M, and complex formation is compatible with binding of the UCM-1–N-UBM peptide and the presence of 15 U RNA and AMP-PNP. Lane 6 is additionally shown in Fig. 3d. **b**, UAP56 can interact with the N-UBM, C-UBM or UCM-1 when bound to the TREX-2^M complex. Biotinylated N-UBM, C-UBM or UCM-1 peptide is immobilized on neutravidin agarose beads and incubated with recombinant TREX-2^M and UAP56. After several washes, bead-bound complexes are eluted using low pH and analysed by SDS-PAGE followed by Coomassie staining. UAP56 in the UAP56–TREX-2 complex can bind the N-UBM, C-UBM and UCM-1 (lanes 3-5), and this interaction occurs through UAP56, since isolated TREX-2^M does not bind these peptides (lanes 6-8). UAP56 and TREX-2^M do not bind unspecifically to the beads (lane 9). **c**, Complex reconstitution scheme for the UAP56–TREX-2^M complex for cryo-EM. MBP-GANP residues 582-1004 was

incubated with a complex consisting of the PCID2–UAP56–UCM–N-UBM fusion protein and SEM1, 15 poly-Uridine RNA and the non-hydrolysable ATP analogue AMP-PNP and the formed complex separated by size exclusion chromatography, with a peak fraction shown on a Coomassie-stained SDS-PAGE gel. **d**, Denoised cryo-EM micrograph of the UAP56–TREX-2^M sample. Scale bar, 100 Å. **e**, Three-dimensional image classification tree, with representative 2D classes shown below. From two datasets, containing 6,839 and 9,374 (collected at 20-degree stage tilt) micrographs, 1,673,226 and 2,378,841 particles were picked using a custom trained BoxNet in WARP and processed in cryoSPARC. The final particle stacks contained 57,499 particles for TREX-2^M (Map C) and 37,692 particles for UAP56–TREX-2^M (Map D) and were each refined to 3.5 Å resolution. The Map D particle stack was further classified using a GANP–UAP56 RecA2 mask, yielding a particle stack of 7,741 particles with observable RecA2 density which was refined to 4.22 Å resolution (Map E). **f**, Gold-standard Fourier shell correlation (FSC = 0.143) of the TREX-2^M complex (Map C), UAP56 RecA1–TREX-2^M complex (Map D), and the UAP56 RecA1-RecA2–TREX-2^M complex (Map E) cryo-EM maps. **g**, Orientation distribution plots, as visualized in cryoSPARC, for all particles contributing to the TREX-2^M complex (Map C), the UAP56 RecA1–TREX-2^M complex (Map D), and the UAP56 RecA1-RecA2–TREX-2^M complex (Map E) cryo-EM maps.

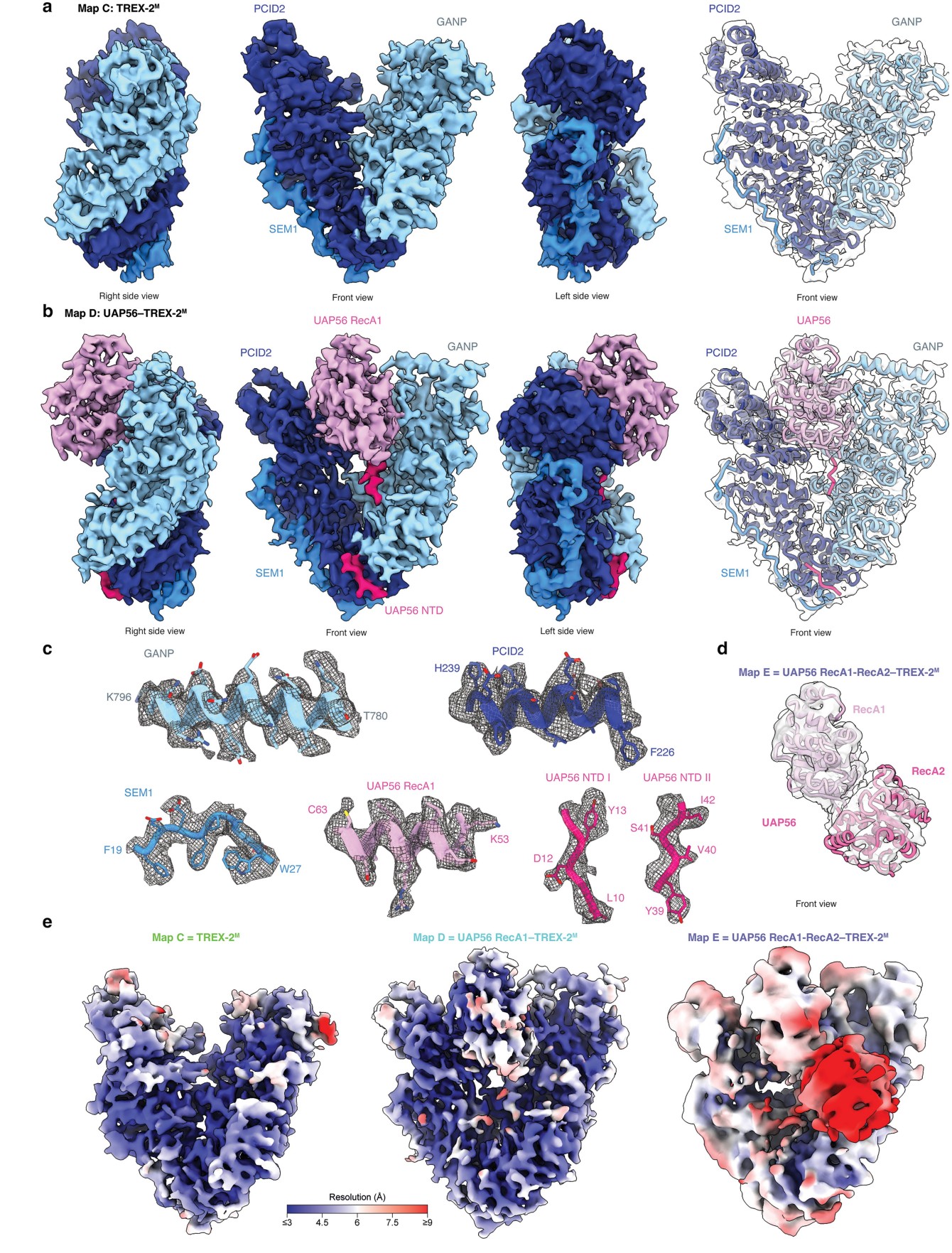

**Extended Data Fig. 8** | See next page for caption.

**Extended Data Fig. 8 | TREX-2$^M$ and UAP56–TREX-2$^M$ complex cryo-EM analysis. a**, TREX-2$^M$ complex (Map C) cryo-EM density, shown in left, front, and right side views and coloured by subunit (GANP, light blue; PCID2, dark blue; SEM1, dodger blue). Shown alongside on the very right is the superposition of the TREX-2$^M$ model, in cartoon representation and again coloured by subunit, superimposed on the cryo-EM Map C. **b**, UAP56 RecA1–TREX-2$^M$ complex (Map D) cryo-EM density, shown in left, front, and right side views and coloured by subunit (UAP56 RecA1, pink; GANP, light blue; PCID2, dark blue; SEM1, blue). Shown alongside on the very right is the superposition of the UAP56 RecA1–TREX-2$^M$ model, in cartoon representation and again coloured by subunit, superimposed on the cryo-EM Map D. **c**, Representative segments of GANP, PCID2, SEM1, and UAP56 NTD and RecA1 from Map D superimposed on the respective cryo-EM densities. **d**, Superposition of the UAP56 RecA1 RecA2 model, in cartoon representation and coloured in pink, on the cryo-EM Map E. **e**, TREX-2$^M$ complex (Map C), UAP56 RecA1–TREX-2$^M$ complex (Map D), and UAP56 RecA1 RecA2–TREX-2$^M$ complex (Map E) cryo-EM maps coloured by local resolution.

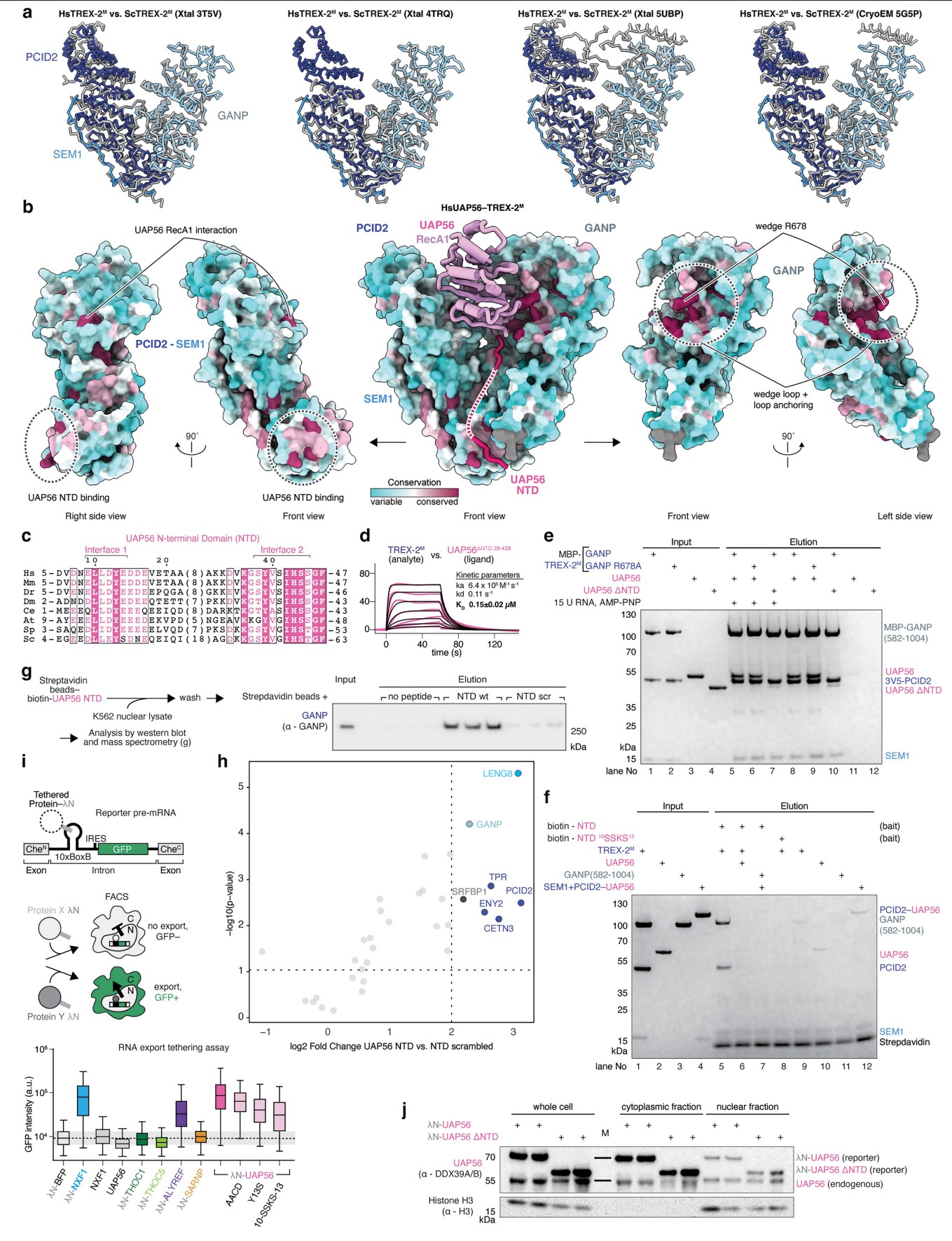

**Extended Data Fig. 9** | See next page for caption.

**Extended Data Fig. 9 | Structural analysis of TREX-2$^M$ and UAP56–TREX-2$^M$, UAP56 NTD interactome and controls for the RNA export tethering assay.**
**a**, The human TREX-2$^M$ complex cryo-EM structure is highly similar to previous crystal structures and a cryo-EM structure of the yeast TREX-2$^M$ complex. Shown are superpositions of the human TREX-2$^M$ model determined in this study, in front view and as Cα trace coloured by subunit (GANP, light blue; PCID2, dark blue; SEM1, dodger blue), on previous structures of the yeast TREX-2$^M$ complex (in grey) determined through crystallography (left, PDB ID 3T5V; middle left, PDB ID 4TRQ; middle right, PDB ID 5UBP) or cryo-EM (right, PDB ID 5G5P)[50–53]. **b**, Structure of the UAP56–TREX-2$^M$ complex, with UAP56 in pink and cartoon representation and PCID2, SEM1 and GANP in surface representation and coloured by sequence conservation (maroon, conserved; cyan, variable). A front view of the entire complex is shown in the centre, flanked by front and side views of PCID2–SEM1 (left) or GANP (right). Surfaces patches involved in complex formation, such as the UAP56 RecA1 proximal patch in PCID2, the UAP56 NTD binding site, the GANP wedge or the PCID2–GANP interface, show a high degree of sequence conservation. **c**, Multiple sequence alignment of the UAP56 N-terminal domain (NTD) from *H. sapiens* (Hs, Uniprot ID Q13838), *M. musculus* (Mm, Q9Z1N5), *D. rerio* (Dr, Q803W0), *D. melanogaster* (Dm, Q27268), *C. elegans* (Ce, Q18212*), A. thaliana* (AtRH56, Q9LFN6), *S. pombe* (SpSub2, O13792) and *S. cerevisiae* (ScSub2, Q07478). Residues invariant or conserved among these species are highlighted in pink or light pink, respectively. **d**, GCI-derived binding kinetics for TREX-2$^M$ (immobilized) probed with UAP56ΔNTD1 (residues 28-428). Sensogram (pink line), the fitted model (black), and a summary table of the binding kinetics is shown. Related to Fig. 4c. **e**, In vitro pulldown assay as in Extended Data Fig. 7a, but using wild type TREX-2$^M$ or the wedge mutant TREX-2$^M$ (containing GANP R678A) and wild type UAP56 or UAP56 ΔNTD (residues 44-428). A UAP56–TREX-2$^M$ complex is formed with the wedge loop mutant TREX-2$^M$, while the deletion of the UAP56 NTD abolishes complex formation. **f**, In vitro pulldown assay probing TREX-2$^M$ binding of the isolated UAP56 NTD. Biotinylated UAP56 NTD peptides (residues 1-21, wild type or mutant E9K, L10S, L11A, D12K, Y13S) are immobilized on streptavidin beads and incubated with TREX-2$^M$ in the presence or absence of UAP56, or with TREX-2$^M$ with UAP56 fused to the C-terminus of PCID2. Protein complexes are eluted and visualized on Coomassie stained SDS-PAGE gels. Wild type, but not mutant, NTD peptide forms a complex with TREX-2$^M$ (compare lane 8 to 5), and complex formation is abolished in the presence of full length UAP56. TREX-2$^M$ with UAP56 fused to the PCID2 C-terminus also does not form a complex with NTD peptide, suggesting that UAP56 is TREX-2$^M$-bound in the fusion construct.

**g**, UAP56 NTD protein interactome. A C-terminally biotinylated UAP56 NTD peptide (residues 1-21) was immobilized on strepdavidin beads, with a scrambled peptide serving as the control. Beads were incubated with nuclear K562 lysate, washed and the NTD peptide's interactome analysed by western blot, probing for the TREX-2 subunit GANP, or by mass spectrometry (see panel **h**). **h**, Volcano plot showing the log2 fold-changes of the protein interactome of wildtype versus scrambled UAP56 NTD. Proteins with a log2 fold-change over two and a -log10 p-value over one are labelled. SRFBP1 is a nucleolar protein required for ribosome biogenesis with no known function in mRNA biogenesis. Data (*n* = 3) were analysed using a two-sided Welch t-test with FDR correction for multiple testing. **i**, RNA export tethering assay, related to Fig. 4d with λN-BFP, λN-NXF1 and λN-UAP56 from Fig. 4d shown for comparison. We split an mCherry open reading frame in two halves (exon 1, exon 2) by inserting an intron containing ten BoxB RNA aptamers, an IRES, and a GFP open reading frame (top). λN-tagged proteins are transiently expressed and bind the reporter RNA through the RNA aptamers. Export of the reporter RNA allows GFP production, which is quantified through Fluorescence-activated cell sorting (FACS). While the direct UAP56-tethering to the reporter RNA bypasses preceding pathway steps and does not replicate UAP56's native mRNA binding dynamics, this assay allowed us to investigate the export-promoting features of UAP56. Mutations interfering with UAP56's ATPase activity (D196A, E197A, 'AACD') do not substantially alter the export promoting effect, while mutating key residues in the UAP56 NTD (Y13S or L10S, L11S, D12K, Y13S, '10-SSKS-13') leads to a reduced effect. As additional controls, we also tethered the THO complex via THOC1 or THOC5, or SARNP, which showed no effect, as well as ALYREF, which modestly stimulated reporter pre-mRNA export. Boxplots include the median (centre), interquartile range (25th–75th percentiles) as the height of the box, and whiskers extending to the 5th and 95th percentiles. *n* = min. 40,000 cells examined over three independent experiments.
**j**, λN-UAP56 and λN-UAP56 ΔNTD are expressed and imported to the nucleus at similar levels. Two replicates of λN-UAP56 or λN-UAP56 ΔNTD expressing K562 cells are fractionated into nucleus and cytoplasm, proteins separated by SDS-PAGE and analysed by western blotting, probing for UAP56 (top) and Histone H3 (bottom, fractionation control) on the same membrane; the full membrane is shown. Both constructs are expressed to similar levels (as judged from the whole cell extract, compared to endogenous UAP56 to showcase equal loading). Equal amounts of each construct are important into the nucleus, and nuclear levels of the λN-tagged proteins are comparable to levels of endogenous nuclear UAP56.

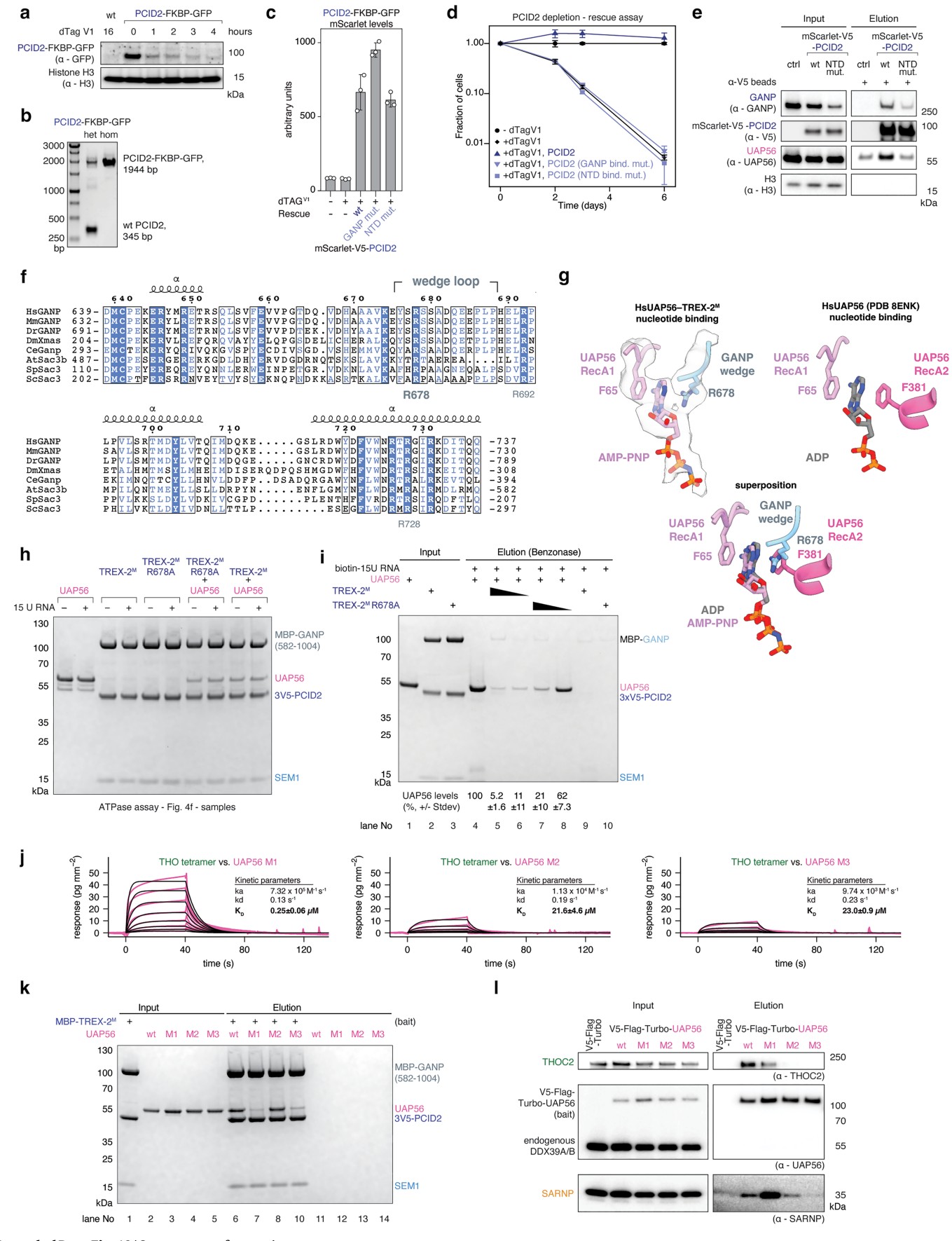

**Extended Data Fig. 10 |** See next page for caption.

**Extended Data Fig. 10 | In vivo probing of PCID2 interfaces, GANP wedge loop analysis, controls for UAP56 ATPase assays and analysis of UAP56 THO/TREX-2 binding mutants. a-b,** Analysis of the PCID2-FKBP-GFP cell line, showing by western blot that tagged PCID2 is efficiently depleted upon the addition of dTAG-V1 **(a)** and by PCR that PCID2 is tagged homozygously **(b)**. het, heterozygous population; hom, homozygous clonal cell line. **c-d,** Human K562 cell growth competition assays to probe the interaction of the UAP56 NTD with the TREX-2 subunit PCID2 and between PCID2 and the TREX-2 subunit GANP. Wild type or mutant mScarlet-V5-PCID2 is ectopically expressed from a doxycycline (dox)-inducible promoter in human K562 cells, where endogenous PCID2 is tagged with the dTAG-degron (FKBP12$^{F36V}$). Addition of the dTAG-V1 compound leads to the rapid depletion of endogenous PCID2. Addition of dox initiates the ectopic expression of the respective rescue construct, as monitored by mScarlet levels two days after induction **(c)**. Dox and dTAG-V1 or control treated cells are mixed at a 1:1 ratio with wild type K562 cells at day 0. Cell growth is monitored by FACS (**d**, see Methods for details); error bars, mean ± s.d. from $n$ = 3 independent samples. **e**, PCID2 wild type or mutant immunoprecipitation after the rapid depletion of endogenous dTAG-PCID2 using the dTAG-V1 compound. Human K562 cells, as used in panel **c,d**, were harvested 8 h after the treatment with dox and dTAG-V1 or dox and DMSO. Cells were fractionated into nuclei and cytoplasm, and PCID2 was immunoprecipitated using anti-V5 magnetic beads. After washes, bead-bound proteins were eluted by boiling in SDS-PAGE and western blotted for TREX-2 (GANP, PCID2), UAP56, and Histone H3. GANP protein levels are reduced upon the ectopic expression of the PCID2 NTD-binding mutant for unclear reasons. Input samples are shown (left). **f**, Multiple sequence alignment of the GANP wedge loop region of GANP proteins from *H. sapiens* (Hs, Uniprot ID O60318), *M. musculus* (Mm, Q9WUU9), *D. rerio* (Dr, F1Q712), *D. melanogaster* (Dm, Q9U3V9), *C. elegans* (Ce, Q19643*), A. thaliana* (At, F4JAU2), *S. pombe* (Sp, O74889) and *S. cerevisiae* (Sc, P46674), coloured by conservation (blue background, invariant residue), and with secondary structure elements depicted on top. Highlighted are the wedge loop with the key and invariant residue R678 (numbering according to human GANP), as well as the invariant residues R692 and R728 which might stabilize the wedge loop and UAP56 RecA1 interactions and are implicated in gene

gating[51]. **g**, The GANP wedge binding mimics the ATP-binding pocket in clamped UAP56. Shown are the UAP56-bound nucleotide and nucleotide base stacking residues in sticks representation, coloured by heteroatom, and with UAP56 in pink, GANP in light blue and the nucleotide in pink for the UAP56–TREX-2$^M$ structure (left, superimposed on the cryo-EM density), and in grey for the clamped human UAP56 structure (right, PDB ID 8ENK)[33]. In the UAP56–TREX-2$^M$ structure the adenine moiety of the non-hydrolyzable ATP analogue forms stacking interactions with F65 in the UAP56 RecA1 lobe and the GANP wedge residue R678 (left). In clamped UAP56, the adenine base of bound ADP stacks with F65 of UAP56's RecA1 lobe and F381 of the RecA2 lobe (right). Superposition of both structures reveals that the GANP wedge residue R678 substitutes the RecA2 lobe residue F381 (bottom). **h**, Protein samples used in the UAP56 ATPase assay in Fig. 4f. An aliquot of each reaction was separated by SDS-PAGE and visualized by Coomassie-staining. **i**, UAP56–RNA unclamping assay. UAP56 is incubated with bead-immobilized 15 nucleotide poly-Uridine RNA (15U RNA) in the presence of ATP, to form UAP56–ADP-P$_i$–RNA complexes. After removing unbound UAP56 and excess ATP, TREX-2$^M$ complexes containing GANP wild type or the 'wedge' mutant (R678A) are added. UAP56–ADP-P$_i$–RNA complexes remaining on the beads after incubation are eluted by digestion of the RNA (benzonase) and analysed by Coomassie-stained SDS-PAGE. **j**, GCI-derived binding kinetics of the THO tetramer (immobilized) probed with wildtype or UAP56 mutants M1, M2, or M3. Sensogram (pink line), the fitted model (black), and a summary table of the binding kinetics are shown. **k**, In vitro pulldown assay probing the interaction of TREX-2$^M$ with UAP56 mutants M1, M2 or M3. TREX-2$^M$ complex (MBP-GANP residues 582-1004, PCID2, SEM1), immobilized on amylose beads through the MBP on the GANP, is incubated with wildtype or mutant UAP56. Bound proteins are eluted with a maltose containing buffer and visualized in Coomassie stained SDS-PAGE gels. **l**, Western blot analysis of the UAP56 wildtype and mutant IP samples that were analysed by mass spectrometry in Fig. 5b. GANP is not shown as it was not detected in the western blot. wt, wildtype; M1, UAP56 TREX-2 binding mutant; M2, UAP56 THO-binding mutant; M3, combined UAP56 THO- & TREX-2-binding mutant. See Extended Data Fig. 11a for a detailed description of the experiment outcomes.

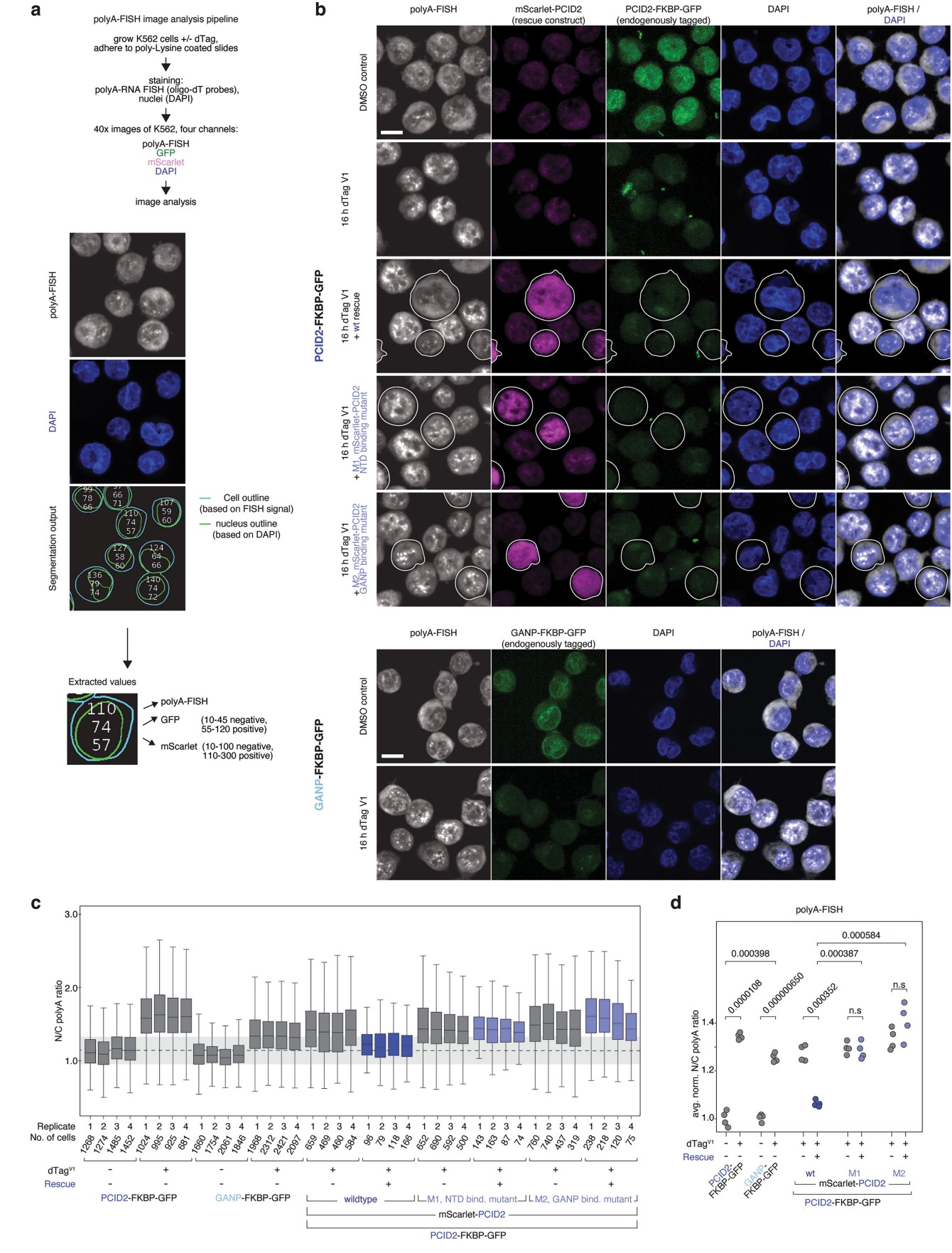

**Extended Data Fig. 11** | See next page for caption.

**Extended Data Fig. 11 | Poly(A) RNA FISH experiments and analysis.**
**a**, Poly(A) RNA FISH sample preparation and image analysis workflow, with a representative image shown together with a segmented image used for analysis. **b**, Representative images for all analysed samples. Shown are the poly(A) RNA FISH channel, the mScarlet channel (indicating the expression of mScarlet-PCID2 rescue constructs), the GFP channel (indicating the effect of dTag V1 depletion of PCID2-FKBP-GFP (top) or GANP-FKBP-GFP (bottom)), the DAPI channel showing nuclei, and an overlay of the poly(A) RNA FISH and DAPI channels, illustrating the nuclear/cytoplasmic distribution of the poly(A) RNA FISH signal. Cells expressing mScarlet are highlighted with a white outline. Scale bar = 10 μm. **c**, Boxplots depicting the results of the poly(A) RNA FISH image analysis for each replicate. We obtained four replicates per condition, each with >70 cells quantified. For experiments, where the mScarlet-PCID2 rescue construct is expressed upon the depletion of the endogenous PCID2-FKBP-GFP protein, cells were classified into mScarlet positive or negative, to quantify '−' and '+' rescue construct expression conditions within the same field of view. Boxplots include the median (centre), interquartile range (25th–75th percentiles) as the height of the box, and whiskers extending to the 10th and 90th percentiles. **d**, Results of the poly(A) RNA FISH quantification, showing data from Fig. 4e alongside additional controls. Shown are the averages of each replicate experiment, with p-values indicating the significance of condition comparisons using two-sided Welch t-tests with FDR correction for multiple testing.

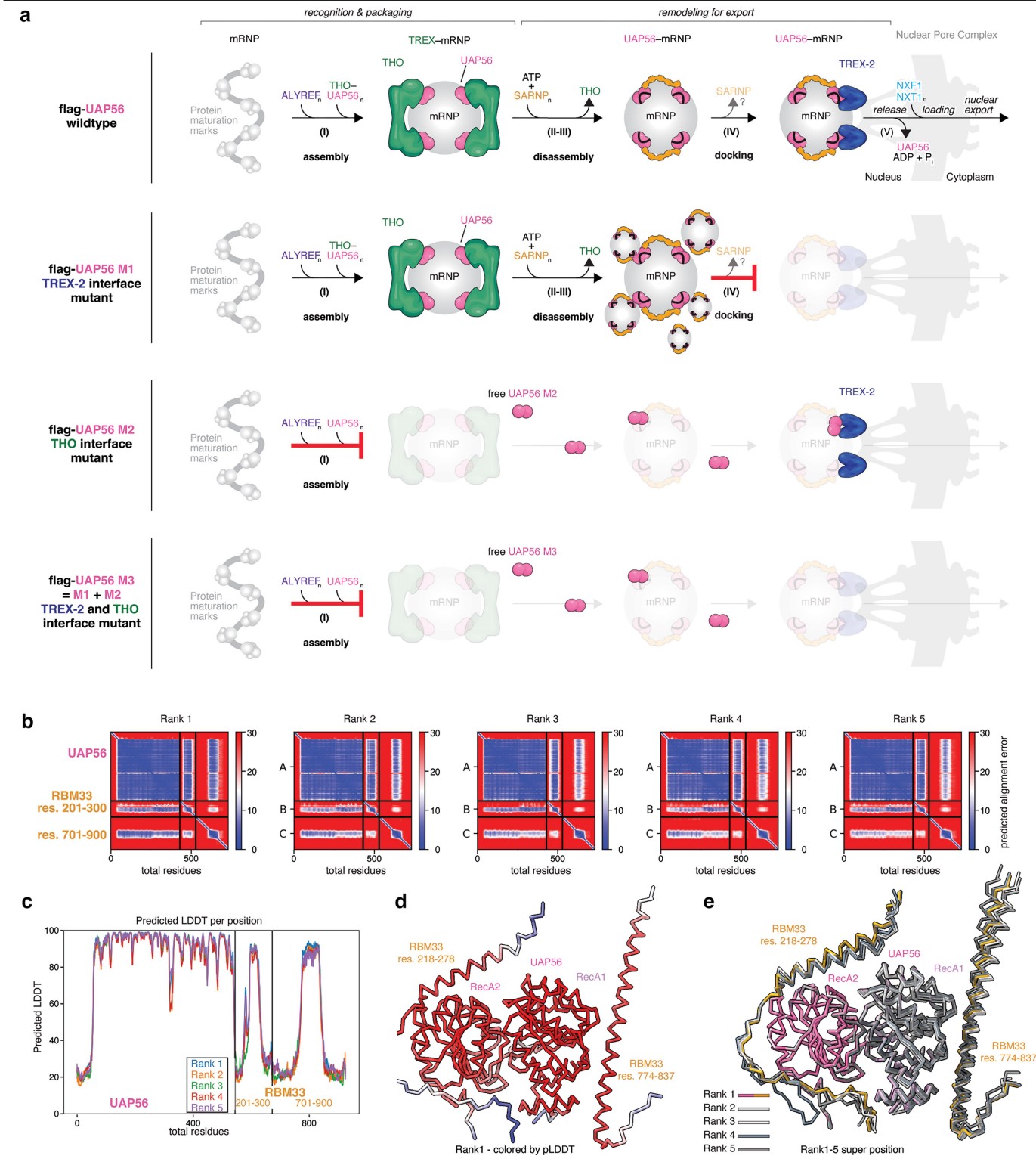

**Extended Data Fig. 12** | See next page for caption.

**Extended Data Fig. 12 | Predicted effects of UAP56 mutants and AlphaFold2 prediction of a UAP56–RBM33 complex. a**, Predicted effects of mutations M1-3 in UAP56 on the general model for mRNA nuclear export (related to Fig. 5b). When binding of UAP56 to TREX-2 is prevented (M1), UAP56–mRNP docking at the NPC and release of UAP56 from mRNPs is impaired. This would result in the accumulation of UAP56 M1 on mRNPs together with SARNP. Because the experiment is performed in a wildtype background, with the levels of the ectopically expressed mutant UAP56 being lower than endogenous UAP56 and finite, this would lead to a mild reduction of THO complex levels in the UAP56 M1 protein interactome: the more UAP56 M1 accumulates on mRNPs, the less UAP56 M1 would be available to bind the THO complex. For UAP56 M2, where THO binding is impaired, the pathway is blocked at step (I), which would lead to the depletion of the THO complex, mRNP proteins, and of SARNP in the UAP56 M2 protein interactome. This mutant would lead to a relatively high concentration of free UAP56 M2 protein, which could allow UAP56 M2 to engage TREX-2 with however no functional consequence. When UAP56 M1 and M2 mutations are combined in M3, the mRNA export pathway would again be blocked at step (I), but now the free UAP56 M3 protein would also be unable to engage TREX-2. Unlike in M1, mRNP proteins would not accumulate with UAP56 in M3, because the binding of UAP56 to mRNPs is THO-dependent. **b-e**, Diagnostic plots for the AlphaFold2 Multimer prediction of RBM33–UAP56. RBM33, which aids the export of intronless mRNAs[67], is predicted to bind UAP56 via two novel peptides. In this prediction, one of the two RBM33 peptides would bind to the UAP56 UCM-binding site, although it is distinct from the UCM. Shown are **b**, the PAE plot, **c**, the pLDDT plot, **d**, the structure of the top ranked model in Cα trace coloured by pLDDT (shown are only the ordered and interacting elements: UAP56 RecA1 and RecA2, residues 40-428; RBM33 interacting regions 1 and 2, residues 218-278 and 774-837), and **e**, a superposition of the structures of all five models, rank 1-5, as in **d**, but coloured for rank 1 with UAP56 in pink and RBM33 in yellow, and in shades of grey for rank 2-5.

## Extended Data Table 1 | Cryo-EM data collection and refinement statistics

**A**

| | THO–UAP56 tetramer (Map A) | THO–UAP56–ALYREF N-UBM (Map B) | TREX2[M] (Map C) | UAP56–TREX2[M] (Map D) | UAP56 RecA2 –TREX2[M] (Map E) |
|---|---|---|---|---|---|
| **Data collection** | | | | | |
| Particles | 204,147 | 204,147 | 57,499 | 37,692 | 7,741 |
| Pixel Size (Å) | 0.84 | 0.84 | 0.749 | 0.749 | 0.749 |
| Defocus range (μm) | −0.75 to −1.25 | −0.75 to −1.25 | −1.0 to −2.5 | −1.0 to −2.5 | −1.0 to −2.5 |
| Voltage (kV) | 300 | 300 | 300 | 300 | 300 |
| Electron dose (e⁻/Å²) | 60 | 60 | 50 | 50 | 50 |
| **Reconstruction (cryoSPARC)** | | | | | |
| Resolution (Å) | 5.9 | 4.1 | 3.5 | 3.5 | 4.2 |
| Map sharpening B-factor (Å²) | −409 | −178 | −128 | −108 | −98 |
| **Model composition** | | | | | |
| Non-hydrogen atoms | | 14,961 | 6,042 | 7,984 | |
| Protein residues | | 1,622 | 754 | 1,012 | |
| Nucleotide residues | | 0 | 0 | 0 | |
| Ligands | | 0 | 0 | 1 | |
| **Refinement (PHENIX)** | | | | | |
| Map CC (around atoms) | | 0.68 | 0.73 | 0.71 | |
| B-factor (overall) | | 146 | 125 | 126 | |
| **Rms deviations** | | | | | |
| Bond lengths (Å) | | 0.01 | 0.01 | 0.01 | |
| Bond angles (°) | | 1.62 | 1.38 | 1.54 | |
| **Validation** | | | | | |
| Molprobity score | | 1.76 | 1.68 | 1.85 | |
| All-atom clashscore | | 5.63 | 4.86 | 4.72 | |
| Rotamer outliers (%) | | 0.06 | 0.3 | 1.57 | |
| C-beta deviations | | 0.1 | 0.1 | 0.2 | |
| **Ramachandran plot** | | | | | |
| Outliers (%) | | 0.11 | 0.41 | 0.71 | |
| Allowed (%) | | 6.97 | 6.10 | 6.57 | |
| Favoured (%) | | 92.92 | 93.50 | 92.73 | |
| **Data Deposition** | | | | | |
| EMDB ID | EMD-18979 | EMD-18980 | EMD-18977 | EMD-18978 | EMD-18981 |
| PDB ID | | 8R7L | 8R7J | 8R7K | |

**B**

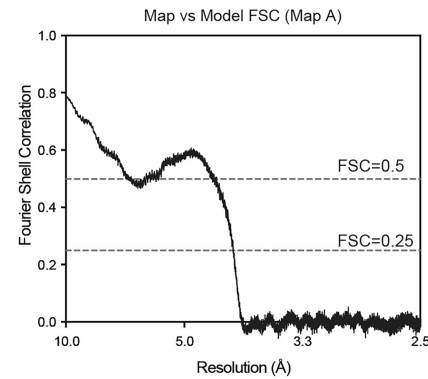
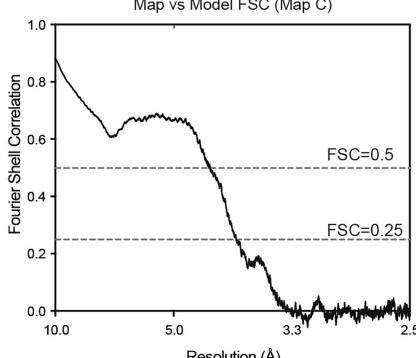
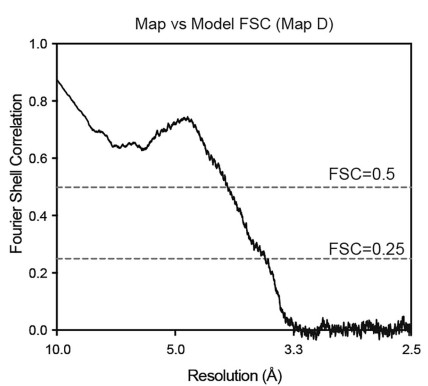

**a**, Cryo-EM data collection and refinement statistics for the THO–UAP56–ALYREF-N-UBM complex as well as TREX-2[M] and UAP56–TREX-2[M]

**b**, Fourier shell correlations between the cryo-EM densities of the THO–UAP56–ALYREF-N-UBM complex (Map B, left), TREX-2[M] (Map C, middle) and UAP56–TREX-2[M] (Map D, right) with the respective refined coordinate models using phenix.mtriage.

# Reporting Summary

Nature Research wishes to improve the reproducibility of the work that we publish. This form provides structure for consistency and transparency in reporting. For further information on Nature Research policies, see our Editorial Policies and the Editorial Policy Checklist.

## Statistics

For all statistical analyses, confirm that the following items are present in the figure legend, table legend, main text, or Methods section.

| n/a | Confirmed | |
|---|---|---|
| ☐ | ☒ | The exact sample size (*n*) for each experimental group/condition, given as a discrete number and unit of measurement |
| ☒ | ☐ | A statement on whether measurements were taken from distinct samples or whether the same sample was measured repeatedly |
| ☐ | ☒ | The statistical test(s) used AND whether they are one- or two-sided<br>*Only common tests should be described solely by name; describe more complex techniques in the Methods section.* |
| ☒ | ☐ | A description of all covariates tested |
| ☒ | ☐ | A description of any assumptions or corrections, such as tests of normality and adjustment for multiple comparisons |
| ☐ | ☒ | A full description of the statistical parameters including central tendency (e.g. means) or other basic estimates (e.g. regression coefficient) AND variation (e.g. standard deviation) or associated estimates of uncertainty (e.g. confidence intervals) |
| ☐ | ☒ | For null hypothesis testing, the test statistic (e.g. *F*, *t*, *r*) with confidence intervals, effect sizes, degrees of freedom and *P* value noted<br>*Give P values as exact values whenever suitable.* |
| ☒ | ☐ | For Bayesian analysis, information on the choice of priors and Markov chain Monte Carlo settings |
| ☒ | ☐ | For hierarchical and complex designs, identification of the appropriate level for tests and full reporting of outcomes |
| ☒ | ☐ | Estimates of effect sizes (e.g. Cohen's *d*, Pearson's *r*), indicating how they were calculated |

*Our web collection on statistics for biologists contains articles on many of the points above.*

## Software and code

Policy information about availability of computer code

| Data collection | Cryo-EM data were collected with EPU 3. |
|---|---|
| Data analysis | CryoEM data were analysed with WARP 1, RELION 3 and 4, cryoSPARC 4, Coot 0.9, Phenix 1.2, ISOLDE 1.6 and ChimeraX 1.7-1.10. All other data were analysed using FIJI with ImageJ 2.9, GraphPad Prism8 and R 4.0.<br>HT-Colabfold is free open-source software (MIT) and available at https://gitlab.com/BrenneckeLab/ht-colabfold. |

For manuscripts utilizing custom algorithms or software that are central to the research but not yet described in published literature, software must be made available to editors and reviewers. We strongly encourage code deposition in a community repository (e.g. GitHub). See the Nature Research guidelines for submitting code & software for further information.

## Data

Policy information about availability of data

All manuscripts must include a data availability statement. This statement should provide the following information, where applicable:

- Accession codes, unique identifiers, or web links for publicly available datasets
- A list of figures that have associated raw data
- A description of any restrictions on data availability

Three-dimensional cryo-EM density maps of the TREX–EJC–ALYREF complex, TREX-2M and UAP56–TREX-2M have been deposited into the Electron Microscopy Data Bank under the accession numbers EMD-18980 (Map-A) and EMD-18979 (Map-B), EMD-18977 (Map-C), EMD-18978 (Map-D) and EMD-18981 (Map-E) respectively. The coordinate files of the TREX–EJC–ALYREF, TREX-2M and UAP56–TREX-2M have been deposited into the Protein Data Bank under the accession numbers 8R7L, 8R7J, and 8R7K. The coordinate file of the TREX–mRNA complex was updated in the Protein Data Bank under the accession number 7ZNK. Proteomics data have been deposited to the ProteomeXchange Consortium via the PRIDE101 partner repository under the accession number PXD069399.

# Field-specific reporting

Please select the one below that is the best fit for your research. If you are not sure, read the appropriate sections before making your selection.

[X] Life sciences [ ] Behavioural & social sciences [ ] Ecological, evolutionary & environmental sciences

For a reference copy of the document with all sections, see nature.com/documents/nr-reporting-summary-flat.pdf

# Life sciences study design

All studies must disclose on these points even when the disclosure is negative.

| | |
|---|---|
| Sample size | CryoEM data set sizes were chosen to achieve the desired resolution. All other sample sizes were chosen based on pilot experiments to detect biologically meaningful differences with statistical robustness. |
| Data exclusions | No data were excluded. |
| Replication | All experiments, except cryoEM data set collection and processing, were performed at least three times independently with comparable outcome. Sample preparation and the analysis were highly reproducible. |
| Randomization | For 3D refinement the cryo-EM data were split randomly into two halves for gold-standard FSC determination. All other experiments were not randomized but do contain appropriate controls. |
| Blinding | For the export tethering and RNA FISH experiments sample identities were revealed only after the analysis finished. For all other experiments Blinding is impractical. However, to minimize potential bias samples are always analyzed with the same pipeline and/or parameters for all samples and controls. |

# Reporting for specific materials, systems and methods

We require information from authors about some types of materials, experimental systems and methods used in many studies. Here, indicate whether each material, system or method listed is relevant to your study. If you are not sure if a list item applies to your research, read the appropriate section before selecting a response.

## Materials & experimental systems

| n/a | Involved in the study |
|---|---|
| [ ] | [X] Antibodies |
| [ ] | [X] Eukaryotic cell lines |
| [X] | [ ] Palaeontology and archaeology |
| [X] | [ ] Animals and other organisms |
| [X] | [ ] Human research participants |
| [X] | [ ] Clinical data |
| [X] | [ ] Dual use research of concern |

## Methods

| n/a | Involved in the study |
|---|---|
| [X] | [ ] ChIP-seq |
| [X] | [ ] Flow cytometry |
| [X] | [ ] MRI-based neuroimaging |

## Antibodies

| | |
|---|---|
| Antibodies used | Primary: anti-V5 antibody (Thermo Scientific, 37-7500, 2 µg ml-1 for GCI), anti-THOC2 (ab129485; Abcam; 1:1000), anti-UAP56 (ab181059; Abcam; 1:1000), anti-histone H3 (ab1791; Abcam; 1:1000), anti-UAP56 antibody (Cell Signaling Technology #47258; 1:1000), anti-GANP (ab113295; Abcam; 1:1000), anti-SARNP (PA5-56585; Invitrogen; 1:1000), anti-ALYREF (ab202894; Abcam; 1:1000), anti-V5 (2F11F7; Invitrogen; 1:1000), anti-Histone H3 (17168-1-AP; Proteintech; 1:1000), anti-GFP (CAS A11122, ThermoFisher, 1:1000), anti-EIF4A3 (AB180519, Abcam, 1:1000), anti-Histone H3-HRP (5192S, Cell Signalling Technologies; 1:1000) . Secondary: goat-anti-rabbit antibody coupled to HRP (Thermo Scientific 31466, diluted 1:5000), goat-anti-mouse antibody coupled to HRP (Thermo Scientific G-21040, diluted 1:5000) |
| Validation | All antibodies are validated by the manufacturer. In addition, anti-THOC, anti-SARNP, anti-UAP56, anti-GANP, anti-GFP and anti-V5 are validated assessing the detection of tagged proetins and/or the depletion of the respective protein in K562 cell lysate. |

## Eukaryotic cell lines

Policy information about cell lines

| | |
|---|---|
| Cell line source(s) | Leukemia cell line K562 (source: ATCC) |
| Authentication | The cell line was authenticated by short tandem repeat analysis (see Muhar et al. 2018, Science, DOI: 10.1126/science.aao2793) |

| Mycoplasma contamination | The cell line was confirmed to be negative for mycoplasma. |
| Commonly misidentified lines (See ICLAC register) | No commonly misidentified cell line was used in this study. |

