## [Peer Review File · Nature]

An ATP-gated molecular switch orchestrates human messenger RNA export

Corresponding Author: Dr Clemens Plaschka

Version 0:

Reviewer comments:

Referee #1

(Remarks to the Author)

In this study, Plaschka and colleagues report the central role of UAP56 in orchestrating RNA nuclear export. Based mainly on structural studies of UAP56 with other proteins, they suggest that apart from its known role in packaging into mRNP along with other TREX components, UAP56 also forms two separate complexes with SARNP, a previously identified TREX component, and with GANP and PCID2, central components of the TREX2 complex primarily located at the nuclear pore. Structural data indicate that UAP56 interacts with ALYREF-THO, SARNP, and GANP/PCID2 in a mutually exclusive manner. Consequently, they propose that UAP56 acts as a central molecular switch directing mRNPs through the TREX and TREX2 complexes.

The manuscript is generally well-written and easy to follow, presenting novel findings on mRNA export and potentially introducing an interesting concept about mRNA export. However, it is important to note that most of the conclusions drawn in the study are based on in vitro data and prediction/artificial experiment design. Additionally, the major conclusion suggesting that UAP56 forms into different complexes stepwise requires further direct evidence.

--The authors proposed that SARNP replaces THO to facilitate THO disassembly from the mRNP. However, this conclusion does not stand with the evidence presented. Recent structural studies have shown that SARNP forms a complex with UAP56, suggesting that SARNP and THO interact with UAP56 in a mutually exclusive manner. The authors confirmed this competition between SARNP and THO for UAP56 interaction using recombinant proteins. However, no evidence supports the proposal that SARNP is recruited to UAP56 following ALYREF/THO. It remains possible that SARNP could be recruited to mRNA ahead of THO or form an independent complex with UAP56-ALYREF-THO. Based on the published and A-fold2-mediated predicted structure of UAP56-SARNP, they speculate that SARNP and ALYREF could interact with UAP56 simultaneously. The authors did not examine this possibility, but instead, took it as reality, and made the SARNP-UCM-ALYREF-UBM fused peptide to study the competition with THO. In Nat Commun. 2024 Jan 15;15(1):455 and Mol Biol Cell. 2010 Aug 15;21(16):2953-65., SARNP has been shown to form a separate complex with UAP56 in the absence of ATP. Thus, the authors need to provide direct evidence whether SARNP can form a complex with UAP56 and ALYREF using purified proteins and endogenous mRNP. Otherwise, the usage of the SARNP-UCM-ALYREF-UBM peptide does not make sense and could lead to inappropriate conclusions.

--In Dufu et al., Genes Dev 2010, THO and SARNP interacted with each other as efficiently as they interacted with UAP56 in the presence of ATP. The authors should test whether depletion of SARNP affects UAP56-THO interaction.

--The interactions of UAP56 with TREX-2 proteins GANP and Sac3 are the only new complex identified in this study and present the most significant findings. Again, in the complex, the authors used the fused SARNP-UCM-ALYREF-UBM that brings artifact. The authors should at least present evidence that GANP/PCID2 could form a complex with UAP56 and ALYREF (with or without SARNP) without artificially fusing ALYREF with SARNP. These interactions of endogenous proteins should be validated using immunoprecipitation from cells followed by western blot analysis. The authors propose that the TREX-2-UAP56 interaction is essential for competing with SARNP. Direct assessment of RNA export using fluorescence in situ hybridization (FISH) would be beneficial.

--The authors proposed that TREX-2 stimulates the ATPase activity of UAP56 and thus releases UAP56 from the mRNP. Does depletion of TREX2 or mutation of the TREX2-UAP56 interaction affect UAP56 release from the mRNP? It is known that RNA, ALYREF, and THO together stimulate the ATPase activity. In support of this, UAP56 does not clamp on RNA in the presence of ALYREF and THO. What leads the authors to believe that only after TREX2 binding, the ATPase activity of

UAP56 is stimulated? Also, de-clamp of UAP56 on RNA does not mean its disassembly from the RNA. In the presence of ALYREF and THO, UAP56 does not clamp on RNA, but still associates with the mRNP.

Referee #2

(Remarks to the Author)

Before export from their nuclear site of production into the cytoplasm, mRNAs are packaged into particles (mRNP) that undergo several maturation and remodelling steps. This packaging serves as a quality control step to license correctly processed mRNAs and offers also a mode of control of gene expression. Although research on this highly conserved process has been going on for decades, many basic mechanistic questions remain unanswered. In their current manuscript, Hohmann, Brennecke, Plaschka and colleagues study some of the mRNP remodelling steps that precede mRNA export, using a combination of biochemical assays and structural studies. The focus lies on the RNA dependent ATPase UAP56, which forms a platform for sequential, mutually exclusive binding events of mRNA export factors. The study is of interest for researchers working on RNA biology and gene regulation.

The paper contains overall high quality cryoEM and biochemical data, the text is well structured and figures are intuitively accessible. However, I have three main points of criticism:

1. Going through the available literature, I was left with the impression that prior knowledge was not accurately reflected possibly with the wish to increase "novelty". This affects particularly the first half of the manuscript.

Examples of previously published points, that appear presented as if new:

A structure of yeast UAP56 bound by yeast ALYREF was published by Ren et al. eLife 2017. This paper also already showed that ALYREF binding to UAP56 increases RNA affinity.

A structure of UAP56 bound by yeast SARNP was published by Xie et al. Cell Rep. 2023. This paper also already showed that SARNP binding to UAP56 increases RNA affinity, that SARNP has multiple UAP56 binding motifs in all eukaryotes (they called them DIMs) and that SARNP binding to UAP56 had to occur after THO complex release because their binding sites are mutually exclusive.

2. The biological relevance of the structures and resulting models is not tested. The only cell based assay is a tethering export assay, where different proteins are artificially bound to an mRNA via stem-loop-aptamer links. On top of this, the assay is problematic for the proteins studied here, because UAP56 normally has to be released from mRNAs before export. This can no longer occur when UAP56 is tethered, which draws the validity of this whole assay into question (and the authors actually make a big point about this putative function of TREX-2M on UAP56 release from mRNAs). Originally this assay was conceived to study the effect of splicing/EJC deposition on export, but it is not a good tool for the factors studied here.

Furthermore, none of the interactions are tested in cells, neither are any mutants shown to affect interactions in cells. Literature data indicates that these are relevant points. In a minimalized in vitro system of reconstituted complexes certain effects might appear clear cut, but in the more complex environment of the nucleus, factors might be in competition with others, and their effects may be much less important. E.g. Xie et al. Cell Rep. 2023 found that only a minority of mRNAs (i.e. ~200 vs. 12,000) were less well (1.5x) exported upon SARNP depletion. In contrast, from the model presented in the current manuscript one would expect SARNP to be crucially important for export of all mRNAs.

3. a) Some aspects of the biochemical assays are problematic. b) Also some conclusions are drawn from structures but are not supported by any experimental evidence.

a) In the first half of the paper several binding and biochemical assays depend on using a fusion peptide of the UAP56 binding-motifs of SARNP and ALYREF. As a reason for this artificial method the low affinities of the individual peptides is given by the authors. However, in cells these low affinities would have to suffice if the proposed hand-over models of the authors are true. The authors might argue that in the cellular environment there is multiplicity in all of the interactions which provides avidity effects (THO tetramer, SARNP with multiple binding motifs, ALYREF with 2 UAP56 binding motifs). Then why is this never tested? Instead of using the fusion peptide full length SARNP could be used in the competition assays. Longer RNAs can be in vitro transcribed to allow for multiple copies of UAP56/ALYREF/THO to bind to the same molecule. The danger is that by using the fusion peptide an artificial enrichment is achieved that never occurs in cells, and that cells require actually additional/other factors to carry out TREX disassembly. This danger is increased by the lack of cellular assays to verify the mechanistic models proposed here.

b) The main advance of the paper comes in its second half, where the previously unknown binding of UAP56 to TREX-2M is identified and structurally characterized by cryoEM. This structure suggests that TREX-2M binding to UAP56 leads to mRNA release. Unfortunately, this interesting hypothesis is not experimentally verified at all, neither in vitro nor in cellular assays. Does TREX-2M binding actually lead to release of RNA in vitro? Is UAP56-TREX-2M complex affinity for RNA lower than UAP56 alone? Are UAP56-deltaN mutants stuck on mRNAs in the nucleus of cells?

Along a similar line, it is unlikely that the GANP "wedge" with its small contact can release the RNA clamp of UAP56 directly. It seems more likely that ATP hydrolysis has to occur first to release the RecA1-2 clamp and then the GANP "wedge" can insert itself bridged by a new ATP molecule, thereby preventing renewed RNA binding. This could easily be tested in vitro.

Other points:

4. The title is rather vague and not very descriptive of the paper content.

5. Can C-UBM of ALYREF still bind on UAP56 in presence of TREX-2M? This is not visible from the structure figures.

(Remarks to the Author)

The manuscript of Hohmann et al. "A molecular switch orchestrates the export of human messenger RNA" provides essential and novel insights into the mRNA export pathway, specifically by which factor mature mRNPs are recognised as such and then handed-over to the NPC via TREX-2. Generally, this manuscript builds on and discusses appropriately previous findings and reconsolidates them in light of the discoveries made here. The dead-box RNA helicase UAP56 turns out to be a molecular switch, which interacts with ALYREF and recruits SARNP to release TREX and allow interaction with the NPC-coupled TREX-2. The insightful final model on mRNP export is based on new and re-analysed cryo-EM data, mass-spectrometry-based proteomics to identify SARNP as additional modulator, pull-down assays, grating-coupled interferometry, and additional biochemical assays. While I think that the provided insights merit publication in Nature, three major concerns need to be addressed prior acceptance. Firstly, although it might appear as a detail I find the usage of the word licensing in the abstract and later (at altogether four instances) not ideal. To me this implies a certain quality control process. However, I think the work does not describe exactly how UAP56 can distinguish between a correctly assembled mRNP and an immature or improper mRNP. In this regard, it would be also good, if the authors could shortly describe the limitations of the study at the end of the manuscript and refer to literature about what is known about these "licensing" steps and how this could be linked to their findings. Secondly, the experiments performed to postulate cooperativity between ALYREF and SARNP are not ideal or enough to be convincing. In general the data behind Figure 2 and related Ext. Data Figs would need to be extended and cross-validated (See below for more details) especially with regards to statistics. Thirdly, the language needs to be improved and the figure arrangement and referrals. The current arrangement didn't make it easy to review this manuscript. Especially the faulty figure referrals were confusing.

Major concerns:

Concerns around Figure 2 (mostly) but also Figure 1 and related Ext. Data Figs: Statistics for all GCI measurements are missing. Are there no replicates? Especially regarding the small differences in most points made. I think this is essential to estimate the validity of the assumptions made. I am not familiar with the technique and it is hard to judge whether this method is accurate enough to allow for only one measurement to make such assumptions. It would be ideal to at least in some instances use an alternative biophysical method to cross-validate.

In Ext. Data Fig. 1f, why wasn't C-UBM and N-UBM not used together in an additional experiment?

In experiments of Figure 1g, I wonder if it would be possible to do the comparison with ATP or RNA only.

The experimental difference between Figure 1g and Ext. Data Fig. 3 is not really clear. Recombinant vs. Non-recombinant? Ext. Data Fig. 3b should be done with only ATP and only RNA as additional controls.

Experiments with SARNPs UCM motifs leave too many questions open, which could be easily answered in this work. In line 164, the authors speculate that multivalency has evolved here to disassemble multivalent tetrameric complexes. This could be tested. Also, they could just increase binding to UAP56, which also could be tested by mutations of residues in UCM2-5 corresponding to R106 in UCM1 (one at a time and multiple at once). Actually in Ext. Data Fig. 4h the authors test UCM4, but this is not mentioned in main text (If I haven't missed it).

Concerning cooperativity: Why has C-UBM been ignored with respect to cooperativity to UCMs? Could be allosteric? I understand that there are challenges with low-affinity complexes and that fusing two systems helps. However, this is also very artificial. The authors should at least show an experiment where they add both, N-UBM (possibly also C-UBM) and UCMs as separate constructs (not fused to UAP56) (UCM1, the other UCMs and UCM1-5 or derivatives). These are all important control experiments. Especially since differences in affinities in Figure 2h do not infer "cooperativity" but possibly synergistic binding at best. Especially here statistics (replicates) are needed. A two-fold increase in affinity is not cooperative. Controls should still be made with unfused UBM/UCMs. Ext. Data Fig. 5a does not help as it is rather ambiguous, while lane 9 has a clear intensity increase compared to lane 7, lane 12 vs. lane 11 does not. Or is this rather seen in direct context to Figure 2h, where RNA binding is tested (missing in Ext. Data Fig. 5?). If so, then it is not clear. Why not using GCI for experiments in Ext. Data Fig. 5a? Figure 2f also lacks the comparison with both peptides (UBM (N and C?) and UCM) added non-fused. Also, should the band for MBP-UAP56 not keep the same intensity over all lanes? It seems to also decrease with the THOC bands. All this leaves me wondering about the role of the SAP domain in SARNP and the RRM domain in ALYREF. For the latter I guess more is known but it could be mentioned somewhere. However, both could further increase cooperative effects, possibly binding RNA and reaching into the mRNP granule?

Minor concerns:

Why not call UAP56 an RNA helicase, but an ATPase. RNA helicases have ATPase function, but I would assume that UAP56 also has RNA helicase function in this complex at a certain step of this process? (Which would be interesting to follow-up upon or discuss shortly).

2. The "newly-revised" in caption to Figure 1b is confusing at this stage. Leaves the reader wondering. It should be mentioned somehow in the main text before referring to this figure to make this clear.

3. The EM density for ALYREF N-UBM should be shown in Ext. Fig. 1 to illustrate why it has been missed and how good this helix fits as done for other structural elements. Compare old vs. new data. I think Ext. Data Fig. 2g shows something along the lines but only for the new data?

4. The sequence alignment of ALYREF N- and C- UBM should be added to Figure 1c (Enough space left) to better illustrate the point made in lines 65-70.

5. There is enough space to show the actual residue contacts between C-UBM-UAP56 and N-UBM-UAP56. I realize that the EM density resolution is low at especially these regions. However, the authors made mutations to ensure that these interactions are relevant. In light of the sentence that mentions that it has been suggested earlier, that N-UBM and C-UBM could bind the same site but with higher affinity due to avidity it would be a useful illustration.

6. Line 342: DNA-tethered? I do not understand this. Is this to consolidate with the before mentioned hypothesis by Günter Blobel? To me it seems that DNA does not need to be involved?

7. Linked to major concern 3. Not clear if in experiment to Fig. 1g cell extract is used.

8. Ext. Data Fig. 5b: Why using eIF4A3? It is nowhere explained.

9. Ext. Data Fig. 2f, please indicate the domains with labels to help orient the reader.

10. Ext. Data Fig. 8c. It would be useful to show the density for SEM1 also.

11. Ext. Data Fig. 9f. Show EM-densities also.

12. Language (please check for more than listed here):

Line 46: "we here", sounds clumsy.

Line 54 and 55: please change "yeast Sub2" to "yeast: Sub2" for better reading flow

Line 69: "AlphFold2" should be AlphaFold2

Line 626: Caption to Fig. 1a is confusing. Now it means that TREX-mRNP must disassemble TREX. Shouldn't it be: "...of a TREX-mRNO complex, which prior to mRNP export must disassemble and the mRNP remodelled via an unknown mechanism."?

Line 633: RecA2 and RecA2 lobes. One of them should be RecA1.

Line 640 and 642: I find the usage of the word coordinated not ideal. It is rather an induction of conformational change to reorient the two domains.

13. Figure referral (should be also checked carefully beyond from which is listed here): Sometimes a later Figure panel is referred before an earlier one, e.g.:

Line 58, Figure 1d before Figure 1b,c

Line 96, Ext. Data Fig. 3b is referred to before 3a

Line 1236, referral to Figure 1d is wrong. It should be 1g I assume.

Line 629-630, referral to left and middle panel in 1b and right panel in c (c has only one panel) is confusing.

Line 233-236: Figure arrangement should be improved to prevent the reader from too much jumping in between.

Version 1:

Reviewer comments:

Referee #1

(Remarks to the Author)

The manuscript presents intriguing work on the role of the DExD ATPase UAP56 in various aspects of mRNA export, particularly through its interactions with multiple factors during assembly and disassembly. The study provides valuable insights into the regulation of mRNA export and gene expression. In the revised manuscript, the authors have addressed most of my previous concerns. However, a few issues remain unresolved, and I have additional questions regarding the newly added data.

1. In lines 99-105, the GCI data demonstrate that THO efficiently disassembles from UAP56 in the presence of ATP and RNA, suggesting that THO dissociates from UAP56 after binding RNA. However, in lines 119-121, the authors report only inefficient disassembly of TREX upon the addition of ATP and RNA. This result seems contradictory. Could the authors clarify the reasons for these differing outcomes in what appears to be similar experimental conditions?

2. In the section title (line 127), they state, "SARNP is a multivalent TREX disassembly factor." However, the data presented to support this conclusion are not entirely convincing. In Figure 2f, the authors examine the effect of different proteins/peptides on the UAP56-THO interaction. While the western blot data were quantified, it appears that different amounts of UAP56 were pulled down under different conditions. Additionally, less THO was detected when less UAP56 was pulled down, which undermines the reliability of the data. More importantly, when the authors used a degron system to induce rapid degradation of SARNP, the UAP56-THO interaction was not affected. This result does not support a role for SARNP in promoting TREX disassembly *in vivo*.

3. The title of this section should be revised to "SARNP Stabilizes UAP56-RNA Complexes *In Vitro*." First, the data presented are entirely *in vitro*, and this should be reflected in the section title. Second, the role of SARNP in stabilizing UAP56 binding to RNA seems limited, as its depletion only affects the nuclear export of a small population of RNAs. This suggests that SARNP may not play a central role in UAP56 binding to RNAs, and the section title should better reflect the data presented.

4. The authors provide three lines of evidence to support the functional importance of UAP56-TREX2. However, it remains questionable whether these data can be used to conclusively support this function, as alternative explanations exist. In lines 315-318, the authors show that truncation of the UAP56 N-terminal domain (NTD), which interacts with TREX2 in the

structure, impairs interaction with TREX2 and causes a growth defect in K562 cells. However, this mutant also exhibits reduced interaction with SARNP and likely other UAP56 partners, such as ALYREF. Therefore, this result cannot be considered strong evidence specifically for the role of UAP56-TREX2. Similarly, the PCID2 mutant used to assess UAP56-PCID2 functionality also shows significantly reduced interaction with GANP, raising further concerns about the function of UAP56-TREX2.

5. While this work focuses on the mechanism of mRNA export, no experiments were conducted to directly examine the nuclear export of RNAs. The authors rely on protein tethering-GFP intensity or growth curves to infer changes in mRNA export. However, these are indirect measures. To provide more direct evidence, the authors could investigate changes in the nucleocytoplasmic distribution of polyA⁺ RNAs, which would offer a better reflection of mRNA export.

Referee #2

(Remarks to the Author)

The revised version leaves this reviewer very much torn.

The structural and biochemical work remains solid, it is high quality data and an impressive experimental effort. It is also clear that the authors made a valiant attempt to address reviewers' comments. Indeed, the in vitro model of the paper is strengthened by the newly added evidence, and the concerns raised by using artificial fusion proteins are fully resolved.

However, a central issue remains: The cellular experiments fail to prove the relevance of the proposed model for nuclear export of mRNPs in cells (or even seem to contradict it, see below).

Firstly, some experiments using UAP56/PCID2 mutants only test effects on cell viability, therefore these experiments show that mutations in the structure based interfaces are important for cell survival. This is a good start, but it is a "black box" result, which does not show whether the suggested functions of UAP56/SARNP/TREX-2 hold really true in cells.

Secondly, the tools exist: E.g. it is a pity that the authors do not make better use of their degron SARNP and PCID2 cell lines to show whether upon transient protein depletion, THO/UAP56/SARNP is enriched on mRNPs, and mRNAs are stuck in the nucleus. These cell lines, and other analogous cell lines for transient THO/ALYREF/UAP56/TREX-2 depletion are also the better tool for testing effects of mutants on complex compositions/changes to protein localization/mRNA export defects.

Thirdly, the argument that several other labs have used the tethering assay in the past is not truly valid, because the technical possibilities of the present are different and today's genome editing tools allow for completely different approaches. The goal should be to find out whether the model mechanism that comes from the minimal in vitro system is actually happening in cells. The reductive system may be missing important components, it may be explaining only some aspects of mRNA export, or it may be entirely correct, but this remains open.

Lastly, potential contradictions between cellular experiments and in vitro model: Upon transient SARNP depletion THOC2 is not enriched in with UAP56 in the IP (Ext. Fig 5i). This runs counter to expectations based on the proposed mechanism. If SARNP is required for TREX disassembly, its removal should trap THO on UAP56. Analogously, if UAP56 cannot bind SARNP (UCM mut) it should enrich THO, however this is not observed in the IP (Ext. Fig. 5h). In contrast, the UAP56 deltaNTD mutant should bind less GANP and should get stuck at the SARNP-bound stage i.e. enrich SARNP, which is also not observed in the IP (Ext. Fig. 5h).

Minor points:

Ext. Fig. 5i: why is SARNP not detected (or not shown?) in FKBP-GFP-tagged cell line? An upward mass shift is expected and would confirm successful endogenous tagging. Is the band cut off? It would be helpful to indicate in the panel which protein is used as bait.

Fig. 3, Ext. Fig. 7e, Ext. Fig. 8e: The density of UAP56RecA2 is so poorly defined that it seems unreasonable to model also the N-UBM/UCM-1 peptides. Are they visible in the density with sufficient clarity to allow their placement? A close-up of the density in this region should be shown as for UAP56 in Ext. Fig. 8d or the peptides should be removed.

Referee #3

(Remarks to the Author)

The authors have satisfyingly erased all my concerns and I congratulate them to this impactful work.

Version 2:

Reviewer comments:

Referee #1

(Remarks to the Author)

During the second round of revision, the authors provided explanations based on their own data and literature from other groups in response to my questions, which has alleviated some of my concerns. However, several issues remain.

1. Regarding my question on SARNP's role in promoting TREX disassembly in vivo, one explanation was that THO is 40-fold less abundant than UAP56, with nearly all THO associated with UAP56-mRNP complexes. However, in ED Fig. 5i, the

co-IP efficiency of THOC2 is significantly lower than that of UAP56 and ALYREF, which contradicts the claim that all THO is bound by UAP56 under normal conditions—at least in this experimental setting. If SARNP indeed facilitates TREX-mRNP disassembly *in vivo*, an enhanced THO-UAP56 interaction should be observed. Contrary to this prediction, less THOC2 was co-IP'd with UAP56 upon SARNP degradation, while more THOC2 was present in the input. Even more puzzling, why were THOC2 and ALYREF not co-IP'd with UAP56 in WT cells?

2. Regarding SARNP's role in UAP56 binding to mRNA and mRNA export, the authors cited Xie et al., 2023, to support its involvement in export. However, while this study reported mild poly(A) accumulation in the nuclei of SARNP KD cells, RNA-seq data indicated that only a small subset of mRNAs was retained. Given that the degron system is superior to KD in avoiding indirect effects, compensation, and inefficiencies, I am perplexed as to why the authors did not test it. Additionally, while they speculate that the limited effect of KD might be due to redundancy with other UCM-containing proteins, they also cite SARNP as a "common essential" gene in the human DepMap project to underscore its importance. This appears selective, as the authors seem to collect supporting evidence without fully striving for scientific accuracy. If they suspect redundancy, they should directly test it through co-KD experiments.

3. In response to my question about the functional importance of UAP56-TREX2 interactions, the authors stated that they do not understand why truncation of UAP56's NTD, which interacts with TREX2, also weakens its interaction with SARNP. This raises the possibility that the NTD is important for UAP56's interactions with multiple proteins, such as ALYREF. To clarify this, the authors should compare the interactome of the NTD deletion mutant with the WT protein. Additionally, regarding the functional importance of the identified interactions in mRNA export, the authors rely solely on literature references. However, in these studies, export defects do not necessarily imply disruption of the identified interactions. For example, in Saguez et al., 2013, mutation of yeast Y12 (human Y13) led to nuclear accumulation of poly(A) RNA, but it remains unclear whether this mutation significantly impacts UAP56-TREX2 interaction. This should be directly tested in mammalian cells by assessing both mRNA export and UAP56-TREX2 interaction.

4. Finally, the authors claim that UAP56 interacts with NPC-bound TREX2. However, how can they be certain it is NPC-bound without imaging? High- or super-resolution imaging should be used to determine whether UAP56 binds TREX2/mRNA when NXF1/NPC function is blocked. Additionally, it is crucial to examine whether UAP56 remains associated with mRNPs in TREX2-depleted cells to directly assess its role in releasing UAP56 from mRNA *in vivo*.

Referee #2

(Remarks to the Author)

The paper by Hohmann, Plaschka and colleagues deals with the mechanism of nuclear mRNA export in human cells. In previous papers, the group had established the structure of THO-UAP56 bound mRNPs. In their current work they aim to address how mRNPs could be remodelled and prepared for export through the pore. Based on biochemical experiments as well as structural evidence from minimal reconstituted protein/RNA complexes of factors linked to mRNA export, the study suggests a directional handover mechanism of mRNPs that is governed by the ATPase UAP56. Specifically, the authors propose that SARNP first displaces THO from UAP56, which then clamps onto the mRNA. As a second step TREX-2M (GANP-PCID2-SEM1 complex) is suggested to release UAP56 from the mRNA, which could then be exported.

This is a tempting mechanism and both the biochemical and structural data are sound, and well controlled. A main concern of this reviewer (and similarly of reviewer 1), voiced in the initial and second review, is that the *in vitro* model is not sufficiently validated in cells, and may therefore not reflect the actual cellular mechanism of mRNP export. In the 2nd revision the authors provide no new experimental evidence and respond to this point by reiterating their own and literature findings. The authors also argue that experiments which would test their proposed mechanism in cells are beyond the scope of the current paper.

After considering the authors' arguments and checking the cited previous studies my concerns remain. It worries me that alternative scenarios are not sufficiently tested and cellular data that do not support (or even seem to contradict) the proposed *in vitro* model seem to be weighed less strongly.

Below, I am attempting to address the authors' points and highlight data that substantiate the concerns. Also, I try to suggest a few experiments that could work towards model confirmation in cells.

* Rescue cell viability assays: These assays show that the interfaces on UAP56 are important for cell survival. They do not show that the mRNP handover mechanism, the directionality of interactions or their sequential nature happens in cells. Furthermore, the data suggest that the UAP56 UCM binding site (i.e. the point where SARNP would bind to compete out THO) is less crucial for cell viability than the UAP56 N-term (i.e. the contact for TREX-2M binding, which would trigger release of the mRNP from UAP56) [Ext.Fig. 5g, 10a]. Therefore, this data could also be interpreted such that SARNP and/or other UCM site binding factors are less important than TREX-2M. But this would not be predicted according to the proposed model, where they act sequentially and should be equally important.

* Immunoprecipitations: As done currently, these assays only show that the structurally characterized interfaces between UAP56 and SARNP or PCID2 are involved in cellular interactions of these proteins. They do not support (or even partially contradict) that the mRNP handover mechanism, the directionality of interactions or their sequential nature happens in cells. While the V5-PCID2 mutant that should no longer bind UAP56 indeed loses UAP56 interaction in IPs (i.e. consistent with structural data), it also shows weaker GANP interaction (i.e. inconsistent with structural data). While the UAP56 UCM mutant interacts less well with SARNP and GANP (i.e. consistent with handover model), the UAP56 Δ NTD mutant does not only lose GANP interaction (i.e. consistent with structure) but also has reduced SARNP interaction instead of the expected

enrichment (i.e. contradicting handover model).

* Tethering assays: Because proteins are artificially bound to mRNAs without the potential for their release, these assays are difficult to interpret and problematic, especially in the context of a proposed model where factors hand over the mRNP and sequentially release from the mRNP. Even if we forget these fundamental issues, then the data supports a part of the mechanism and contradicts other parts (see also section on "alternative scenarios" below). UAP56 increases export of the reporter mRNA and this effect is decreased (but not abolished), when UAP56 cannot hydrolyze ATP or bind TREX-2M (i.e. partially consistent with model). In contrast, while ALYREF tethering increases reporter mRNA export, neither THO nor SARNP tethering has such an effect (i.e. inconsistent with model). This is particularly surprising since SARNP binds UAP56 and UAP56 tethering does increase export. Furthermore, tethering of different TREX-2 factors only weakly promotes or even hampers (PCID2) reporter mRNA export, which could also be seen as contradicting the model.

* MS data: Why is TREX-2 not listed in the MS data figures? Is it not detected (which would be unexpected) or not included (which should be changed)? Based on which criteria was the list of shown factors decided? It is possible that by selecting interactor subsets that are shown in the heat map, a bias is introduced.

* UAP56 abundance: If UAP56 is strongly overly abundant (the authors state 40-fold) then what would prevent it from binding ALYREF without THO, or to TREX-2 without any of the other factors?

* Previous literature:

The cited papers confirm that the studied mRNA export proteins are involved in mRNA export, and that the UAP56-TREX-2 interface is important for mRNA export. They do not show that the mRNP handover mechanism, the directionality of interactions or their sequential nature happens in cells.

* Alternative scenarios (playing "devils advocate" here, these are meant as examples based on the proposed model, not a comprehensive list):

Why is THO needed to bind UAP56 first? ALYREF could also recruit UAP56 directly to mRNPs without THO. All proposed consecutive steps could happen without this first step. The authors state that UAP56 is even 40-fold more abundant than THO. ALYREF tethering does increase reporter mRNA export, while THO tethering does not (see, why this assay is problematic).

Why is SARNP needed to release THO from UAP56 first? Based on the provided data TREX-2 could directly compete with THO for UAP56 binding (e.g. by first latching onto the UAP56 N-term), TREX-2 has a higher affinity than THO for UAP56 (0.07uM vs. 0.2-0.3uM). SARNP and TREX-2 could be parallel pathways or only co-incide on some transcripts.

Why would it matter if UAP56 clamps or unclamps from mRNA in terms of mRNP delivery (i.e. the molecular switch from the title)? UAP56 is bound by ALYREF constantly in the model and ALYREF is bound to the mRNP. Therefore, UAP56's hold onto the mRNP should not be lost by mRNA unclamping.

There are also "unknown unknowns", i.e. factors or steps that are important in cells but are not part of the reconstituted system and are therefore not probed here. E.g. mass spec and AlphaFold2 data indicates that there are many more interactors (Fig. 2b, 2g, 3b, and MS datasets). If the effect of the studied factors on mRNA export is not tested in cells then such holes in our knowledge cannot be recognized. The observation that SARNP depletion (Xie et al., Cell reports 2023: siRNA 48h, A549 cells, 16% SARNP remaining) impairs only 1.2% of mRNAs in export could suggest that alternative explanations and untested factors exist. So does the (albeit problematic) tethering assay, and above mentioned MS data.

Potential experiments:

Using IPs (UAP56, SARNP, TREX-2, ALYREF, EJC) upon transfections of mutants in degron cell lines (i.e. 1-2 days after depletion & transfection when cells are still alive) as well as upon overexpression of mutant proteins the authors could at least start to test whether their model is valid in cells. Mutants and overexpression constructs could be chosen such that they are expected to disrupt consecutive steps, or move the equilibrium into a specific direction and enrichment of intermediate complexes should be detected, EJC components can be probed as a proxy for mRNP binding. Similarly, FRET assays, or localization studies (i.e. enrichment at nuclear periphery via NPC anchored TREX-2) could be used to show stalling of intermediate species.

Potential mutants e.g. UAP56 ATP hydrolysis deficient mutant, UAP56 ATP binding mutant, UAP56 SARNP binding mutant, SARNP UCM mutant, TREX-2 UAP56 binding mutant, UAP56 TREX-2 binding mutant...

Other point:

Rescue assays - Extended figures 3b, 5g, 10a: Please provide western blots that show depletion of DDX39A/B or PCID2 and expression levels of rescue constructs (at a time point when cells can still be harvested e.g. days 2-3), since lack of rescue could also be explained by lack of rescue construct expression.

Version 3:

Reviewer comments:

Referee #1

(Remarks to the Author)

During the last round of revisions, the authors performed additional biochemical and functional assays, which have addressed most of my concerns. However, a few key issues remain:

1. The authors combined acute depletion of PCID2-GFP-FKBP with doxycycline-inducible expression of either wild-type or mutant PCID2 rescue constructs to assess the role of TREX2-UAP56 interactions in mRNA export (Fig. 4e). However, the manuscript only displays polyA+ RNA distribution in a single cell with or without PCID2 degradation. For the critical rescue experiments, no representative images are provided—only quantification data. To strengthen the findings, PolyA+ FISH images of the rescue experiments should be included. Furthermore, images showing a population of cells (rather than just one example) should be presented to ensure reproducibility and robustness.
2. Regarding interaction Analysis Between UAP56 Mutants and TREX-2 (Extended Data Fig. 10I), the authors used three UAP56 mutants to investigate the sequential functional relationship between TREX and TREX-2. However, the interactions with TREX-2 components were not shown. To support their conclusion regarding the sequential binding of THO and TREX-2 to UAP56, the authors should include data on how these mutants affect UAP56's interaction with TREX-2, particularly whether TREX-2 binding is disrupted when THO binding is lost.
3. Correction Needed (Extended Data Fig. 11a):
The label "M3" is incorrect.

(Remarks on code availability)

Referee #2

(Remarks to the Author)

The authors have now addressed my concerns. In particular, I appreciate their experimental efforts to verify their mechanistic model in cells and the more nuanced discussion of their results. These changes have in my view significantly strengthened the paper. Few open questions are left (e.g. how important is SARNP for the sequential hand-over, as the UAP56-SARNP binding mutant was not tested in the new IP-MS experiments), but these are for future studies to address. What remains are only smaller technical points that should not stand in the way of overall accepting the paper.

Fig. 4e - FISH data: Details are missing in the figure legend, some data is not shown and no statistical analysis for significance is provided. How many cells were quantified per replicate and what is the spread of the detected signal between different cells per replicate and between replicates (violin or box plots)? Are the observed changes significant? Why is the mCherry-PCID rescue plasmid expression signal not shown? For the whole experiment please provide IF images (and if available western blots) for rescue construct expression.

Please note that the z-plane used for the representative images shows very little of the cytoplasm and is dominated by the nucleus, which may be due to the rounded shape of the K562 cell line but likely results in a small dynamic range for the assay.

Fig. 5b: Please state in figure legend which mutations UAP56 M1 and M2 mutants contain.

ED Fig. 5i: Why was the UAP56deltaN mutant data now selectively deleted from the gel while the other IP is still shown? The data should be shown fully or replaced by a different IP under the new conditions because otherwise one gets an impression of selective data presentation/removal.

Supplementary Data Tables 1 and 4: It was not possible to open the Excel files (corrupted?).

Data availability: So far only the structural data is mentioned as publicly available. The MS datasets should also be submitted to a FAIR data repository (e.g. PRIDE).

(Remarks on code availability)

List of responses to reviewer comments

“An ATP-gated molecular switch orchestrates human messenger RNA export”

(Ulrich Hohmann, Max Graf, Ulla Schellhaas, Belén Pacheco-Fiallos, Laura Fin, László Tirián, Dominik Handler, Alex W. Phillips, Daria Riabov-Bassat, Thomas Pühringer, Michael-Florian Szalay, Julius Brennecke, and Clemens Plaschka)

Nature Manuscript 2024-02-02833

Reviewer comments are in *blue*.

Responses are in *black*.

A summary of the responses to all three referees:

We are grateful to all three reviewers for their constructive and insightful comments on our manuscript, which led us to majorly strengthen the work through extensive new cellular and biochemical experiments and comprehensive revisions of the manuscript text and figures. We briefly summarize our main additions to the revised manuscript:

(a) *We conducted extensive experiments in human cells, showing that mutations in the interfaces between UAP56 and ALYREF, SARNP, or TREX-2 are important for cell function. We confirmed that key mutations in UAP56–SARNP and UAP56–TREX-2 interfaces disrupted their interactions in nuclear extract. New cellular data also adds evidence that SARNP acts downstream of THO and upstream of TREX-2.*

(b) *New biochemical data confirm UAP56 interactions with ALYREF, SARNP, and THO, support a role for SARNP's multivalency, and validate UAP56-protein fusion constructs.*

(c) *New biochemical assays show that RNA-clamped UAP56 complexes are stable and contain ADP-Pi, not ATP. This means that TREX-2 promotes ADP-Pi product release from UAP56 rather than directly stimulating ATP hydrolysis, a finding that offers new insights into UAP56's mechanism.*

(d) *A new biochemical assay confirms that TREX-2 removes UAP56 from RNA.*

(e) *We extensively revised the manuscript text and figures, and conducted additional biochemical experiments, including on the TREX-2 complex.*

In summary, our new in vitro and cellular data greatly strengthen the proposed model of the mRNA export pathway and add new insights on UAP56's regulation. We hope that the referees will find this revised manuscript to be improved and suitable for publication.

Referees' comments:

Referee #1 (Remarks to the Author):

In this study, Plaschka and colleagues report the central role of UAP56 in orchestrating RNA nuclear export. Based mainly on structural studies of UAP56 with other proteins, they suggest that apart from its known role in packaging into mRNP along with other TREX components, UAP56 also forms two separate complexes with SARNP, a previously identified TREX component, and with GANP and PCID2, central components of the TREX2 complex primarily located at the nuclear pore. Structural data indicate that UAP56 interacts with ALYREF-THO, SARNP, and GANP/PCID2 in a mutually exclusive manner. Consequently, they propose that UAP56 acts as a central molecular switch directing mRNPs through the TREX and TREX2 complexes.

The manuscript is generally well-written and easy to follow, presenting novel findings on mRNA export and potentially introducing an interesting concept about mRNA export. However, it is important to note that most of the conclusions drawn in the study are based on in vitro data and prediction/artificial experiment design. Additionally, the major conclusion suggesting that UAP56 forms into different complexes stepwise requires further direct evidence.

We thank the referee for the insightful and encouraging comments. In the revised manuscript, we have added substantial new in vitro and cellular data, which helped us to strengthen our manuscript.

--The authors proposed that SARNP replaces THO to facilitate THO disassembly from the mRNP. However, this conclusion does not stand with the evidence presented. Recent structural studies have shown that SARNP forms a complex with UAP56, suggesting that SARNP and THO interact with UAP56 in a mutually exclusive manner. The authors confirmed this competition between SARNP and THO for UAP56 interaction using recombinant proteins. However, no evidence supports the proposal that SARNP is recruited to UAP56 following ALYREF/THO. It remains possible that SARNP could be recruited to mRNA ahead of THO or form an independent complex with UAP56-ALYREF-THO. Based on the published and A-fold2-mediated predicted structure of UAP56-SARNP, they speculate that SARNP and ALYREF could interact with UAP56 simultaneously. The authors did not examine this possibility, but instead, took it as reality, and made the SARNP-UCM-ALYREF-UBM fused peptide to study the

competition with THO. In Nat Commun. 2024 Jan 15;15(1):455 and Mol Biol Cell. 2010 Aug 15;21(16):2953-65., SARNP has been shown to form a separate complex with UAP56 in the absence of ATP. Thus, the authors need to provide direct evidence whether SARNP can form a complex with UAP56 and ALYREF using purified proteins and endogenous mRNP. Otherwise, the usage of the SARNP-UCM-ALYREF-UBM peptide does not make sense and could lead to inappropriate conclusions.

We have re-examined the literature and our data and provide additional in vitro and in vivo data that collectively support a role for SARNP in the mRNA export pathway following the dissociation of the THO complex and in promoting the RNA-clamping of UAP56. As requested by the reviewer, we now show in the revision that SARNP can form a complex with ALYREF and UAP56 using non-fused peptides (Extended Data Fig. 4g-j) or full-length recombinant proteins (Extended Data Fig. 4e,f). In addition, we show that full-length SARNP can act in vitro to promote THO release from RNA-clamped UAP56 or from endogenous TREX-mRNPs (Fig. 2f, Extended Data Fig. 5f). We now also provide new cellular data that are consistent with a role for SARNP downstream of THO (Extended Data Fig. 5g-i). Details are described below.

As raised by the reviewer, we had used peptide fusion proteins for our in vitro reconstitutions and to biochemically characterize UAP56 and its interactors ALYREF and SARNP. We designed the fusions to overcome the intrinsically low binary affinities of the ALYREF and SARNP peptides with UAP56, but we appreciate the reviewer's concerns about the fusions.

To briefly summarize our findings: We have shown in the submission that the UAP56-peptide fusions themselves functioned as intended in Extended Data Fig. 5c: UAP56 fused to the ALYREF N-UBM no longer binds the isolated N-UBM peptide (lanes 8, 10 vs. 7), and that UAP56 fused to the SARNP UCM-1 no longer binds the isolated UCM-1 peptide (lanes 13,14 vs. 11). This indicated that the respective peptides in the fusion constructs were bound to their cognate binding site. We further showed that a UAP56-N-UBM fusion can bind the SARNP UCM-1 (lane 12), and the UAP56-UCM-1 fusion binds the N-UBM (lane 9), suggesting that both the N-UBM and UCM-1 binding sites in UAP56 can be occupied simultaneously. Consistent with this, a UAP56-UCM-1-N-UBM fusion no longer binds to either the N-UBM (lane 10) or the UCM-1 (lane 14), indicating that both peptides are bound to their cognate binding site in the UAP56 fusion construct.

Following the reviewer's request, we performed additional experiments to further probe the simultaneous interactions of UAP56 with the ALYREF N-UBM, C-UBM, and the SARNP UCM. For these assays we used non-fused peptides or proteins, which are fully consistent with our earlier biochemical data (Extended Data Fig. 4). Specifically, we now show, first, that UAP56 can bind simultaneously to full-length recombinant ALYREF and full-length recombinant SARNP (Extended Data Fig. 4e-f). Second, we show that UAP56 can simultaneously bind ALYREF (residues 106-257, which contain the ALYREF C-UBM), and an ALYREF N-UBM peptide and a SARNP UCM-1 peptide. This second experiment shows that the N-UBM, C-UBM and UCM-1 peptides can bind UAP56 simultaneously without being fused (Extended Data Fig. 4g-h). Third, we also show that UAP56 can simultaneously bind ALYREF (residues 1-182, containing the N-UBM), and an ALYREF C-UBM peptide and a SARNP UCM-1 peptide. This again shows that all three peptides can bind UAP56 simultaneously (Extended Data Fig. 4i-j). Fourth, we show that also full-length SARNP can promote THO release from UAP56 in our *in vitro* assay (Fig. 2f), and that full-length SARNP can also promote THO release from native mRNPs (Extended Data Fig. 4f). Collectively, these data show that our previous results and conclusions using fusion of ALYREF or SARNP peptides with UAP56 are consistent with our new data where we use non-fused peptides or proteins. A role of SARNP with RNA-clamped UAP56 would predict that SARNP (i) would not affect steps in the export pathway prior to UAP56 RNA-clamping and (ii) that SARNP would not need to bind mRNA on its own since mRNP-specificity is given by clamped UAP56–mRNP complexes.

- (i) We carried out two *in vivo* experiments for the revision, to test if SARNP might function upstream or downstream of THO–UAP56, in experiments (A) and (B).

Experiment (A): We generated a human K562 cell line where endogenous SARNP was CRISPR-Cas9-tagged with a dTAG-degron for its rapid depletion. We could then show that the efficient, near-complete, and rapid depletion of SARNP did not change the levels of THO–UAP56 or of UAP56–ALYREF, which bind to mRNPs prior to SARNP in our model (Extended Data Fig. 5i). We showed this by doing a UAP56 immunoprecipitation before or after SARNP depletion and then western blotting for the THO subunit THOC2 and ALYREF (Extended Data Fig. 5i).

Experiment (B): We also generated an in vivo cellular system for CRISPR-Cas9 knockout-rescue experiments, to mutate key interfaces of UAP56 in cells. We established a cell line where we could acutely induce Cas9-editing to knock out both DDX39A and DDX39B genes using two guides targeting each gene (four guides in total). Cas9 expression was doxycycline (dox) inducible, to control the timing of editing. We also expressed dox-inducible wild-type or mutant UAP56, to assess the in vivo the functional relevance of the newly described UAP56–interactor interfaces (Extended Data Figs 3b, 5g). For our response to this comment, we focus only on the mutant of the UAP56–SARNP UCM interface. While cells die rapidly after the knockout of DDX39A and DDX39B, growth can be rescued by expressing wild-type DDX39B (UAP56). However, a point mutant (D283R) in UAP56, which specifically mutates only the UCM interface, only partially rescued UAP56 function compared to wild-type UAP56 (Extended Data Fig. 5g). This shows that the UCM surface is necessary for normal UAP56 function. We then also immunoprecipitated either wild-type rescue or UCM-interface mutant rescue UAP56 and probed UAP56’s interactions via western blotting with SARNP, THO (THOC2), and TREX-2 (GANP) (Extended Data Fig. 5h). We find that while wild-type UAP56 immuno-precipitated all three interactors, the UAP56 UCM mutant immunoprecipitated substantially less SARNP and less TREX-2 (GANP), but immunoprecipitated unchanged levels of THO (THOC2). These data are consistent with our model for the placement of SARNP in the pathway. Because UAP56–mRNP complexes are presumably less stable in the UAP56 UCM mutant condition, we saw the impaired interaction with the TREX-2 subunit GANP (Extended Data Fig. 4h).

These two new cellular experiments are consistent with our model, where SARNP acts in between THO and TREX-2 complexes in the pathway. We however do not exclude that there could be redundancies for SARNP in cells, since other UCM-peptide containing proteins (such as NCBP3, Extended Data Fig. 6j) might substitute for its function.

We identified SARNP due to its enrichment in UAP56 immunoprecipitates but its absence from THO–UAP56–bound mRNPs. The same was recently reported in yeast, where THO–UAP56-bound mRNPs are also devoid of

yeast SARNP (*Tho1*) (Bonneau et al., *Genes Dev* 2023). In addition, in both yeast and humans, complexes containing UAP56 and SARNP but lacking THO could be detected (here and Bonneau et al., *Genes Dev* 2023). Hence, although yeast and human SARNP act genetically as conserved and general mRNA pathway factors, SARNP does not associate with THO–UAP56-bound endogenous mRNPs at steady state. However, our biochemical observations and new *in vivo* data, suggest that it can associate with UAP56 that it is RNA-clamped after dissociation of the THO complex. Past *in vivo* observations in yeast are also consistent with this model. It was shown by the Aguilera lab (Jimeno et al., *Mol. Cell Biol*, 2006) that the association of highly conserved yeast SARNP protein (*Tho1*) to chromatin is THO-complex- and RNA-dependent, indicating that SARNP binds downstream of THO, which we now cite in line 227-229.

- (ii) We also showed that full-length SARNP does not bind RNA *in vitro* in the absence of UAP56 (Extended Data Fig. 5j). This would be consistent with a model, where UAP56 binding to mRNPs is a requirement for SARNP to associate with mRNPs. We wish to briefly summarize, that in the general human export pathway, ALYREF would recognize mRNPs first, to deliver THO–UAP56 to mRNPs. The THO complex delivers UAP56 in a highly specific conformation of the two UAP56 RecA lobes, that is molecularly primed for RNA-binding (here and Ren et al., *eLife* 2017; Pühringer et al., *eLife* 2020; Schuller et al., *eLife* 2020; Xie et al., *eLife* 2021; Pacheco-Fiallos et al., *Nature*, 2023). This requires the THOC2 MIF4G domain, which positions the two RecA lobes for RNA-clamping. This is the same mechanism by which the spliceosome subunit CWC22 (which also contains a MIF4G domain) is thought to load the exon junction complex DExD-box ATPase EIF4A3 onto RNA. Thus, efficient RNA loading of UAP56 and EIF4A3 depends on a MIF4G-domain partner. Taken together, we currently find it challenging to envision an orthogonal pathway, were SARNP would (i) mechanistically substitute for the THO complex subunit THOC2 MIF4G domain for loading UAP56 onto RNAs and (ii) for how SARNP would specifically recognize and bind mRNPs prior to UAP56 for mRNA selection. We instead favor a model, consistent with our new cellular data, where SARNP acts after THO but before TREX-2 in the pathway (Extended Data Fig. 5h, i). While beyond our current manuscript, we think future studies on

SARNP and other UCM-containing proteins will be interesting to understand how UCM-proteins might interplay, perhaps with a similar logic to UBM-containing mRNA export adapters.

We thus place SARNP in the general mRNA export pathway after the clamping of UAP56 onto mRNA and after the dissociation of the THO complex, but before TREX-2. We hope that the reviewer agrees to the value of our newly added data and extensive revisions of the main manuscript text.

--In Dufu et al., Genes Dev 2010, THO and SARNP interacted with each other as efficiently as they interacted with UAP56 in the presence of ATP. The authors should test whether depletion of SARNP affects UAP56-THO interaction.

Following the reviewers request, we probed the SARNP–UAP56 interaction genetically in cells in two experiments (see also our response to comment #1). Both show that SARNP does not influence the cellular THO–UAP56 interaction.

In brief, we first tagged SARNP endogenously with a degron (FKBP12^{F36V} for use of the dTAG system) (Nabet et al., Nat. Chem. Biol. 2018; Nabet et al., Nat. Com. 2020) in human K562 cells for its acute depletion. We then depleted SARNP by adding the dTAGV1 compound to SARNP levels that were undetectable by western blot. We then immunoprecipitated UAP56 using an anti-UAP56 antibody to analyse the UAP56-interacting proteins by western blotting, including SARNP, THO (THOC2), and ALYREF (Extended Data Fig. 5i). This shows that acute SARNP depletion does not impact the interaction of UAP56 with the THO complex (THOC2) or with ALYREF.

In a second experiment, we developed an inducible CRISPR-Cas9 knockout-rescue assay, where we could rescue the simultaneous knockout of endogenous UAP56/DDX39A with either wild-type or an UCM-interface mutant of UAP56. In this assay we also used western blotting of key interactors (SARNP, THO (THOC2), and TREX-2 (GANP)) after the immunoprecipitation of the wild-type or UCM-mutant UAP56 (Extended Data Fig. 5g,h). This experiment also showed that the UAP56 UCM mutant did not affect the THO–UAP56 interaction. However, the UAP56 UCM mutant did impair the association with SARNP, and with TREX-2 (GANP), consistent with a function of SARNP in stabilizing UAP56–mRNP complexes.

We would also like to point out the key experimental differences to Dufu et al., which might explain their different results. In contrast to our IP-western or IP-MS experiments, the authors in Dufu et al. for their experiments first pre-incubated their nuclear extract at an elevated temperature (30°C for 30 min), added high concentrations of ATP, and added an ATP-regeneration system (500 μM ATP, 20 mM creatine phosphate). The rationale for this pre-incubation is unclear to us. However, these conditions used by Dufu et al. could induce artificial UAP56-clamping on any RNPs in the nuclear extract and could generate a mRNPs, which bind to THO, through unclamped UAP56, and bind to SARNP, through artificially induced and clamped UAP56 molecules. The results by Dufu et al. contrast with our experiments, where we purified native TREX–mRNPs that did not contain SARNP (Fig. 2b, data from Pacheco-Fiallos et al., Nature, 2023). In these experiments, we purified TREX–mRNPs directly from nuclear extract, but without the incubation at 30°C for 30 min, added ATP, and an added ATP regeneration system. Our results are also consistent with recent data from the Conti lab (Bonneau et al., Genes Dev 2024). There, they showed that yeast THO–UAP56 and UAP56–SARNP complexes are mutually exclusive in IP experiments. Specifically, they showed that when they immunoprecipitated UAP56, they could detect SARNP. However, when they first immunoprecipitated UAP56 and then THO (via the THOC1 / yeast: Hpr1 subunit), then a fraction of UAP56 and SANRP are in the flowthrough, indicating that UAP56–SARNP complexes are not THO-bound.

--The interactions of UAP56 with TREX-2 proteins GANP and Sac3 are the only new complex identified in this study and present the most significant findings. Again, in the complex, the authors used the fused SARNP-UCM-ALYREF-UBM that brings artifact. The authors should at least present evidence that GANP/PCID2 could form a complex with UAP56 and ALYREF (with or without SARNP) without artificially fusing ALYREF with SARNP. These interactions of endogenous proteins should be validated using immunoprecipitation from cells followed by western blot analysis. The authors propose that the TREX-2-UAP56 interaction is essential for competing with SARNP. Direct assessment of RNA export using fluorescence in situ hybridization (FISH) would be beneficial.

Following the reviewer's comments, we now carried out additional biochemical and in vivo experiments to probe the UAP56–TREX-2 interaction, which are fully consistent with our initial characterizations using in vitro pulldown assays and the cell-based RNA tethering assay.

We note that our initial intention for the UAP56–TREX-2 cryo-EM structure was to obtain a structure with UAP56-clamped onto RNA together with the ALYREF N-UBM and SARNP UCM. We had therefore reconstituted the cryo-EM sample using a UAP56–N-UBM–UCM fusion protein. In the structure, to our initial surprise, we instead observed that TREX-2 unclamped UAP56 from RNA, which we then followed up in other assays. We also showed in the submitted manuscript that the isolated full-length, recombinant UAP56 binds to TREX-2 in vitro (Extended Data Fig. 7a, Fig. 4d) and that TREX-2 stimulates the UAP56 ATPase activity (Fig. 4f). Thus, neither the binding of UAP56 to TREX-2 nor the TREX-2 mediated ATPase stimulation of UAP56 required SARNP or ALYREF. We now also show for the revision that isolated, non-fused ALYREF N-UBM, C-UBM or SARNP UCM peptides can bind to UAP56 in a UAP56–TREX-2 complex (Extended Data Fig. 7b). This shows that all three peptide binding sites on UAP56 are in principle available in UAP56 when bound to TREX-2 with non-fused constructs. This is consistent with our UAP56–TREX-2 cryo-EM structure, where we predict no mutual exclusivity for peptide–UAP56 interfaces with UAP56–TREX-2 interfaces. We wish to highlight that we currently do not know at which point SARNP dissociates from UAP56 in the pathway.

To probe the functions and interactions of UAP56 with TREX-2 in vivo, we carried out two different genetic experiments (A) and (B).

(A) First, we tested the importance of the UAP56–TREX-2 interfaces from the side of UAP56. The UAP56–TREX-2 interaction is controlled through the UAP56 NTD (N-terminal domain) that mainly binds the TREX-2 subunit PCID2 (Fig. 4b,c). We therefore made use of the in vivo CRISPR-Cas9 knockout-rescue assay, described in detail in our response to reviewer #1 comment #1, to probe this interface genetically (Extended Data Fig. 5g, h). Specifically, we show in vivo that the DDX39A/DDX39B knockout could be rescued by wild-type UAP56, but that it could not be rescued by a mutant UAP56 lacking the NTD (UAP56 Δ NTD). In addition, we then immunoprecipitated the UAP56 Δ NTD mutant or wild-type UAP56 from the nuclear fraction of these cells, and probed for changes in interactions with THO (THOC2), SARNP, and TREX-2 (GANP). This showed a slight reduction in the signals for THO and SARNP, but a larger signal reduction for TREX-2 (Extended Data Fig. 10b, compare lanes 6 to 4), consistent with our model. We note that since the UAP56 Δ NTD cells rapidly die, carrying out

experiments with this condition are challenging. However, we wish to note that the importance of the UAP56 NTD is also consistent with our cell-based RNA-tethering assay, where we showed that a UAP56 NTD truncation impairs the activity of UAP56 in this assay (Fig. 4e).

(B) Second, we probed the relevance of the UAP56 NTD interaction with the TREX-2 subunit PCID2, from the perspective of TREX-2 (Extended Data Fig. 10a, b). For this, we endogenously tagged the TREX-2 subunit PCID2 in human K562 cells with a FKBP12^{F36V} degron, which allowed for the acute and rapid depletion of PCID2 upon the addition of the dTAGV1 molecule (the dTAG system is described here, Nabet et al., Nat. Com. 2020). Using this cell line, we now first show that depletion of TREX-2 subunit PCID2 leads to a severe cell growth defect of human K562 cells (Extended Data Fig. 10a). This phenotype can however be fully rescued by the ectopic expression of wild-type PCID2, but cannot be rescued by a PCID2 mutant in the interface to the UAP56 NTD (PCID2 K374D, K388D). Further, we also show that disruption of TREX-2, with the expression of a PCID2 mutant that is mutated in the PCID2–GANP interface also cannot rescue (Extended Data Fig. 10a). The integrity of TREX-2 and the UAP56–PCID2 interface is thus required for the normal cellular function of PCID2. We further show using co-immunoprecipitation and western blot analysis of PCID2, via an engineered V5 tag, that wild-type PCID2 can immunoprecipitate the TREX-2 subunit GANP and UAP56, while PCID2 mutated in the UAP56 NTD-binding site loses the interaction with UAP56 (and possibly with GANP) (Extended Data Fig. 10b).

Although beyond the scope of our current manuscript, future studies could make use of these mutants to establish quantitative and genome-wide SLAM-seq transcriptomics with cell fractionation to study mRNA dependencies, and to carry out quantitative cellular imaging using poly-A FISH and single-transcript-specific probes (which could be identified through SLAM-seq data).

Taken together, these newly added data support the relevance of the UAP56–TREX-2 interaction *in vitro* and *in vivo*, in addition to our biochemical and other *in cell* data from the cell-based RNA tethering assay.

--The authors proposed that TREX-2 stimulates the ATPase activity of UAP56 and thus releases UAP56 from the mRNP. Does depletion of TREX2 or mutation of the TREX2-UAP56 interaction affect UAP56 release from the mRNP? It is known that RNA, ALYREF, and THO together stimulate the ATPase activity. In support of this, UAP56 does not clamp on RNA in the presence of ALYREF and THO. What leads the authors to believe that only after TREX2 binding, the ATPase activity of UAP56 is stimulated? Also, de-clamp of UAP56 on RNA does not mean its disassembly from the RNA. In the presence of ALYREF and THO, UAP56 does not clamp on RNA, but still associates with the mRNP.

The reviewer highlights very interesting questions on the UAP56 ATPase mechanism and its regulation, which we have now significantly expanded on with new biochemical data in the revision. In brief, we now first show that TREX-2 stimulates the UAP56 ATPase activity in vitro by promoting ADP-Pi and RNA product release instead of stimulating ATP hydrolysis (Figs 1h, i, 4f, Extended Data Fig. 10f). Second, we find that once UAP56 is RNA-clamped, only TREX-2 but not THO can unclamp UAP56 from RNA (Fig. 4g, Extended Data Fig. 10f). Thus, TREX-2 has the unique biochemical activity to unclamp UAP56 from RNA through wedging (see Fig. 4c) in between the two UAP56 RecA lobes and promoting ADP-Pi product release. These new data add to a better understanding of the UAP56 ATPase activity and its regulation. Details are described below.

Indeed, the UAP56 ATPase activity has been shown to be stimulated by RNA (here and Shen et al., JBC 2007; Ren et al., eLife 2017), the recombinant yeast and human THO complexes (Ren et al., eLife 2017, Pühringer et al., eLife 2020), yeast ALYREF (Ren et al., eLife 2017), and TREX-2 (this manuscript). However, the respective effects on ATPase stimulation are mechanistically different and stimulation is exerted to different degrees. Yeast ALYREF was reported to stimulate UAP56 two-fold (Ren et al., eLife 2017) while THO stimulates UAP56 two- to three-fold (Ren et al., eLife 2017; Pühringer et al., eLife 2020). In contrast, TREX-2 stimulates UAP56 more than fifty-fold, more than an order of magnitude greater than THO.

The low two-fold ATPase stimulation of UAP56 by the THO complex is consistent with THO's function in loading UAP56 onto RNA, where THO would aid to position the two UAP56 RecA lobes for RNA binding, which in an in vitro assay could yield such a low apparent stimulation of the ATPase activity.

In contrast, the mechanism by which TREX-2 stimulates UAP56 sixty-fold is not understood. TREX-2 could either stimulate the release of ADP-Pi after ATP hydrolysis or could stimulate ATP hydrolysis itself. To distinguish between these possibilities in the revision, we first examined the RNA-clamped UAP56 complex biochemically. It was previously reported that some DExD-box ATPases hydrolyze ATP immediately after RNA clamping, yielding a stable Protein-ADP-Pi-RNA complex instead of a Protein-ATP-RNA complex (Henn et al., JMB 2008; Nielsen et. al., RNA 2008; Cao et. al., JMB 2011). We used UV-crosslinking experiments with radio-labelled ATP (at either the α - or γ -phosphate position of ATP) to determine that the DExD-box ATPase UAP56 hydrolyzes ATP immediately upon RNA-clamping (Fig. 1h). We further show that the resultant complex of UAP56-ADP-Pi-RNA complex is stable over the course of a stringent size exclusion chromatography experiment, but not UAP56 samples prepared with RNA and ADP (Fig. 1i, Extended Data Fig. 3e). This stability is further consistent with UAP56's low ATPase rate (Fig. 4f) and stability of bead-immobilized RNA at room temperature (Fig. 4g, Extended Data Fig. 10f). These data indicate that the substrate for TREX-2 is a UAP56-ADP-Pi-RNA complex, and that TREX-2 acts mechanistically on UAP56 to release the products ADP-Pi and RNA.

This mechanistic explanation is consistent with our UAP56-TREX-2 cryo-EM structure. We had observed that the GANP 'wedge' loop inserts between the two RecA lobes of UAP56, whereby the wedge substitutes the UAP56 residue F381 (of the RecA2 lobe) in the nucleotide base binding site (Extended Data Fig. 10d). In this manner, TREX-2 would first release the UAP56-clamped RNA, and presumably through on-/off-kinetics would release UAP56 and the remaining ADP subsequently, a state we captured by cryo-EM (Fig. 3e, 4c, Extended Data Figs 7,8). This mechanism is also consistent with data on other DExD-box ATPases, including EIF4A3, DbpA, Mss116, and DDX1, where product release (ADP-Pi) was also shown to be the rate limiting step for the turnover of these RNA-clamped protein complexes (Henn et al., JMB 2008, Nielsen et. al., RNA 2008, Cao et. al., JMB 2011, Kellner et al., NAR 2015).

These mechanistic data also suggest that only TREX-2, but not the THO complex, should be able to unclamp UAP56 from RNA. To test this, we formed UAP56-ADP-Pi-RNA complexes on beads, by immobilizing the RNA. We then incubated these RNA-clamped UAP56 complexes with either the recombinant THO or TREX-2 complex. Indeed, we found that while TREX-2 can unclamp nearly all UAP56 from RNA, the THO

complex had no effect (Fig. 4g). In addition, we used the same assay to now also show that the 'wedge' mutant of TREX-2 (GANP R678A) is less effective at UAP56-unclamping, consistent with a ten-fold reduction in the UAP56 ATPase stimulation by the GANP R678A mutant (Extended Data Fig. 10f).

On the THO complex. We show in our manuscript that RNA-clamped UAP56 has no measurable affinity to THO (Fig. 1g), and now also show that THO has no effect on RNA-clamped UAP56 (Fig. 4g). Consistent with our structural data, this indicates that the THO complex would 'prime' UAP56 for RNA binding, by optimally positioning its two RecA lobes, and that this can explain the observed 2-fold increase of the UAP56 ATPase rate in *in vitro* assays. In native TREX-mRNPs, THO is mRNP associated through non-clamped UAP56, which in turn binds the mRNP through its interactions with N-UBM and C-UBM containing mRNA export adapters, such as ALYREF. The high valency of UBM-containing proteins on mRNPs is thus important to overcome the individual low affinities of UBMs to UAP56. Once UAP56 clamps stochastically onto RNA with ATP inside TREX-mRNPs, THO would release, aided by UCM-proteins such as SARNP. We speculate that TREX-2 is efficient at unclamping UAP56 from native mRNPs, because of its wedge-mechanism promoting the release of the reaction products ADP and Pi. Once RNA is unclamped from UAP56, the low affinity of UBM and UCM peptides alone would be insufficient to keep UAP56 tethered to the mRNP via ALYREF and/or SARNP.

We are again thankful to the reviewer for this comment, since it helped us to better understand the TREX-2 mechanism and the regulation of the UAP56 ATPase cycle.

Referee #2 (Remarks to the Author):

Before export from their nuclear site of production into the cytoplasm, mRNAs are packaged into particles (mRNP) that undergo several maturation and remodelling steps. This packaging serves as a quality control step to license correctly processed mRNAs and offers also a mode of control of gene expression. Although research on this highly conserved process has been going on for decades, many basic mechanistic questions remain unanswered. In their current manuscript, Hohmann, Brennecke, Plaschka and colleagues study some of the mRNP remodelling steps that precede mRNA export, using a combination of biochemical assays and structural studies. The focus lies on the RNA dependent ATPase UAP56, which forms a platform for sequential, mutually exclusive binding events of mRNA export factors. The study is of interest for researchers working on RNA biology and gene regulation.

The paper contains overall high quality cryoEM and biochemical data, the text is well structured and figures are intuitively accessible. However, I have three main points of criticism:

We thank the reviewer for the constructive feedback, which helped us to improve our manuscript.

1. Going through the available literature, I was left with the impression that prior knowledge was not accurately reflected possibly with the wish to increase "novelty". This affects particularly the first half of the manuscript.

Examples of previously published points, that appear presented as if new:

A structure of yeast UAP56 bound by yeast ALYREF was published by Ren et al. eLife 2017. This paper also already showed that ALYREF binding to UAP56 increases RNA affinity.

A structure of UAP56 bound by yeast SARNP was published by Xie et al. Cell Rep. 2023. This paper also already showed that SARNP binding to UAP56 increases RNA affinity, that SARNP has multiple UAP56 binding motifs in all eukaryotes (they called them DIMs) and that SARNP binding to UAP56 had to occur after THO complex release because their binding sites are mutually exclusive.

We apologize for having left this impression, which was not our intention. We address each point raised by the reviewer, below, and highlight the major gaps of knowledge

that our data addresses beyond previous findings. We accordingly revised relevant manuscript parts, which we hope the reviewer finds satisfactory. Specifically on the work by Xie et al., which we cited in several manuscript parts and was published while our manuscript was in preparation, we clarify our key and novel data and discoveries.

*(A) On the structural data of prior partial complexes of yeast *S. cerevisiae* Yra1–Sub2 and partial human ALYREF–UAP56.*

Thus far partial structures of yeast Yra1 C-UBM–Sub2 and human ALYREF C-UBM–UAP56 had been determined, by respectively by Ren et al. eLife 2017 in the yeast system and by our lab (Pacheco-Fiallos et al. Nature 2023) in the human system. We cited these structures together (refs. 2, 24; formerly 1, 23) in line 61-63 (formerly 57-58), but noted that “only the C-UBM had been observed in [these] structures^{2,24}”. However, Yra1/ALYREF contains a second highly conserved UBM, the N-UBM.

In our current manuscript we report that, unlike previously thought, the ALYREF N-UBM binds to a distinct site in UAP56. This novel N-UBM-binding site in UAP56 had not been identified in any prior structural, biochemical, or genetic data. The identification of the N-UBM binding site is significant for two reasons:

First, this new insight allowed us to propose that the ALYREF N- and C-UBMs, which bind to two distinct binding sites on UAP56, may be functionally different. We then show that the newly identified ALYREF N-UBM binding site is located directly adjacent to the binding site of the SARNP UCM. Our biochemical data show that these two ALYREF and SARNP peptides act in synergy to bind UAP56 and promote RNA-clamping by UAP56. There was no prior evidence for synergy between ALYREF N-UBM and SARNP UCM binding. New in vivo data, which we added for the revision, also show that mutations of UAP56 in the C-UBM, N-UBM, and UCM interfaces lead to growth defects in human K562 cells in a genetic CRISPR-Cas9 knockout-rescue experiment (Extended Data Figs 3b, 5g). We thus demonstrate for the first time, to our knowledge, that each peptide–UAP56 interface (the N-UBM interface being unique to our manuscript) are necessary for the normal in vivo function of UAP56. Thus our data on the ALYREF N-UBM are novel.

Ren et al., 2017 had reported an increased binding of Sub2 to RNA in presence of a larger C-terminal region of Yra1. This region included Yra1's unstructured RNA binding domain (termed the C-terminal variable region, C-vr), that comes before the C-UBM, in addition to the C-UBM. The authors then showed that the Yra1 RNA-binding C-vr region was fully required for the reported increase in Sub2's RNA binding. The authors had thus concluded that "Sub2 and Yra1-C, by spatially juxtaposing their RNA binding regions, cooperatively bind to RNA". In contrast, in our work, we describe that the N-UBM and UCM directly affect RNA-binding instead of indirectly as reported by Ren et al., where in their study the RNA-binding effect did not come from a direct modulation of UAP56 by the C-UBM but instead from ALYREFs adjacent RNA-binding domain.

Second, the identification of a unique binding site of the N-UBM on UAP56 further suggests that other metazoan UBM-containing proteins should functionally be – likely – categorized more explicitly. For example, while UIF, LUZP4, PHAX and POLDIP3 contain N-UBMs, CHTOP, RBM26, RBM27 (the latter two are identified in our work) contain C-UBMs. While UBMs had previously been thought to be interchangeable, our data suggest that they act differently in the pathway. The implications of this for understanding other metazoan, UBM-containing export adapters will be interesting to explore in future studies beyond our current manuscript.

Taken together, our ALYREF N-UBM findings, which we highlight also together with the synergy with the SARNP UCM, are novel and have not been covered by the published literature.

(B) On the SARNP–UAP56 interaction and its implications in comparison to Xie et al., Cell Reports, 2023 (ref. 45, formerly 39).

We first cite Xie et al. in lines 134-135 (formerly lines 108-111) to comment that "[...] Additionally, SARNP has been shown to bind human mRNAs in vivo^{46,47} and RNA-clamped UAP56 in vitro^{48,45}". We also cite Xie et al., where we first describe the SARNP-UAP56 binding in lines 138-140 (formerly 113-116) to say that our AlphaFold2 prediction of human UAP56- and human SARNP "was indistinguishable from a recent crystal structure of a chimeric human UAP56–

yeast SARNP complex exhibiting an RMSD of 0.67 Å across 370 atom pairs⁴⁵.” We agree with the reviewer that we should also cite Xie et al., when we describe UCM sequence conservation and multivalency, which we now added in the revised manuscript, line 145.

The authors in Xie et al. had noted that the SARNP UCM overlaps with THOC2-binding to UAP56. However, they had not tested this model either in vitro using recombinant THO, UAP56, SARNP UCM, ALYREF NUBM interactions or using endogenous TREX–mRNPs. This statement had thus remained as an untested speculation. In contrast, we here provide evidence that the overlap of SARNP and THOC2-binding sites on UAP56 confers an activity on SARNP to help dissociate UAP56 from THO. However, we wish to note that these data are in support of a key advance in our manuscript: We discovered that the RNA-clamping of UAP56 drives the remodeling of TREX–mRNPs to become competent for export. UAP56 RNA-clamping is aided by binding of the ALYREF N-UBM and SARNP UCM to disassemble TREX to release the THO complex. This discovery suggested that UAP56 acts as an mRNA maturation mark that, similar to the exon junction complex in pre-mRNA splicing, marks the completion of mRNP packaging and export competence. RNA-clamped UAP56 then allows the mRNP to subsequently dock at the nuclear pore to TREX-2 to promote the final steps of mRNA export.

Taken together, we are excited that our proposed conserved mRNA export model, is consistent with the past observations in the yeast and human mRNA export fields, and hope the reviewer agrees that we now better reference and refer to the past literature.

2. The biological relevance of the structures and resulting models is not tested. The only cell based assay is a tethering export assay, where different proteins are artificially bound to an mRNA via stem-loop-aptamer links. On top of this, the assay is problematic for the proteins studied here, because UAP56 normally has to be released from mRNAs before export. This can no longer occur when UAP56 is tethered, which draws the validity of this whole assay into question (and the authors actually make a big point about this putative function of TREX-2M on UAP56 release from mRNAs). Originally this assay was conceived to study the effect of splicing/EJC deposition on export, but it is not a good tool for the factors studied here. Furthermore, none of the interactions are tested in cells, neither are any mutants shown to affect interactions in cells. Literature

data indicates that these are relevant points. In a minimalized in vitro system of reconstituted complexes certain effects might appear clear cut, but in the more complex environment of the nucleus, factors might be in competition with others, and their effects may be much less important. E.g. Xie et al. Cell Rep. 2023 found that only a minority of mRNAs (i.e. ~200 vs. 12,000) were less well (1.5x) exported upon SARNP depletion. In contrast, from the model presented in the current manuscript one would expect SARNP to be crucially important for export of all mRNAs.

We appreciate the reviewer's comment that additional in vivo probing of the newly identified interactions would strengthen our manuscript. We have thus invested to establish in vivo assays to test the contribution of these interactions in human cells, in addition to the cell-based tethering assay that we comment on at the end of the response.

First, we have now established in human K562 cells an acute CRISPR-Cas9 knockout-rescue assay to deplete endogenous UAP56 and rescue with either wild-type or mutant UAP56 (Extended Data Fig. 3b, 5g). Because UAP56 is essential for human cell viability, we monitor in this assay for cell survival. For efficient knock-out editing and owing to there being two genes for UAP56 in human cells (DDX39A and DDX39B), we established a dual guide CRISPR-Cas9 knockout strategy (two guides per gene) in the same cell. In these cells, Cas9 is inducible by doxycycline (dox), such that the beginning of Cas9-editing can be experimentally controlled. Prior to inducing editing with +dox, we also introduced either wild-type UAP56 or mutant UAP56 under the control of a dox-inducible promoter. We used DDX39B (named UAP56 below) for these rescues, since DDX39B is broadly essential in human cell lines according to the depmap project, in contrast to DDX39A (<https://depmap.org/>).

In UAP56 we first individually mutated the UAP56-binding sites to the N-UBM (R276E, A302R, E309S), to the C-UBM (R208S, Q212A, R216S, K241S) and to the SARNP-UCM (R283D), or the UAP56 ATPase activity (DECD-box mutated to AACD, D196A, E197A) (Extended Data Fig. 3b, 5g). These mutations tested specific interactions described in the first part of the manuscript. Notably, no mutant UAP56 could rescue the UAP56 knockout to the levels of wild-type UAP56. This shows that each tested mutation in UAP56-interactor interfaces is necessary for the normal cellular function of UAP56. Further, we truncated the UAP56 NTD (N-terminal extension), which we observed to be critical for binding to TREX-2 in the second manuscript half. This

mutant could also not rescue UAP56 function (Extended Data Fig. 5g). To show that the *in vivo* tested mutant of the TREX-2–UAP56 interface (UAP56 Δ NTD) impairs the TREX-2–UAP56 interaction, as predicted by our *in vitro* data (Fig. 4d), we also carried out anti-UAP56 IPs via a UAP56-specific antibody in nuclear extract from UAP56 Δ NTD mutant cells (Extended Data Fig. 10b). As expected, while the rescue with wild-type UAP56 could precipitate the TREX-2 subunit GANP, the rescue with UAP56 Δ NTD mutant was significantly impaired in the TREX-2 interaction.

Taken together, our newly added *in vivo* data on UAP56 support that each important UAP56-protein interface described in the manuscript (ALYREF N- or C-UBM, SARNP UCM, or TREX-2) contributes to its *in vivo* function in human cells, consistent with our structural and biochemical data.

Second, we also tested the binding site of the UAP56 NTD with TREX-2 *in vivo*, but from the TREX-2 perspective. For this experiment, we mutated the TREX-2 subunit PCID2, which directly interacts with the UAP56 NTD which is necessary for the UAP56–TREX-2 interaction (Fig. 4c, d). We first modified the endogenous PCID2 by CRISPR-Cas9 engineering with a degron tag (dTAG, FKBP12^{F36V}) for its acute depletion using the dTAGV1 compound (Nabet et al., Nat. Com. 2020). In this setup, we could acutely deplete PCID2 and rescue at the same time by the dox-inducible expression of either wild-type or mutant PCID2 (Extended Data Fig. 10a) and again assess cell growth. We mutated either the PCID2–UAP56 NTD interface (K374D, K388D) or the PCID2–GANP interface (D356R, A365F) as a control. These experiments showed that both, the interaction between PCID2 and the UAP56 NTD and the intact TREX-2 complex, are essential for PCID2 to function *in vivo*. In addition, we also show by westerns from nuclear cell extracts that mutant PCID2 (PCID2–UAP56 NTD interface) immunoprecipitated UAP56 at background levels in comparison to wild-type UAP56 and a mock control (we also note a reduction in the GANP signal) (compare Lanes 6 to 4 in Extended Data Fig. 10b). Thus, consistent with our biochemical data, the UAP56–TREX-2 interaction is also important *in vivo*.

Taken together, our new *in vivo* experiments show that the UAP56–protein interfaces we discuss and identify newly in the manuscript are necessary for the functions of UAP56 and of TREX-2.

On the cell-based tethering assay. We share the reviewer's view that it is surprising that an RNA export-promoting effect is observed for factors that should be removed during or upon the completion of mRNA nuclear export. While the field incompletely understands the limitations of this assay, we wish to note that this assay has yielded insightful data when used for example by the labs of Brian Cullen, Stuart Wilson, Frédéric Allain, and Elisa Izaurralde to study mRNA export proteins, including the mRNA export factor NXF1 and ALYREF (Braun et al., JBC, 2001; Wiegand et al., PNAS 2003; Hargous et al., EMBO J 2006; Hautbergue et al., PNAS 2008; Hautbergue et al., Current Bio 2009; Viphakone et al., NAR 2015). Inspired by the previous work in the field, which had used luciferase-based read-outs, we generated a similar reporter that instead coupled to a GFP-based readout. We unfortunately cannot comment to the release of mRNP proteins on the cytoplasmic face of the nuclear pore complex for NXF1, which was previously studied with this assay, or on UAP56 that we focus on here. However, the previous data on NXF1 and here on UAP56 show that this assay provides meaningful insights in functionally important regions of these proteins. We hope the reviewer will agree that this assay, together with our newly added in vivo genetic data, consistently shows that in every assay we have designed, whether biochemical, tethering assay, or genetic in vivo experiments, the tested UAP56–TREX-2 interfaces contribute to UAP56 function.

We agree that our data describe a mechanistic framework for the pathway, but that we expect that additional complexities of the pathway will be discovered in the future and will influence the efficiencies of specific steps in vivo. We now better discuss the limitations of our mechanistic framework, and exciting future direction in the manuscript discussion in lines 400-415. For example, added complexity in vivo could arise from potential redundancy in two respects: (i) Other N- or C-UBM-containing proteins, in addition to ALYREF, may have partially overlapping functions in the pathway. (ii) Other UCM-containing proteins in addition to SARNP, as we for example newly identified also in the protein NCBP3 (Extended Data Fig. 6j), or as yet undiscovered UCM-containing proteins may also be partially redundant with SARNP. We now discuss this in the manuscript discussion.

In this context, we wish to highlight two key limitations of the work by Xie et al., where the authors reported that only a subset of cellular RNAs require SARNP for their export. First, Xie et al. used an siRNA-mediated SARNP knockdown for the RNA seq experiments. This lead to a reduction of SARNP levels to 16% after 48h (Fig. 6B in Xie

et al. Cell Reports, 2023). Owing to the incomplete depletion of SARNP, the authors study the effects of low SARNP levels but not of the complete removal of SARNP. The data by Xie et al. therefore cannot distinguish if the non-affected transcripts are independent or, alternatively, super-dependent on SARNP. Second, owing to the long-term depletion, secondary effects may accumulate to 'survive' under these new conditions, which mask primary and direct effects of SARNP on mRNA export. A clear conclusion on which mRNAs depend on SARNP would therefore require (a) a more efficient depletion over (b) a shorter time scale of the perturbation. Nascent RNA metabolic labelling, unlike the steady-state RNA seq measurements reported by Xie et al., would be necessary to examine direct effects of SARNP. We find such experiments exciting future directions, but think that these go significantly beyond the scope of our current manuscript. We thus currently think that THO–UAP56 and ALYREF may be needed for the vast majority of mRNAs, and that SARNP might also affect many mRNAs, also owing to its general essentiality for human cell viability (<https://depmap.org/>). But we do not exclude that SARNP might be more important for some transcripts than others.

We hope the reviewer agrees that the explicit discussion on limitations of the current mechanistic framework, addresses this point and will hopefully excite others to further investigate transcriptome-wide dynamic RNA metabolism aspects in the future for SARNP and other members of the mRNA export pathway.

3. a) Some aspects of the biochemical assays are problematic. b) Also some conclusions are drawn from structures but are not supported by any experimental evidence.

a) In the first half of the paper several binding and biochemical assays depend on using a fusion peptide of the UAP56 binding-motifs of SARNP and ALYREF. As a reason for this artificial method the low affinities of the individual peptides is given by the authors. However, in cells these low affinities would have to suffice if the proposed hand-over models of the authors are true. The authors might argue that in the cellular environment there is multiplicity in all of the interactions which provides avidity effects (THO tetramer, SARNP with multiple binding motifs, ALYREF with 2 UAP56 binding motifs). Then why is this never tested? Instead of using the fusion peptide full length SARNP could be used in the competition assays. Longer RNAs can be in vitro transcribed to allow for multiple copies of UAP56/ALYREF/THO to bind to the same molecule. The danger is that by using the fusion peptide an artificial enrichment is achieved that never occurs in cells, and that cells require actually additional/other factors to carry out TREX disassembly. This

danger is increased by the lack of cellular assays to verify the mechanistic models proposed here.

We appreciate the reviewer's concern about the usage of fusion peptides to overcome low binary affinities. We have carried out several new experiments for the revision to address these comments, which suggest that the fusion peptides are reasonable proxies for non-fused or full-length ALYREF and SARNP. First, despite technical challenges, we now repeated important in vitro experiments on UAP56 without fusion peptides, which yielded results fully consistent with our fusion constructs (Extended Data Fig. 4g-j). Details are described below. Second, we carried out selected experiments, that were technically feasible for us, using recombinant full-length ALYREF and SARNP (Extended Data Figs 4e,f, 5f), that also support the conclusions from peptide only experiments, as described below. Third, in addition to these data we now also provide in vivo mutational data on UAP56 and TREX-2 that support our in vitro conclusions (see our response #2 for details). Altogether, these data are consistent with the mechanistic framework of the mRNA export pathway that we proposed.

Generally, we wish to note that high-quality in vitro experiments with full-length ALYREF and SARNP are technically challenging or not possible, since both full-length proteins behave poorly biochemically and are prone to precipitate in vitro below the protein and salt concentrations required for our in vitro experiments. We had thus decided, to be able to perform high-quality experiments, to establish the highly controlled and well-behaved in vitro system using the minimal components or peptides that are essential to the pathway based on prior or our data. However, we appreciate that fusion peptides do not fully capture the in vivo situation, where also multivalent interactions can occur. We have thus carried out the following experiments that we include in the revision:

- (i) First, we have now added several in vitro experiments (Extended Data Fig. 4 g-j), to demonstrate the isolated but non-fused ALYREF N-UBM, C-UBM, and SARNP UCM peptides can simultaneously engage with UAP56 in vitro. We now also show in the in vitro THO–UAP56 dissociation assay, that the isolated N-UBM and UCM peptides elicit a synergistic effect on the dissociation of UAP56 compared to the isolated peptides (Fig. 2f). While N-UBM alone (74±10%), UCM-1 alone (72±11%) can release UAP56, the combined addition of isolated N-UBM and UCM-1 together increases this*

effect ($53\pm 9\%$). This is less than the fusion of N-UBM and UCM-1, owing presumably to the added avidity effect that the fusion brings.

- (ii) To assess the effect of multivalency, we first now show the effect of multivalent, full-length recombinant SARNP in the *in vitro* THO–UAP56 release. In this experiment, we now show that full-length SARNP dissociated UAP56 to $46\pm 9\%$ in comparison to $75\pm 11\%$ for the isolated UCM-1 (Fig. 2f). We note that owing to the poor biochemical behavior of SARNP, we were unable to go to test higher concentrations of SARNP in the assay, and we may thus underestimate the effect. We also show in additional *in vitro* pulldown experiments, that both full-length SARNP and ALYREF can simultaneously bind UAP56 (Extended Data Fig. 4e,f). We also show that a full-length recombinant SARNP, where each of the 5 UCMs is mutated (the key UCM motif R mutated to D), cannot bind UAP56 anymore *in vitro*, suggesting that only the UCMs bind UAP56 (Extended Data Fig. 4d). Consistent with this, truncation of the SARNPs SAP domain, whose function is unknown, does also not affect SARNP binding to UAP56 (Extended Data Fig. 4d). Finally, we also show, using a western blot readout, that multivalent, full-length SARNP can dissociate endogenously purified TREX–mRNPs (Extended Data Fig. 5f).

Others and us have noted that proteins within mRNPs, especially evidenced for human ALYREF (and its homolog, yeast Yra1), rely on multivalent interactions (Pacheco-Fiallos et al., *Nature*, 2023; Bonneau et al., *Genes Dev* 2023). For human ALYREF this concerns interactions with RNA, itself, ERH, the exon junction complex, the cap-binding complex, and possibly other proteins (Portman et al., *RNA* 1997; Pacheco-Fiallos et al., *Nature*; Bonneau et al., *Genes Dev* 2023; Clarke et al., *eLife* 2024). However, proteins stoichiometries of ALYREF, SARNP and other mRNP proteins on mRNPs within human or yeast cell nuclei *in vivo* are currently unknown. We also note that we cannot currently reconstitute entire yeast or human mRNPs *in vitro*, and that we have particularly encountered challenges with reconstituting *in vitro* interactions using longer RNAs owing to aggregation *in trans*. Full human mRNP reconstitution is thus an exciting part of the long-term vision of our group, but it is currently out of reach and beyond a ten-year goal of our group. We believe, and hope the reviewer agrees, that the added experiments, revisions to the manuscript text on the impact of multivalency of SARNP

and ALYREF, and the new *in vivo* genetic experiments strengthen the mechanistic framework for the export pathway that we propose.

b) The main advance of the paper comes in its second half, where the previously unknown binding of UAP56 to TREX-2M is identified and structurally characterized by cryoEM. This structure suggests that TREX-2M binding to UAP56 leads to mRNA release. Unfortunately, this interesting hypothesis is not experimentally verified at all, neither *in vitro* nor in cellular assays.

Does TREX-2M binding actually lead to release of RNA *in vitro*? Is UAP56-TREX-2M complex affinity for RNA lower than UAP56 alone? Are UAP56-deltaN mutants stuck on mRNAs in the nucleus of cells?

The reviewer raises an interesting point, which we have now further addressed experimentally in vitro and in vivo.

First, we tested if recombinant TREX-2^M can remove UAP56 from RNA in vitro (Fig. 4g, Extended Data Fig. 10f). For this experiment, we immobilized RNA on beads, and added UAP56 and ATP to form RNA-clamped UAP56 complexes. We then washed away excess UAP56 and ATP. We then challenged the immobilized UAP56–RNA–ATP complexes with or without TREX-2, and afterwards eluted the remaining UAP56–RNA–ATP with the nuclease Benzonase. We observed that TREX-2 indeed can remove UAP56 almost completely from UAP56–RNA–ATP complexes (Fig. 4g, Extended Data Fig. 10f). This is consistent with our cryo-EM data: Despite assembling TREX–UAP56–RNA–ATP complexes that contained RNA, in the structure we only observed the RNA-unbound form of UAP56. Further, we also show now that in the UAP56–RNA–ATP immobilization assay, the addition of the recombinant THO complex has no effect on the persistence of immobilized UAP56–RNA–ATP complexes (Fig. 4g). This data shows collectively that TREX-2 but not the THO complex can release UAP56 from RNA complexes in vitro.

Second, we now show in vivo, using genetic assays, that removal of the UAP56 NTD or mutation of the UAP56 NTD binding region in PCID2, both interfere with the essential cellular functions of UAP56 and PCID2, respectively (Extended Data Figs 5g, 10a). For details on the in vivo experiments, please see our response to point #2. Based on the known biochemistry of UAP56, we think that UAP56 Δ NTD could engage with nuclear RNA, since (i) UAP56 Δ NTD is localized to the nucleus (Extended Data Figs 9g), (ii) and

because the NTD is connected to UAP56 by a flexible linker that would not be expected to influence the spatially distant UAP56 RecA1 and RecA2 lobes that clamp onto RNA. This suggests that UAP56 Δ NTD indeed would be enriched on RNA in the nucleus in UAP56/DDX39A knockout cells, where cell viability is severely perturbed and these cells are thus hard to analyze. If UAP56 Δ NTD would be present in cells expressing wild type levels of UAP56 we expect the protein not to accumulate in the nucleus as it likely still binds nuclear RNA but then might be exported with the mRNA to the cytoplasm as it is not removed by TREX-2 at the NPC.

We have updated the manuscript accordingly with these new data.

Along a similar line, it is unlikely that the GANP "wedge" with its small contact can release the RNA clamp of UAP56 directly. It seems more likely that ATP hydrolysis has to occur first to release the RecA1-2 clamp and then the GANP "wedge" can insert itself bridged by a new ATP molecule, thereby preventing renewed RNA binding. This could easily be tested in vitro.

The reviewer highlights a very interesting unknown aspect of the TREX-2 mechanism, which we have now investigated further for the revision. In brief, as the reviewer predicted, we now have evidence that TREX-2 acts on UAP56 after ATP hydrolysis, instead of directly stimulating ATP hydrolysis (Figs 1h, i, 4g, Extended Data Fig. 10f).

Specifically, we asked in what state the ATP nucleotide (ATP or ADP-Pi or ADP) within RNA-clamped UAP56–RNA–ATP complexes is, which we for example used in the in vitro TREX-2 RNA release assay described above (Fig. 4g, Extended Data Fig. 10f). We therefore performed two experiments. First, we carried out UAP56 RNA-clamping experiments in the presence of radioactively labelled 32 P-ATP nucleotide, where the 32 P label was included either in the α or γ -phosphate position of ATP (Fig. 1h). This allowed us to distinguish whether RNA-clamped UAP56 had hydrolyzed ATP. We then crosslinked the nucleotide to UAP56 through UV-light at $\lambda=280$ nm. These data show that RNA-clamped UAP56 complexes contained hydrolyzed ATP, corresponding to either ADP-Pi or ADP. Size exclusion experiments with UAP56, RNA, and ATP or ADP mixtures demonstrated that only RNA+ATP but not RNA+ADP mixtures formed stable complexes with UAP56 (Fig. 1i and ADP controls in Extended Data 3b). The two assays together reveal that stable, RNA-clamped UAP56 complexes contain ADP-Pi. This finding is consistent with previously reports for other DExD-box ATPases, DbpA, EIF4A3 or Mss-116 that ATP is rapidly hydrolyzed upon RNA binding, and that this

converts the complex into the most stable intermediate, the RNA-clamped state that contains ADP-Pi (Henn et al., JMB 2008, Nielsen et. al., RNA 2008, Cao et. al., JMB 2011). We also show through the in vitro immobilization of RNA-clamped UAP56 (Fig. 4g, Extended Data Fig. 10f) and our newly added size exclusion chromatography experiment, that RNA-clamped UAP56–ADP-Pi, but not UAP56+ADP samples, yield stable complexes over the time-course of our experiments (Fig. 1i). Thus we conclude, that TREX-2 acts to release UAP56 from ATP hydrolyzed UAP56–RNA–ADP-Pi complexes. This is also consistent with a prior observation, that UAP56 has a higher affinity for ADP than for ATP (Henn et al., JMB 2008, Nielsen et. al., RNA 2008, Cao et. al., JMB 2011, Kellner et al., NAR 2015). Hence, product release (ADP, Pi) from UAP56 is rate-limiting for the UAP56 ATPase cycle, which is stimulated in our in vitro ATPase assay (Fig. 4f) by TREX-2.

This mechanism for TREX-2 is also consistent with our cryoEM structure, where we observed that the GANP R678 in the ‘wedge loop’ replaces UAP56 F381 in the nucleotide-binding site (Extended Data Fig. 10d). This could explain how GANP would weaken the binding of the nucleotide to UAP56, by competing with the UAP56 RNA-clamped conformation.

We are grateful for this comment, which added new insights into the mechanism of TREX-2 in RNA release from UAP56.

Other points:

4. The title is rather vague and not very descriptive of the paper content.

We have adjusted our manuscript title and provide our rationale for this below. We edited the title to the following:

“An ATP-gated molecular switch orchestrates human messenger RNA export”.

We feel that the major advances of our study revolve around UAP56 as an ATP-gated molecular switch, which is regulated throughout multiple steps of the mRNA export pathway, and led us to propose a novel mechanistic framework for this pathway. In the revision, we have added significant new insights on UAP56 as an ATP-gated RNA clampase, and thus edited the title to highlight this mechanistic ‘ATP-gated’ aspect. We also felt that the inclusion of ‘molecular switch’ points to the mechanistic approaches of our manuscript (structure, biochemistry, and genetics). The new title also fits into the

character limit, 69/75 characters (including spaces). We hope the reviewer agrees with the edited title.

5. Can C-UBM of ALYREF still bind on UAP56 in presence of TREX-2M? This is not visible from the structure figures.

From our experimental UAP56–TREX-2^M cryo-EM structure, the C-UBM should be able to bind. For the revision, we now test this and show in Extended Data Fig. 7b that the C-UBM can bind in an in vitro pulldown assay to the recombinant UAP56–TREX-2^M complex.

Referee #3 (Remarks to the Author):

The manuscript of Hohmann et al. “A molecular switch orchestrates the export of human messenger RNA” provides essential and novel insights into the mRNA export pathway, specifically by which factor mature mRNPs are recognised as such and then handed-over to the NPC via TREX-2. Generally, this manuscript builds on and discusses appropriately previous findings and reconsolidates them in light of the discoveries made here. The dead-box RNA helicase UAP56 turns out to be a molecular switch, which interacts with ALYREF and recruits SARNP to release TREX and allow interaction with the NPC-coupled TREX-2. The insightful final model on mRNP export is based on new and re-analysed cryo-EM data, mass-spectrometry-based proteomics to identify SARNP as additional modulator, pull-down assays, grating-coupled interferometry, and additional biochemical assays. While I think that the provided insights merit publication in Nature, three major concerns need to be addressed prior acceptance.

We are grateful to the reviewer for the careful evaluation of our work and insightful comments, which helped us improve our manuscript. In addition to the new biochemical data and revisions in figures and text, we wish to also highlight our efforts on newly added in vivo experiments, where we established a genetic CRISPR-Cas9 knockout-rescue assay (Extended Data Figs. 3b, 5g, h, 10a, b). These in-cell data show that the interfaces of UAP56 with N-UBM, C-UBM, and UCM and of UAP56 with PCID2 are essential for human cell function.

Firstly, although it might appear as a detail I find the usage of the word licensing in the abstract and later (at altogether four instances) not ideal. To me this implies a certain quality control process. However, I think the work does not describe exactly how UAP56 can distinguish between a correctly assembled mRNP and an immature or improper mRNP. In this regard, it would be also good, if the authors could shortly describe the limitations of the study at the end of the manuscript and refer to literature about what is known about these “licensing” steps and how this could be linked to their findings.

We agree with the referee’s comment on the use of ‘licensing’ and have rephrased all instances to ‘remodeling for mRNA export’. Owing to the central function of UAP56 in the mRNA export pathway, we do indeed also anticipate that UAP56 could be important for mRNP quality control. We have added our speculations on such possible quality control links and future directions beyond our work in the manuscript discussion, Lines

400-415. In brief, our *in silico* protein-protein prediction screen for putative UAP56 interactors revealed additional proteins that have been linked to RNA decay (RBM26, RBM27; Silla et al., NAR 2020), to mRNA biogenesis (NCBP3, RBM33; Gebhardt et al., Nat. Com. 2015; Thomas et al., Genes Dev. 2022), and to unknown functions (LENG8, SAC3D1). We are excited to explore these putative interactors in the future, which might reveal how UAP56 contributes to mRNP quality control. In the context of TREX-2, we are particularly interested to explore the LENG8 protein beyond this current manuscript, which is conserved and essential in human cell lines (<https://depmap.org/portal/>) and a paralog of the TREX-2 subunit GANP. Taken together, we hope the reviewer agrees to the rephrasing of 'licensing' and the expanded discussion on UAP56-linked mRNP quality control.

Secondly, the experiments performed to postulate cooperativity between ALYREF and SARNP are not ideal or enough to be convincing. In general the data behind Figure 2 and related Ext. Data. Figs would need to be extended and cross-validated (See below for more details) especially with regards to statistics. Thirdly, the language needs to be improved and the figure arrangement and referrals. The current arrangement didn't make it easy to review this manuscript. Especially the faulty figure referrals were confusing.

We thank the reviewer for these comments, which helped us to improve the manuscript. Specifically, to address the reviewer's concerns, we (i) improved the figure arrangements and referrals, (ii) rephrased our conclusions on ALYREF and SARNP biochemistry, (iii) included triplicate repeats of all GCI experiments and (iv) carried out novel biochemical and cell-based genetic experiments to address UAP56–ALYREF and –SARNP interactions. For details, please see our detailed responses below.

Regarding specifically ALYREF and SARNP, we rephrased that these two proteins likely synergize with UAP56 and carried out additional supportive experiments (see points 6-9). Our initially reported and new biochemical data for the revision (Extended Data. Fig. 4b, d-i) and new added *in vivo* genetic data (Extended Data. Figs 3b, 5g, h) show that the interfaces between ALYREF and SARNP with UAP56 are important for their essential cellular functions. To remain conservative in our interpretation, we have rephrased manuscript parts to highlight our main discovery more explicitly: to reveal how UAP56 functions in the mRNA export pathway by (i) regulated mRNA-clamping and (ii) regulated interactions with THO, ALYREF, SARNP, and TREX-2. Newly added

biochemical data further support that UAP56 acts as an RNA clampase in the pathway (Fig. 1h, i).

Major concerns:

1. Concerns around Figure 2 (mostly) but also Figure 1 and related Ext. Data Figs: Statistics for all GCI measurements are missing. Are there no replicates? Especially regarding the small differences in most points made. I think this is essential to estimate the validity of the assumptions made. I am not familiar with the technique and it is hard to judge whether this method is accurate enough to allow for only one measurement to make such assumptions. It would be ideal to at least in some instances use an alternative biophysical method to cross-validate.

We apologize for not having been clearer on the technical quality of the GCI experiments. The GCI methodology is highly reproducible and sensitive to determine binding kinetics across a wide range of affinities (nM to μ M). We fully appreciate the reviewers' concerns, and now report in each figure the measured K_{DS} as the average (\pm standard deviations) of three independent experiments for all data points in the manuscript. The small standard deviations show that comparisons of K_{DS} between experiments of related conditions are meaningful. In addition, we note that we performed all individual experiments, which we directly compare in the manuscript (e.g. among Fig. 1g and Extended Data Fig. 3c, or among Fig. 2h, or among Extended Data Fig. 3a or 5a, or among Fig. 4d and Extended Data Fig. 9c), at the same time and on the same chip to minimize experimental variation. We further note that the reported K_{DS} agree well with qualitative in vitro pulldown experiments throughout the manuscript (compare for example Extended Data Fig. 4a, b and 5a for SARNP UCMs or Fig. 4d to Extended Data Fig. 9d for TREX-2).

2. In Ext. Data Fig. 1f, why wasn't C-UBM and N-UBM not used together in an additional experiment?

For technical reasons we did not use C- and N-UBMs together in one GCI experiment: Because of the individually high off-rates and low affinities of the ALYREF C-UBM and N-UBM to UAP56 (K_{DS} in the range of 15-30 μ M), and because the GCI experiment reads out only a change in detected mass on the assay chip, we are unable to distinguish C-UBM from N-UBM binding. However, to address the reviewers' comment

on whether C-UBM and N-UBM can jointly bind to UAP56, and if the two UBMs might influence each other, we carried out new *in vitro* pulldown assays (Extended Data Fig. 4g-h). These assays collectively show that UAP56, in RNA-clamped and non-clamped conditions, can simultaneously engage isolated and non-fused peptides of ALYREF N-UBM, C-UBM, and SARNP UCM.

3. In experiments of Figure 1g, I wonder if it would be possible to do the comparison with ATP or RNA only.

Following the reviewer's suggestion, we have carried out these additional control experiments and measured the THO–UAP56 interaction in the presence of either only ATP or only RNA (Extended Data Fig. 3d). For completeness, we also carried out these experiments in the context of the recombinant 24-subunit THO tetramer, the 12-subunit THO dimer, and a 5-subunit THO monomer, and each experiment in triplicates. We find that neither only ATP or only RNA substantially impact the THO–UAP56 interaction, consistent with the expectation that the simultaneous binding of RNA and ATP to UAP56 are necessary for stable mRNA-clamping by UAP56. In contrast to non-detectable binding/ K_D s for the THO–UAP56 interaction in the presence of ATP+RNA, the THO tetramer bound UAP56 with a K_D of $0.24 \pm 0.1 \mu\text{M}$ in absence of both ATP and RNA, with a K_D of $0.39 \pm 0.7 \mu\text{M}$ with RNA, and $0.34 \pm 0.4 \mu\text{M}$ with ATP. We conclude that RNA and ATP alone have very little effect compared to RNA+ATP together.

4. The experimental difference between Figure 1g and Ext. Data Fig. 3 is not really clear. Recombinant vs. Non-recombinant?

Thank you for pointing this out. All THO complexes used in the GCI assay were recombinantly made. The difference between the Fig. 1g and Extended Data Fig. 3c experiments is the specific, recombinant THO construct used in the assay. The THO complex forms a 24-subunit tetramer that comprises THOC1, -2, -3, -5, -6, and -7. With these experiments, we wanted to assess if THO complex oligomerization (tetramer, dimer, monomer) would influence the measured THO–UAP56 affinity, for example via cooperative binding of multiple UAP56 molecules. In brief, our results showed no evidence for cooperativity. For space reasons, we initially decided to show the experiment with the 12-subunit THO dimer in Extended Data Fig. 3a (now Extended Data Fig. 3c). To better clarify this, we have decided to show only the data for the THO tetramer in the main text Fig. 1g. We moved the THO monomer to the THO dimer data

into Extended Data Fig. 3c. We also added small THO complex icons to these panels, to clearly illustrate which specific THO complex oligomeric was used for which experiment. We also edited this in the main text line 100.

5. Ext. Data Fig. 3b should be done with only ATP and only RNA as additional controls.

As requested, we now performed the in vitro THO–UAP56 release experiment with ATP only or RNA only as additional controls in triplicate (shown now in Extended Data Fig. 3f). We find that neither ATP only nor RNA only lead to substantial release of THO from UAP56.

6. Experiments with SARNPs UCM motifs leave too many questions open, which could be easily answered in this work. In line 164, the authors speculate that multivalency has evolved here to disassemble multivalent tetrameric complexes. This could be tested. Also, they could just increase binding to UAP56, which also could be tested by mutations of residues in UCM2-5 corresponding to R106 in UCM-1 (one at a time and multiple at once). Actually in Ext. Data Fig. 4h the authors test UCM4, but this is not mentioned in main text (If I haven't missed it).

We agree with the reviewer that we cannot exclude that UCM multivalency in SARNP functions to increase the binding of SARNP to UAP56, instead or in addition to promoting multivalent THO tetramer dissociation. To address this comment, we have both rephrased our speculation to include this possibility, lines 202-205, and carried out five new experiments to probe (i) the role of UCM motif multivalency, (ii) possible differences between different UCM motifs (UCM-1-5), and (iii) the in vivo requirement of the SARNP–UAP56 interaction (Fig. 2f, Extended Data Figs 4b, d, 5f-h).

First, we now show that also multivalent SARNP can dissociate THO from endogenous mRNPs using a western blot readout (Extended Data Fig. 5f). Here we used full-length SARNP at concentrations to adjust to molarity of the SARNP UCMs to that of the isolated UCM-1 peptide (1x full-length SARNP compared to 5x UCM-1). This experiment shows that full-length SARNP alone promotes THO release from endogenous mRNPs. To assay mRNPs in this western readout, we probed for the exon junction complex (EJC) subunit EIF4A3, which is among the most abundant mRNP proteins (Pacheco-Fiallos et al., Nature, 2023) (see also response to minor comment #8). In this assay, the THO release efficiency by full-length SARNP is increased in the

presence of the ALYREF N-UBM, approaching THO release efficiency that we achieved with the fusion peptide of the SARNP UCM-1 and ALYREF N-UBM. This indicates that under these conditions the UCM multivalency in SARNP did not noticeably influence the release efficiency of the THO complex from mRNPs.

Second, we now show using recombinant proteins that also the in vitro reconstituted THO–UAP56 complex can be dissociated using full-length SARNP (Fig. 2f). Here, the SARNP concentrations were again adjusted to that of UCM-1 (1x SARNP to 5x UCM-1), showing that the in vitro dissociation of THO from UAP56 is more efficient ($46\pm 9\%$ for SARNP and $75\pm 11\%$ for the isolated UCM-1) for multivalent, full-length SARNP than for the concentration adjusted UCM-1 peptide. This result shows that UCM-multivalent SARNP can better dissociate THO–UAP56 complexes than concentration adjusted UCM-1 peptide, indicating that multivalent binding of UAP56 molecules or increased binding to UAP56 prevent UAP56 re-association with THO.

Third, to ensure our experiments are not influenced by possible differences between the five SARNP UCMs, we now show that the five SARNP UCMs bind to UAP56 with similar affinity. For this, we performed in vitro pulldowns of UAP56 with each individual SARNP UCM peptide (UCM-1, -2, -3, -4, and -5) (Extended Data Fig. 4b). We had previously also shown in a GCI experiment that UCM-4 binds to UAP56 with comparable affinities to UCM-1 (in the 10-14 μM range, Extended Data Fig. 5a). We apologize for omitting a reference to this experiment in the main text, and now mention also the UCM-4 experiment explicitly in line 157-159. Taken together, we conclude that the UCMs act biochemically redundant.

Fourth, to ensure that only the UCMs but not the other structured SARNP domain, the SAP domain, binds UAP56, we show that only the SARNP UCMs but not the SARNP SAP domain bind UAP56 in an in vitro pulldown experiment using a truncated or all-UCM mutated SARNP (Extended Data Fig. 4d). We further show that full-length SARNP does not appreciably bind to RNA (Extended Data Fig. 5j). Together, the data suggest that SARNP likely acts solely through its UCMs that bind UAP56, and that SARNP does not noticeably bind to RNA in our assays.

Fifth, we established and carried out in vivo experiments, whereby we show that the acute CRISPR-Cas9 knockout of the two paralogs of UAP56 (DDX39A and DDX39B) can be rescued by wild-type UAP56 (DDX39B), but not by UAP56 carrying the UCM-

binding mutant (D283R), which we had previously also characterized in vitro (Extended Data Fig. 5g).

Generally, we wish to note that our in vitro biochemical experiments can to our knowledge not distinguish the functional consequences of mono- or multi-valent UCM–UAP56 interactions. That is because the physiological effect (increased binding over multivalent disassembly) would depend on the mRNP-local, sub-cellular concentrations of SARNP and UAP56.

We wish to note that to avoid complications in our in vitro biochemical assays, owing to UCM multi-valency, we chose to focus our studies largely on one UCM of SARNP (UCM-1). The newly added experiments indicate our detailed analysis of UCM-1 will extend to all other UCMs, where for UCM-1 we demonstrated that it (a) binds to UAP56, (b) promotes mRNA clamping by UAP56, (c) promotes the dissociation of UAP56 from THO in vitro, and (d) promotes the dissociation of endogenous mRNPs from THO. This is consistent with the high degree of UCM amino-acid conservation (Extended Data Fig. 4c). We are thankful to the referee’s comment, based on which we rephrased our speculation on the possible functions of multivalent SARNP.

7. Concerning cooperativity: Why has C-UBM been ignored with respect to cooperativity to UCMs? Could be allosteric? I understand that there are challenges with low-affinity complexes and that fusing two systems helps. However, this is also very artificial. The authors should at least show an experiment where they add both, N-UBM (possibly also C-UBM) and UCMs as separate constructs (not fused to UAP56) (UCM-1, the other UCMs and UCM-1-5 or derivatives). These are all important control experiments. Especially since differences in affinities in Figure 2h do not infer “cooperativity” but possibly synergistic binding at best. Especially here statistics (replicates) are needed. A two-fold increase in affinity is not cooperative. Controls should still be made with unfused UBM/UCMs.

Following the reviewers suggesting we further investigated the binding between SARNP, ALYREF, and UAP56 in biochemical assays. For technical reasons of low affinity (in the >10 μ M range), we could not carry out UAP56 binding experiments with multiple isolated peptides using the GCI method. We have however now carried out several new in vitro pulldown experiments, which altogether show that the individual peptides of the ALYREF N-UBM, C-UBM, and SARNP UCM can all simultaneously

engage UAP56 with or without RNA (Extended Data Fig. 4g-j). In these experiments we did not observe any synergistic effects of the ALYREF C-UBM on UAP56 interactions with the ALYREF N-UBM, SARNP UCM, or RNA. In addition, as recommended by the reviewer, we also rephrased the effect of N-UBM and UCM-binding to UAP56 as ‘synergistic’ instead of cooperative.

Specifically, for the revision, we now additionally show that full-length SARNP with 5xUCM motifs and full-length ALYREF can simultaneously bind UAP56 in vitro (Extended Data Fig. 4e,f).

Together, our data illustrate that all peptides are compatible with each other to bind UAP56 at the same time, consistent with our cryo-EM data and AlphaFold2 predictions (Fig. 1b,c, 2d).

We now also show in vitro that the non-fused peptides of the SARNP UCM-1 and of the ALYREF N-UBM act synergistically to promote the dissociation of recombinant THO from UAP56. While isolated UCM-1 and N-UBM respectively dissociate THO to $74\pm 10\%$ and $72\pm 11\%$, when combined, the two isolated non-fused peptides dissociate THO to $52\pm 9\%$ (Fig. 2f). We note that also the isolated peptides of UCM-1 and N-UBM precipitate at concentrations above what we used in the assay. This limits the maximal concentrations we could use in the experiment, leaving the possibility open, that at higher concentrations even more efficient THO release might be observed. However, this does not change our conclusion that in this assay the isolated UCM-1 and N-UBM peptides act synergistically.

As discussed in our response to point #3, we now also show the average binding kinetics from three experiments for all GCI data. Thus, while we see only a two-fold difference between the RNA binding of UAP56–UCM-1(fused)-ATP in comparison to UAP56–N-UBM–UCM-1(fused)–ATP (K_{Ds} of $0.27\pm 0.06 \mu\text{M}$ versus to $0.12\pm 0.03 \mu\text{M}$), the synergistic effect of UCM-1 and N-UBM is significant in the GCI experiment and consistent with the corresponding in vitro pulldown scenario (Fig. 2h, Extended Data Fig. 5j).

Taken together with the newly included data, and as suggested by the reviewer, we rephrased all manuscript mentions of ‘cooperative’ to ‘synergistic’ binding. We hope the referee agrees to these changes.

8. Ext. Data Fig. 5a does not help as it is rather ambiguous, while lane 9 has a clear intensity increase compared to lane 7, lane 12 vs. lane 11 does not. Or is this rather seen in direct context to Figure 2h, where RNA binding is tested (missing in Ext. Data Fig. 5?). If so, then it is not clear. Why not using GCI for experiments in Ext. Data Fig. 5a?

In the experiment in Extended Data Fig. 5c (former ED Fig. 5a) we explored two biochemical aspects on UAP56: First, we showed that at the technical level the peptides fused in the respective UAP56–peptide fusion proteins occupied the intended binding sites. Second, we observed synergistic binding of the N-UBM and UCM to UAP56. As the reviewer pointed out, we observed synergistic binding between the UAP56–UCM fusion to the N-UBM peptide (Lane 7 versus 9), but not detectably in the inverse experiment with an UAP56–N-UBM fusion to the UCM peptide (lane 11 versus 12). We speculate that in this inverse experiment, we may be less sensitive to detecting such synergistic binding, because UAP56 has different affinities to the isolated UCM peptide compared to the isolated N-UBM peptide. Supported by the following additional evidence, we speculate in the revised manuscript text synergistic binding of N-UBM and UCM peptides could be important in the context of mRNA or mRNP complexes:

- *Using GCI experiments, we previously observed synergistic binding of UAP56 to RNA in the UAP56-UCM1-N-UBM fusion in Fig. 2h.*
- *For the revision, we now show that the isolated, non-fused N-UBM and UCM-1 peptides act synergistically to aid in the dissociation of recombinant THO–UAP56 complexes, beyond the effect of each peptide alone (Fig. 2f).*
- *For the revision, we also show that non-fused recombinant SARNP acts synergistically with the isolated ALYREF N-UBM peptide to aid the dissociation of THO from native TREX-mRNPs.*
- *Beyond these data, we also showed that all peptides (N-UBM, C-UBM, and UCM) can simultaneously bind to UAP56 (Extended Data Fig. 4g-j), supporting the formation of a UAP56–ALYREF–SARNP–RNA-ATP complex.*

Generally, we find a further exploration of interactions of N-UBM, C-UBM and UCM with UAP56 highly interesting. While significantly beyond our current manuscript, we think that a deeper molecular understanding on these interactions and possible allosteric effects, will require experimental atomic X-ray crystallography or cryo-EM structural data of the 49 kDa UAP56 (45 kDa of the structured RecA1 and RecA2 lobes) with

combinations of these peptides, with and without RNA and ATP. For the revision, we hope the additional data and rephrasing of the observed effect to 'synergistic' address the referee's concerns.

Figure 2f also lacks the comparison with both peptides (UBM (N and C?) and UCM) added non-fused. Also, should the band for MBP-UAP56 not keep the same intensity over all lanes? It seems to also decrease with the THOC bands.

We now show in Fig. 2f that the combined addition of the isolated N-UBM and isolated UCM-1 peptides in the *in vitro* THO-UAP56 release assay further increases the THO release efficiency: N-UBM alone ($74\pm 10\%$), UCM-1 alone ($72\pm 11\%$), isolated N-UBM and UCM-1 together ($52\pm 9\%$). We note that for technical reasons, we likely do not achieve saturation binding of the individual peptides to UAP56 in this assay, owing to the low affinities and low solubility at the required peptide concentrations. Hence, we likely underestimate the full effects of N-UBM and UCM-1, when inside complete mRNPs, where multivalency of ALYREF molecules and SARNP molecules will also contribute to higher local peptide concentrations. Since the C-UBM does not bind near the UCM-binding site, and because the C-UBM does not influence N-UBM or UCM binding (see the newly added systematic *in vitro* pulldown experiments of N-UBM, C-UBM, UCM-1 to UAP56, Extended Data Fig. 4g-j), we did not include it in the THO-UAP56 release assay.

Regarding the MBP-UAP56 levels. For, to us, unknown technical reasons we indeed reproducibly observed a slight decrease in immobilized MBP-UAP56 with increasing THO release (Fig. 2f). To account for this loss, we normalized UAP56 levels in the respective condition prior to the quantification of THO levels. Hence, differences in UAP56 immobilization levels do not affect the reported quantification of THO release and our conclusions. To clarify this technical aspect, we added that we normalized the lanes by UAP56 levels prior to THO quantification in the Fig. 2f figure legend.

9. All this leaves me wondering about the role of the SAP domain in SARNP and the RRM domain in ALYREF. For the latter I guess more is known but it could be mentioned somewhere. However, both could further increase cooperative effects, possibly binding RNA and reaching into the mRNP granule?

The reviewer raises interesting points, which we now address in the main text for both the ALYREF RRM and SARNP SAP domain and with an experiment for the SARNP SAP domain.

We apologize for the omission on ALYREF RRM details. Indeed, available data show that neither the human ALYREF nor the yeast Yra1 RRM bind to RNA (Zenklusen et al., MCB, 2001; Pacheco-Fiallos et al., Nature, 2023). Instead, it was shown by cryo-EM for the human ALYREF RRM domain, that it binds in a mutually exclusive manner to either the mRNP-bound Cap-binding complex (Clarke et al., eLife 2024) or the mRNP-bound exon junction complex, EJC (Pacheco-Fiallos et al., Nature, 2023). Hence the RRM most likely serves for ALYREF recruitment, and additional interactors beyond CBC and EJC may exist. We highlight these details in line 60-61.

*The function of the SAP domain is currently unknown and will be of interest for future studies. To test if the SARNP SAP domain may affect the UCM–UAP56 interaction, we now tested in Extended Data Fig. 4d that the binding of recombinant full-length SARNP or of SARNP lacking the SAP domain with UAP56 are indistinguishable from each other. We also note that the SAP domain, but not UCMs, are absent from plant SARNP (*Arabidopsis thaliana*, Mos11), suggesting that the SAP domain may be dispensable for SARNP’s main mRNA export activity.*

Minor concerns:

1. Why not call UAP56 an RNA helicase, but an ATPase. RNA helicases have ATPase function, but I would assume that UAP56 also has RNA helicase function in this complex at a certain step of this process? (Which would be interesting to follow-up upon or discuss shortly).

Thank you for this comment. Indeed, UAP56 is an RNA-dependent DExD-box ATPase. For the reasons describe below, we prefer to refer to UAP56 as an RNA-dependent ATPase or ‘RNA clampase’ since its biochemical activities are distinct from those expected of the processive, translocating DExH-box RNA ATPases.

The DExD-box protein family are non-processive ATPase-dependent RNA binders, in contrast to the DExH-box protein family of RNA-dependent ATPases, which are processive.

DExD-box family proteins are biochemically able to clamp onto RNA in the presence of ATP, but they cannot act as processive helicases because the RNA substrate is lost upon unclamping, which is coupled to ADP-Pi product release. This RNA binding is facilitated by their two RecA lobes. Historically, DExD-box family proteins had been termed 'RNA helicases', because, when sufficiently short double stranded RNA substrates are used, their RNA clamping cycle can displace one of the two RNA strands (Yang et al., Mol Cell, 2007).

In contrast, DExH-box family proteins harbour RNA binding domains in addition to their two RecA lobes, allowing them to hold on to the RNA substrate during the ATPase cycle. This yields a processive RNA translocase that can in vitro unwind long (beyond the RNA-binding footprint of the protein) double-stranded RNAs (see for example DHX15, a DExH-box ATPase in Vorlaender et al., Nature, 2024).

Thus, while a single ATPase cycle of any DExD-box proteins will involve only clamping and unclamping from RNA, the life-time of each step in the ATPase cycle can vary widely. Relevant to our work in particular is the apparently long life-time of UAP56 clamped onto RNA. For the revision, we now show that RNA-clamped UAP56 is stable during size exclusion chromatography experiments (Fig. 1i), a stringent assay for long-term RNA binding. We further show that RNA-clamped UAP56 complexes contain hydrolysed ATP (ADP-Pi) (Fig. 1h, i, Extended Data Fig. 3e), consistent with evidence that this is the highest affinity state of DExD-box ATPases (Henn et al., JMB 2008, Nielsen et. al., RNA 2008, Cao et. al., JMB 2011). We now also show that bead-immobilized RNA remains stably clamped by UAP56, unless incubated with TREX-2 (Fig. 4g, Extended Data Fig. 10f). These new data together suggest that RNA-clamped UAP56 is a stable complex and indicated that the function of TREX-2 is to promote product release (ADP-Pi) from UAP56 instead of promoting ATP hydrolysis per se.

The stability of the UAP56-RNA complex suggests that UAP56 does not act through a processive, translocating activity in the export pathway, but instead acts by RNA-clamping. We would thus like to keep the term of either DExD-box ATPase or RNA clampase in the manuscript. We added a short discussion on this in the manuscript,

where we discuss the new revision data on UAP56 acting as an RNA clampase (Lines 106-114, 416-425).

2. The “newly-revised” in caption to Figure 1b is confusing at this stage. Leaves the reader wondering. It should be mentioned somehow in the main text before referring to this figure to make this clear.

We apologize for the confusion and have edited the Figure 1b legend to clarify that the meaning of ‘revised’ is the result of a re-analysis of cryo-EM single particle data of the endogenous TREX–mRNP complex and of the dataset of a reconstituted TREX–EJC complex.

3. The EM density for ALYREF N-UBM should be shown in Ext. Fig. 1 to illustrate why it has been missed and how good this helix fits as done for other structural elements. Compare old vs. new data. I think Ext. Data Fig. 2g shows something along the lines but only for the new data?

Thank you for the comment, we added a density image to Extended Data Fig. 1b, as we had it previously already in Extended Data Fig. 2g. This density had been missed in earlier studies to previously lower map-quality. This density however improved with a new re-analysis of the data.

4. The sequence alignment of ALYREF N- and C- UBM should be added to Figure 1c (Enough space left) to better illustrate the point made in lines 65-70.

We now include multiple sequence alignments for the ALYREF N-UBM and C-UBM in Extended Data Fig. 1a, since we did not fit into to the main text figures owing to the newly added data in Fig. 1.

5. There is enough space to show the actual residue contacts between C-UBM-UAP56 and N-UBM-UAP56. I realize that the EM density resolution is low at especially these regions. However, the authors made mutations to ensure that these interactions are relevant. In light of the sentence that mentions that it has been suggested earlier, that N-UBM and C-UBM could bind the same site but with higher affinity due to avidity it would be a useful illustration.

Thank you for the suggestion. We now highlight critical interface residues in Extended Data Fig. 1c, owing to space constraints because of newly added data in Fig. 1.

6. Line 342: DNA-tethered? I do not understand this. Is this to consolidate with the before mentioned hypothesis by Günter Blobel? To me it seems that DNA does not need to be involved?

The reviewer is correct, the original hypothesis by Günter Blobel posited that a DNA-tethering element would be critical, but our data allows the speculation that that it is chromatin-associated nascent mRNA would be the tether. We rephrased this sentence clarify this better.

7. Linked to major concern 3. Not clear if in experiment to Fig. 1g cell extract is used.

We apologize for the lack of clarity. We used recombinant THO complexes and recombinant UAP56 for all GCI experiments, which we now clearly stated in the main text and Fig. 1g figure legend, lines 100 and 793.

8. Ext. Data Fig. 5b: Why using eIF4A3? It is nowhere explained.

We used EIF4A3 as a proxy for mRNPs, since EIF4A3 is among the most abundant proteins in mRNPs according to mass spectrometry data of TREX-mRNPs (see also Pacheco-Fiallos, Nature, 2023). We updated the Extended Data Fig. 5e (formerly ED Fig. 5b) legend to clarify this.

9. Ext. Data Fig. 2f, please indicate the domains with labels to help orient the reader.

Done.

10. Ext. Data Fig. 8c. It would be useful to show the density for SEM1 also.

Done.

11. Ext. Data Fig. 9f. Show EM-densities also.

We thank the reviewer for this suggestion and now include the cryoEM density for the nucleotide base binding residues in Ext. Data Fig. 10d (formerly ED Fig. 9f).

12. Language (please check for more than listed here):

Line 46: “we here”, sounds clumsy.

Done.

Line 54 and 55: please change “yeast Sub2” to “yeast: Sub2” for better reading flow

Done.

Line 69: “AlphFold2” should be AlphaFold2

Done.

Line 626: Caption to Fig. 1a is confusing. Now it means that TREX-mRNP must disassemble TREX. Shouldn't it be: “...of a TREX-mRNO complex, which prior to mRNP export must disassemble and the mRNP remodelled via an unknown mechanism.”?

Thank you for pointing this out, we changed the text to clarify this.

Line 633: RecA2 and RecA2 lobes. One of them should be RecA1.

We corrected the error.

Line 640 and 642: I find the usage of the word coordinated not ideal. It is rather an induction of conformational change to reorient the two domains.

We rephrased the text accordingly.

13. Figure referral (should be also checked carefully beyond from which is listed here):

Sometimes a later Figure panel is referred before an earlier one, e.g.:

Line 58, Figure 1d before Figure 1b,c

Line 96, Ext. Data Fig. 3b is referred to before 3a

Line 1236, referral to Figure 1d is wrong. It should be 1g I assume.

Line 629-630, referral to left and middle panel in 1b and right panel in c (c has only one panel) is confusing.

Line 233-236: Figure arrangement should be improved to prevent the reader from too much jumping in between.

Thank you for bringing this to our attention, we have carefully gone through the figure referrals and updated the relevant main text parts, and remodeled figures to reflect the order in which they appear in the text (this particularly affected Extended Data Figs 1, 3, 9, and 10).

List of responses to reviewer comments

“An ATP-gated molecular switch orchestrates human messenger RNA export”

(Ulrich Hohmann, Max Graf, Ulla Schellhaas, Belén Pacheco-Fiallos, Laura Fin, László Tirián, Dominik Handler, Alex W. Phillips, Daria Riabov-Bassat, Thomas Pühringer, Michael-Florian Szalay, Julius Brennecke, and Clemens Plaschka)

Nature Manuscript 2024-02-02833

Reviewer comments are in *blue*.

Responses are in *black*.

Referees' comments:

Referee #1 (Remarks to the Author):

The manuscript presents intriguing work on the role of the DExD ATPase UAP56 in various aspects of mRNA export, particularly through its interactions with multiple factors during assembly and disassembly. The study provides valuable insights into the regulation of mRNA export and gene expression. In the revised manuscript, the authors have addressed most of my previous concerns. However, a few issues remain unresolved, and I have additional questions regarding the newly added data.

We thank the reviewer for the critical assessment of the newly added data. The constructive comments helped us to clarify the outstanding concerns, and their incorporation along with the other suggested changes further strengthened the revised manuscript.

1. In lines 99-105, the GCI data demonstrate that THO efficiently disassembles from UAP56 in the presence of ATP and RNA, suggesting that THO dissociates from UAP56 after binding RNA. However, in lines 119-121, the authors report only inefficient disassembly of TREX upon the addition of ATP and RNA. This result seems contradictory. Could the authors clarify the reasons for these differing outcomes in what appears to be similar experimental conditions?

We apologize for not being sufficiently clear on this in the manuscript. The perceived contradiction stems from the different experimental designs of the two assays, that each had a different aim.

In the GCI experiment (Fig. 1 and ED Fig. 3c,d), we assess whether the THO complex can bind to a preformed UAP56–ATP–RNA complex or to UAP56 alone (we also tested UAP56 with only ATP or with only RNA added). The results show that preformed UAP56–ATP–RNA complexes, where UAP56 is in the RNA-clamped state, do not show any measurable binding affinity to the THO complex. In contrast, UAP56 alone or UAP56 preincubated with only ATP or with only RNA, all of which are not RNA-clamped, can readily bind the THO complex. Based on these results we hypothesized that, after THO-UAP56 is recruited to the mRNP via UBM motifs, RNA clamping by UAP56 results in the release of THO from the mRNP.

In contrast, in the in vitro disassembly assay (ED Fig. 3f), we used a different experimental setup: here, THO–UAP56 complexes were preformed on beads and then challenged with RNA and ATP afterwards. This assay demonstrated that disassembly of preformed THO–UAP56 complexes is inefficient under these conditions. Thus, RNA and ATP alone are insufficient to induce UAP56 to transition to the RNA-clamped conformation, when it is bound in preformed THO–UAP56 complexes. This observation led us to hypothesize that additional release factor(s) may aid this process in cells (e.g. SARNP), where we had previously shown that THO–UAP56–mRNPs complexes in which UAP56 is not RNA-clamped (‘TREX–mRNPs’) exist (Pacheco-Fiallos et al., 2023). This assay may thus more closely mirror the native context, which we further tested in our manuscript through the analysis of THO release from endogenously purified THO–UAP56–mRNPs (Fig. 2f,g).

We have edited the main manuscript text to better clarify the difference between the two experimental setups, in lines 99-103, 106, 122.

2. In the section title (line 127), they state, “SARNP is a multivalent TREX disassembly factor.” However, the data presented to support this conclusion are not entirely convincing. In Figure 2f, the authors examine the effect of different proteins/peptides on the UAP56-THO interaction. While the western blot data were quantified, it appears that different amounts of UAP56 were pulled down under different conditions. Additionally, less THO was detected when less UAP56 was pulled down, which undermines the reliability of the data. More importantly, when the authors used a degron system to induce rapid degradation of SARNP, the UAP56-THO interaction was not affected. This result does not support a role for SARNP in promoting TREX disassembly in vivo.

We thank the reviewer for the thoughtful feedback and appreciate the opportunity to address these points, which improved the manuscript.

The reviewer raises a concern regarding the varying UAP56 levels and their potential impact on the quantification of THO release in Fig. 2f. We indeed reproducibly observe a slight decrease in immobilized MBP-UAP56 levels as the efficiency of THO release increases. While the reasons for this are unclear to us, they appear to be technical in nature. We have accounted for variations in UAP56 immobilization in the original manuscript, by normalizing UAP56 levels across these conditions before quantifying

THO levels and thus release efficiency. This ensures that differences in UAP56 immobilization did not affect the reported results or conclusions. We also describe this normalization procedure in the legend of Fig. 2f. We hope this additional information resolves the reviewer's concerns.

Regarding the *in vivo* degron depletion experiment for SARNP. We appreciate the reviewer's concerns and agree that we should better contextualize the interpretation. For the interpretation of this experiment, it is important to note that at steady state, almost all cellular *THO* complexes are bound within TREX–mRNP complexes (i.e., *THO*–UAP56–mRNPs), which we showed in our previous work (Pacheco-Fiallos et al., 2023 in Extended Data Fig. 10b) and reproduced below.

Pacheco-Fiallos et al., 2023: Extended Data Fig. 10b:

Edited here to also indicate the THO–UAP56 peak.

Free *THO* complexes are nearly undetectable in this and similar experiments. Moreover, the *THO* complex subunits (*THOC1-7*) are approximately 40-fold less abundant than UAP56 in human cellular proteomics data (Schwanhäusser et al., 2011; Heath, Viphakone and Wilson, 2016). Additionally, the *THO* complex is among the least abundant nucleoplasmic mRNA export proteins, especially when compared to UAP56, *ALYREF* or *SARNP* (Schwanhäusser et al., 2011; Heath, Viphakone and Wilson, 2016). These findings indicate that any small amounts of free *THO* complexes are likely bound to UAP56 at cellular steady state.

Considering these data, the total levels of THO–UAP56 complexes cannot increase further, following SARNP depletion in our experiment, because the THO complex pool is already saturated with UAP56 at steady state. To ensure this important point is clear, we have edited the manuscript (lines 214-217) to emphasize this and clarify why no change in THO–UAP56 levels is observed in our experiment.

As a side note, the fact that nearly all THO complexes are associated with mRNPs at steady state underscores the importance of regulated and efficient dissociation of THO from these complexes, to ensure THO’s availability for newly synthesized mRNPs. When considered with the rest of our data, the in vivo SARNP degron experiment remains consistent with the proposed model.

3. The title of this section should be revised to “SARNP Stabilizes UAP56–RNA Complexes in Vitro.” First, the data presented are entirely in vitro, and this should be reflected in the section title. Second, the role of SARNP in stabilizing UAP56 binding to RNA seems limited, as its depletion only affects the nuclear export of a small population of RNAs. This suggests that SARNP may not play a central role in UAP56 binding to RNAs, and the section title should better reflect the data presented.

We appreciate the reviewer’s suggestion regarding the section title and have revised it to “SARNP Stabilizes UAP56–RNA Complexes In Vitro”. Below, we provide additional context to clarify SARNP’s role in mRNA export and its functional relevance.

Previous work by the groups of Beatriz Fontoura and Yi Ren (Xie et al., 2023) showed that siRNA-mediated depletion of SARNP results in the nuclear accumulation of poly(A)-positive RNA—a phenotype shared by other key mRNA export factors such as THO complex components (Chi et al., 2013), UAP56 (Hautbergue et al., 2009; Yamazaki et al., 2010; Saguez et al., 2013), ALYREF (Hautbergue et al., 2009; Jani et al., 2012; Chi et al., 2013), and TREX–2 (Wickramasinghe et al., 2010; Jani et al., 2012; Umlauf et al., 2013). In addition, SARNP is classified as a “common essential” gene in the human DepMap project (<https://depmap.org/portal/gene/SARNP>), highlighting its functional importance in human cells. In their study, Xie et al. speculated that SARNP binds to UAP56 in a mutually exclusive manner with THO. Our manuscript provides direct experimental evidence by showing that SARNP promotes the disassembly of both in vitro reconstituted THO–UAP56 complexes and endogenously purified human TREX–

mRNP complexes, yielding SARNP–UAP56–RNA complexes. These findings demonstrate SARNP’s activity in TREX disassembly and provide direct insight into its molecular function.

Functional redundancy of SARNP: While SARNP is highly conserved across eukaryotes, its function in human cells appears to be partially redundant with other UCM- or UCM-like domain-containing proteins. For example, we report a predicted UCM in the mRNA biogenesis factor NCBP3 (Extended Data Fig. 6j). Additionally, RBM33, which facilitates the nuclear export of intronless RNAs, is predicted to bind UAP56 at a site overlapping with the UCM-binding interface (Extended Data Fig. 10g-j). These observations suggest that, alongside SARNP, other proteins such as NCBP3 and RBM33 likely act as alternative TREX disassembly factors or contribute to UAP56–RNA clamping. Further studies will be necessary to dissect the biochemical and functional redundancies and specificities among these factors.

The report by Xie et al. that a subset of mRNAs depends on SARNP for their nuclear export would be consistent with a scenario where TREX disassembly involves partially redundant factors. However, we wish to note that Xie et al. used siRNA-mediated knockdown, which reduced SARNP levels to approximately 16% over 48 hours (Fig. 6B in Xie et al., 2023). Incomplete protein depletion over this extended experiment time could leave residual SARNP protein and/or allow for the buffering of immediate and direct cellular effects, which might obscure the full extent of SARNP’s role in mRNA export. To address these open questions, future studies are needed that combine highly efficient and acute depletion strategies (e.g., degron-based approaches) with the metabolic labeling of nascent RNA to quantify the contributions of SARNP and other factors to mRNA export. Such studies, which are beyond the scope of this manuscript, will help elucidate whether additional factors, such as for instance NCBP3 and RBM33, act redundantly or in parallel with SARNP in TREX disassembly and mRNP biogenesis dynamics.

4. The authors provide three lines of evidence to support the functional importance of UAP56-TREX2. However, it remains questionable whether these data can be used to conclusively support this function, as alternative explanations exist. In lines 315-318, the authors show that truncation of the UAP56 N-terminal domain (NTD), which interacts with TREX2 in the structure, impairs interaction with TREX2 and causes a growth defect

in K562 cells. However, this mutant also exhibits reduced interaction with SARNP and likely other UAP56 partners, such as ALYREF. Therefore, this result cannot be considered strong evidence specifically for the role of UAP56-TREX2. Similarly, the PCID2 mutant used to assess UAP56-PCID2 functionality also shows significantly reduced interaction with GANP, raising further concerns about the function of UAP56-TREX2.

We thank the reviewer for the careful analysis of our revision experiments. As pointed out by the reviewer we show by IP-western blot that deletion of the UAP56 NTD reduces the association with TREX-2, while it also reduces the levels of SARNP (ED Fig. 5h). While we currently cannot explain why SARNP levels are reduced in this experiment, the reduced TREX-2 binding is consistent with our model. We additionally showed that mutating the UAP56 NTD-binding residues in the PCID2 protein reduces co-immunoprecipitated UAP56 to background levels. This PCID2 mutant still forms a complex with GANP, although GANP levels are reduced in the experiment. At this point, we can only speculate that this may be due to reduced stability of the TREX-2 complex (specifically GANP) in this PCID2 mutant (ED Fig. 10b).

We would like to point out that these IP-western blot experiments are one of a series of orthogonal experiments, ranging from in vitro interaction studies, enzymatic assays, structural biology to in cell experiments, which collectively are consistent and led us to propose that the UAP56–TREX-2 interaction is an important element in mRNA export. In brief we:

- *show biochemically that UAP56 and TREX-2 form a complex in vitro (Fig. 3d, ED Fig. 7a,b);*
- *solve the cryo-EM structure of UAP56–TREX-2^M to reveal the mode of interaction (Figs 3e, 4c, ED Figs 7,8,9a-b). Our structure further shows that the UAP56 interacting patches in both PCID2 and GANP are highly conserved across species (ED Fig. 9b);*
- *reveal that UAP56 interacting surfaces in both PCID2 and GANP are highly conserved;*
- *show that TREX-2 stimulates the ATPase activity of UAP56 (Fig. 4f, ED Fig. 10e);*
- *demonstrate that TREX-2 can unclamp UAP56 from RNA in vitro (Fig. 4g, ED Fig. 10f);*

- use an in-cell RNA tethering assay to demonstrate that these newly observed interfaces between UAP56–TREX-2 are important for the stimulation of mRNA export (Fig. 4e, ED Fig. 9f). In this experiment, we had also tested a UAP56 mutant based on the UAP56–TREX-2 structure, where the interaction of UAP56 RecA1 and PCID2 is perturbed. The reduction in the export promoting effect of this mutant is as severe as the UAP56 Δ NTD mutant, providing additional evidence for the UAP56–TREX-2 interaction independent of the UAP56 NTD–TREX-2 interface.
- show in cells that perturbing the UAP56–TREX-2 interaction leads to severe growth defects (ED Figs 5g, 10a);
- and show that mutations in the UAP56–TREX-2 interface perturb complex formation in cells (as detailed above, ED Figs 5h, 10b)

We find the enzymatic coupling between UAP56 and TREX-2 particularly striking. We show that UAP56 can form RNA-clamped complexes that are stable for prolonged periods (Fig. 1i), which would exceed the time required for the nuclear export of an average mRNA (Mor et al., 2010; Ietswaart et al., 2024). These UAP56–ADP–Pi–RNA complexes cannot bind the THO complex anymore (Fig. 1g). This suggests that to continue to engage actively in nuclear mRNA export, UAP56 would need to unclamp from RNA, a process that is catalyzed by TREX-2.

In addition, published data in the literature (which we have cited in the relevant manuscript parts) support the *in vivo* relevance of the UAP56 NTD and of the UAP56-binding interfaces in TREX-2. For example, Torben Heick Jensen’s lab showed that deleting or mutating the UAP56 NTD leads to mRNA export defects in yeast (Saguez et al., 2013). The groups of Alwin Köhler and Murray Stewart demonstrated in separate manuscripts that mutating residues in PCID2 or GANP, which we observed to be at the interface of the UAP56 NTD and TREX-2, cause mRNA export defects (Ellisdon et al., 2012; Schneider et al., 2015). In the latter study the authors also observed mRNA export defects for GANP mutations, which we can now rationalize, since they would perturb the positioning of the wedge loop in TREX-2 and thus impair TREX-2’s ability to promote unclamping of UAP56. It had been further shown that depletion of the TREX-2 subunit SEM1 leads to the accumulation of UAP56 on poly-A-RNA in yeast (Wilmes et al., 2008), in line with our finding that TREX-2 removes UAP56 from RNA.

Taken together, our data provides strong support for a direct mechanistic link between UAP56 and TREX-2 and we currently cannot envision alternative explanations that would be congruent with our data and the previous literature.

5. While this work focuses on the mechanism of mRNA export, no experiments were conducted to directly examine the nuclear export of RNAs. The authors rely on protein tethering-GFP intensity or growth curves to infer changes in mRNA export. However, these are indirect measures. To provide more direct evidence, the authors could investigate changes in the nucleocytoplasmic distribution of polyA⁺ RNAs, which would offer a better reflection of mRNA export.

We thank the reviewer for this thoughtful comment. However, we would like to emphasize that our work builds upon decades of research in the mRNA export field, which has previously established that the proteins studied here are involved in mRNA export. Specifically:

1. *All proteins we discussed in our model (THO, UAP56, ALYREF, SARNP, TREX-2) are highly conserved from yeast to humans, and their essential roles in mRNA export have been extensively documented in the literature, for instance in: (Portman, O'Connor and Dreyfuss, 1997; Piruat and Aguilera, 1998; Gatfield et al., 2001; Jensen et al., 2001; Sträßer and Hurt, 2001; Fischer et al., 2002; Sträßer et al., 2002; Zenklusen et al., 2002; Masuda et al., 2005; Dufu et al., 2010; Wickramasinghe et al., 2010; Jani et al., 2012; Saguez et al., 2013; Ren, Schmiede and Blobel, 2017; Pacheco-Fiallos et al., 2023; Xie et al., 2023).*
2. *The knockout or the depletion of each protein implicated in our model has been shown to result in mRNA export defects, as assessed by poly-A⁺ or single-molecule RNA FISH in previous studies. For example:*
 - *The accumulation of poly-A RNA in the nucleus had been shown for the THO complex through the knockout of yeast THO subunits Hpr1, Tho2, Mft1 or Thp2 (human THOC1, -2, -5, -7) (Sträßer et al., 2002) or shRNA-mediated depletion of each subunit of the THO complex in human 293FT cells (Chi et al., 2013).*
 - *For UAP56, nuclear poly-A RNA accumulation was shown upon dsRNA-mediated depletion in Drosophila S2 cells (Gatfield et al., 2001), in yeast for thermo sensitive UAP56 alleles at the restrictive temperature (Jensen et al., 2001; Sträßer and Hurt, 2001) as well as for knockout and mutant*

strains (Saguez et al., 2013), for UAP56 mutant alleles in *Drosophila* egg chambers (Meignin and Davis, 2008), upon inducible miRNA mediated depletion in human FLP-In T-REX 293 cells (Hautbergue et al., 2009) or upon siRNA mediated depletion in human Hela cells (Yamazaki et al., 2010).

- In the case of ALYREF, nuclear poly-A RNA accumulation has been observed in thermo sensitive yeast alleles at the restrictive temperature (Sträßer and Hurt, 2000) as well as upon inducible miRNA mediated depletion in human FLP-In T-REX 293 cells (Hautbergue et al., 2009), shRNA-mediated depletion in human 293FT cells (Chi et al., 2013) or siRNA mediated depletion in human HCT116 cells (Jani et al., 2012).
- For the TREX-2 complex, poly-A RNA accumulation had been reported in yeast upon the knockout of PCID2 (*ScThp1*), GANP (*ScSac3*) and SEM1 (Fischer et al., 2002; Wilmes et al., 2008) or in addition also for PCID2 and GANP point mutants (see also below) (Ellisdon et al., 2012; Schneider et al., 2015), for the siRNA-mediated depletion of GANP (*Xmas-2*) or ENY2 (*DmE(y)2*) in *Drosophila* S2 cells (Kurshakova et al., 2007) and for the depletion of the subunits GANP, PCID2, ENY2 or CENTRIN2 using siRNA in human HCT116 cells (Wickramasinghe et al., 2010; Jani et al., 2012; Umlauf et al., 2013),

3. Most relevant to the reviewer's specific comment on the UAP56–TREX-2 interaction: several key protein-protein interfaces central to the UAP56–TREX-2 interactions have been directly linked to mRNA export defects when mutated, as also outlined in response to comment #4. For example:

- Deleting or mutating the UAP56 NTD (yeast *Sub2* Δ 6-17 and Y12S, corresponding to HsUAP56 residues 7-18 and Y13) leads to mRNA export defects in yeast (Saguez et al., 2013).
- Mutations in PCID2 or GANP, for which we know show that they directly impact the interface of TREX-2 with the UAP56 NTD, (yeast *ScThp1* R414D or K427D [human PCID2 K374 and K388]; *ScSac3* K467/468D [human GANP R895]) cause similar mRNA export defects (Ellisdon et al., 2012).
- Mutations in GANP which will affect the coordination of the 'wedge loop' (*ScSac3* R256A or R288D, corresponding to HsGANP R692 and R728),

and thus also impact the functional interaction of TREX-2 and UAP56, do also lead to an mRNA export defect (Schneider et al., 2015).

The central advance of our work lies in providing a mechanistic framework that directly rationalizes these well-documented phenotypes. Our data can rationalize a molecular basis for these previously observed defects, marking a significant step forward in understanding the mechanism of mRNA nuclear export.

We agree with the reviewer that additional analyses of mRNA export defects—such as those following rapid depletion of key components or upon perturbation of protein-protein interfaces— will be an important next step. We strongly believe that impactful future insights will be gained by focusing on the combination of rapid depletion experiments with unbiased, quantitative approaches like SLAM-seq, to evaluate mRNA export kinetics at high temporal and molecular resolution. We view these experiments as significantly beyond the scope of the current study, but consider them exciting future directions.

Referee #2 (Remarks to the Author):

The revised version leaves this reviewer very much torn.

The structural and biochemical work remains solid, it is high quality data and an impressive experimental effort. It is also clear that the authors made a valiant attempt to address reviewers' comments. Indeed, the in vitro model of the paper is strengthened by the newly added evidence, and the concerns raised by using artificial fusion proteins are fully resolved. However, a central issue remains: The cellular experiments fail to prove the relevance of the proposed model for nuclear export of mRNPs in cells (or even seem to contradict it, see below).

We sincerely thank the reviewer for the thoughtful and detailed feedback. We are pleased to hear that the biochemical, structural, and in vitro work is regarded as high quality, and we greatly appreciate the recognition of our efforts to address the reviewers' comments. We are particularly glad to hear that the newly added evidence has further strengthened the model and resolved the concerns previously raised regarding the use of peptide fusion proteins.

Regarding the reviewer's remaining central concern that the cellular experiments fail to fully support—or may even appear to contradict—the relevance of our proposed model for mRNP export in cells, we welcome the opportunity to clarify these experiments and findings in detail below.

Firstly, some experiments using UAP56/PCID2 mutants only test effects on cell viability, therefore these experiments show that mutations in the structure based interfaces are important for cell survival. This is a good start, but it is a "black box" result, which does not show whether the suggested functions of UAP56/SARNP/TREX-2 hold really true in cells.

While we understand the concerns raised, we respectfully disagree with the statement that our cellular data fail to support the relevance of our proposed mRNA export model in cells. Our study builds on a strong foundation of prior work that has already established the requirement of all factors in our model for mRNA nuclear export using various assays (see among others: (Portman, O'Connor and Dreyfuss, 1997; Piruat and Aguilera, 1998; Gatfield et al., 2001; Jensen et al., 2001; Sträßer and Hurt, 2001; Fischer et al., 2002; Sträßer et al., 2002; Zenklusen et al., 2002; Masuda et al., 2005;

Dufu et al., 2010; Wickramasinghe et al., 2010; Jani et al., 2012; Saguez et al., 2013; Ren, Schmiege and Blobel, 2017; Pacheco-Fiallos et al., 2023; Xie et al., 2023). Based on this existing knowledge, we designed our experiments to test whether the protein-protein interactions identified in vitro are also relevant in cellular contexts.

The following cellular evidence supports our model:

- 1. Knockout-Rescue Cell Viability Experiments: Our cell viability experiments (ED Figs. 3b, 5g, 10a) show that the mutations tested have functional relevance in vivo. While these assays do not directly address mRNA export, they demonstrate that disrupting the protein-protein interfaces identified in vitro of mRNA export-relevant proteins has significant consequences for cellular function, supporting their biological importance.*
- 2. The cellular viability data goes hand in hand with immunoprecipitation-western blot experiments: Here, we validate the identified interfaces in cells through immunoprecipitation-western blot experiments, probing how the identified mutations affect protein-protein interactions (ED Figs. 5h, 10b). These experiments reveal:*
 - UCM Binding Site in UAP56: Mutations in the UCM binding site reduce SARNP binding and diminish the interaction of UAP56 with TREX-2. This is consistent with our model, which predicts that reduced THO-released UAP56 limits its availability for TREX-2 interaction (ED Fig. 5h).*
 - UAP56 NTD Deletion: Deleting the UAP56 NTD reduces its interaction with TREX-2 (ED Fig. 5h), confirming this entirely novel protein-protein interface.*
 - PCID2 Mutations: Mutating residues in PCID2 that interact with the UAP56 NTD reduces UAP56 levels pulled down by PCID2 to background levels (ED Fig. 10b).*

All of these findings align closely with our in vitro results, demonstrating that the interfaces we identified at the molecular level are relevant in a cellular context.

Additionally, our findings provide molecular explanations for mRNA export defects observed in prior cell-based assays:

- Torben Heick Jensen's group showed that mutating or deleting the UAP56 NTD leads to mRNA export defects in yeast (Saguez et al., 2013).*

- *Alwin Köhler's and Murray Stewart's groups separately analyzed PCID2 and GANP mutants, which we can now rationalize as they are critical for either UAP56-NTD-binding or the coordination of the GANP wedge loop. These mutants would thus impair the functional interaction with UAP56. These mutants were also shown to exhibit mRNA export defects in vivo (Ellisdon et al., 2012; Schneider et al., 2015).*

By uncovering the molecular basis for these defects, our work bridges the gap between structural/biochemical insights and cellular phenotypes, offering a cohesive explanation for previously observed in vivo phenotypes.

Secondly, the tools exist: E.g. it is a pity that the authors do not make better use of their degron SARNP and PCID2 cell lines to show whether upon transient protein depletion, THO/UAP56/SARNP is enriched on mRNPs, and mRNAs are stuck in the nucleus. These cell lines, and other analogous cell lines for transient THO/ALYREF/UAP56/TREX-2 depletion are also the better tool for testing effects of mutants on complex compositions/changes to protein localization/mRNA export defects.

We thank the reviewer for this thoughtful suggestion and agree that the degron cell lines we have established during the revision will be valuable tools for future highly quantitative transcriptomics experiments to investigate mRNA export kinetics (by combining these cell lines with metabolic labeling and SLAM-seq approaches). However, regarding the specific expectations raised by the reviewer, we would like to highlight two important observations that suggest that cellular outcomes of perturbing this pathway are much more complex than we would have initially anticipated:

1. Steady-state saturation of THO complexes

We have shown in our previous work (Pacheco-Fiallos et al., 2023, Extended Data Fig. 10b) that at steady state, nearly all cellular THO complexes are incorporated into TREX-mRNP complexes (i.e., THO-UAP56-mRNPs) and almost no freely available THO complexes are detected (see our response to reviewer #1 on this). Additionally, cellular UAP56 levels are approximately 40-fold higher than THO levels (Schwanhäusser et al., 2011; Heath, Viphakone and Wilson, 2016), suggesting that even the little free THO complex is likely UAP56 bound at steady state. Consequently, following SARNP depletion, the total levels of THO-UAP56 complexes remain unchanged because the THO pool is already

saturated with UAP56 in cells. We apologize for not having been clear on this point previously. This observation is however consistent with our model and we have now edited the revised manuscript in lines 214-217 to clarify this.

2. Functional redundancy of the release step

We here describe SARNP as a release factor that contributes to the regulated disassembly of THO–UAP56 complexes. SARNP is identified as a “common essential” gene in human cells by the DepMap project (<https://depmap.org/portal/gene/SARNP>) and is broadly conserved across eukaryotes. However, it appears that SARNP may have functional redundancy with other UCM- or UCM-like domain-containing proteins. For instance, we identified that the mRNA biogenesis factor NCBP3 also contains a predicted UCM (Extended Data Fig. 6j). We further identified that RBM33, a protein that is important for the nuclear export of intronless and/or GC-rich RNAs, is predicted to bind UAP56 at a binding site overlapping the UCM interface (Extended Data Fig. 10g-j). This suggests that these or other proteins could act as partially redundant TREX disassembly factors, which complicates the interpretation of SARNP depletion experiments.

In light of these observations, we hope the reviewer agrees that the presented cellular data in the revised manuscript together with existing literature provide robust support for our proposed model and are consistent with our biochemical findings. We agree that degron-based approaches offer exciting opportunities for future investigations, but the intricate cellular context described above highlights the need for highly controlled and quantitative RNA sequencing methods, which go substantially beyond the scope of the current work.

Thirdly, the argument that several other labs have used the tethering assay in the past is not truly valid, because the technical possibilities of the present are different and today's genome editing tools allow for completely different approaches. The goal should be to find out whether the model mechanism that comes from the minimal in vitro system is actually happening in cells. The reductive system may be missing important components, it may be explaining only some aspects of mRNA export, or it may be entirely correct, but this remains open.

We thank the reviewer for their comment and appreciate the opportunity to clarify the role and interpretation of the tethering assay within the manuscript. We used this assay as one of multiple cellular assays that altogether support the proposed model, rather than as a standalone method to probe the pathway. In the revised manuscript, we use the tethering assay specifically to assess the effect of mutations in UAP56 that impact its interface with the TREX-2 complex. The results from this assay align with and are further corroborated by our knockout-rescue experiments, which independently probed and supported the importance of the newly identified UAP56–TREX-2 interfaces in cells.

We agree that advances in genome editing have provided new opportunities for studying these pathways. Indeed, we have incorporated state-of-the-art tools, such as degron-based approaches, into our study during the revision. We believe in this context, that the tethering assay provides a unique and direct readout of mRNA nuclear export upon mutational probing of UAP56-TREX2 interfaces, which is distinct from the endpoints measured in other cellular assays. As such, it provides complementary evidence that strengthens our conclusions and serves as a useful complement to the knockout-rescue experiments. This is particularly relevant in light of the reviewer's earlier comment (point 1) regarding the need for experiments that directly address mRNA export. If the reviewer feels strongly, we are however open to moving the tethering assay data to the supplement.

We respectfully disagree with the comment that our model is based solely on a minimal in vitro system. Our work integrates extensive and complementary data, including biochemical, structural, and cellular approaches. These layers of evidence extend well beyond a purely reductive system, providing both mechanistic insights and cellular relevance. The factors implicated in our model represent the minimal set of mRNA factors conserved from yeast to human: the THO complex, UAP56, ALYREF, SARNP and TREX-2. We do acknowledge in the manuscript in several contexts that more complex regulation and redundancies (e.g. for UBMs and UCMs) may exist. Our experiments outline a sequential pathway involving (1) TREX-mRNP remodelling and UAP56 clamping, and (2) TREX-2 binding at the NPC. These are two cornerstone steps of the pathway, for which we had lacked a mechanistic understanding. These steps further would likely be irreversible, because RNA-clamped UAP56 has no affinity for the THO complex and has to be unclamped before it can re-associate with the THO complex. We describe how SARNP can aid UAP56 clamping and THO release, but discuss that there in this step there might indeed be redundancy with other factors: we

found for instance a predicted UCM in the mRNA biogenesis factor NCBP3 (Extended Data Fig. 6j). In addition RBM33, implicated in the nuclear export of intronless RNAs, is predicted to bind UAP56 at a site overlapping the UCM-binding interface (Extended Data Fig. 10g-j). These findings suggest that other proteins such as NCBP3 and RBM33 likely act as additional TREX disassembly factors, contributing to UAP56–RNA clamping. We discuss this potential redundancy with SARNP in lines 416-419.

We hope these clarifications underscore the robustness of our study and its significant contribution to advancing the mechanistic understanding of mRNA export.

Lastly, potential contradictions between cellular experiments and in vitro model: Upon transient SARNP depletion THOC2 is not enriched in with UAP56 in the IP (Ext. Fig 5i). This runs counter to expectations based on the proposed mechanism. If SARNP is required for TREX disassembly, its removal should trap THO on UAP56. Analogously, if UAP56 cannot bind SARNP (UCM mut) it should enrich THO, however this is not observed in the IP (Ext. Fig. 5h). In contrast, the UAP56 deltaNTD mutant should bind less GANP and should get stuck at the SARNP-bound stage i.e. enrich SARNP, which is also not observed in the IP (Ext. Fig. 5h).

We thank the reviewer for raising these important points and apologize for not addressing them more explicitly in the previous version of our manuscript (the first two points raised here also overlap with Reviewer 1, point #2). We appreciate the opportunity to clarify these observations in greater detail.

SARNP Degron Experiment and UAP56 UCM Mutant:

We understand the reviewer’s concerns and agree that the interpretation of these experiments needs more context in the manuscript. We had demonstrated in our previous work (Pacheco-Fiallos et al., 2023, Extended Data Fig. 10b) that at steady state nearly all cellular THO complexes are incorporated into TREX–mRNP complexes (i.e., THO–UAP56–mRNPs), leaving available almost no free THO complex.

Additionally and consistent with this, the THO complex is the least abundant mRNA export pathway factor in human cells, with UAP56 being present at approximately 40-fold higher levels (Schwanhäusser et al., 2011; Heath, Viphakone and Wilson, 2016). This suggests that the very little free THO complex is likely also UAP56-bound at steady-state. Consequently, following the acute SARNP depletion, we observed that the total levels of THO–UAP56 complexes remain unchanged because the THO pool is

already saturated with UAP56 in cells. Similarly, when the UCM-binding site on UAP56 is mutated, the THO complex is not enriched because there is no significant pool of UAP56-free THO available at steady-state conditions. We have revised the manuscript text to better clarify these important points, which remains consistent with our model (lines 214-217). We note in addition that the near-complete saturation of THO complexes in TREX–mRNPs may further underscore the importance of regulated and efficient dissociation of THO from mRNPs, to make THO complexes available for engaging newly synthesized mRNPs. The SARNP degron experiment is therefore consistent with and supportive of our model.

UAP56 NTD Experiment:

We agree that UAP56 lacking the NTD (Δ NTD) mutant does not enrich for SARNP in immunoprecipitation experiments (ED Fig. 5h). In line with our expectations, we did observe a reduction in TREX-2 levels, when UAP56 Δ NTD is immunoprecipitated, consistent with a disruption of the TREX-2 interaction. However, the absence of SARNP enrichment in UAP56 Δ NTD IPs is unexpected and remains unexplained at present. One possible explanation is the relative instability of the UAP56–SARNP interaction in the context of IP experiments, which we have observed previously. Future work will be needed to address this observation in greater detail.

Minor points:

Ext. Fig. 5i: why is SARNP not detected (or not shown?) in FKBP-GFP-tagged cell line? An upward mass shift is expected and would confirm successful endogenous tagging. Is the band cut off? It would be helpful to indicate in the panel which protein is used as bait.

We apologize for any misunderstanding. In ED Fig. 5i, the anti-SARNP antibody is used to detect both the wild-type protein (lowest row) and the FKBP-GFP tagged protein (second to lowest row). As expected, we observe the following: (1) wild-type SARNP is present only in the wild-type cell line, (2) tagged SARNP is present in the tagged cell line, and (3) no SARNP is detected in the tagged SARNP cell line treated with dTAG-V1 for SARNP depletion.

Following the reviewer's suggestion, we have now explicitly indicated the bait protein in the figure panels for this figure, as well as for all other IP-MS experiments and in vitro

pull-down experiments. We agree that this addition improves the clarity and interpretability of the figures. Previously, this information was included only in the figure legend, for example, in ED Fig. 5i: “UAP56 is immunoprecipitated with or without the rapid, dTAG-V1-dependent depletion of cellular SARNP.”

Fig. 3, Ext. Fig. 7e, Ext. Fig. 8e: The density of UAP56RecA2 is so poorly defined that it seems unreasonable to model also the N-UBM/UCM-1 peptides. Are they visible in the density with sufficient clarity to allow their placement? A close-up of the density in this region should be shown as for UAP56 in Ext. Fig. 8d or the peptides should be removed.

We wish to clarify that we had indicated that UCM and N-UBM peptides were putative fits, owing to the underlying low-resolution cryo-EM density (Fig. 3e legend: [...] the UAP56 RecA2 lobe, UCM-1, and N-UBM are putatively fitted based on a low-resolution density”). We in addition did not include UCM and N-UBM peptides in the submitted PDB model for the same reason, to remain conservative. Because UCM and N-UBM were compatible with the binding of UAP56 in TREX-2 complexes in vitro (Fig. 3d, ED Fig. 7a,b), we had felt it helpful to visualize these. However, we agree with the reviewer’s suggestion to remove the peptides, and have done this in the revised manuscript and revised Fig. 3e.

Referee #3 (Remarks to the Author):

The authors have satisfyingly erased all my concerns and I congratulate them to this impactful work.

We thank the reviewer for their comments and support.

References

- Chi, B. *et al.* (2013) 'Aly and THO are required for assembly of the human TREX complex and association of TREX components with the spliced mRNA', *Nucleic Acids Research*, 41(2), pp. 1294–1306. Available at: <https://doi.org/10.1093/nar/gks1188>.
- Dufu, K. *et al.* (2010) 'ATP is required for interactions between UAP56 and two conserved mRNA export proteins, Aly and CIP29, to assemble the TREX complex', *Genes & Development*, 24(18), pp. 2043–2053. Available at: <https://doi.org/10.1101/gad.1898610>.
- Ellisdon, A.M. *et al.* (2012) 'Structural basis for the assembly and nucleic acid binding of the TREX-2 transcription-export complex', *Nature Structural & Molecular Biology*, 19(3), pp. 328–336. Available at: <https://doi.org/10.1038/nsmb.2235>.
- Fischer, T. *et al.* (2002) 'The mRNA export machinery requires the novel Sac3p–Thp1p complex to dock at the nucleoplasmic entrance of the nuclear pores', *The EMBO Journal*, 21(21), pp. 5843–5852. Available at: <https://doi.org/10.1093/emboj/cdf590>.
- Gatfield, D. *et al.* (2001) 'The DExH/D box protein HEL/UAP56 is essential for mRNA nuclear export in *Drosophila*', *Current biology: CB*, 11(21), pp. 1716–1721.
- Hautbergue, G.M. *et al.* (2009) 'UIF, a New mRNA Export Adaptor that Works Together with REF/ALY, Requires FACT for Recruitment to mRNA', *Current Biology*, 19(22), pp. 1918–1924. Available at: <https://doi.org/10.1016/j.cub.2009.09.041>.
- Heath, C.G., Viphakone, N. and Wilson, S.A. (2016) 'The role of TREX in gene expression and disease', *The Biochemical Journal*, 473(19), pp. 2911–2935. Available at: <https://doi.org/10.1042/BCJ20160010>.
- letswaard, R. *et al.* (2024) 'Genome-wide quantification of RNA flow across subcellular compartments reveals determinants of the mammalian transcript life cycle', *Molecular Cell*, 84(14), pp. 2765-2784.e16. Available at: <https://doi.org/10.1016/j.molcel.2024.06.008>.
- Jani, D. *et al.* (2012) 'Functional and structural characterization of the mammalian TREX-2 complex that links transcription with nuclear messenger RNA export', *Nucleic Acids Research*, 40(10), pp. 4562–4573. Available at: <https://doi.org/10.1093/nar/gks059>.
- Jensen, T.H. *et al.* (2001) 'The DECD box putative ATPase Sub2p is an early mRNA export factor', *Current Biology*, 11(21), pp. 1711–1715. Available at: [https://doi.org/10.1016/S0960-9822\(01\)00529-2](https://doi.org/10.1016/S0960-9822(01)00529-2).
- Kurshakova, M.M. *et al.* (2007) 'SAGA and a novel *Drosophila* export complex anchor efficient transcription and mRNA export to NPC', *The EMBO Journal*, 26(24), pp. 4956–4965. Available at: <https://doi.org/10.1038/sj.emboj.7601901>.
- Masuda, S. *et al.* (2005) 'Recruitment of the human TREX complex to mRNA during splicing', *Genes & Development*, 19(13), pp. 1512–1517. Available at: <https://doi.org/10.1101/gad.1302205>.

- Meignin, C. and Davis, I. (2008) 'UAP56 RNA helicase is required for axis specification and cytoplasmic mRNA localization in *Drosophila*', *Developmental Biology*, 315(1), pp. 89–98. Available at: <https://doi.org/10.1016/j.ydbio.2007.12.004>.
- Mor, A. *et al.* (2010) 'Dynamics of single mRNP nucleocytoplasmic transport and export through the nuclear pore in living cells', *Nature Cell Biology*, 12(6), pp. 543–552. Available at: <https://doi.org/10.1038/ncb2056>.
- Pacheco-Fiallos, B. *et al.* (2023) 'mRNA recognition and packaging by the human transcription–export complex', *Nature*, 616(7958), pp. 828–835. Available at: <https://doi.org/10.1038/s41586-023-05904-0>.
- Piruat, J.I. and Aguilera, A. (1998) 'A novel yeast gene, THO2, is involved in RNA pol II transcription and provides new evidence for transcriptional elongation-associated recombination', *The EMBO Journal*, 17(16), pp. 4859–4872. Available at: <https://doi.org/10.1093/emboj/17.16.4859>.
- Portman, D.S., O'Connor, J.P. and Dreyfuss, G. (1997) 'YRA1, an essential *Saccharomyces cerevisiae* gene, encodes a novel nuclear protein with RNA annealing activity.', *RNA*, 3(5), pp. 527–537.
- Ren, Y., Schmiege, P. and Blobel, G. (2017) 'Structural and biochemical analyses of the DEAD-box ATPase Sub2 in association with THO or Yra1', *eLife*. Edited by K. Weis, 6, p. e20070. Available at: <https://doi.org/10.7554/eLife.20070>.
- Saguez, C. *et al.* (2013) 'Mutational analysis of the yeast RNA helicase Sub2p reveals conserved domains required for growth, mRNA export, and genomic stability', *RNA*, 19(10), pp. 1363–1371. Available at: <https://doi.org/10.1261/rna.040048.113>.
- Schneider, M. *et al.* (2015) 'The Nuclear Pore-Associated TREX-2 Complex Employs Mediator to Regulate Gene Expression', *Cell*, 162(5), pp. 1016–1028. Available at: <https://doi.org/10.1016/j.cell.2015.07.059>.
- Schwanhäusser, B. *et al.* (2011) 'Global quantification of mammalian gene expression control', *Nature*, 473(7347), pp. 337–342. Available at: <https://doi.org/10.1038/nature10098>.
- Sträßer, K. *et al.* (2002) 'TREX is a conserved complex coupling transcription with messenger RNA export', *Nature*, 417(6886), pp. 304–308. Available at: <https://doi.org/10.1038/nature746>.
- Sträßer, K. and Hurt, E. (2000) 'Yra1p, a conserved nuclear RNA-binding protein, interacts directly with Mex67p and is required for mRNA export', *The EMBO Journal*, 19(3), pp. 410–420. Available at: <https://doi.org/10.1093/emboj/19.3.410>.
- Sträßer, K. and Hurt, E. (2001) 'Splicing factor Sub2p is required for nuclear mRNA export through its interaction with Yra1p', *Nature*, 413(6856), pp. 648–652. Available at: <https://doi.org/10.1038/35098113>.
- Umlauf, D. *et al.* (2013) 'The human TREX-2 complex is stably associated with the nuclear pore basket', *Journal of cell science*, 126(Pt 12), pp. 2656–2667. Available at: <https://doi.org/10.1242/jcs.118000>.

Wickramasinghe, V.O. *et al.* (2010) 'mRNA Export from Mammalian Cell Nuclei Is Dependent on GANP', *Current Biology*, 20(1), pp. 25–31. Available at: <https://doi.org/10.1016/j.cub.2009.10.078>.

Wilmes, G.M. *et al.* (2008) 'A Genetic Interaction Map of RNA Processing Factors Reveals Links Between Sem1/Dss1-Containing Complexes and mRNA Export and Splicing', *Molecular cell*, 32(5), pp. 735–746. Available at: <https://doi.org/10.1016/j.molcel.2008.11.012>.

Xie, Y. *et al.* (2023) 'Structural basis for high-order complex of SARNP and DDX39B to facilitate mRNP assembly', *Cell Reports*, 42(8), p. 112988. Available at: <https://doi.org/10.1016/j.celrep.2023.112988>.

Yamazaki, T. *et al.* (2010) 'The Closely Related RNA helicases, UAP56 and URH49, Preferentially Form Distinct mRNA Export Machineries and Coordinately Regulate Mitotic Progression', *Molecular Biology of the Cell*, 21(16), pp. 2953–2965. Available at: <https://doi.org/10.1091/mbc.e09-10-0913>.

Zenklusen, D. *et al.* (2002) 'Stable mRNP Formation and Export Require Cotranscriptional Recruitment of the mRNA Export Factors Yra1p and Sub2p by Hpr1p', *Molecular and Cellular Biology*, 22(23), pp. 8241–8253. Available at: <https://doi.org/10.1128/MCB.22.23.8241-8253.2002>.

List of responses to reviewer comments

“An ATP-gated molecular switch orchestrates human messenger RNA export”

(Ulrich Hohmann, Max Graf, László Tirián, Belén Pacheco-Fiallos, Ulla Schellhaas, Laura Fin, Dominik Handler, Alex W. Phillips, Daria Riabov-Bassat, Rupert Faraway, Thomas Pühringer, Michael-Florian Szalay, Elisabeth Roitinger, Julius Brennecke, and Clemens Plaschka)

Nature Manuscript 2024-02-02833

Reviewer comments are in *blue*.

Responses are in *black*.

Summary response to referees #1 and #2:

We thank both reviewers for their thoughtful and constructive evaluations throughout this review process. Their feedback has been important in strengthening the manuscript and refining our model. In response to the latest comments, we carried out additional in vitro and cellular experiments that further probe the key findings. Specifically, these new data provide additional cellular evidence for the placement of SARNP in the pathway, clarify the function of the UAP56–TREX-2 interaction in mRNA export, and support the sequential connection between TREX and TREX-2 complexes. We revised the manuscript to incorporate these findings alongside other changes, including a more balanced discussion of SARNP. We are grateful for the reviewers' time and efforts, and believe that these substantial additions and clarifications address their concerns and hope that they render the revised manuscript suitable for publication.

We first summarize the major experimental additions below and provide detailed responses, along with corresponding manuscript revisions, in the later point-by-point rebuttal to each comment.

Major experimental additions:

(1) Mapping the protein interactome of the UAP56 N-terminal domain (NTD).

To explore the binding partners of the UAP56 N-terminal domain (NTD) beyond its association with TREX-2, we synthesized a biotinylated UAP56 NTD peptide (residues 1–21) along with a scrambled control peptide. These peptides were immobilized on streptavidin beads, incubated with human K562 nuclear lysate, and the precipitated proteins were analyzed by western blotting and mass spectrometry (ED Figs. 9g,h). This approach confirmed the specific interaction between the UAP56 NTD and the TREX-2 complex (GANP, PCID2, ENY2, CETN3). Mass spectrometry also identified TPR, a key TREX-2 interactor that anchors the complex to the nuclear pore complex (NPC), strengthening the conclusion that the UAP56–TREX-2 interaction could take place in the context of the NPC.

Notably, we also detected LENG8 as a putative UAP56 NTD interactor. LENG8 is a poorly characterized protein with a SAC3 domain reminiscent of GANP in TREX-2. In collaboration with Torben Heick Jensen's group (Aarhus University), an expert in RNA turnover, we have recently characterized LENG8 and its binding partners. In contrast to

TREX-2, LENG8 does not play a role in mRNA export but instead plays a role in mRNP quality control (unpublished results). Importantly, the unpublished results from this collaboration show that the UAP56–LENG8 interaction does not conflict with the NTD's key role in linking mRNPs to TREX-2 for export.

Our analysis identified TREX-2 and LENG8 as the primary high-confidence interactors of the UAP56 NTD. While other proteins were detected (e.g. the nucleolar protein SRFBP1), none of these proteins are predicted to bind UAP56 by AlphaFold 2 and 3 or to have roles in mRNA export, indicating that these proteins are false positives and that TREX-2 and LENG8 are the major functional interactors of the NTD.

(2) Testing the functional relevance of the UAP56–TREX-2 interaction in mRNA export.
To further probe the role of the UAP56 NTD–TREX-2 interaction in cells, we established a poly(A) RNA FISH assay in K562 cells. We had previously shown that this interaction is essential for TREX-2 immunoprecipitation in cells (ED Fig. 9g,h), for maintaining cell viability (ED Fig. 5g,h), for the UAP56–TREX-2 interaction in vitro (Fig. 4c, ED Fig. 9d,f), and for efficient export of a reporter RNA (Fig. 4d).

For this new RNA FISH experiment, we used our PCID2 depletion cell line, in which PCID2 is tagged with FKBP12 (dTag) to enable rapid degradation using a dTAG small molecule. We combined the acute depletion of PCID2-GFP-FKBP with the doxycycline-inducible expression of either wild-type or mutant PCID2 rescue constructs (new Fig. 4e). Depletion of PCID2 caused a strong nuclear accumulation of poly(A) RNA, consistent with a block in mRNA export. The magnitude of this accumulation was comparable to that observed upon GANP depletion, another TREX-2 subunit. Importantly, the accumulation of poly(A) RNA FISH signal owing to the PCID2-depletion could be fully rescued by the dox-induced expression of wildtype PCID2. In contrast, mutant PCID2 constructs unable to bind to either GANP or the UAP56 NTD failed to restore the normal poly(A) RNA signal distribution.

These results provide cellular evidence that the UAP56–TREX-2 interaction is important for mRNA export, complementing our biochemical data and supporting the functional importance of this interface.

(3) Demonstrating the sequential functional connection between TREX and TREX-2.
To test the predicted sequential roles of TREX and TREX-2 in our mRNA export model, we performed the quantitative mass spectrometry analysis of UAP56 wildtype and mutant interactomes in cells. Our model predicts that selectively disrupting UAP56's

binding to THO (TREX) or TREX-2 would differentially impact its association with mRNPs and SARNP. Specifically, loss of the UAP56–TREX-2 interaction (while retaining THO-binding) should lead to accumulation of UAP56 on mRNPs together with SARNP, whereas loss of the UAP56–THO (TREX) interaction should instead decrease UAP56 association with mRNPs and SARNP, because TREX-2 cannot substitute for THO's mRNP-loading function. For more details on the expected outcomes, please see the new ED Fig. 11a and its figure legend.

To test these predictions, we engineered UAP56 point mutants that selectively disrupt binding to TREX-2 (M1: D49R, L51D), THO (M2: F336E, R339D), or to both (M3: M1+M2). Since the protein interfaces of UAP56–THO and UAP56–TREX-2 show some overlap, we first confirmed in vitro that these new UAP56 mutants behaved as expected, disrupting the intended interaction with one but not with the other complex for mutants M1 and M2 or with both for mutant M3 (ED Fig. 10j,k). We then expressed 3xFlag-tagged wild-type or mutant UAP56 constructs in K562 cells and performed triplicate anti-Flag immunoprecipitations under optimized conditions for detecting labile interactions, followed by quantitative mass spectrometry (new Fig. 5b, ED Fig. 10l).

The results show that:

- Loss of TREX-2 binding (M1) led to enrichment of mRNP proteins and SARNP with UAP56, indicating that UAP56 remains clamped on mRNPs when it cannot efficiently engage TREX-2.*
- Loss of THO binding (M2) caused a marked depletion of mRNP proteins and SARNP from UAP56, consistent with THO's role in loading UAP56 onto mRNPs. We note that UAP56 (M2) still enriched for TREX-2, likely reflecting increased association of free, unbound UAP56 (M2) with TREX-2.*
- Loss of both interactions with THO and TREX-2 (M3) produced a profile similar to M2, supporting the model that THO acts upstream of TREX-2 in the export pathway.*

Together, these data strongly support a sequential model of mRNA export (Fig. 5a):

- 1. THO (as part of TREX) loads UAP56 onto mRNPs in the nucleoplasm.*
- 2. SARNP associates with these UAP56–mRNP complexes.*
- 3. TREX-2 at the NPC then unloads UAP56 from mRNPs, enabling export.*

The close similarity of the UAP56 interactomes in the THO-binding mutant (M2) and the double mutant (M3) further support that THO and TREX-2 act in sequence, rather than in parallel, to drive mRNA nuclear export.

In conclusion, these complementary new biochemical and cellular experiments provide new functional evidence for the placement of SARNP in the pathway, clarify the role of the UAP56–TREX-2 interaction, and support the sequential handoff of mRNPs between TREX and TREX-2 in mRNA export. Taken together, these findings substantially strengthen our revised export model.

Referee comments:

Referee #1 (Remarks to the Author):

During the second round of revision, the authors provided explanations based on their own data and literature from other groups in response to my questions, which has alleviated some of my concerns. However, several issues remain.

1. Regarding my question on SARNP's role in promoting TREX disassembly in vivo, one explanation was that THO is 40-fold less abundant than UAP56, with nearly all THO associated with UAP56-mRNP complexes. However, in ED Fig. 5i, the co-IP efficiency of THOC2 is significantly lower than that of UAP56 and ALYREF, which contradicts the claim that all THO is bound by UAP56 under normal conditions—at least in this experimental setting. If SARNP indeed facilitates TREX-mRNP disassembly in vivo, an enhanced THO-UAP56 interaction should be observed. Contrary to this prediction, less THOC2 was co-IP'd with UAP56 upon SARNP degradation, while more THOC2 was present in the input. Even more puzzling, why were THOC2 and ALYREF not co-IP'd with UAP56 in WT cells?

We thank the reviewer for raising these points regarding Extended Data Fig. 5i (now ED Fig. 5k). For this experiment we used different antibodies to detect the different proteins and loaded the inputs and elution samples on the same SDS-PAGE gel. To visualize each protein, we selected different exposure times for the respective protein owing to the different quality of the respective antibody in providing a western blot signal. We assembled the final figure with these different exposures, best suited to each antibody's signal intensities. Thus, while we can compare the signal intensities among each respective protein within the inputs alone or the elutions alone, we cannot directly compare signal intensities between the inputs or elutions of these proteins. While this experiment remains consistent with SARNP acting downstream of THO–UAP56 and not

upstream, we think that our new experiment addition (Major experiment #3, see summary to referees) demonstrates this more clearly: By interfering with the binding of UAP56 to THO (mutant M1), and thus preventing the loading of UAP56 onto mRNPs and the predicted subsequent association of SARNP, SARNP levels indeed deplete substantially in the UAP56 M1 IP experiment (Fig. 5b). Together with our biochemical results that SARNP can help to dissociate THO from recombinant THO–UAP56 complexes (Fig. 2f) or endogenous THO–UAP56–mRNP complexes (Fig. 2g), and its increased affinity for UAP56–RNA complexes in vitro (Fig. 2h), these results are consistent with the placement of SARNP in the export pathway model (Fig. 5a).

Towards the last comment by the reviewer, we apologize for the suboptimal labelling of the western blot. In this experiment we immunoprecipitated UAP56 with a UAP56-specific antibody that we had immobilized on protein-G beads. As a control for protein background binding from cell extracts to the protein-G beads, we used protein-G beads without immobilizing the UAP56-specific antibody and assessed background binding using wildtype nuclear extract. This showed that protein-G beads alone, lacking the UAP56-specific antibody, did not immunoprecipitate UAP56 or any of its interactors, as expected. We thank the reviewer for pointing this out, based on which we edited the figure labels and the corresponding Methods section to clearly indicate when the UAP56-specific antibody was immobilized on the protein-G beads.

2. Regarding SARNP's role in UAP56 binding to mRNA and mRNA export, the authors cited Xie et al., 2023, to support its involvement in export. However, while this study reported mild poly(A) accumulation in the nuclei of SARNP KD cells, RNA-seq data indicated that only a small subset of mRNAs was retained. Given that the degron system is superior to KD in avoiding indirect effects, compensation, and inefficiencies, I am perplexed as to why the authors did not test it. Additionally, while they speculate that the limited effect of KD might be due to redundancy with other UCM-containing proteins, they also cite SARNP as a "common essential" gene in the human DepMap project to underscore its importance. This appears selective, as the authors seem to collect supporting evidence without fully striving for scientific accuracy. If they suspect redundancy, they should directly test it through co-KD experiments.

We thank the reviewer for these comments and agree that our discussion of the role of SARNP was not sufficiently balanced. We have revised the manuscript as follows, which we hope addresses these concerns.

We have removed the statement on SARNP essentiality in line 139, since although SARNP does score as ‘essential’ in 511 of the 1183 tested cell lines (depmap.com), the mean GeneEffect score for SARNP – the measure for cell essentiality – is more modest compared to the other mRNA export proteins (SARNP, –0.5, THO subunits, mean –1.3, UAP56 (here DDX39B, but in presence of DDX39A paralog), –0.8, ALYREF, –1.2, TREX-2 subunits (excluding paralogs CETN2/3), mean –1.2). To us, this more modest effect together with the previously reported effect of transcript-specific export defects following the siRNA depletion of SARNP (Xie et al., 2023), suggests that the action of SARNP assisting in THO dissociation from TREX–mRNPs may be (i) not as important in cells as other steps of the export pathway, (ii) may be differently required in different cell states or contexts, or (iii) may be redundant with other factors (see last paragraph). We note that the Reed group previously reported that SARNP overexpression has a dominant negative effect on mRNA export in cells (Dufu et al., 2010), highlighting the regulatory role of the SARNP–UAP56 interaction.

We share the reviewer’s enthusiasm for using the SARNP-degron system to probe its acute effects on mRNA export. However, to meaningfully address this would require combining SARNP depletion with transcriptome-wide metabolic labeling approaches (e.g., SLAM-seq with subcellular fractionation) to directly quantify nuclear export kinetics. These experiments constitute a substantial project in their own right and are beyond the current scope but represent an exciting direction for future work.

With regards to SARNP redundancy and probing double-knockouts or double-depletions. In addition to SARNP, we report an additional UCM-containing protein, NCBP3, and one other protein, RBM33, that binds UAP56’s UCM-binding site but with a peptide different to UAP56’s UCM. However, unlike SARNP, which is well-characterized by its multiple UCMs and role in mRNA export, NCBP3 is described to have additional functions in the mRNA biogenesis (Gebhardt et al., 2015; Dou et al., 2020), while RBM33 is intron-less mRNA specific (Thomas et al., 2022). In double-knockout or double-depletion experiments would thus be unable to discriminate UAP56-specific mRNA export defects from pleiotropic phenotypes at other mRNA biogenesis steps or to specific substrates, making the interpretation of such experiments considerably more challenging.

Taking these comments together, we have revised the manuscript to account for a more balanced view on SARNP (line numbers 139ff, 446ff) and hope the referee agrees to the changes.

3. In response to my question about the functional importance of UAP56-TREX2 interactions, the authors stated that they do not understand why truncation of UAP56's NTD, which interacts with TREX2, also weakens its interaction with SARNP. This raises the possibility that the NTD is important for UAP56's interactions with multiple proteins, such as ALYREF. To clarify this, the authors should compare the interactome of the NTD deletion mutant with the WT protein.

We thank the reviewer for the suggestion to investigate the possibility of other UAP56 NTD interacting proteins, beyond TREX-2. As detailed in the new experiment additions (Major experiment #1, summary to referees), we now show in a cellular assay using a mass spectrometry read out that the UAP56 NTD peptide is sufficient to bind TREX-2 and a TREX-2-like complex harboring the SAC3-domain containing protein LENG8 (ED Fig. 9g,h), which plays a role in mRNP quality control but not in mRNA export (unpublished results in collaboration with the Jensen lab). This shows that the UAP56 NTD targets TREX-2 complexes, but does likely not have additional interactions in cells beyond those reported here (ED Fig. 9g,h).

Additionally, regarding the functional importance of the identified interactions in mRNA export, the authors rely solely on literature references. However, in these studies, export defects do not necessarily imply disruption of the identified interactions. For example, in Saguez et al., 2013, mutation of yeast Y12 (human Y13) led to nuclear accumulation of poly(A) RNA, but it remains unclear whether this mutation significantly impacts UAP56-TREX2 interaction. This should be directly tested in mammalian cells by assessing both mRNA export and UAP56-TREX2 interaction.

We thank the reviewer for raising this important point about directly testing the functional consequences of the identified UAP56-TREX-2 interactions in mammalian cells. We have addressed this in several ways in the revised manuscript.

(1) *New experimental evidence supporting the UAP56-TREX-2 interaction:*

We now include a new UAP56 N-terminal domain (NTD) immunoprecipitation experiment using human cell extracts (ED Fig. 9g,h). This shows that wild-type

UAP56 NTD, but not a scrambled version, binds cellular TREX-2. These data support that the UAP56 NTD mediates a specific interaction with TREX-2 in cell extracts. Importantly, this result offers a mechanistic explanation for the nuclear poly(A) RNA accumulation observed in Saguez et al. (2013) upon mutation of the corresponding yeast UAP56 NTD: disruption of this domain likely impairs the UAP56–TREX-2 interaction, thereby blocking mRNA export.

(2) Additional evidence for the cellular relevance of the UAP56–TREX-2 interaction:

- *Reporter RNA tethering assay: Mutation of the UAP56–TREX-2 interface, either within the UAP56 NTD or at UAP56’s PCID2-binding site, significantly reduces the export-promoting effect of UAP56 when tethered to a reporter RNA (Fig. 4d, Extended Data Fig. 9i).*
- *Cell viability assays: Mutation of this interface on either UAP56 or TREX-2 impairs cell growth (Extended Data Figs. 5g,h, 10a–d) and reduces the UAP56–TREX-2 interaction (Extended Data Fig. 10e).*
- *New poly(A) RNA FISH experiment: Following the reviewer’s suggestion, we now provide additional evidence that disrupting the UAP56–TREX-2 interaction in human cells (via a TREX-2 mutant) leads to nuclear accumulation of poly(A) RNA FISH signal (Major experiment #2, detailed in the summary to referees). This result is consistent with a defect in mRNA export.*

Taken together, our new and existing results together support the cellular relevance of the UAP56–TREX-2 interface. Specifically, these data show that the UAP56 NTD mediates a functionally critical interaction with TREX-2 in human cells, consistent with prior data from yeast (Saguez et al. 2013). Our newly added experiments thereby link the disruption of this interface to defective mRNA export, strengthening our mechanistic model.

4. Finally, the authors claim that UAP56 interacts with NPC-bound TREX2. However, how can they be certain it is NPC-bound without imaging? High- or super-resolution imaging should be used to determine whether UAP56 binds TREX2/mRNA when NXF1/NPC function is blocked. Additionally, it is crucial to examine whether UAP56 remains associated with mRNPs in TREX2-depleted cells to directly assess its role in releasing UAP56 from mRNA in vivo.

We thank the reviewer for these additional comments. We think that the following evidence supports that UAP56 interacts with NPC-tethered TREX-2:

(1) Evidence that UAP56 interacts with NPC-bound TREX-2:

The new and prior data support that UAP56 binds the NPC-tethered form of TREX-2:

- New UAP56 NTD IP–MS data: In our newly added experiment (Extended Data Fig. 9g,h), immunoprecipitation of the UAP56 NTD from human cell extracts enriched both TREX-2 and TPR. Since TPR is the nucleoporin that anchors TREX-2 to the NPC, this strongly suggests that UAP56 associates with the NPC-tethered form of TREX-2.
- Full-length UAP56 IP–MS: Our wild-type UAP56 IP–MS dataset (Table S1) likewise enriched TREX-2 subunits and NPC components, including TPR, NUP42, and RAE1.
- Consistency with published studies: Previous work showed that NPC-optimized IPs co-purify UAP56 and TREX-2 (Oeffinger et al., 2007). In addition, human TREX-2 has been reported to be NPC-associated at steady state, implying that the UAP56–TREX-2 interaction normally occurs at the NPC.

Together, these findings support our interpretation that the functionally relevant UAP56–TREX-2 interaction would take place at the NPC.

(2) Functional rationale for an NPC-specific interaction:

The spatial separation of THO and TREX-2 would help to coordinate sequential steps in export: THO recruits UAP56 to mRNPs in the nucleoplasm, while TREX-2, which is restricted to the NPC, mediates UAP56 docking and release only once the mRNP reaches the NPC. This prevents the premature dissociation of UAP56 from mRNPs in the nucleoplasm. Supporting this, overexpression of a soluble, NPC-detached GANP fragment in yeast causes dominant-negative growth effects (Fischer et al., 2002), likely by inducing premature UAP56 release from mRNPs.

(3) On the reviewer’s imaging suggestion:

We agree that high- or super-resolution imaging of mRNPs and their associated proteins during export would provide exciting insights. However, we consider

such experiments substantially beyond the scope of this study. They would require simultaneous tagging of reporter mRNAs, UAP56, TREX-2, and/or NPC components in cells with acute NXF1 depletion, alongside live-cell imaging to capture transient mRNP–NPC interactions. Notably, prior acute NXF1 depletion studies did not show stable mRNA accumulation at the NPC (Segref et al., 1997; Herold, Klymenko and Izaurralde, 2001; Ben-Yishay et al., 2019; Aksenova et al., 2020), likely due to the short-lived nature of these interactions. While we are enthusiastic about pursuing this in future studies, these experiments remain technically challenging and are outside the current manuscript’s scope.

(4) New evidence for TREX-2–mediated UAP56 release in cells:

To directly probe the role of TREX-2 in releasing UAP56 from mRNPs, we performed a new IP–MS experiment (Major experiment #3, Fig. 5b; Extended Data Fig. 10j–l, 11a). We designed and validated UAP56 mutants that specifically disrupt binding to TREX-2 (M1), THO (M2), or to both TREX-2 & THO (M3). Comparing the interactomes of these mutants to wild-type UAP56 revealed that the TREX-2–binding–deficient mutant (M1) accumulates significantly more mRNP proteins and SARNP with UAP56, consistent with TREX-2 acting to release UAP56 from mRNPs at the NPC.

Taken together, our biochemical, proteomic, and literature-based evidence supports a model in which UAP56 engages TREX-2 at the NPC to mediate its release from mRNPs, coupling the nuclear loading of UAP56 by THO to UAP56–mRNP docking and export at the NPC.

Referee #2 (Remarks to the Author):

The paper by Hohmann, Plaschka and colleagues deals with the mechanism of nuclear mRNA export in human cells. In previous papers, the group had established the structure of THO-UAP56 bound mRNPs. In their current work they aim to address how mRNPs could be remodelled and prepared for export through the pore. Based on biochemical experiments as well as structural evidence from minimal reconstituted protein/RNA complexes of factors linked to mRNA export, the study suggests a directional handover mechanism of mRNPs that is governed by the ATPase UAP56. Specifically, the authors propose that SARNP first displaces THO from UAP56, which then clamps onto the mRNA. As a second step TREX-2M (GANP-PCID2-SEM1 complex) is suggested to release UAP56 from the mRNA, which could then be exported.

This is a tempting mechanism and both the biochemical and structural data are sound, and well controlled.

We thank the reviewer for the positive assessment of our molecular work!

A main concern of this reviewer (and similarly of reviewer 1), voiced in the initial and second review, is that the in vitro model is not sufficiently validated in cells, and may therefore not reflect the actual cellular mechanism of mRNP export. In the 2nd revision the authors provide no new experimental evidence and respond to this point by reiterating their own and literature findings. The authors also argue that experiments which would test their proposed mechanism in cells are beyond the scope of the current paper.

After considering the authors' arguments and checking the cited previous studies my concerns remain. It worries me that alternative scenarios are not sufficiently tested and cellular data that do not support (or even seem to contradict) the proposed in vitro model seem to be weighed less strongly.

Below, I am attempting to address the authors' points and highlight data that substantiate the concerns. Also, I try to suggest a few experiments that could work towards model confirmation in cells.

We thank the reviewer for their helpful comments. We hope the reviewer finds the new cellular experiments and other manuscript revisions satisfactory.

* Rescue cell viability assays: These assays show that the interfaces on UAP56 are important for cell survival. They do not show that the mRNP handover mechanism, the directionality of interactions or their sequential nature happens in cells. Furthermore, the data suggest that the UAP56 UCM binding site (i.e. the point where SARNP would bind to compete out THO) is less crucial for cell viability than the UAP56 N-term (i.e. the contact for TREX-2M binding, which would trigger release of the mRNP from UAP56) [Ext.Fig. 5g, 10a]. Therefore, this data could also be interpreted such that SARNP and/or other UCM site binding factors are less important than TREX-2M. But this would not be predicted according to the proposed model, where they act sequentially and should be equally important.

We appreciate the reviewer's thoughtful comments. These points focused on the (1) evidence for the sequential nature of the proposed mRNP handover model and (2) the differing severity of phenotypes observed when mutating UAP56–interactor interfaces.

(1) On the sequential nature of the mRNA export model:

We thank the reviewer for highlighting the need for stronger cellular evidence to support the proposed sequential mRNP handover mechanism. In our revised manuscript, we now include new data (Major experiments #1–3, detailed in the summary to referees) that address this point. Specifically, we show that perturbations of core UAP56 interactions lead to the expected accumulation of defined mRNP intermediates with UAP56 (Fig. 5b). Importantly, SARNP behaves in a manner consistent with the sequential model: it accumulates with UAP56 when the TREX-2–UAP56 interaction is disrupted (UAP56 mutant M1) but is depleted from UAP56 when the THO–UAP56 interaction is impaired (UAP56 mutant M2). Furthermore, the combined THO– and TREX-2–UAP56 double mutant (M3) reinforces these findings. Together, these results support a stepwise model of mRNP handover between THO, SARNP, and TREX-2.

(2) On the relative importance of UAP56 interfaces:

We agree with the reviewer that the rescue assays show a less pronounced growth defect upon mutation of the UAP56 UCM-binding site (SARNP-binding interface) compared to the TREX-2–UAP56 interface. We now mention in the

manuscript that SARNP may be a less important step of mRNA nuclear export in comparison to other UAP56 interactions (such as UAP56–TREX-2) (line numbers 446ff). This is also consistent with large-scale genetic dependency data from the DepMap project, where SARNP exhibits a weaker essentiality score (mean GeneEffect: –0.5 across 1,183 human cell lines) compared to other key export pathway components (e.g., THO subunits: –1.3; UAP56/DDX39B (note that DDX39A is still present): –0.8; ALYREF: –1.2; TREX-2 subunits: –1.2). Thus, while SARNP is highly conserved, it appears to act as a factor that enhances the efficiency of the pathway rather than as an indispensable core component. We do not exclude redundancy with other factors (see also our related response to reviewer #1, comment #2).

Our biochemical data show that SARNP binds UAP56 via its UCMs, facilitates THO dissociation from UAP56–mRNP complexes (Fig. 2f,g), and stabilizes RNA-bound UAP56 (Fig. 2h). These activities are likely to enhance the efficiency of mRNA export, but our data and model would also allow for the possibility that THO can dissociate from UAP56–mRNPs independently of SARNP, possibly on their own or aided by redundant or context-specific factors. Indeed, proteins such as RBM33 (Extended Data Fig. 11), which functions in intronless mRNA export, is predicted to bind with partial overlap to UAP56’s UCM-binding interface. We now mention this possibility in the revised manuscript (line numbers 446ff).

In summary, our revisions provide a more balanced discussion of SARNP’s role: while it aids in pathway efficiency and may be crucial under specific transcript, stress, or cellular contexts, it is not as essential for cell viability compared to the other mRNA export factors. We hope these clarifications and the new data satisfactorily address the reviewer’s concerns.

* Immunoprecipitations: As done currently, these assays only show that the structurally characterized interfaces between UAP56 and SARNP or PCID2 are involved in cellular interactions of these proteins. They do not support (or even partially contradict) that the mRNP handover mechanism, the directionality of interactions or their sequential nature happens in cells. While the V5-PCID2 mutant that should no longer bind UAP56 indeed loses UAP56 interaction in IPs (i.e. consistent with structural data), it also shows weaker GANP interaction (i.e. inconsistent with structural data). While the UAP56 UCM mutant interacts less well with SARNP and GANP (i.e. consistent with handover model),

the UAP56 Δ NTD mutant does not only lose GANP interaction (i.e. consistent with structure) but also has reduced SARNP interaction instead of the expected enrichment (i.e. contradicting handover model).

We thank the reviewer for these detailed observations. We address them in two parts, (1) to clarify the apparent discrepancies in the PCID2-binding mutant experiment and (2) to describe new experiments that address the directionality and sequential nature of the proposed mRNP handover mechanism.

(1) On the PCID2-binding mutant (Extended Data Fig. 10e):

We appreciate the reviewer's careful analysis of the PCID2 mutant data. We agree that the reduced GANP signal in the PCID2-binding mutant immunoprecipitations could appear inconsistent with the structural model. However, we would like to point out that GANP protein levels are already markedly reduced in the input of the PCID2 mutant overexpression compared to wild-type PCID2. We interpret this as a technical effect of mutant expression impacting GANP stability or co-complex formation, rather than as evidence against the model. This point is now mentioned in the revised manuscript in the relevant ED Fig. 10e legend.

(2) On the UAP56 Δ NTD mutant and probing of the mRNP handover mechanism:

We acknowledge the reviewer's concern that, in our previous western blot analysis (former Extended Data Fig. 5h), the UAP56 Δ NTD mutant displayed a reduction, rather than the expected enrichment, of SARNP interaction. To resolve this and to probe the sequential handover mechanism, we have undertaken two key steps for the revised manuscript:

(i) We optimized the buffer conditions for immunoprecipitations to preserve more labile interactions, increasing sensitivity to dynamic complexes.

(ii) We performed a comprehensive, quantitative immunoprecipitation–mass spectrometry (IP–MS) experiment (Fig. 5b, Major experiment #3) using newly designed UAP56 mutants.

These improved experiments provide a quantitative picture of SARNP: SARNP shows the expected behavior under conditions where the UAP56–TREX-2 interaction is

impaired, accumulating with UAP56 as predicted by our model. This supports the sequential transfer of mRNPs between these complexes.

In summary, we believe these optimizations and the addition of quantitative IP-MS experiment substantially strengthen our evidence for the directional and sequential nature of the mRNP handover mechanism. We thank the reviewer for prompting these improvements, which have significantly enhanced the manuscript.

* Tethering assays: Because proteins are artificially bound to mRNAs without the potential for their release, these assays are difficult to interpret and problematic, especially in the context of a proposed model where factors hand over the mRNP and sequentially release from the mRNP. Even if we forget these fundamental issues, then the data supports a part of the mechanism and contradicts other parts (see also section on "alternative scenarios" below). UAP56 increases export of the reporter mRNA and this effect is decreased (but not abolished), when UAP56 cannot hydrolyze ATP or bind TREX-2M (i.e. partially consistent with model). In contrast, while ALYREF tethering increases reporter mRNA export, neither THO nor SARNP tethering has such an effect (i.e. inconsistent with model). This is particularly surprising since SARNP binds UAP56 and UAP56 tethering does increase export. Furthermore, tethering of different TREX-2 factors only weakly promotes or even hampers (PCID2) reporter mRNA export, which could also be seen as contradicting the model.

We agree that the RNA tethering assay has inherent limitations for studying a mechanism that relies on the exchange of proteins. We now acknowledge these limitations in the revised manuscript (line numbers 311-313). However, we do not agree that the results contradict our model. Instead, they can be rationalized within our model, as detailed below.

(1) Why some tethered factors fail to promote export:

- THO (THOC1, THOC5): Because the THO complex is nuclear and not exported with mRNPs, RNA tethered to THO with high affinity would also remain nuclear. This is consistent with the observed lack of RNA export (Extended Data Fig. 9i).*
- SARNP: Two aspects likely explain the absence of a strong effect: (i) The affinity of SARNP for UAP56 is low (Kd 10–14 μ M, Extended Data Fig. 5a), making stable recruitment of UAP56 unlikely when tethered in isolation. (ii) Even if UAP56 would bind, this may not be sufficient to generate an RNA-clamped*

UAP56 conformation, which in cells likely requires THO. Importantly, the SARNP–UAP56 binding site overlaps with the THO–UAP56 interface (Fig. 2e), meaning SARNP cannot recruit the THO–UAP56 complex. These two aspects can explain the lack of an RNA export effect.

- *TREX-2 (ENY2, CETN3, PCID2): TREX-2 is anchored to the NPC via TPR. When tethered, the reporter RNA would be effectively immobilized at the nuclear basket, which would likely block productive RNA export instead of promoting it.*

(2) Why some tethered factors do promote export:

- *NXF1: Since NXF1 is not constitutively tethered to the NPC, but considered a “mobile nucleoporin”, it can bypass upstream quality control steps when tethered to an RNA (Grüter et al., 1998; Herold et al., 2000) and directly drive reporter RNA export.*
- *UAP56: UAP56 is not constitutively associated with the THO complex or with the NPC, allowing it to promote mRNA export when tethered. It is possible that RNA-clamping by UAP56 is not as critical when UAP56 itself is RNA-tethered (as suggested by the AACD-mutant) since multiple UAP56 molecules are brought to the reporter RNA simultaneously. In addition, RNA-tethered (concentrated) UAP56 may be able to engage with the NPC-tethered TREX-2 complex owing to multivalent UAP56–TREX-2 interactions. Mutation of specific interfaces of UAP56 with TREX-2 impaired but did not abolish reporter RNA export. This indicates that these isolated point mutations in UAP56 are important for efficient reporter RNA export. In this way, these results are orthogonal to our other cellular experiments, despite not being stand-alone sufficient.*
- *ALYREF: Tethering ALYREF enriches UBMs on the reporter RNA, which can recruit the THO–UAP56 complex, thereby enabling UAP56 loading and promoting reporter RNA export.*

Overall, we agree that this minimal tethering system does not replicate the ordered dissociation events and pathway timing that occurs in cells. Instead, this assay bypasses preceding steps of the pathway by tethering individual factors to the RNA. Nonetheless, when interpreted in this light, the observed outcomes are consistent with our mechanistic model: factors that can induce (UAP56, ALYREF) or mimic (NXF1) export-competence can promote export, whereas those that are nuclear-retained (THO), weakly interacting with UAP56 (SARNP), or NPC-tethered (TREX-2) do not.

In summary, while we agree that the tethering assay is an artificial system with limitations, when interpreted in the context of our broader dataset, it aligns with the proposed handover model. We now clarify the assay limitations in the revised manuscript (line numbers 311-313).

* MS data: Why is TREX-2 not listed in the MS data figures? Is it not detected (which would be unexpected) or not included (which should be changed)? Based on which criteria was the list of shown factors decided? It is possible that by selecting interactor subsets that are shown in the heat map, a bias is introduced.

The reviewer raises important points regarding the mass spectrometry (MS) data presentation that we wish to clarify.

Detection of TREX-2:

TREX-2 subunits were indeed detected in our UAP56 immunoprecipitation–MS experiments (Fig. 2b, Fig. 5b), including GANP, which is labeled in the newly added Fig. 5b and has also been added to Fig. 2b in response to this comment. In contrast, and aligning with the proposed model, we did not detect TREX-2 subunits in experiments where THO was immunoprecipitated (TREX–mRNP MS in Fig. 2b and TREX–mRNP disassembly assay in Fig. 2g). This is consistent with expectations, as THO acts upstream in UAP56-clamping, while TREX-2, which is stably anchored at the NPC, acts downstream by releasing UAP56 from mRNPs. Hence, no co-purification of TREX-2 is expected in THO-based IP-MS experiments.

Criteria for selecting proteins in heatmaps:

The heatmaps in Fig. 2b were designed to provide a concise and interpretable overview of the data. We prioritized proteins that are (i) mRNP-associated, (ii) evolutionarily conserved, and (iii) representative of key functional groups in the mRNA export pathway. This allowed us to simplify the large dataset for visual presentation without overwhelming the figure. However, we wish to emphasize that the complete, unfiltered MS datasets for all experiments are provided in the Supplementary Tables (UAP56 IP-MS: Table 1; mRNP disassembly assay: Table 2; UAP56 mutant IP-MS: Table 4), for full transparency and to allow readers to examine all detected proteins.

In the AlphaFold2 multimer predictions (Fig. 3b), both GANP and PCID2 (TREX-2

subunits) scored as predicted UAP56 interactors, consistent with their detection in our UAP56 IP-MS experiments. For completeness, we note that the TREX–mRNP MS data shown in Fig. 2b was previously published (Pacheco-Fiallos et al., Nature 2023) and is cited accordingly. In summary, TREX-2 was detected in the relevant UAP56 IP-MS datasets, and we have updated the manuscript figures to make this clearer. We hope these clarifications and additions address the reviewer’s concern.

* UAP56 abundance: If UAP56 is strongly overly abundant (the authors state 40-fold) then what would prevent it from binding ALYREF without THO, or to TREX-2 without any of the other factors?

This is a great question that relates very closely to three other comments below. We wish to refer the reviewer to our detailed responses to the questions “Why is THO needed to bind UAP56 first?” and “Why is SARNP needed to release THO from UAP56 first?” and “Why would it matter if UAP56 clamps or unclamps from mRNA in terms of mRNP delivery [...]?”.

* Previous literature:

The cited papers confirm that the studied mRNA export proteins are involved in mRNA export, and that the UAP56-TREX-2 interface is important for mRNA export. They do not show that the mRNP handover mechanism, the directionality of interactions or their sequential nature happens in cells.

We agree that the previously cited papers demonstrate the involvement of these mRNA export factors, including the UAP56–TREX-2 interface, in mRNA export but do not directly establish the handover mechanism, directionality, or sequential nature of these interactions in cells. Our intent in citing these studies was to place our work in the context of prior cellular evidence for the functional relevance of these proteins and interfaces. However, we recognize that these earlier studies did not directly probe the sequential handover mechanism we propose.

To address this gap, we have added new data in the revised manuscript (Major experiment addition #3, detailed in our summary response to referees and presented in the new Fig. 5b). These experiments examine the consequences of disrupting key UAP56 interfaces on the association of pathway components in cells, providing

functional evidence for the sequential connections between TREX-2 and earlier steps in the export pathway.

We also elaborate on these points in our responses to the 'alternative scenarios', where we clarify how our new findings strengthen the support for a stepwise RNA export model.

* Alternative scenarios (playing "devils advocate" here, these are meant as examples based on the proposed model, not a comprehensive list):

Why is THO needed to bind UAP56 first? ALYREF could also recruit UAP56 directly to mRNPs without THO. All proposed consecutive steps could happen without this first step. The authors state that UAP56 is even 40-fold more abundant than THO. ALYREF tethering does increase reporter mRNA export, while THO tethering does not (see, why this assay is problematic).

We find this a very interesting question. We think there may be three reasons why UAP56 recruitment to mRNPs would require its prior binding to THO, rather than being directly recruited by ALYREF or other UBM-containing proteins.

(1) Multivalent recruitment overcomes weak UAP56–ALYREF affinity:

The binding affinity of UAP56 for ALYREF is relatively low (15–30 μ M; Extended Data Fig. 3a). The human THO complex tetramerizes and can simultaneously engage up to four UAP56 molecules. This multivalent arrangement allows for cooperative binding of THO–UAP56 to ALYREF-rich mRNPs, effectively overcoming the weak individual UAP56–UBM interactions. Thus, THO likely enhances both the efficiency and stability of UAP56 loading onto mRNPs.

(2) THO primes UAP56 for RNA clamping:

THO binds the two UAP56 RecA lobes in a conformation that favors subsequent RNA engagement by UAP56. This "priming" promotes UAP56 RNA-clamping, which is in turn required for THO's release from UAP56, a prerequisite for export.

(3) THO protects genome integrity during transcription:

THO has an established role in preventing harmful R-loops that form during transcription (Piruat and Aguilera, 1998; Chávez and Aguilera, 1997; Huertas and Aguilera, 2003). This protective function depends on its recruitment to nascent

mRNPs, a process mediated efficiently through UAP56. Directly delivering THO–UAP56 to the transcriptome may thus provide a dual benefit: safeguarding genome integrity and initiating the formation of export-competent mRNPs.

New experimental support:

Our newly added cellular data (Major experiment addition #3, Fig. 5b) provide direct evidence that THO binding is necessary for efficient UAP56 association with mRNPs: mutation of the UAP56–THO interface (mutant M2) significantly reduces UAP56–mRNP interactions. While this establishes the functional importance of THO in recruiting UAP56, we now also point out in the manuscript discussion that a subset of UAP56 molecules may associate with mRNPs independently of THO (line numbers 428-430).

On the RNA tethering assay:

We wish to note that the RNA tethering assay results on THO- and ALYREF-tethering follow expectations, as outlined above. The THO complex is nuclear and does not export with mRNPs across the NPC and into the cytoplasm. The tethering of THO to our reporter RNA (with high affinity aptamers) would therefore be predicted to retain the reporter RNA in the nucleus, preventing RNA export. This expectation is consistent with the lack of RNA export signal observed in the assay upon THO-tethering (ED Fig. 9i). In contrast, the tethering of several ALYREF molecules to the reporter RNA does promote RNA export, as we would have expected, since a high local concentration of ALYREF molecules (UBMs on the reporter RNA) would efficiently recruit THO–UAP56 for UAP56 loading onto the reporter RNA to promote export.

Why is SARNP needed to release THO from UAP56 first? Based on the provided data TREX-2 could directly compete with THO for UAP56 binding (e.g. by first latching onto the UAP56 N-term), TREX-2 has a higher affinity than THO for UAP56 (0.07 μ M vs. 0.2–0.3 μ M). SARNP and TREX-2 could be parallel pathways or only co-incide on some transcripts.

We agree that the relative affinities of UAP56 for TREX-2 (0.07 μ M) and THO (0.2–0.3 μ M) would in principle allow TREX-2 to compete with THO for UAP56 binding. However, we believe that spatial organization and pathway timing can explain why SARNP plays a privileged role in releasing THO from UAP56 before TREX-2 engages.

(1) Spatial compartmentalization prevents direct competition:

In human cells, TREX-2 is tethered to NPC at steady state, while THO and SARNP are nucleoplasmic. Thus, during mRNP formation, UAP56 is first recruited by THO to nascent transcripts in the nucleoplasm, where it can then engage nucleoplasmic SARNP. In this compartment, TREX-2 is not available to compete for UAP56 binding, as it is NPC-tethered. Only later, as the mRNP diffuses to the NPC, does TREX-2 gain access to UAP56–mRNP complexes.

(2) Sequential roles of SARNP and TREX-2:

Our new cellular data (Major experiment addition #3, detailed in our ‘summary to the referees’ and shown in Fig. 5b) support this sequence: SARNP associates with UAP56 after its delivery by THO but before its release by TREX-2. This is consistent with SARNP stabilizing RNA-clamped UAP56 on the mRNP, while TREX-2, at the NPC, acts later to promote UAP56 dissociation and export through the NPC.

(3) Context-dependent variation during gene gating:

We note that alternative scenarios may occur under specific conditions, such as during “gene gating”, where a gene locus is positioned at the nuclear periphery. In such cases, TREX-2 may gain earlier access to the mRNP and bypass or diminish the role of SARNP.

Why would it matter if UAP56 clamps or unclamps from mRNA in terms of mRNP delivery (i.e. the molecular switch from the title)? UAP56 is bound by ALYREF constantly in the model and ALYREF is bound to the mRNP. Therefore, UAP56's hold onto the mRNP should not be lost by mRNA unclamping.

We also find this an interesting question. We agree that UAP56 remains associated with ALYREF on mRNPs in our model. In this model, we think that RNA-clamping by UAP56 would significantly influence the stability and handover dynamics of UAP56–mRNP complexes for three reasons:

(1) RNA clamping strengthens UAP56–mRNP association:

The intrinsic protein–protein affinity between UAP56 and ALYREF is relatively low (15–30 μM ; Extended Data Fig. 3a). When UAP56 dissociates THO (which tetramerized UAP56 and enhanced its avidity), the remaining UAP56–ALYREF

interaction alone may not be sufficient to stably retain the UAP56–ALYREF interaction. RNA clamping provides a compensating, high-affinity interaction between UAP56 and its RNA substrate, effectively anchoring UAP56 to the mRNP in the absence of THO and independent of the ALYREF–UAP56 interaction.

(2) SARNP further stabilizes RNA-clamped UAP56:

Once UAP56 clamps onto RNA, SARNP binding can further stabilize this RNA-bound complex, maintaining UAP56 association with the mRNP until its docking to TREX-2. This stepwise stabilization is supported by our new cellular data (Major experiment addition #3, Fig. 5b), which show that SARNP binds UAP56 after THO release but before TREX-2 engagement.

(3) Controlled release at the NPC:

RNA-clamping by UAP56 also provides a mechanism for controlled UAP56 release from mRNPs at the NPC. When TREX-2 binds RNA-clamped UAP56 at the NPC, it triggers UAP56 dissociation from the mRNA and the mRNP. At this stage, the low-affinity UAP56–ALYREF interaction may be insufficient to retain UAP56, leading to its effective release.

In summary, RNA-clamping by UAP56 provides the molecular “switch” that allows UAP56 to stably associate with the mRNP after THO release and to remain engaged until the UAP56–mRNPs reach the NPC, where TREX-2 binding promotes UAP56 dissociation.

There are also "unknown unknowns", i.e. factors or steps that are important in cells but are not part of the reconstituted system and are therefore not probed here. E.g. mass spec and AlphaFold2 data indicates that there are many more interactors (Fig. 2b, 2g, 3b, and MS datasets). If the effect of the studied factors on mRNA export is not tested in cells then such holes in our knowledge cannot be recognized. The observation that SARNP depletion (Xie et al., Cell reports 2023: siRNA 48h, A549 cells, 16% SARNP remaining) impairs only 1.2% of mRNAs in export could suggest that alternative explanations and untested factors exist. So does the (albeit problematic) tethering assay, and above mentioned MS data.

We thank the reviewer for highlighting the limitations of reconstituted biochemical systems relative to the complexity of intact cells. We agree that reconstituted systems can only capture aspects of a cellular process, that they do not necessarily capture all factors of cellular mRNA export, and that “unknown unknowns” may remain. We address these concerns in the following four points.

(1) Value of our biochemical and structural approach:

Despite the limitations of reconstituted studies, we believe that focusing on core mRNA export pathway factors (THO, UAP56, ALYREF, SARNP, TREX-2) using biochemical, kinetic, and structural approaches has been essential to uncover unexpected activities of THO, SARNP, and TREX-2 that modulate the UAP56 RNA-binding cycle and vice versa. These insights would have been extremely challenging to obtain in intact cellular contexts but provide a mechanistic framework for how mRNP handover between these factors may occur. Our choice of proteins was guided by decades of genetic and biochemical evidence, together with our own unbiased IP-MS data and AlphaFold2 predictions (Fig. 2b, Fig. 3b), which identified these proteins and their complexes as conserved and functionally relevant mRNA export factors.

(2) Strengthening the cellular evidence:

In response to reviewer feedback, we have added substantial cellular experiments to test key aspects of our model. In this revision, we introduced a new IP-MS-based assay (Major experiment addition #3, Fig. 5b) that supports the functional, sequential connection between THO and TREX-2 and positions SARNP downstream of THO but upstream of TREX-2.

(3) On SARNP’s contribution:

The data suggest that SARNP’s role in mRNA export is less important than THO and TREX-2 and that it may be context-dependent or that additional factors participate in transcript-specific or condition-specific pathways. As noted by the reviewer, the relatively modest effect of SARNP depletion in some datasets (e.g., Xie et al. Cell Reports 2023, despite technical limitations) highlights that alternative or redundant mechanisms may exist. We have now revised the manuscript to provide a more balanced discussion of SARNP’s role (see also our expanded response to Reviewer 1, Comment #2). In the future, we plan to use approaches such as nascent RNA metabolic labeling (e.g., SLAM-seq) with

cellular fractionation to further define SARNP's contribution across transcript classes, but these experiments are beyond the current manuscript's scope.

(4) Outlook and limitations:

Finally, we agree with the reviewer that additional, as-yet-uncharacterized factors (including those identified in our MS datasets) may fine-tune mRNA nuclear export. We mention in the revised manuscript discussion that these proteins represent exciting targets for future work (line numbers 446ff).

In summary, while we recognize the limitations inherent to biochemical reconstitution, our combined structural, biochemical, and cellular experiments together support the proposed mechanistic framework of the mRNA export pathway. This also leaves open exciting avenues for future discoveries of additional regulatory layers.

Potential experiments:

Using IPs (UAP56, SARNP, TREX-2, ALYREF, EJC) upon transfections of mutants in degron cell lines (i.e. 1-2 days after depletion & transfection when cells are still alive) as well as upon overexpression of mutant proteins the authors could at least start to test whether their model is valid in cells. Mutants and overexpression constructs could be chosen such that they are expected to disrupt consecutive steps, or move the equilibrium into a specific direction and enrichment of intermediate complexes should be detected, EJC components can be probed as a proxy for mRNP binding. Similarly, FRET assays, or localization studies (i.e. enrichment at nuclear periphery via NPC anchored TREX-2) could be used to show stalling of intermediate species.

Potential mutants e.g. UAP56 ATP hydrolysis deficient mutant, UAP56 ATP binding mutant, UAP56 SARNP binding mutant, SARNP UCM mutant, TREX-2 UAP56 binding mutant, UAP56 TREX-2 binding mutant...

We thank the reviewer for additional experiment suggestions that would probe the proposed mechanistic framework of the mRNA export pathway. The core features of the proposed pathway are the roles of THO in loading UAP56 onto mRNPs and the docking of mRNPs at the NPC-tethered TREX-2 complex, which would establish a sequential model for THO and TREX-2 actions on mRNPs relayed through the RNA-binding cycle of UAP56. Based on suggestions from both reviewers 1 & 2, we have for this revision round added new cellular experiments (Major experiment additions #1-2 and #3, summary to referees), that show that (#3) the steady-state protein interactome of

UAP56 changes as would be predicted from the export model (Fig. 5b, ED Fig. 10l, 11a), and (#1-2) that the UAP56 NTD–TREX-2 interaction is important in cells in an RNA FISH assay (Fig. 4e), complementing the previously provided CRISPR-knockout and RNA tethering assays. We hope that these new experiments together with the text revisions address the reviewer's concerns.

Other point:

Rescue assays - Extended figures 3b, 5g, 10a: Please provide western blots that show depletion of DDX39A/B or PCID2 and expression levels of rescue constructs (at a time point when cells can still be harvested e.g. days 2-3), since lack of rescue could also be explained by lack of rescue construct expression.

Thank you for this valuable suggestion. We have now added data to confirm protein depletion and rescue construct expression in the referenced experiments.

- *DDX39A/B: We provide western blots demonstrating effective depletion in our knockout experiments (Extended Data Fig. 3b).*
- *PCID2: Commercial antibodies failed to detect wildtype PCID2; therefore, we now show (i) depletion of the endogenously tagged PCID2-FKBP-GFP by anti-GFP western blot (Extended Data Fig. 10a), (ii) loss of GFP fluorescence by confocal microscopy (Fig. 4e), and (iii) a genotyping PCR confirming homozygous, endogenous tagging (Extended Data Fig. 10b), consistent with cell lethality upon PCID2 depletion without rescue (Extended Data Fig. 10d).*
- *Rescue construct expression: For all DDX39B (UAP56) and PCID2 variants, we used the FACS-based quantification of mScarlet-tagged constructs (N-terminal fusion for PCID2, C-terminal P2A-fusion for UAP56) two days after depletion or knockout. These rescue construct expression levels are now shown alongside the respective experiments (Extended Data Figs. 3c, 5g, 10c), confirming that all rescue constructs were expressed comparably.*

References

Aksenova, V. et al. (2020) 'Nucleoporin TPR is an integral component of the TREX-2 mRNA export pathway', *Nature Communications*, 11(1), p. 4577. Available at: <https://doi.org/10.1038/s41467-020-18266-2>.

Ben-Yishay, R. et al. (2019) 'Imaging within single NPCs reveals NXF1's role in mRNA export on the cytoplasmic side of the pore', *Journal of Cell Biology*, 218(9), pp. 2962–2981. Available at: <https://doi.org/10.1083/jcb.201901127>.

Chávez, S. and Aguilera, A. (1997) 'The yeast HPR1 gene has a functional role in transcriptional elongation that uncovers a novel source of genome instability', *Genes & Development*, 11(24), pp. 3459–3470. Available at: <https://doi.org/10.1101/gad.11.24.3459>.

Dou, Y. et al. (2020) 'NCBP3 positively impacts mRNA biogenesis', *Nucleic Acids Research*, 48(18), pp. 10413–10427. Available at: <https://doi.org/10.1093/nar/gkaa744>.

Dufu, K. et al. (2010) 'ATP is required for interactions between UAP56 and two conserved mRNA export proteins, Aly and CIP29, to assemble the TREX complex', *Genes & Development*, 24(18), pp. 2043–2053. Available at: <https://doi.org/10.1101/gad.1898610>.

Fischer, T. et al. (2002) 'The mRNA export machinery requires the novel Sac3p–Thp1p complex to dock at the nucleoplasmic entrance of the nuclear pores', *The EMBO Journal*, 21(21), pp. 5843–5852. Available at: <https://doi.org/10.1093/emboj/cdf590>.

Gebhardt, A. et al. (2015) 'mRNA export through an additional cap-binding complex consisting of NCBP1 and NCBP3', *Nature Communications*, 6(1), p. 8192. Available at: <https://doi.org/10.1038/ncomms9192>.

Grüter, P. et al. (1998) 'TAP, the Human Homolog of Mex67p, Mediates CTE-Dependent RNA Export from the Nucleus', *Molecular Cell*, 1(5), pp. 649–659. Available at: [https://doi.org/10.1016/S1097-2765\(00\)80065-9](https://doi.org/10.1016/S1097-2765(00)80065-9).

Herold, A. et al. (2000) 'TAP (NXF1) Belongs to a Multigene Family of Putative RNA Export Factors with a Conserved Modular Architecture', *Molecular and Cellular Biology*, 20(23), pp. 8996–9008. Available at: <https://doi.org/10.1128/MCB.20.23.8996-9008.2000>.

Herold, A., Klymenko, T. and Izaurralde, E. (2001) 'NXF1/p15 heterodimers are essential for mRNA nuclear export in *Drosophila*.', *RNA*, 7(12), pp. 1768–1780.

Huertas, P. and Aguilera, A. (2003) 'Cotranscriptionally Formed DNA:RNA Hybrids Mediate Transcription Elongation Impairment and Transcription-Associated Recombination', *Molecular Cell*, 12(3), pp. 711–721. Available at: <https://doi.org/10.1016/j.molcel.2003.08.010>.

Oeffinger, M. et al. (2007) 'Comprehensive analysis of diverse ribonucleoprotein complexes', *Nature Methods*, 4(11), pp. 951–956. Available at: <https://doi.org/10.1038/nmeth1101>.

Piruat, J.I. and Aguilera, A. (1998) 'A novel yeast gene, THO2, is involved in RNA pol II transcription and provides new evidence for transcriptional elongation-associated

recombination.’, *The EMBO Journal*, 17(16), pp. 4859–4872. Available at:
<https://doi.org/10.1093/emboj/17.16.4859>.

Segref, A. et al. (1997) ‘Mex67p, a novel factor for nuclear mRNA export, binds to both poly(A)+ RNA and nuclear pores’, *The EMBO Journal*, 16(11), pp. 3256–3271. Available at:
<https://doi.org/10.1093/emboj/16.11.3256>.

Thomas, A. et al. (2022) ‘RBM33 directs the nuclear export of transcripts containing GC-rich elements’, *Genes & Development [Preprint]*. Available at:
<https://doi.org/10.1101/gad.349456.122>.

Xie, Y. et al. (2023) ‘Structural basis for high-order complex of SARNP and DDX39B to facilitate mRNP assembly’, *Cell Reports*, 42(8), p. 112988. Available at:
<https://doi.org/10.1016/j.celrep.2023.112988>.

List of responses to reviewer comments

“An ATP-gated molecular switch orchestrates human messenger RNA export”

(Ulrich Hohmann, Max Graf, László Tirián, Belén Pacheco-Fiallos, Ulla Schellhaas, Laura Fin, Dominik Handler, Alex W. Philipps, Daria Riabov-Bassat, Rupert W. Faraway, Thomas Pühringer, Michael-Florian Szalay, Elisabeth Roitinger, Julius Brennecke, and Clemens Plaschka)

Nature Manuscript 2024-02-02833

Reviewer comments are in *blue*.

Responses are in *black*.

Referee comments:

Referee #1 (Remarks to the Author):

During the last round of revisions, the authors performed additional biochemical and functional assays, which have addressed most of my concerns. However, a few key issues remain:

We thank the reviewer for the encouraging comments.

1. The authors combined acute depletion of PCID2-GFP-FKBP with doxycycline-inducible expression of either wild-type or mutant PCID2 rescue constructs to assess the role of TREX2-UAP56 interactions in mRNA export (Fig. 4e). However, the manuscript only displays polyA+ RNA distribution in a single cell with or without PCID2 degradation. For the critical rescue experiments, no representative images are provided—only quantification data. To strengthen the findings, PolyA+ FISH images of the rescue experiments should be included. Furthermore, images showing a population of cells (rather than just one example) should be presented to ensure reproducibility and robustness.

Thank you for the suggestion. We now provide the new Extended Data Fig. 11, where we provide a visual description of the poly(A) RNA FISH experiment and analysis pipeline that was used for all experiments (ED Fig. 11a), representative images for each poly(A) RNA FISH experiment (ED Fig. 11b), and additional plots and statistical analyses of the data (ED Fig. 11c, d).

2. Regarding interaction Analysis Between UAP56 Mutants and TREX-2 (Extended Data Fig. 10I), the authors used three UAP56 mutants to investigate the sequential functional relationship between TREX and TREX-2. However, the interactions with TREX-2 components were not shown. To support their conclusion regarding the sequential binding of THO and TREX-2 to UAP56, the authors should include data on how these mutants affect UAP56's interaction with TREX-2, particularly whether TREX-2 binding is disrupted when THO binding is lost.

Thank you for the comment. We note that in this experimental setup, where we immunoprecipitated 3x-Flag-UAP56 (wildtype and mutants), the levels of TREX-2

(GANP) were below the western blot detection limit. However, we did detect TREX-2 using mass spectrometry as a quantitative and sensitive readout (as shown in Fig. 5b). We noted (Extended Data Fig. 12a) that for the UAP56 M2 mutant (THO-binding deficient), we had observed that the levels of immunoprecipitated GANP (TREX-2) increased. We speculated (Extended Data Fig. 12a) that TREX-2 increased because in this experiment there would be a substantial pool of free UAP56 M2 mutant protein since the mutant protein cannot engage with THO and therefore cannot be efficiently loaded onto mRNPs (Fig 5b). This pool of free UAP56 M2 mutant protein could freely engage the native TREX-2 complex. The interaction of the free UAP56 M2 mutant protein with TREX-2 would likely be driven by the specific UAP56–TREX-2 binding sites, which are not mutated in the THO-binding mutant M2. This is in agreement with our data that purified UAP56 binds TREX-2 in vitro also when not bound to RNA (Figs 3d, 4c, Extended Data Fig. 7a).

Overall, this experiment (Fig. 5b, Extended Data Fig. 10l) probes the linear pathway model using the association of UAP56 with mRNPs as the primary readout, upon the mutation of the interface of UAP56 with either TREX-2 (M1), THO (M2), or both (M3). The results are fully consistent with this linear model, as described in the previous revision round (main text, Extended Data Fig. 12a). Briefly, the UAP56 mutant M1 (TREX-2 binding deficient) accumulated mRNP proteins including SARNP with UAP56. The UAP56 mutant M2 (THO-binding deficient), depleted mRNP proteins and SARNP from UAP56. The UAP56 mutant M3 (M1+M2), also depleted mRNP proteins and SARNP from UAP56.

Consistent with the specificity of our mutants in vivo, the TREX-2 increase observed with the UAP56 M2 mutant was lost in the UAP56 M3 mutant, in which the TREX-2-binding interface was mutated in combination with the THO-binding interface. Importantly, we had previously also carried out in vitro interaction experiments to test the specificity of the UAP56 mutants M1, M2, and M3 used in Fig. 5b for their interactions with either the THO complex or TREX-2 complex using purified proteins (Extended Data Fig. 10j, k). These in vitro experiments confirmed the intended specificity of the used UAP56 mutants, enabling the cellular experiments in Fig. 5b.

3. Correction Needed (Extended Data Fig. 11a):

The label "M3" is incorrect.

Thank you for pointing this out. The label is corrected to 'M1+M2'. The relevant figure was renamed to Extended Data Fig. 12a, owing to the addition of Extended Data Fig. 11.

Referee #2 (Remarks to the Author):

The authors have now addressed my concerns. In particular, I appreciate their experimental efforts to verify their mechanistic model in cells and the more nuanced discussion of their results. These changes have in my view significantly strengthened the paper. Few open questions are left (e.g. how important is SARNP for the sequential hand-over, as the UAP56-SARNP binding mutant was not tested in the new IP-MS experiments), but these are for future studies to address. What remains are only smaller technical points that should not stand in the way of overall accepting the paper.

We thank the reviewer for the positive assessment of our revision efforts.

Fig. 4e - FISH data: Details are missing in the figure legend, some data is not shown and no statistical analysis for significance is provided. How many cells were quantified per replicate and what is the spread of the detected signal between different cells per replicate and between replicates (violin or box plots)? Are the observed changes significant? Why is the mCherry-PCID rescue plasmid expression signal not shown? For the whole experiment please provide IF images (and if available western blots) for rescue construct expression.

Please note that the z-plane used for the representative images shows very little of the cytoplasm and is dominated by the nucleus, which may be due to the rounded shape of the K562 cell line but likely results in a small dynamic range for the assay.

Thank you for these comments. We have made the following additions to address them:

- (1) We now provide additional information in the Fig. 4e legend (Lines 921-929), which includes (i) the PCID2 mutants used, (M1, PCID2 NTD-binding mutant, K374D and K388D; M2, PCID2 GANP-binding mutant, D356R and A365F) and (ii) the pairwise testing of statistical significance between conditions using the stringent and robust Welch t-test. This shows that observed effects were statistically significant with a p-value of at least <0.001.*
- (2) We added Extended Data Fig. 11. In this figure we first visualize the image analysis pipeline (also see Methods) and an example of cell segmentation (Extended Data Fig. 11a). The nucleus corresponds to the majority of the cellular area in K562 cells, as the reviewer predicted, likely due to these being non-adherent cells. We nevertheless observed highly reproducible poly(A) RNA FISH*

nuclear to cytoplasmic ratios in K562 cells in the various experiment conditions and replicates (Extended Data Fig. 11b, c, d). For the rescue construct expression experiments, cells were classified based on mScarlet fluorescence, a proxy for doxycycline-induced rescue construct expression (see Extended Data Fig. 11a, b), confirming the rescue constructs were expressed.

We now also provide box plots and the cell numbers for each of the four biological replicates of each experiment condition (Extended Data Fig. 11c). This includes controls in addition to those in Fig. 4e, including the GANP-FKBP-GFP RNA FISH nuclear to cytoplasm ratio without dTag treatment and the RNA FISH nuclear to cytoplasm ratios of mScarlet negative cells from each rescue condition. The cell numbers analyzed differs between experiments because the number of cells with mScarlet fluorescence, indicating expression of the respective rescue construct, is a subset of all cells in the imaged field of view. The spread of the RNA FISH signals among cells of the same condition was highly similar between replicates. We tested for statistical significance between conditions in Extended Data Fig. 11d using Welch t-tests. For completeness, in that same panel 11d, we also include the data shown in Fig. 4e and the additional controls from panel 11c. We also updated Fig. 4e to show statistical significance of the observed effects. We found that all relevant observed changes were significant, with p-values between < 0.001 to <0.00001.

Fig. 5b: Please state in figure legend which mutations UAP56 M1 and M2 mutants contain.

We updated the figure 5b legend to include the mutations: for M1 (D49R, L51D) and for M2 (F336E, R339D).

ED Fig. 5i: Why was the UAP56 Δ N mutant data now selectively deleted from the gel while the other IP is still shown? The data should be shown fully or replaced by a different IP under the new conditions because otherwise one gets an impression of selective data presentation/removal.

We apologize for having given this impression, which was not our intention. We removed the UAP56 Δ NTD mutant condition, because this experiment was superseded by the new and improved experiment in Fig. 5b, which uses more precise interface mutants in new cell lines, an improved protocol, and an MS readout. We had in the last

revision kept the UAP56 UCM mutant in ED Fig. 5i to show that mutation of UAP56's UCM-binding site impaired SARNP binding in cells. Yet we appreciate that this was not ideal. Because we still wanted to test the UAP56–SARNP interaction in cells, we have now made a new cell line, expressing the UAP56 UCM mutant with a 3x-Flag tag (as done for Fig. 5b) and repeated the UAP56 UCM-binding mutant IP and western blot. Consistent with the earlier result, mutation of the UAP56 UCM-binding site reduces SARNP binding. We note that this experiment is performed in a wildtype K562 cell background and that we observed co-precipitation of endogenous wildtype UAP56. The presence of endogenous wildtype UAP56 may explain the residual SARNP levels detected in the UAP56 UCM-binding mutant immunoprecipitation, since wildtype UAP56 can still bind SARNP. We wish to note that in vitro, the purified UAP56 UCM mutant no longer binds to a UCM peptide (Extended Data Fig. 4a). To improve the main text flow for the reader, we have edited the text to move the cellular UAP56 UCM mutant experiment (now Extended Data Fig. 4d) together with the related CRISPR-Cas9 knockout-rescue data (now Extended Data Fig. 4b,c) as to complement the in vitro characterization of the UAP56–SARNP interaction (Lines 148-153).

Supplementary Data Tables 1 and 4: It was not possible to open the Excel files (corrupted?).

We apologize for this inconvenience. We believe this might have been due to Supplementary Data Tables 1 and 4 being macro-containing Excel files (.xlsm format), which might cause problems with certain programs. We have converted these files into standard Excel files (.xlsx format) to avoid this issue. We thank you for bringing this to our attention.

Data availability: So far only the structural data is mentioned as publicly available. The MS datasets should also be submitted to a FAIR data repository (e.g. PRIDE).

Thank you for the comment. We have now submitted the MS datasets reported in Figures 2b, 2g, 5b, and in Extended Data Figures 9g and h to the PRIDE with the accession number PXD069399. This is now also referred to in the data availability statement.